# Sub-exponential time Sum-of-Squares lower bounds for Principal Components Analysis

**Aaron Potechin***
University of Chicago
potechin@uchicago.edu

**Goutham Rajendran***
University of Chicago
goutham@uchicago.edu

## Abstract

Principal Components Analysis (PCA) is a dimension-reduction technique widely used in machine learning and statistics. However, due to the dependence of the principal components on all the dimensions, the components are notoriously hard to interpret. Therefore, a variant known as sparse PCA is often preferred. Sparse PCA learns principal components of the data but enforces that such components must be sparse. This has applications in diverse fields such as computational biology and image processing. To learn sparse principal components, it's well known that standard PCA will not work, especially in high dimensions, and therefore algorithms for sparse PCA are often studied as a separate endeavor. Various algorithms have been proposed for Sparse PCA over the years, but given how fundamental it is for applications in science, the limits of efficient algorithms are only partially understood. In this work, we study the limits of the powerful Sum of Squares (SoS) family of algorithms for Sparse PCA. SoS algorithms have recently revolutionized robust statistics, leading to breakthrough algorithms for long-standing open problems in machine learning, such as optimally learning mixtures of gaussians, robust clustering, robust regression, etc. Moreover, it is believed to be the optimal robust algorithm for many statistical problems. Therefore, for sparse PCA, it's plausible that it can beat simpler algorithms such as diagonal thresholding that have been traditionally used. In this work, we show that this is not the case, by exhibiting strong tradeoffs between the number of samples required, the sparsity and the ambient dimension, for which SoS algorithms, even if allowed sub-exponential time, will fail to optimally recover the component. Our results are complemented by known algorithms in literature, thereby painting an almost complete picture of the behavior of efficient algorithms for sparse PCA. Since SoS algorithms encapsulate many algorithmic techniques such as spectral or statistical query algorithms, this solidifies the message that known algorithms are optimal for sparse PCA. Moreover, our techniques are strong enough to obtain similar tradeoffs for Tensor PCA, another important higher order variant of PCA with applications in topic modeling, video processing, etc.

## 1 Introduction

Principal components analysis (PCA) [62] is a popular data processing and dimension reduction routine that is widely used. It has numerous applications in Machine Learning, Statistics, Engineering, Biology, etc. Given a dataset, PCA projects the data to a lower dimensional space spanned by the principal components. The intuition is that PCA sheds lower order information such as noise

---

*Equal contribution

†A.P. was supported in part by NSF grant CCF-2008920 and G.R. was supported in part by NSF grants CCF-1816372 and CCF-2008920

but importantly preserves much of the intrinsic information present in the data that are needed for downstream tasks.

However, despite great optimality properties, PCA has its drawbacks. Firstly, because the principal components are linear combinations of all the original variables, it's notoriously hard to interpret them [84]. Secondly, it's well known that PCA does not yield good estimators in high dimensional settings [13, 97, 61].

To address these issues, a variant of PCA known as Sparse PCA is often used. Sparse PCA searches for principal components of the data with the added constraint of sparsity. Concretely, consider given data $v_1, v_2, \ldots, v_m \in \mathbb{R}^d$. In Sparse PCA, we want to find the top principal component of the data under the extra constraint that it has sparsity at most $k$. That is, we want to find a vector $v \in \mathbb{R}^d$ that maximizes $\sum_{i=1}^{m} \langle v, v_i \rangle^2$ such that $\|v\|_0 \leq k$.

Sparse PCA has enjoyed applications in a diverse range of fields ranging from medicine, computational biology, economics, image and signal processing, finance and of course, machine learning and statistics (e.g. [117, 89, 85, 115, 31, 2]). It's worth noting that in some of these applications, other algorithms are also often used to learn statistical models with sparse structure, such as greedy algorithms (e.g. [60, 81, 59, 124]) and score-based algorithms (e.g. [28, 90, 107]) but in this work, we focus on the widely used sparse PCA technique. Sparse PCA comes with the important benefit that the learnt components are easier to interpret. A notable example of this is to recover topics from documents [32, 95]. Moreover, this has important benefits for algorithmic fairness in machine learning.

A large volume of research has been devoted to study Sparse PCA and its variants. Algorithms have been proposed and studied by several works, e.g. [4, 83, 73, 33, 118, 20, 82, 34, 54, 23, 35, 29, 36]. For example, simple variants of PCA such as thresholding on top of standard PCA [61, 29] work well in certain parameter settings. This leads to the natural question whether more sophisticated algorithms can do better either for these settings or other parameter settings.

On the other hand, there have been works from the inapproximability perspective as well (e.g. [20, 54, 23, 73, 35, 118], see Section 3.1 for a more detailed overview) In particular, a lot of these inapproximability results have relied on various other conjectures, due to the difficulty of proving unconditional lower bounds. Despite these prior works, exactly understanding the limits of efficient algorithms to this problem is still an active research area. This is natural considering the importance of sparse PCA and how fundamental it is to a multitude of applications.

In this work, we focus on the powerful Sum-of-Squares (SoS) family of algorithms [113, 92, 96, 48] based on semidefinite programming relaxations. SoS algorithms have recently revolutionized robust machine learning, a branch of machine learning where the underlying dataset is noisy, with the noise being either random or adversarial. Robust machine learning has gotten a lot of attention in recent years because of its wide variety of use cases in machine learning and other downstream applications, including safety-critical ones like autonomous driving. For example, there has been a high volume of practical works in computer vision [114, 47, 121, 50, 112, 122, 42, 76] and speech recognition [57, 119, 106, 108, 78, 3, 91, 94]. In this important field, SoS has recently lead to breakthrough algorithms for long-standing open problems [16, 80, 51, 70, 43, 72, 14, 15, 111]. Highlights include

- Robustly learning mixtures of high dimensional Gaussians. This is an extremely important problem that has been subjected to intense scrutiny, with a long line of work culminating in [16, 80].
- Efficient algorithms for the fundamental problems of regression [70], moment estimation [72], clustering [14] and subspace recovery [15] in the presence of outliers. Also known as robust machine learning, this setting is more akin to real life data which almost always has outliers or corrupted data.

Moreover, SoS algorithms are believed to be the optimal robust algorithm for many statistical problems. In a different direction, SoS algorithms have led to the design of fast algorithms for problems such as tensor decomposition [53, 111].

Put more concretely, SoS algorithms, also known as the SoS hierarchy or the Lasserre hieararchy, offers a series of convex semidefinite programming (SDP) based relaxations to optimization problems. Due to its ability to capture a wide variety of algorithmic techniques, it has become a fundamental tool in algorithms and optimization. It was and still remains an extremely versatile tool for combinatorial

optimization [46, 9, 49, 102]) but recently, it is being extensively used in Statistics and Machine Learning (apart from the references above, see also [17, 18, 52, 100]).

Therefore, we ask (also raised by and posed as an open problem in the works [82, 54, 55])

*Can Sum-of-Squares algorithms beat known algorithms for Sparse PCA?*

In this work, we show that SoS algorithms cannot beat known spectral algorithms, even if we allow sub-exponential time! Therefore, this suggests that currently used algorithms such as thresholding or other spectral algorithms are in a sense optimal for this problem.

To prove our results, we will consider random instances of Sparse PCA and show that they are naturally hard for SoS. In particular, we focus on the Wishart random model of Sparse PCA. This model is a more natural modeling assumption compared to other random models that have been studied before, such as the Wigner random model.

Note importantly that our model assumptions only strengthen our results because we are proving impossibility results. In other words, if SoS algorithms do not work for this restricted version of sparse PCA, then it will not work for more general models, e.g. with general covariance or multiple spikes. We now describe the model.

The Wishart model of Sparse PCA, also known as the Spiked Covariance model, was originally proposed by [61]. In this model, we observe $m$ vectors $v_1, \ldots, v_m \in \mathbb{R}^d$ from the distribution $\mathcal{N}(0, I_d + \lambda uu^T)$ where $u$ is a $k$-sparse unit vector, that is, $\|u\|_0 \leq k$ and we would like to recover the principal component $u$. Here, the sparsity of a vector is the number of nonzero entries and $\lambda$ is known as the signal-to-noise ratio.

As the signal to noise ratio $\lambda$ gets lower, it becomes harder and maybe even impossible to recover $u$ since the signature left by $u$ in the data becomes fainter. But it's possible that this may be mitigated if the number of samples $m$ grows. Therefore, there is a tradeoff between $m, d, k$ and $\lambda$ at play here. Algorithms proposed earlier have been able to recover $u$ at various regimes. For example, if the number of samples is really large, namely $m \gg \max(\frac{d}{\lambda}, \frac{d}{\lambda^2})$, then standard PCA will work. But if this is not the case, we may still be able to recover $u$ by assuming that the sparsity is not too large compared to the number of samples, namely $m \gg \frac{k^2}{\lambda^2}$. To do this, we use a variant of standard PCA known as diagonal thresholding. Similar results have been obtained for various regimes, while some regimes have resisted attack to algorithms.

Our results here complete the picture by showing that in the regimes that have so far resisted attack by efficient algorithms, the powerful Sum of Squares algorithms also cannot recover the principal component. We now state our theorem informally, with the formal statement in Theorem 3.1.

**Theorem 1.1.** *For the Wishart model of Sparse PCA, sub-exponential time SoS algorithms fail to recover the principal component when the number of samples $m \ll \min(\frac{d}{\lambda^2}, \frac{k^2}{\lambda^2})$.*

In particular, this theorem resolves an open problem posed by [82] and [54, 55].

In almost all other regimes, algorithms to recover the principal component $u$ exist. We give a summary of such algorithms in Section 3, captured succinctly in Fig. 1. We say almost all other regimes because there is one interesting regime, namely $\frac{d}{\lambda^2} \leq m \leq \frac{\min(d,k)}{\lambda}$ marked by light green in Fig. 1, where we can show that information theoretically, we cannot recover $u$ but it's possible to do hypothesis testing of Sparse PCA. That is, in this regime, we can distinguish purely random unspiked samples from the spiked samples. However, we will not be able to recover the principal component even if we use an exponential time bruteforce algorithm.

We use our techniques to also obtain strong results for the related Tensor Principal components analysis (Tensor PCA) problem. Tensor PCA, originally introduced by [110], is a generalization of PCA to higher order tensors. Formally, given an order $k$ tensor of the form $\lambda u^{\otimes k} + B$ where $u \in \mathbb{R}^n$ is a unit vector and $B \in \mathbb{R}^{[n]^k}$ has independent Gaussian entries, we would like to recover the principal component $u$. Here, $\lambda$ is known as the signal-to-noise ratio.

Tensor PCA is a remarkably useful statistical and computational technique to exploit higher order moments of the data. It was originally envisaged to be applied in latent variable modeling and indeed, it has found multiple applications in this context (e.g. [5, 68, 69, 6]). Here, a tensor containing statistics of the input data is computed and then it's decomposed in order to recover the latent variables. Because of the technique's versatility, it has gathered a lot of attention in machine

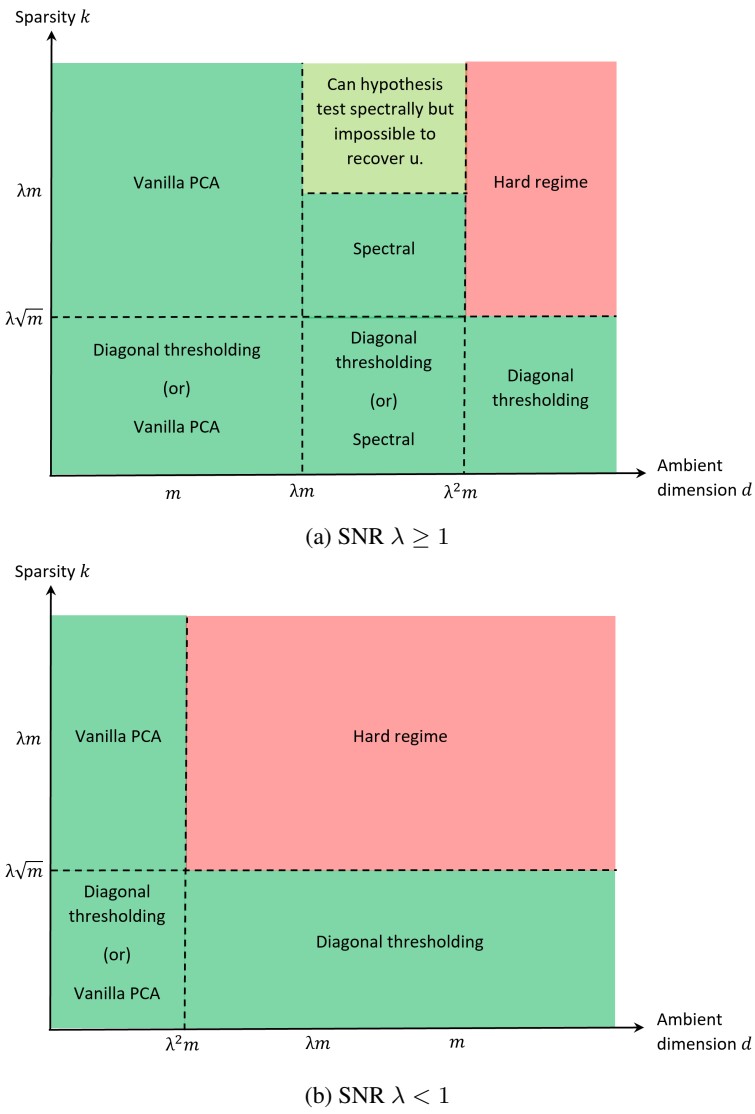

(a) SNR $\lambda \geq 1$

(b) SNR $\lambda < 1$

Figure 1: Computational barrier diagram for Sparse PCA

learning with applications in topic modeling, video processing, collaborative filtering, community detection, etc. (see e.g. [56, 7, 110, 5, 6, 38, 79] and references therein.)

For Tensor PCA, similar to sparse PCA, there has been wide interest in the community to study algorithms (e.g. [11, 22, 52, 53, 110, 125, 120, 67, 8]) as well as approximability and hardness (e.g. [88, 75, 24, 54], see Section 3.2 for a more detailed overview). It's worth noting that many of these hardness results are conditional, that is, they rely on various conjectures, sometimes stronger than P $\neq$ NP. Moreover, there has been widespread interest from the statistics community as well, e.g. [58, 98, 77, 26, 27], due to fascinating connections to random matrix theory and statistical physics.

In this work, we study the performance of sub-exponential time Sum of Squares algorithms for Tensor PCA. Our main result is stated informally below and formally in Theorem 3.2.

**Theorem 1.2.** *For Tensor PCA, sub-exponential time SoS algorithms fail to recover the principal component when the signal to noise ratio* $\lambda \ll n^{\frac{k}{4}}$.

In particular, this resolves an open question posed by the works [52, 22, 54, 55].

Therefore, our main contributions can be summarized as follows

1. Despite the huge breakthroughs achieved by Sum-of-Squares algorithms in recent works on high dimensional statistics, we show barriers to it for the fundamental problems of Sparse PCA and Tensor PCA.

2. We achieve optimal tradeoffs compared to known algorithms, thereby painting a full picture of the computational thresholds of tractable algorithms. This suggests that existing algorithms are preferrable for PCA and its variants.

3. Prior lower bounds for these problems have either focused on weaker classes of algorithms or were obtained assuming other hardness conjectures, whereas we prove high degree sub-exponential time SoS lower bounds without relying on any conjectures.

**Acknowledgements and Bibliographic note**   We thank Sam Hopkins, Pravesh Kothari, Prasad Raghavendra, Tselil Schramm, David Steurer and Madhur Tulsiani for helpful discussions. We also thank Sam Hopkins and Pravesh Kothari for assistance in drafting the informal description of the machinery (Section C). Parts of this work have also appeared in [99, 104].

## 2   Sum-of-Squares algorithms

The Sum of Squares (SoS) hierarchy is a powerful class of algorithms that utilizes the power of semidefinite programming for optimization problems, which has achieved breakthrough algorithms for many problems in machine learning and statistics. In this section, we briefly describe the sum of squares hierarchy of algorithms. For a more detailed treament with an eye towards applications to machine learning and statistics, see the ICM survey [102] or the monograph [43].

Given an optimization problem given by a program with polynomial constraints, the SoS hierarchy of algorithms gives a family of convex relaxations parameterized by an integer known as its degree. As the degree gets higher, the running time to solve the convex relaxation increases but on the other hand, the relaxation gets stronger and hence serves as a better algorithm. This offers a smooth tradeoff between running time and the quality of approximation. In general, we can solve degree-$D_{sos}$ SoS in $n^{O(D_{sos})}$ time [†]. Therefore, constant degree SoS corresponds to polynomial time algorithms which in general translates to efficient algorithms. In this work, we focus on and show limitations of degree $n^\varepsilon$ SoS which corresponds to subexponential running time.

Suppose we are given multivariate polynomials $p, g_1, \ldots, g_m$ on $n$ variables $x_1, \ldots, x_n$ (denoted collectively by $x$) taking real values. Consider the task:

$$\text{maximize } p(x) \text{ such that } g_1(x) = 0, \ldots, g_m(x) = 0$$

In general, we could also allow inequality constraints, e.g., $g_i(x) \geq 0$. In this work, we only have equality constraints but much of the theory generalizes when we have inequality constraints instead.

We now formally describe the Sum of Squares hierarchy of algorithms, via the so-called pseudo-expectation operators.

**Definition 2.1** (Pseudo-expectation values). *Given multivariate polynomial constraints $g_1 = 0, \ldots, g_m = 0$ on $n$ variables $x_1, \ldots, x_n$, degree $D_{sos}$ pseudo-expectation values are a linear map $\tilde{\mathbb{E}}$ from polynomials of $x_1, \ldots, x_n$ of degree at most $D_{sos}$ to $\mathbb{R}$ satisfying the following conditions:*

*1. $\tilde{\mathbb{E}}[1] = 1$,*

*2. $\tilde{\mathbb{E}}[f \cdot g_i] = 0$ for every $i \in [m]$ and polynomial $f$ such that $\deg(f \cdot g_i) \leq D_{sos}$.*

*3. $\tilde{\mathbb{E}}[f^2] \geq 0$ for every polynomial $f$ such that $\deg(f^2) \leq D_{sos}$.*

Any linear map $\tilde{\mathbb{E}}$ satisfying the above properties is known as a degree $D_{sos}$ pseudoexpectation operator satisfying the constraints $g_1 = 0, \ldots, g_m = 0$.

The intuition behind pseudo-expectation values is that the conditions on the pseudo-expectation values are conditions that would be satisfied by any actual expectation operator that takes expected values over a distribution of true optimal solutions, so optimizing over pseudo-expectation values gives a relaxation of the problem.

---

[†]In pathological cases, there may be issues with bit complexity but that will not appear in our settings. For details, see [93, 101]

**Definition 2.2** (Degree $D_{sos}$ SoS)**.** *The degree $D_{sos}$ SoS relaxation for the polynomial optimization problem*

$$\text{maximize } p(x) \text{ such that } g_1(x) = 0, \ldots, g_m(x) = 0$$

*is the program that maximizes $\tilde{\mathbb{E}}[p(x)]$ over all degree $D_{sos}$ pseudoexpectation operators $\tilde{\mathbb{E}}$ satisfying the constraints $g_1 = 0, \ldots, g_m = 0$.*

The main advantage is that the SoS relaxation can be efficiently solved via convex programming! In particular, Item 3 in Definition 2.1 is equivalent to a matrix being positive semidefinite, therefore the degree $D_{sos}$ SoS relaxation can be done via semidefinite programming [116]. This meta-algorithm is known as a degree-$D_{sos}$ SoS algorithm. This algorithm runs in $n^{O(D_{sos})}$ time[1] Therefore, constant degree SoS can be solved in polynomial time. In the next section, we apply SoS on PCA and formally state our results.

## 2.1 Related algorithmic techniques

**Statistical Query algorithms** Statistical query algorithms are another popular restricted class of algorithms introduced by [66]. In this model, for a given data distribution, we are allowed to query expected value of functions. Concretely, for a dataset $D$ on $\mathbb{R}^n$, we have access to it via an oracle that given as query a function $f : \mathbb{R}^n \to [-1, 1]$ returns $\mathbb{E}_{x \sim D} f(x)$ up to some additive adversarial error. SQ algorithms capture a broad class of algorithms in statistics and machine learning and have been used to study information-computation tradeoffs [109, 40, 30]. There has also been significant work trying to understand the limits of SQ algorithms (e.g. [40, 41, 34]). Formally, SQ algorithms and SoS are in general incomparable. However, the recent work [25] showed that under mild conditions, low-degree polynomial algorithms (defined next) and statistical query algorithms have equivalent power. But also, under these conditions, it's easy to see that SoS is a more powerful algorithm than low degree algorithms and hence, SoS algorithms are stronger than statistical query algorithms. Therefore, SoS lower bounds as shown in this work give strictly stronger evidence of hardness than SQ lower bounds.

**Low degree polynomial algorithms** In statistics, a hypothesis testing problem is a problem where the input is sampled from one of two distributions and we would like to identify which distribution it was sampled from. In this setting, a low degree polynomial algorithm is to compare the expectation of a low-degree polynomial to try and distinguish the two distributions. This method has been used to conjecture hardness thresholds for various problems [54, 55, 75]. However, under mild conditions, the SoS hierarchy of algorithms is more powerful than low degree polynomial algorithms [54] and therefore potentially yields better algorithms. Therefore, the SoS lower bounds shown in this work are stronger than low degree polynomial lower bounds as well.

# 3 Lower bounds for Sparse Principal Components Analysis

In this section, we will state our main results for Sparse PCA and Tensor PCA.

## 3.1 Sparse PCA

We recall the setting of the Wishart model of Sparse PCA: We are given $v_1, \ldots, v_m \in \mathbb{R}^d$ sampled from $\mathcal{N}(0, I_d + \lambda u u^T)$ where $u$ is a $k$-sparse unit vector and we wish to recover $u$.

We will further assume that the entries of $u$ are in $\{-\frac{1}{\sqrt{k}}, 0, \frac{1}{\sqrt{k}}\}$ chosen such that the sparsity is $k$ (and hence, the norm is 1). Note importantly that this assumption is only strengthening our result: If SoS cannot solve this problem even for this specific $u$, it cannot do any better for the general problem with arbitrary $u$.

Let the vectors from the given dataset be $v_1, \ldots, v_m$. Let them form the rows of a matrix $S \in \mathbb{R}^{m \times d}$. Let $\Sigma = \frac{1}{m} \sum_{i=1}^{m} v_i v_i^T$ be the sample covariance matrix. Then the standard PCA objective is to maximize $x^T \Sigma x$ and recover $x = \sqrt{k}u$. Therefore, the sparse PCA problem can be rephrased as

$$\text{maximize } \frac{m}{k} \cdot x^T \Sigma x = \frac{1}{k} \sum_{i=1}^{m} \langle x, v_i \rangle^2 \text{ such that } x_i^3 = x_i \text{ for all } i \leq d \text{ and } \sum_{i=1}^{d} x_i^2 = k$$

where the program variables are $x_1, \ldots, x_d$. The constraint $x_i^3 = x_i$ enforces that the entries of $x$ are in $\{-1, 0, 1\}$ and along with these constraints, the last condition $\sum_{i=1}^{d} x_i^2 = k$ enforces $k$-sparsity.

Now, we will consider the series of convex relaxations for Sparse PCA obtained by SoS algorithms. In particular, we will consider SoS degree of $d^\varepsilon$ for a small constant $\varepsilon > 0$. Note that this corresponds to SoS algorithms of subexponential running time in the input size $d^{O(1)}$.

Our main result states that for choices of $m$ below a certain threshold, when the vectors $v_1, \ldots, v_m$ are sampled from the unspiked standard Gaussian $\mathcal{N}(0, I_d)$, then sub-exponential time SoS algorithms will have optimal value at least $m + m\lambda$. This is also the optimal value of the objective in the case when the vectors $v_1, \ldots, v_m$ are indeed sampled from the spiked Gaussian $\mathcal{N}(0, I_d + \lambda uu^T)$ and $x = \sqrt{k}u$. Therefore, SoS is unable to distinguish $\mathcal{N}(0, I_d)$ from $\mathcal{N}(0, I_d + \lambda uu^T)$ and hence cannot solve sparse PCA. Formally,

**Theorem 3.1.** *For all sufficiently small constants $\varepsilon > 0$, suppose $m \leq \frac{d^{1-\varepsilon}}{\lambda^2}, m \leq \frac{k^{2-\varepsilon}}{\lambda^2}$, and for some $A > 0$, $d^A \leq k \leq d^{1-A\varepsilon}, \frac{\sqrt{\lambda}}{\sqrt{k}} \leq d^{-A\varepsilon}$, then for an absolute constant $C > 0$, with high probability over a random $m \times d$ input matrix $S$ with Gaussian entries, the sub-exponential time SoS algorithm of degree $d^{C\varepsilon}$ for sparse PCA has optimal value at least $m + m\lambda - o(1)$.*

In other words, sub-exponential time SoS cannot certify that for a random dataset with Gaussian entries, there is no unit vector $u$ with $k$ nonzero entries and $m \cdot u^T \Sigma u \approx m + m\lambda$. The proof of Theorem 3.1 is deferred to the appendix.

A few remarks are in order.

1. Note here that $m + m\lambda$ is approximately the value of the objective when the input vectors $v_1, \ldots, v_m$ are indeed sampled from the spiked model $\mathcal{N}(0, I_d + \lambda uu^T)$ and $x = \sqrt{k}u$. Therefore, sub-exponential time SoS is unable to distinguish a completely random distribution from the spiked distribution and hence is unable to solve sparse PCA.

2. The constant $A$ can be thought of as $\approx 0$ and it appears for technical reasons, to ensure that we have sufficient decay in our bounds (see Remark K.8). In particular, most values of $k, \lambda$ fall under the conditions of the theorem.

Informally, our main result says that when $m \ll \min\left(\frac{d}{\lambda^2}, \frac{k^2}{\lambda^2}\right)$, then subexponential time SoS cannot recover the principal component $u$. This is the content of Theorem 1.1

**Prior work on algorithms**   Due to its widespread importance, a tremendous amount of work has been devoted to obtaining algorithms for sparse PCA, both theoretically and practically, [4, 83, 73, 33, 118, 20, 82, 34, 54, 23, 35, 29, 36] to cite a few.

We now place our result in the context of known algorithms for Sparse PCA and explain why it offers tight tradeoffs between approximability and inapproximability. Between this work and prior works, we completely understand the parameter regimes where sparse PCA is easy or conjectured to be hard up to polylogarithmic factors. In Fig. 1a and Fig. 1b, we assign the different parameter regimes into the following categories.

- Diagonal thresholding: In this regime, Diagonal thresholding [61, 4] recovers the sparse vector. Covariance thresholding [73, 33] and SoS algorithms [37] can also be used in this regime. The benefits of these alternate algorithms are that covariance thresholding has better dependence on logarithmic factors and SoS algorithms works in the presence of adversarial errors.

- Vanilla PCA: Vanilla PCA (i.e. standard PCA) can recover the vector, i.e. we do not need to use the fact that the vector is sparse (see e.g. [21, 37]).

- Spectral: An efficient spectral algorithm recovers the sparse vector (see e.g. [37]).

- Can test but not recover: A simple spectral algorithm can solve the hypothesis testing version of Sparse PCA but it is information theoretically impossible to recover the sparse vector [37, Appendix E].

- Hard: A regime where it is conjectured to be hard for algorithms to recover the sparse principal component. We discuss this in more detail below.

In Fig. 1a and Fig. 1b, the regimes corresponding to Diagonal thresholding, Vanilla PCA and Spectral are dark green, while the regimes corresponding to Spectral* and Hard are light green and red respectively.

**Prior work on hardness** Prior works have explored statistical query lower bounds [25], basic SDP lower bounds [73], reductions from conjectured hard problems [21, 20, 23, 44, 118], lower bounds via the low-degree conjecture [35, 37], lower bounds via statistical physics [35, 12], etc. We note that similar threshold behaviors as us have been predicted by [37], but importantly, they assume a conjecture known as the low-degree likelihood conjecture. Similarly, many of these other lower bounds rely on various conjectures. To put in context, the low-degree likelihood conjecture is a stronger assumption than $P \neq NP$. In contrast, our results are unconditional and do not assume any conjectures.

Compared to these other lower bounds, there have only been two prior works on lower bounds against SoS algorithms [73, 21, 82] which are only for degree 2 and degree 4 SoS. In particular, degree 2 SoS lower bounds have been studied in [73, 21] although they don't state it this way. And [82] obtained degree 4 SoS lower bounds but they were very lossy, i.e. they hold for a strict subset of the *Hard* regime $m \ll \frac{k^2}{\lambda^2}$ and $m \ll \frac{d}{\lambda^2}$. Moreover, the ideas used in these prior works do not generalize for higher degrees. The lack of other SoS lower bounds can be attributed to the difficulty in proving such lower bounds. In this paper, we vastly strengthen these known results and show almost-tight lower bounds for SoS algorithms of degree $d^\varepsilon$ which correspond to sub-exponential running time $d^{d^{O(\varepsilon)}}$. We note that SoS algorithms get stronger as the degree increases, therefore our results immediately imply these prior results and even in the special case of degree 4 SoS, we improve the known lossy bounds. In summary, Theorem 3.1 subsumes all these earlier known results and is a vast improvement over prior known SoS lower bounds which provides compelling evidence for the hardness of Sparse PCA in this parameter range.

The work [54] also states SoS lower bounds for Sparse PCA but it differs from our work in three important aspects. First, they handle the related but qualitatively different Wigner model of Sparse PCA. Their techniques fail for the Wishart model of Sparse PCA, which is more natural in practice. We overcome this shortcoming and work with the Wishart model. We emphasize that their techniques are insufficient to handle this generality and overcoming this is far from being a mere technicality. On the other hand, our techniques can easily recover their results. Second, while they sketch a high level proof overview for their lower bound, they don't give a proof. On the other hand, our proofs are fully explicit. Finally, they assume the input distribution has entries in $\{\pm 1\}$, that is, they work with the $\pm 1$ variant of PCA. On the other hand, we work with the more realistic setting where the distribution is $\mathcal{N}(0, 1)$. Again, our techniques can easily recover their results as well.

### 3.2 Tensor PCA

We will now state our main result for Tensor PCA. Let $k \geq 2$ be an integer. We are given an order $k$ tensor $A$ of the form $A = \lambda u^{\otimes k} + B$ where $u \in \mathbb{R}^n$ is a unit vector and $B \in \mathbb{R}^{[n]^k}$ has independent Gaussian entries and we would like to recover the principal component $u$. Tensor PCA can be rephrased by the program

$$\text{maximize } \langle A, x^{\otimes k} \rangle = \langle A, \underbrace{x \otimes \ldots \otimes x}_{k \text{ times}} \rangle \text{ such that } \sum_{i=1}^{n} x_i^2 = 1$$

where the program variables are $x_1, \ldots, x_n$. The principal component $u$ will then just be the returned solution $x$. We will again consider sub-exponential time SoS algorithms, in particular degree $n^\varepsilon$ SoS, for this problem. This is sub-exponential time because the input size is $n^{O(1)}$.

We then show that if the signal to noise ratio $\lambda$ is below a certain threshold, then sub-exponential time SoS for the unspiked input $A \sim \mathcal{N}(0, I_{[n]^k})$ will have optimal value close to $\lambda$, which is also the optimal objective value in the spiked case when $A = \lambda u^{\otimes k} + B, B \sim \mathcal{N}(0, I_{[n]^k})$ and $x = u$. In other words, SoS cannot distinguish the unspiked and spiked distributions and hence cannot recover the principal component $u$.

**Theorem 3.2.** *Let $k \geq 2$ be an integer. For all sufficiently small $\varepsilon > 0$, if $\lambda \leq n^{\frac{k}{4} - \varepsilon}$, for an absolute constant $C > 0$, with high probability over a random tensor $A \sim \mathcal{N}(0, I_{[n]^k})$, the sub-exponential time SoS algorithm of degree $n^{C\varepsilon}$ for Tensor PCA has optimal value at least $\lambda - o(1)$.*

Therefore, sub-exponential time SoS cannot certify that for a random tensor $A \sim \mathcal{N}(0, I_{[n]^k})$, there is no unit vector $u$ such that $\langle A, \underbrace{u \otimes \ldots \otimes u}_{k \text{ times}} \rangle \approx \lambda$. The proof of Theorem 3.2 is deferred to the appendix.

We again remark that when the tensor $A$ is actually sampled from the spiked model $A = \lambda u^{\otimes k} + B$, the optimal objective value is approximately $\lambda$ when $x = u$. Therefore, this shows that sub-exponential time SoS algorithms cannot solve Tensor PCA.

Informally, the theorem says that when the signal to noise ratio $\lambda \ll n^{\frac{k}{4}}$, SoS algorithms cannot solve Tensor PCA, as stated in Theorem 1.2.

**Prior work**   Algorithms for Tensor PCA have been studied in the works [11, 22, 52, 53, 110, 125, 120, 67, 8]. It was shown in [22] that the degree $q$ SoS algorithm certifies an upper bound of $\frac{2^{O(k)}(n \cdot \text{polylog}(n))^{k/4}}{q^{k/4-1/2}}$ for the Tensor PCA problem. When $q = n^{\varepsilon}$ this gives an upper bound of $n^{\frac{k}{4}-O(\varepsilon)}$. Therefore, our result is tight, giving insight into the computational threshold for Tensor PCA.

Lower bounds for Tensor PCA have been studied in various forms including statistical query lower bounds [25, 39], reductions from conjectured hard problems [123, 24], lower bounds from the low-degree conjecture [54, 55, 75], evidence based on the landscape behavior [10, 88], etc. Compared to a lot of these works which rely on various conjectures, we remark that our lower bounds are unconditional and do not rely on any conjectures.

In [54], similar to Sparse PCA, they state a similar theorem for a different variant of Tensor PCA. However, they do not give a proof whereas we give explicit proofs. In particular, they state their result without proof for the $\pm 1$ variant of Tensor PCA whereas we work with the more realistic setting where the distribution is $\mathcal{N}(0, 1)$. We remark that their techniques do not recover our results but on the other hand, our techniques can recover theirs.

# 4   Related work

As stated in their respective sections, there have been some prior works on (degree at most 4) SoS lower bounds on Sparse and Tensor PCA and various other lower bounds that have mostly relied on various hardness conjectures, some of which are stronger than $P \neq NP$. The lack of results on higher degree SoS, compared to other models, can be attributed to the difficulty of proving such lower bounds, which we undertake in this work.

Sum of Squares lower bounds have been obtained for various problems of interest, such as Sherrington-Kirkpatrick Hamiltonian [45, 74, 63, 104], Maximum Cut [87], Maximum Independent Set [64, 104], Constraint Satisfaction Problems [71], Densest $k$-Subgraph [65], etc. The techniques used in this work are closely related to the work [19] which proved Sum of Squares lower bounds for a problem known as Planted Clique. Some of the ideas and techniques we employ in this work, namely pseudo-calibration and graph matrices have also appeared in other works [87, 103, 45, 1, 64, 105, 63, 65]. It's plausible that our generalized techniques could be applied to other high dimensional statistical problems, which we leave for future work.

# 5   Conclusion

In this work, we show sub-exponential time lower bounds for the powerful Sum-of-Squares algorithms for Sparse PCA and Tensor PCA. With the evergrowing research into better algorithms for Sparse PCA [4, 83, 73, 33, 118, 20, 82, 34, 54, 23, 35, 36] and Tensor PCA [11, 22, 52, 53, 110, 125, 120, 67, 8], combined with the recent breakthrough of Sum of Squares algorithms in statistics [16, 80, 51, 70, 43, 72, 14, 15, 111], it's therefore an important goal to understand whether Sum of Squares algorithms can beat state of the art algorithms for these problems.

In this work, we answer this negatively and show that even sub-exponential time SoS algorithms cannot do much better than relatively simpler algorithms. In particular, we settle open problems raised by [82, 54, 55, 52, 22]. Our work does not handle exponential time $\Omega(n)$ degree SoS so analyzing these algorithms is a potential future direction. Another important direction is to understand the limits

of powerful algorithms such as SoS for other statistical problems of importance, such as mixture modeling or clustering. For algorithm designers, our results illustrates the intrinsic difficulty of PCA problems and sheds light on information-computation gaps exhibited by PCA. For practitioners, this result provides strong evidence that existing algorithms work relatively well.

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
