# A   Overview and proof techniques

In this section, we present an overview of the ideas that go into the proofs of our main theorems Theorem 3.1 and Theorem 3.2. To do this, we develop a general meta-theorem that will enable us to show SoS lower bounds for a general class of "noisy" problems and simply invoke it for PCA. We take this approach because we expect this meta-theorem to be applicable to other problems of interest.

To show SoS lower bounds, we have to exhibit a feasible SoS solution, i.e. a pseudo-expectation operator $\tilde{\mathbb{E}}$, for our program that satisfy the constraints. A natural starting point for us is to apply the technique of pseudo-calibration [19] to construct a candidate SoS solution and then argue that it's feasible. We will cover this technique formally in Appendix B but the basic idea is as follows.

**Pseudo-calibration**   Consider a problem where we are trying to extract a structure (such as a sparse principal component in the case of Sparse PCA) from an input distribution (henceforth called the random distribution in this context). Then, pseudo-calibration proposes that we construct a "maximum entropy" planted distribution of inputs which has the given structure. Using this, we can construct candidate pseudo-expectation values $\tilde{\mathbb{E}}$ so that as far as low degree polynomials (of the input) are concerned, $\tilde{\mathbb{E}}$ for the random distribution mimics the behavior of the given structure for the planted distribution. This gives a candidate SoS solution.

Therefore, the first step is to construct a suitable planted distribution. For the problems of Sparse and Tensor PCA, we use the most natural distributions where we take a completely random input and "plant" the desired structure. We describe this formally next and state the results that we show in this appendix, from which our main theorems immediately follow as corollaries.

**Random and planted distributions**   Instate the notations of Theorem 3.1 and Theorem 3.2. For the Wishart model of Sparse PCA, we use the following distributions.

- Random distribution $\nu$: $v_1, \ldots, v_m$ are sampled from $\mathcal{N}(0, I_d)$ and we take $S$ to be the $m \times d$ matrix with rows $v_1, \ldots, v_m$.
- Planted distribution $\mu$: Sample $u$ from $\{-\frac{1}{\sqrt{k}}, 0, \frac{1}{\sqrt{k}}\}^d$ where the values are chosen with probabilites $\frac{k}{2d}, 1 - \frac{k}{d}, \frac{k}{2d}$ respectively. Then sample $v_1, \ldots, v_m$ as follows. For each $i \in [m]$, with probability $\Delta = d^{-\Theta(\varepsilon)}$, sample $v_i$ from $\mathcal{N}(0, I_d + \lambda uu^T)$ and with probability $1 - \Delta$, sample $v_i$ from $\mathcal{N}(0, I_d)$. Finally, take $S$ to be the $m \times d$ matrix with rows $v_1, \ldots, v_m$.

In Appendix E, we compute the SoS solution obtained by pseudo-calibration. We prove the following theorem.

**Theorem A.1.** *There exists a constant $C > 0$ such that for all sufficiently small constants $\varepsilon > 0$, if $m \leq \frac{d^{1-\varepsilon}}{\lambda^2}, m \leq \frac{k^{2-\varepsilon}}{\lambda^2}$, and there exists a constant $A$ such that $0 < A < \frac{1}{4}, d^{4A} \leq k \leq d^{1-A\varepsilon}$, and $\frac{\sqrt{\lambda}}{\sqrt{k}} \leq d^{-A\varepsilon}$, then with high probability, the SoS solution given by pseudo-calibration for degree $d^{C\varepsilon}$ Sum-of-Squares is feasible.*

For Tensor PCA, we use the following distributions. Let $k \geq 2$ be an integer.

- Random distribution $\nu$: Sample $A$ from $\mathcal{N}(0, I_{[n]^k})$.
- Planted distribution $\mu$: Let $\lambda, \Delta = n^{-\Theta(\varepsilon)} > 0$. Sample $u$ from $\{-\frac{1}{\sqrt{\Delta n}}, 0, \frac{1}{\sqrt{\Delta n}}\}^n$ where the values are taken with probabilites $\frac{\Delta}{2}, 1 - \Delta, \frac{\Delta}{2}$ respectively. Then sample $B$ from $\mathcal{N}(0, I_{[n]^k})$. Set $A = B + \lambda u^{\otimes k}$.

In Appendix D, we apply pseudo-calibration and we prove the following theorem.

**Theorem A.2.** *Let $k \geq 2$ be an integer. There exists a constant $C > 0$ such that for all sufficiently small constants $\varepsilon > 0$, if $\lambda \leq n^{\frac{k}{4}-\varepsilon}$, then with high probability, the SoS solution given by pseudo-calibration for degree $n^{C\varepsilon}$ Sum-of-Squares is feasible.*

We remark that in the planted distribution, we resample the coordinates with probability $1 - \Delta$. This resampling and the conditions involving the constant $A$ in Theorem A.1 are needed for technical reasons, see Remark J.5 and Remark K.8.

The two distributions discussed for each problem might remind the reader of hypothesis testing. Indeed, pseudo-calibration harnesses the intuition that it's hard for efficient algorithms to solve the natural hypothesis testing analogue where we have to distinguish between an alternative and a null hypothesis. We now present an overview of the proof techniques.

## A.1 Proof techniques

To show feasibility, it will be convenient to work with the notion of a moment matrix for a given pseudo-expectation operator.

**Definition A.3** (Moment Matrix of $\tilde{\mathbb{E}}$). *Given degree $d$ pseudo-expectation values $\tilde{\mathbb{E}}$, define the associated moment matrix $\Lambda$ to be a matrix with rows and columns indexed by monomials $p$ and $q$ such that the entry corresponding to row $p$ and column $q$ is*

$$\Lambda[p,q] := \tilde{\mathbb{E}}[pq].$$

It is easy to verify that Item 3 in Definition 2.1 equivalent to $\Lambda \succeq 0$, which is in fact why SoS relaxations can be solved via semidefinite programming.

To show feasibility of our constructed SoS solution, we develop a general meta-theorem to show that $\Lambda$ is PSD. The other constraints follow easily from pseudo-calibration (see Appendix B). To show PSDness of $\Lambda$, we construct certain *coefficient matrices* from $\Lambda$ and give conditions on these coefficient matrices which are sufficient to guarantee that $\Lambda$ is PSD with high probability. We now give an informal sketch of our main techniques. Some of these ideas are a generalization of the techniques used to prove the SoS lower bound for planted clique [19] but apart from generalizing their work, we needed to develop various other analysis techniques necessary to handle Gaussian inputs. Importantly, the notion of coefficient matrices are conceptually new and turn out to be essential for us.

**Shapes and graph matrices** We start by describing shapes and graph matrices, which were originally introduced by [19, 86] (also used in the planted clique SoS lower bound [19]) and later generalized in [1] (which we use here). They will be convenient for our analysis.

Shapes $\alpha$ are graphs that contain extra information about the vertices. Corresponding to each shape $\alpha$, there is a matrix-valued function $M_\alpha$ (i.e. a matrix whose entries depend on the input) that we call a graph matrix. Graph matrices are analogous to a Fourier basis, but for matrix-valued functions that exhibit a certain kind of symmetry. In our setting, $\Lambda$ will be such a matrix-valued function, so we can decompose $\Lambda$ as a linear combination of graph matrices $\Lambda = \sum_{\text{shapes } \alpha} \lambda_\alpha M_\alpha$.

Shapes and graph matrices have several properties which make them very useful to work with. First, $\|M_\alpha\|$ can be bounded with high probability in terms of simple combinatorial properties of the shape $\alpha$. Second, if two shapes $\alpha$ and $\beta$ match up in a certain way, we can combine them to form a larger shape $\alpha \circ \beta$. We call this operation shape composition. Third, each shape $\alpha$ has a canonical decomposition into three shapes, the left, middle and right parts of $\alpha$, which we call $\sigma$, $\tau$, and $\sigma'^T$. For this canonical decomposition, we have that $\alpha = \sigma \circ \tau \circ \sigma'^T$ and $M_\alpha \approx M_\sigma M_\tau M_{\sigma'^T}$. This decomposition is crucial for our analysis.

**A general framework for SoS lower bounds** We now sketch our strategy.

1. Decompose the moment matrix $\Lambda$ as a linear combination $\Lambda = \sum_{\text{shapes } \alpha} \lambda_\alpha M_\alpha$ of graph matrices $M_\alpha$.
2. For each shape $\alpha$, decompose $\alpha$ into a left part $\sigma$, a middle part $\tau$, and a right part $\sigma'^T$.
3. Based on the coefficients $\lambda_\alpha$ and the decompositions of the shapes $\alpha$ into left, middle, and right parts, construct coefficient matrices $H_{Id_U}$ and $H_\tau$.
4. Based on the coefficient matrices $H_{Id_U}$ and $H_\tau$, obtain an approximate PSD decomposition of $\Lambda$.
5. Show that the error terms (which we call intersection terms) can be bounded by the approximate PSD decomposition of $\Lambda$.

The strategy is similar to the work of [19] who showed SoS lower bounds for the planted clique problem but they do it in an ad-hoc manner, without defining or using coefficient matrices. As we will see, this abstraction makes the meta-theorem versatile.

We show that this analysis will succeed by distilling it as three conditions on the coefficient matrices. We have attempted to keep our meta-theorem general enough so that it can be used in other SoS lower bounds. The rough blueprint to use our theorem to prove SoS lower bounds is as follows.

1. Construct a candidate moment matrix $\Lambda$.

2. Decompose the moment matrix $\Lambda$ as a linear combination $\Lambda = \sum_{\text{shapes } \alpha} \lambda_\alpha M_\alpha$ of graph matrices $M_\alpha$ (akin to Fourier decomposition) and find the corresponding coefficient matrices.

3. Verify the required conditions on the coefficient matrices.

### A.1.1 A sketch of the intuition behind the conditions

We now motivate and sketch the conditions we present in our meta-theorem.

**Giving an approximate PSD factorization** As discussed above, we decompose the moment matrix $\Lambda$ as a linear combination $\Lambda = \sum_{\text{shapes } \alpha} \lambda_\alpha M_\alpha$ of graph matrices $M_\alpha$. We then decompose each $\alpha$ into left, middle, and right parts $\sigma$, $\tau$, and $\sigma'^T$. We now have that

$$\Lambda = \sum_{\alpha = \sigma \circ \tau \circ \sigma'^T} \lambda_{\sigma \circ \tau \circ \sigma'^T} M_{\sigma \circ \tau \circ \sigma'^T}$$

We first consider the terms $\sum_{\sigma, \sigma'} \lambda_{\sigma \circ \sigma'^T} M_{\sigma \circ \sigma'^T} \approx \sum_{\sigma, \sigma'} \lambda_{\sigma \circ \sigma'^T} M_\sigma M_{\sigma'^T}$ where $\tau$ corresponds to an identity matrix and can be ignored (which are called trivial shapes).

If there existed real numbers $v_\sigma$ for all left shapes $\sigma$ such that $\lambda_{\sigma \circ \sigma'^T} = v_\sigma v_{\sigma'}$, then we would have

$$\sum_{\sigma, \sigma'} \lambda_{\sigma \circ \sigma'^T} M_\sigma M_{\sigma'^T} = \sum_{\sigma, \sigma'} v_\sigma v_{\sigma'} M_\sigma M_{\sigma'^T} = (\sum_\sigma v_\sigma M_\sigma)(\sum_\sigma v_\sigma M_\sigma)^T \succeq 0$$

which shows that the contribution from these terms is positive semidefinite. In fact, this turns out to be the case for the planted clique analysis. However, this may not hold in general. To handle this, we note that the existence of $v_\sigma$ can be relaxed as follows: Let $H$ be the matrix with rows and columns indexed by left shapes $\sigma$ such that $H(\sigma, \sigma') = \lambda_{\sigma \circ \sigma'^T}$. Up to scaling, $H$ will be one of our coefficient matrices. If $H$ is positive semidefinite then the contribution from these terms will also be positive semidefinite. In fact, this will be the PSD mass condition of our main theorem, see Theorem C.37.

**Handling terms with a non-trivial middle part** Unfortunately, we also have terms $\lambda_{\sigma \circ \tau \circ \sigma'^T} M_{\sigma \circ \tau \circ \sigma'^T}$ where $\tau$ is non-trivial. Our strategy will be to charge these terms to other terms. For the sake of simplicity, we will describe how to handle one term. A starting point is the following inequality. For a left shape $\sigma$, a middle shape $\tau$, a right shape $\sigma'^T$, and real numbers $a, b$,

$$(aM_\sigma - bM_{\sigma'} M_{\tau^T})(aM_\sigma - bM_{\sigma'} M_{\tau^T})^T \succeq 0$$

which rearranges to

$$ab(M_\sigma M_\tau M_{\sigma'^T} + (M_\sigma M_\tau M_{\sigma'^T})^T) \preceq a^2 M_\sigma M_{\sigma^T} + b^2 M_{\sigma'} M_{\tau^T} M_\tau M_{\sigma'^T}$$
$$\preceq a^2 M_\sigma M_{\sigma^T} + b^2 \left\| M_\tau \right\|^2 M_{\sigma'} M_{\sigma'^T}$$

If $\lambda_{\sigma \circ \tau \circ \sigma'^T}^2 \left\| M_\tau \right\|^2 \leq \lambda_{\sigma \circ \sigma^T} \lambda_{\sigma' \circ \sigma'^T}$, then we can choose $a, b$ such that $a^2 \leq \lambda_{\sigma \circ \sigma^T}, b^2 \left\| M_\tau \right\|^2 \leq \lambda_{\sigma' \circ \sigma'^T}$ and $ab = \lambda_{\sigma \circ \tau \circ \sigma'^T}$. This will approximately imply

$$\lambda_{\sigma \circ \tau \circ \sigma'^T}(M_{\sigma \circ \tau \circ \sigma'^T} + M_{\sigma \circ \tau \circ \sigma'^T}^T) \preceq \lambda_{\sigma \circ \sigma^T} M_{\sigma \circ \sigma^T} + \lambda_{\sigma' \circ \sigma'^T} M_{\sigma' \circ \sigma'^T}$$

which will give us a way to charge terms with a nontrivial middle part against terms with a trivial middle part.

While we could try to apply this inequality term by term, it is not strong enough to give us our results. Instead, we generalize this inequality to work with the entire set of shapes $\sigma, \sigma'$ for a fixed $\tau$. This will lead us to the middle shape bounds condition.

**Handing intersection terms** There's one technicality in the above calculations. Whenever we decompose $\alpha$ into left, middle, and right parts $\sigma$, $\tau$, and $\sigma'^T$, $M_\sigma M_\tau M_{\sigma'^T}$ is only approximately equal to $M_\alpha = M_{\sigma \circ \tau \circ \sigma'^T}$. All the other error terms have to be carefully handled in our analysis. We call these terms intersection terms.

We exploit the fact that these intersection terms themselves are graph matrices. Therefore, we recursively decompose them into $\sigma_2 \circ \tau_2 \circ \sigma_2'^T$ and apply the previous ideas. To do this methodically, we employ several ideas such as the notion of intersection patterns and the generalized intersection tradeoff lemma (see Appendix G). Properly handling the intersection terms is one of the most technically intensive parts of our work. This analysis leads us to the intersection term bounds condition.

### A.2 Organization of the appendix

The remainder of this appendix is organized as follows. In Appendix B, we describe pseudo-calibration in more detail. In Appendix C, we present the qualitative statement of the main theorem. In Appendix D and Appendix E, we qualitatively verify the conditions for tensor PCA, and sparse PCA respectively. In Appendix F, we introduce more formal definitions and state a quantitative version of the main theorem, with the proof following in the next few appendices. In Appendix J and Appendix K, we complete the proofs of our applications and in particular, we obtain the quantative tradeoffs we desire.

## B  Pseudo-calibration

Psuedo-calibration is a heuristic introduced by [19] to construct candidate pseudo-expectation values on instances of an optimization problem in order to exhibit SoS integrality gaps. It does this almost mechanically by considering a planted distribution supported on instances of the problem with large objective value and uses this planted distribution as a guide to construct the pseudo-expectation values. This has been successful for various high-degree SoS lower bounds in the literature, e.g., Sherrington-Kirkpatrick [45, 87], Planted Clique [19], Max-$k$-CSPs [71, 103], Max-Cut [87], etc. A variant was used in the problem of Independent set [64].

For our applications, psuedocalibration is used to obtain a candidate pseudoexpectation operator $\tilde{\mathbb{E}}$. from the random vs planted problem. This will be the starting point for all our applications. Here, we do not attempt to motivate and describe it in great detail. Instead, we will briefly describe the heuristic, the intuition behind it and show an example of how to use it. A detailed treatment can be found in [19].

Let $\nu$ denote the random distribution and $\mu$ denote the planted distribution. Let $v$ denote the input and $x$ denote the variables for our SoS relaxation. The main idea is that, for an input $v$ sampled from $\nu$ and any polynomial $f(x)$ of degree at most the SoS degree, pseudo-calibration proposes that for any low-degree test $g(v)$, the correlation of $\tilde{\mathbb{E}}[f]$ should match in the planted and random distributions. That is,

$$\mathbb{E}_{v \sim \nu} [\tilde{\mathbb{E}}[f(x)]g(v)] = \mathbb{E}_{(x,v) \sim \mu} [f(x)g(v)]$$

Here, the notation $(x, v) \sim \mu$ means that in the planted distribution $\mu$, the input is $v$ and $x$ denotes the planted structure in that instance. For example, in Sparse PCA, $x$ would be the sparse principal component. If there are multiple, pick an arbitrary one.

Let $\mathcal{F}$ denote the Fourier basis of polynomials for the input $v$. By choosing different basis functions from $\mathcal{F}$ as choices for $g$ such that the degree is at most $n^\varepsilon$ (hence the term low-degree test), we get all lower order Fourier coefficients for $\tilde{\mathbb{E}}[f(x)]$ when considered as a function of $v$. Furthermore, the higher order coefficients are set to be $0$ so that the candidate pseudoexpectation operator can be written as

$$\tilde{\mathbb{E}}f(x) = \sum_{\substack{g \in \mathcal{F} \\ deg(g) \leq n^\varepsilon}} \mathbb{E}_{v \sim \nu} [\tilde{\mathbb{E}}[f(x)]g(v)]g(v) = \sum_{\substack{g \in \mathcal{F} \\ deg(g) \leq n^\varepsilon}} \mathbb{E}_{(x,v) \sim \mu} [[f(x)]g(v)]g(v)$$

The coefficients $\mathbb{E}_{(x,v) \sim \mu}[[f(x)]g(v)]$ can be explicitly computed in many settings, which therefore gives an explicit pseudoexpectation operator $\tilde{\mathbb{E}}$.

One intuition for pseudo-calibration is as follows. The planted distribution is usually chosen to be a maximum entropy distribution which still has the planted structure. This exploits our intuition that random instances are hard for SoS, such as the Gaussian distribution for Tensor PCA. By conditioning on the lower order moments matching such a planted distribution, pseudo-calibration can be interpreted as sort of interpolating between the random and planted distributions by only looking at lower order Fourier characters. This intuition has proven to be successful, since pseudo-calibration been successfully exploited to construct SoS lower bounds for a wide variety of dense as well as sparse problems.

An advantage of pseudo-calibration is that this construction automatically satisfies some nice properties that the pseudoexpectation $\tilde{\mathbb{E}}$ should satisfy. It's linear in $v$ by construction. For all polynomial equalities of the form $f(x) = 0$ that is satisfied in the planted distribution, it's true that $\tilde{\mathbb{E}}[f(x)] = 0$. For other polynomial equalities of the form $f(x, v) = 0$ that are satisfied in the planted distribution, the equality $\tilde{\mathbb{E}}[f(x, v)] = 0$ is approximately satisfied. In most cases, $\tilde{\mathbb{E}}$ can be mildly adjusted to satisfy these exactly.

In our applications, we have $\tilde{\mathbb{E}}[1] = 1 \pm o(1)$ due to the bounds on the signal-to-noise ratio (this is where the actual bounds kick in!). Once we have this, we simply set our final pseudoexpectation operator to be $\tilde{\mathbb{E}}'$ defined as $\tilde{\mathbb{E}}'[f(x)] = \tilde{\mathbb{E}}[f(x)]/\tilde{\mathbb{E}}[1]$.

## B.1 Tensor PCA

We will now pseudo-calibrate with respect to the pair of random and planted distributions described for Tensor PCA (Appendix A, Random and planted distributions). Let the Hermite polynomials be $h_0(x) = 1, h_1(x) = x, h_2(x) = x^2 - 1, \ldots$. For $a \in \mathbb{N}^{[n]^k}$ and variables $A_e$ for $e \in [n]^k$, define $h_a(A) := \prod_{e \in [n]^k} h_e(A_e)$. We will work with this Hermite basis, which is a standard basis for Gaussian inputs (which is what we consider here). Define the slack parameter to be $\Delta = n^{-C_\Delta \varepsilon}$ for a constant $C_\Delta > 0$.

**Lemma B.1.** *Let $I \in \mathbb{N}^n, a \in \mathbb{N}^{[n]^k}$. For $i \in [n]$, let $d_i = \sum_{i \in e \in [n]^k} a_e$. Let $c$ be the number of $i$ such that $I_i + d_i$ is nonzero. Then, if $I_i + d_i$ are all even, we have*

$$\mathbb{E}_\mu[u^I h_a(A)] = \Delta^c \left(\frac{1}{\sqrt{\Delta n}}\right)^{|I|} \prod_{e \in [n]^k} \left(\frac{\lambda}{(\Delta n)^{\frac{k}{2}}}\right)^{a_e}$$

*Else, $\mathbb{E}_\mu[u^I h_a(v)] = 0$.*

*Proof.* When $A \sim \mu$, for all $e \in [n]^k$, we have $A_e = B_e + \lambda \prod_{i \leq k} u_{e_i}$. where $B_e \sim \mathcal{N}(0, 1)$. Let's analyze when the required expectation is nonzero. We can first condition on $u$ and use the fact that for a fixed $t$, $\mathbb{E}_{g \sim \mathcal{N}(0,1)}[h_k(g + t)] = t^k$ to obtain

$$\mathbb{E}_{(u_i, w_e) \sim \mu}[u^I h_a(A)] = \mathbb{E}_{(u_i) \sim \mu}[u^I \prod_{e \in [n]^k} (\lambda \prod_{i \leq k} u_{e_i})^{a_e}] = \mathbb{E}_{(u_i) \sim \mu}[\prod_{i \in [n]} u_i^{I_i + d_i}] \prod_{e \in [n]^k} \lambda^{a_e}$$

Observe that this is nonzero precisely when all $I_i + d_i$ are even, in which case

$$\mathbb{E}_{(u_i) \sim \mu}[\prod_{i \in [n]} u_i^{I_i + d_i}] = \Delta^c \left(\frac{1}{\sqrt{\Delta n}}\right)^{\sum_{i \leq n} I_i + d_i} = \Delta^c \left(\frac{1}{\sqrt{\Delta n}}\right)^{|I|} \prod_{e \in [n]^k} \left(\frac{1}{(\Delta n)^{\frac{k}{2}}}\right)^{a_e}$$

where we used the fact that $\sum_{e \in [n]^k} a_e = k \sum_{i \in [n]} d_i$. This completes the proof. ∎

## B.2 Sparse PCA

We will pseudo-calibrate with respect to the random and planted distributions for Sparse PCA (Appendix A , Random and planted distributions). We will again work with the Hermite basis of polynomials. For $a \in \mathbb{N}^{m \times d}$ and variables $v_{i,j}$ for $i \in [m], j \in [n]$, define $h_a(v) := \prod_{i \in [m], j \in [n]} h_{a_{i,j}}(v_{i,j})$. For a nonnegative integer $t$, define $t!! = \frac{(2t)!}{t! 2^t} = 1 \times 3 \times \ldots \times t$ if $t$ is odd and $0$ otherwise. Define the slack parameter to be $\Delta = d^{-C_\Delta \varepsilon}$ for a constant $C_\Delta > 0$.

**Lemma B.2.** . *Let $I \in \mathbb{N}^d, a \in \mathbb{N}^{m \times d}$. For $i \in [m]$, let $e_i = \sum_{j \in [d]} a_{ij}$ and for $j \in [d]$, let $f_j = I_j + \sum_{i \in [m]} a_{ij}$. Let $c_1$ (resp. $c_2$) be the number of $i$ (resp. $j$) such that $e_i > 0$ (resp. $f_j > 0$). Then, if $e_i, f_j$ are all even, we have*

$$\mathbb{E}_\mu[u^I h_a(v)] = \left(\frac{1}{\sqrt{k}}\right)^{|I|} \left(\frac{k}{d}\right)^{c_2} \Delta^{c_1} \prod_{i \in [m]} (e_i - 1)!! \prod_{i,j} \frac{\sqrt{\lambda}^{a_{ij}}}{\sqrt{k}^{a_{ij}}}$$

*Else, $\mathbb{E}_\mu[u^I h_a(v)] = 0$.*

*Proof.* $v_1, \ldots, v_m \sim \mu$ can be written as $v_i = g_i + \sqrt{\lambda} b_i l_i u$ where $g_i \sim \mathcal{N}(0, I_d), l_i \sim \mathcal{N}(0, 1), b_i \in \{0, 1\}$ where $b_i = 1$ with probability $\Delta$. Let's analyze when the required expectation is nonzero. We can first condition on $b_i, l_i, u$ and use the fact that for a fixed $t$, $\mathbb{E}_{g \sim \mathcal{N}(0,1)}[h_k(g + t)] = t^k$ to obtain

$$\mathbb{E}_{(u,l_i,b_i,g_i) \sim \mu}[u^I h_a(v)] = \mathbb{E}_{(u,l_i,b_i) \sim \mu}[u^I \prod_{i,j} (\sqrt{\lambda} b_i l_i u_j)^{a_{ij}}] = \mathbb{E}_{(u,l_i,b_i) \sim \mu}[\prod_{i \in [m]} (b_i l_i)^{e_i} \prod_{j \in [d]} u_j^{f_j}] \prod_{i,j} \sqrt{\lambda}^{a_{ij}}$$

For this to be nonzero, the set of $c_1$ indices $i$ such that $e_i > 0$, should not have been resampled otherwise $b_i = 0$, each of which happens independently with probability $\Delta$. And the set of $c_2$ indices $j$ such that $f_j > 0$ should have been such that $u_j$ is nonzero, each of which happens independently with probability $\frac{k}{d}$. Since $l_i, u_j$ are have zero expectation in $\nu$, we need $e_i, f_j$ to be even. The expectation then becomes

$$\Delta^{c_1} \left(\frac{k}{d}\right)^{c_2} \mathbb{E}_{(u,l_i) \sim \mu}[\prod_{i \in [m]} l_i^{e_i} \prod_{j \in [d]} u_j^{f_j}] \prod_{i,j} \sqrt{\lambda}^{a_{ij}} = \left(\frac{1}{\sqrt{k}}\right)^{|I|} \left(\frac{k}{d}\right)^{c_2} \Delta^{c_1} \prod_{i \in [m]} (e_i - 1)!! \prod_{i,j} \frac{\sqrt{\lambda}^{a_{ij}}}{\sqrt{k}^{a_{ij}}}$$

The last equality follows because, for each $j$ such that $u_j$ is nonzero, we have $u_j^t = (\frac{1}{\sqrt{k}})^t$ and $\mathbb{E}_{g \sim \mathcal{N}(0,1)}[g^t] = (t-1)!!$ if $t$ is even. ∎

# C   Informal Description of our main theorem

In this section, we informally describe our general theorem for proving sum of squares lower bounds on planted problems. Our goal for this section is to qualitatively state the conditions under which we can show that the moment matrix $\Lambda$ is PSD with high probability (see Theorem C.37). For simplicity, in this section we restrict ourselves to the setting where the input is $\{-1, 1\}^{\binom{n}{2}}$ (e.g. a random graph on $n$ vertices).

## C.1   Fourier analysis for matrix-valued functions: ribbons, shapes, and graph matrices

For our approach, we need the definitions of ribbons, shapes, and graph matrices from [1].

### C.1.1   Ribbons

*Ribbons* lift the usual Fourier basis for functions $\{f : \{\pm 1\}^{\binom{n}{2}} \to \mathbb{R}\}$ to matrix-valued functions.

**Definition C.1** (Simplified ribbons – see Definition F.22). *Let $n \in \mathbb{N}$. A ribbon $R$ is a tuple $(E_R, A_R, B_R)$ where $E_R \subseteq \binom{[n]}{2}$ and $A_R, B_R$ are tuples of elements in $[n]$. $R$ thus specifies:*

  *1. A Fourier character $\chi_{E_R}$.*

  *2. Row and column indices $A_R$ and $B_R$.*

*We think of $R$ as a graph with vertices*

$$V(R) = \{ \text{ endpoints of } (i, j) \in E_R \} \cup A_R \cup B_R$$

*and edges $E(R) = E_R$, where $A_R, B_R$ are distinguished tuples of vertices.*

**Definition C.2** (Matrix-valued function for a ribbon $R$). *Given a ribbon $R$, we define the matrix valued function $M_R : \{\pm 1\}^{\binom{n}{2}} \to \mathbb{R}^{\frac{n!}{(n-|A_R|)!} \times \frac{n!}{(n-|B_R|)!}}$ to have entries $M_R(A_R, B_R) = \chi_{E_R}$ and $M_R(A', B') = 0$ whenever $A' \neq A_R$ or $B' \neq B_R$.*

The following proposition captures the main property of the matrix-valued functions $M_R$ – they are an orthonormal basis. We leave the proof to the reader.

**Proposition C.3.** *The matrix-valued functions $M_R$ form an orthonormal basis for the vector space of matrix valued functions with respect to the inner product*

$$\langle M, M' \rangle = \mathop{\mathbb{E}}_{G \sim \{\pm 1\}^{\binom{n}{2}}} \left[ \mathrm{Tr} \left( M(G)(M'(G))^\top \right) \right].$$

We don't directly utilize this proposition in our work but this gives insight on to the structure of the matrix valued functions we define and motivates the definition of graph matrices, that we use extensively.

**Example C.4.** *In Fig. 2, consider the ribbon $R$ as shown. We have $A_R = (1,3), B_R = (4), V(R) = \{1,2,3,4\}, E_R = \{\{1,2\},\{3,2\},\{2,4\}\}$. The Fourier character is $\chi_{E_R} = \chi_{1,2}\chi_{3,2}\chi_{2,4}$. And finally, $M_R$ is a matrix with rows and columns indexed by tuples of length $|A_R| = 2$ and $|B_R| = 1$ respectively, with exactly one nonzero entry $M_R((1,3),(4)) = \chi_{E_R}$. Succinctly,*

$$M_R = \quad \begin{array}{c} \\ row\ (1,3) \to \end{array} \overset{\displaystyle column\ (4) \atop \displaystyle \downarrow}{\left( \begin{array}{ccc} 0 & \vdots & 0 \\ \cdots\cdots \chi_{1,2}\chi_{3,2}\chi_{2,4} \cdots\cdots\cdots \\ 0 & \vdots & 0 \end{array} \right)}$$

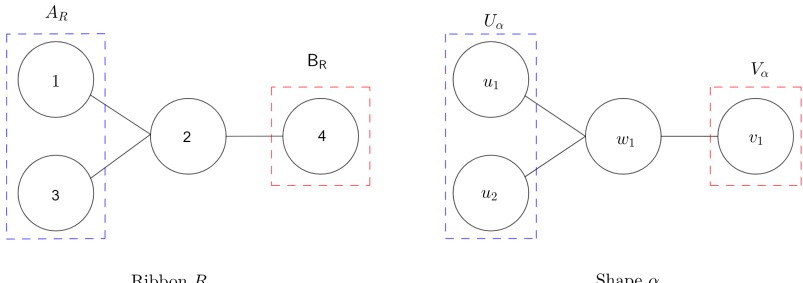

Figure 2: Example of a ribbon and a shape

### C.1.2 Shapes and Graph Matrices

As described above, *ribbons* are an orthonormal basis for matrix-valued functions. However, we will need an orthogonal basis for the subset of those functions which are symmetric with respect to the action of $S_n$. For this, we use *graph matrices*, which are described by *shapes*. The idea is that each ribbon $R$ has a shape $\alpha$ which is obtained by replacing the vertices of $R$ with unspecified indices. Up to scaling, the graph matrix $M_\alpha$ is the average of $M_{\pi(R)}$ over all permutations $\pi \in S_n$.

**Definition C.5** (Simplified shapes – see Definition F.34). *Informally, a shape $\alpha$ is just a ribbon $R$ where the vertices are specified by variables rather than having specific values in $[n]$. More precisely, a shape $\alpha = (V(\alpha), E(\alpha), U_\alpha, V_\alpha)$ is a graph on vertices $V(\alpha)$, with*

1. *Edges $E(\alpha) \subseteq \binom{V(\alpha)}{2}$*

2. *Distinguished tuples of vertices $U_\alpha = (u_1, u_2, \dots)$ and $V_\alpha = (v_1, v_2, \dots)$, where $u_i, v_i \in V(\alpha)$.*

*(Note that $V(\alpha)$ and $V_\alpha$ are not the same object!)*

**Definition C.6** (Shape transposes). *Given a shape $\alpha$, we define $\alpha^\top$ to be the shape $\alpha$ with $U_\alpha$ and $V_\alpha$ swapped i.e. $U_{\sigma^\top} = V_\sigma$ and $V_{\sigma^\top} = U_\sigma$. Note that $M_{\alpha^\top} = M_\alpha^\top$, where $M_\alpha^\top$ is the usual transpose of the matrix-valued function $M_\alpha$.*

**Definition C.7** (Graph matrices)**.** *Let $\alpha$ be a shape. The graph matrix $M_\alpha : \{\pm 1\}^{\binom{n}{2}} \to \mathbb{R}^{\frac{n!}{(n-|U_\alpha|)!} \times \frac{n!}{(n-|V_\alpha|)!}}$ is defined to be the matrix-valued function with $A, B$-th entry*

$$M_\alpha(A, B) = \sum_{\substack{R \text{ s.t. } A_R = A, B_R = B \\ \exists \varphi : V(\alpha) \to [n]: \\ \varphi \text{ is injective}, \varphi(\alpha) = R}} \chi_{E_R}$$

*In other words, $M_\alpha = \sum_R M_R$ where the sum is over ribbons $R$ which can be obtained by assigning each vertex in $V(\alpha)$ a label from $[n]$.*

**Example C.8.** *In Fig. 2, consider the shape $\alpha$ as shown. We have $U_\alpha = (u_1, u_2), V_\alpha = (v_1), V(\alpha) = \{u_1, u_2, v_1, w_1\}$ and $E(\alpha) = \{\{u_1, w_1\}, \{u_2, w_1\}, \{w_1, v_1\}\}$. $M_\alpha$ is a matrix with rows and columns indexed by tuples of length $|U_\alpha| = 2$ and $|V_\alpha| = 1$ respectively. The nonzero entries will have rows and columns indexed by $(a_1, a_2)$ and $b_1$ respectively for all distinct $a_1, a_2, b_1$, with the corresponding entry being $M_\alpha((a_1, a_2), (b_1)) = \sum_{c_1 \in [n] \setminus \{a_1, a_2, b_1\}} \chi_{a_1, c_1} \chi_{a_2, c_1}, \chi_{c_1, b_1}$. Here, the injective map $\varphi$ maps $u_1, u_2, w_1, v_1$ to $a_1, a_2, c_1, b_1$ respectively and we sum over all such maps. Succinctly,*

$$M_\alpha = \quad \text{row } (a_1, a_2) \to \begin{pmatrix} & \text{column } (b_1) \\ & \downarrow \\ & \vdots \\ \cdots\cdots\cdots \sum_{c_1 \in [n] \setminus \{a_1, a_2, b_1\}} \chi_{a_1, c_1} \chi_{a_2, c_1} \chi_{c_1, b_1} \cdots\cdots\cdots \\ & \vdots \end{pmatrix}$$

**Remark C.9.** *The fact that we are summing over all "free" vertices in $V(\alpha) \setminus (U_\alpha \cup V_\alpha)$ is how we are incorporating symmetry into the definition of these graph matrices.*

The following examples illustrate that simple matrices such as the adjacency matrix of a graph and the identity matrix are also graph matrices.

**Example C.10** (Adjacency matrix)**.** *Let $\alpha$ be the shape with two vertices $V(\alpha) = \{u_1, v_1\}$ and a single edge $E(\alpha) = \{\{u_1, v_1\}\}$. The tuples $U_\alpha, V_\alpha$ are $(u_1), (v_1)$, respectively. Then $M_\alpha$ has entries $(M_\alpha)_{i,j}(G) = G_{ij}$ if $i \neq j$ and $(M_\alpha)_{i,i} = 0$. If $G \in \{\pm 1\}^{\binom{n}{2}}$ is thought of as a graph, then $M_\alpha$ is precisely its $\pm 1$ adjacency matrix with zeros on the diagonal.*

**Example C.11** (Identity matrix)**.** *If $V(\alpha) = \{u\}$ is a singleton, $E(\alpha) = \emptyset$, and $U_\alpha = V_\alpha = (u)$, then $M_\alpha(G)$ is identically equal to the $n \times n$ identity matrix, independent of $G$.*

For more examples of graph matrices and why they can be a useful tool to work with, see [1].

**Remark C.12.** *As noted in [1], we index graph matrices by tuples rather than sets so that they are symmetric (as a function of the input) under permutations of $[n]$.*

## C.2 Factoring Graph Matrices and Decomposing Shapes into Left, Middle, and Right Parts

A crucial idea in our analysis is the idea from [19] of decomposing each shape $\alpha$ into left, middle, and right parts. This will allow us to give an approximate factorization of each graph matrix $M_\alpha$.

### C.2.1 Leftmost and Rightmost Minimum Vertex Separators and Decomposition of Shapes into Left, Middle, and Right Parts

For each shape $\alpha$ we will identify three other shapes, which we denote by $\sigma, \tau, \sigma'^T$ and call (for reasons we will see soon) the *left, middle, and right parts of $\alpha$*, respectively. The idea is that $M_\alpha \approx M_\sigma M_\tau M_{\sigma'^T}$. We obtain $\sigma, \tau$, and $\sigma'^T$ by splitting the shape $\alpha$ along the *leftmost and rightmost minimum vertex separators*.

**Definition C.13** (Vertex Separators)**.** *We say that a set of vertices $S$ is a vertex separator of $\alpha$ if every path from $U_\alpha$ to $V_\alpha$ in $\alpha$ (including paths of length $0$) intersects $S$. Note that for any vertex separator $S, U_\alpha \cap V_\alpha \subseteq S$.*

**Definition C.14** (Minimum Vertex Separators)**.** *We say that $S$ is a minimum vertex separator of $\alpha$ if $S$ is a vertex separator of $\alpha$ and for any other vertex separator $S'$ of $\alpha$, $|S| \leq |S'|$.*

**Definition C.15** (Leftmost and Rightmost Minimum Vertex Separators)**.**

1. *We say that $S$ is the leftmost minimum vertex separator of $\alpha$ if $S$ is a minimum vertex separator of $\alpha$ and for every other minimum vertex separator $S'$ of $\alpha$, every path from $U_\alpha$ to $S'$ intersects $S$.*

2. *We say that $T$ is the rightmost minimum vertex separator of $\alpha$ if $T$ is a minimum vertex separator of $\alpha$ and for every other minimum vertex separator $S'$ of $\alpha$, every path from $S'$ to $V_\alpha$ intersects $T$.*

It is not immediately obvious that leftmost and rightmost minimum vertex separators are well-defined. For the simplified setting we are considering here, this was shown by [19]. We now describe how to split $\alpha$ into left, middle, and right parts $\sigma, \tau$, and $\sigma'^T$.

**Definition C.16** (Decomposition Into Left, Middle, and Right Parts)**.** *Let $\alpha$ be a shape and let $S$ and $T$ be the leftmost and rightmost minimum vertex separators of $\alpha$. Given orderings $O_S$ and $O_T$ for $S$ and $T$, we decompose $\alpha$ into left, middle, and right parts $\sigma$, $\tau$, and $\sigma'^T$ as follows.*

1. *The left part $\sigma$ of $\alpha$ is the part of $\alpha$ reachable from $U_\alpha$ without passing through $S$. It includes $S$ but excludes all edges which are entirely within $S$. More formally,*

   (a) *$V(\sigma) = \{u \in V(\alpha) : \text{ there is a path } P \text{ from } U_\alpha \text{ to } u \text{ in } \alpha \text{ such that } (V(P) \setminus \{u\}) \cap S = \emptyset\}$*
   (b) *$U_\sigma = U_\alpha$ and $V_\sigma = S$ with the ordering $O_S$*
   (c) *$E(\sigma) = \{\{u, v\} \in E(\alpha) : u, v \in V(\sigma), u \notin S \text{ or } v \notin S\}$*

2. *The right part $\sigma'^T$ of $\alpha$ is the part of $\alpha$ reachable from $V_\alpha$ without intersecting $T$ more than once. It includes $T$ but excludes all edges which are entirely within $T$. More formally,*

   (a) *$V(\sigma'^T) = \{u \in V(\alpha) : \text{ there is a path } P \text{ from } V_\alpha \text{ to } u \text{ in } \alpha \text{ such that } (V(P) \setminus \{u\}) \cap T = \emptyset\}$*
   (b) *$U_{\sigma'^T} = T$ with the ordering $O_T$ and $V_{\sigma'^T} = V_\alpha$.*
   (c) *$E(\sigma'^T) = \{\{u, v\} \in E(\alpha) : u, v \in V(\sigma'^T), u \notin T \text{ or } v \notin T\}$*

3. *The middle part $\tau$ of $\alpha$ is, informally, the part of $\alpha$ between $S$ and $T$ (including $S$ and $T$ and all edges which are entirely within $S$ or within $T$). More formally, let $U_\tau = S$ with the ordering $O_S$, let $V_\tau = T$ with the ordering $O_T$, and let $E(\tau) = E(\alpha) \setminus (E(\sigma) \cup E(\sigma'))$ be all of the edges of $E(\alpha)$ which do not appear in $E(\sigma)$ or $E(\sigma')$. Then $V(\tau)$ is all of the vertices incident to edges in $E(\tau)$ together with $S, T$.*

**Remark C.17.** *Note that the decomposition into left, middle, and right parts depends on the ordering for the vertices in $S$ and $T$. As we will discuss later (see Section F.8), we will use all possible orderings simultaneously and then scale things by an appropriate constant.*

Because of the minimality and leftmost/rightmost-ness of the vertex separators $S, T$ used to define $\sigma, \tau, \sigma'$, the shapes $\sigma, \tau, \sigma'$ have some special combinatorial structure, which we capture in the following proposition. We defer the proof until Appendix F where we state a generalized version.

**Proposition C.18.** *$\sigma$, $\tau$, and $\sigma'^T$ have the following properties:*

1. *$V_\sigma = S$ is the unique minimum vertex separator of $\sigma$.*

2. *$S$ and $T$ are the leftmost and rightmost minimum vertex separators of $\tau$.*

3. *$T = U_{\sigma'^T}$ is the unique minimum vertex separator of $\sigma'^T$.*

Based on this, we define sets of shapes which can appear as left, middle, or right parts.

**Definition C.19** (Left, Middle, and Right Parts)**.** *Let $\alpha$ be a shape.*

1. *We say that $\alpha$ is a left part if $V_\alpha$ is the unique minimum vertex separator of $\alpha$, all vertices of $\alpha$ are reachable from $U_\alpha$ without passing through $V_\alpha$, and $E(\alpha)$ has no edges which are entirely contained in $V_\alpha$.*

2. *We say that $\alpha$ is a proper middle part if $U_\alpha$ is the leftmost minimum vertex separator of $\alpha$ and $V_\alpha$ is the rightmost minimum vertex separator of $\alpha$*

3. We say that $\alpha$ is a *right part* if $U_\alpha$ is the unique minimum vertex separator of $\alpha$, all vertices of $\alpha$ are reachable from $V_\alpha$ without passing through $U_\alpha$, and $E(\alpha)$ has no edges which are entirely contained in $U_\alpha$.

**Remark C.20.** *For technical reasons, later on we will need to consider middle parts $\tau$ where $U_\tau$ and $V_\tau$ are not the leftmost and rightmost minimum vertex separators of $\tau$ (these $\tau$ are called improper middle parts), which is why we make this distinction here.*

The following proposition is also straightforward from the definitions.

**Proposition C.21.** *A shape $\sigma$ is a left part if and only if $\sigma^T$ is a right part*

### C.2.2 Products of Graph Matrices

We now analyze what happens when we take the products of graph matrices. Roughly speaking, we will have that if $\alpha$ can be decomposed into left, middle, and right parts $\sigma$, $\tau$, and $\sigma'^T$ then $M_\alpha \approx M_\sigma M_\tau M_{\sigma'^T}$.

We begin with a concatenation operation on ribbons.

**Definition C.22** (Ribbon Concatenation). *If $R_1$ and $R_2$ are two ribbons such that $V(R_1) \cap V(R_2) = B_{R_1} = A_{R_2}$ and either $R_1$ or $R_2$ contains no edges entirely within $B_{R_1} = A_{R_2}$ then we define $R_1 \circ R_2$ to be the ribbon formed by glueing together $R_1$ and $R_2$ along $B_{R_1} = A_{R_2}$. In other words,*

1. $V(R_1 \circ R_2) = V(R_1) \cup V(R_2)$

2. $E(R_1 \circ R_2) = E(R_1) \cup E(R_2)$

3. $A_{R_1 \circ R_2} = A_{R_1}$ *and* $B_{R_1 \circ R_2} = B_{R_2}$.

The following proposition is easy to check.

**Proposition C.23.** *Whenever $R_1, R_2$ are ribbons such that $R_1 \circ R_2$ is defined, $M_{R_1} M_{R_2} = M_{R_1 \circ R_2}$*

We have an analogous definition for concatenating shapes:

**Definition C.24** (Shape Concatenation). *If $\alpha_1$ and $\alpha_2$ are two shapes such that $V(\alpha_1) \cap V(\alpha_2) = V_{\alpha_1} = U_{\alpha_2}$ and either $\alpha_1$ or $\alpha_2$ contains no edges entirely within $V_{\alpha_1} = U_{\alpha_2}$ then we define $\alpha_1 \circ \alpha_2$ to be the shape formed by glueing together $\alpha_1$ and $\alpha_2$ along $V_{\alpha_1} = U_{\alpha_2}$. In other words,*

1. $V(\alpha_1 \circ \alpha_2) = V(\alpha_1) \cup V(\alpha_2)$

2. $E(\alpha_1 \circ \alpha_2) = E(\alpha_1) \cup E(\alpha_2)$

3. $U_{\alpha_1 \circ \alpha_2} = U_{\alpha_1}$ *and* $V_{\alpha_1 \circ \alpha_2} = V_{\alpha_2}$.

The next proposition, again easy to check, shows that the shape concatenation operation respects the left/middle/right part decomposition.

**Proposition C.25.** *If $\alpha$ can be decomposed into left, middle, and right parts $\sigma, \tau, \sigma'^T$ then $\alpha = \sigma \circ \tau \circ \sigma'^T$.*

We now discuss why $M_\alpha = M_{\sigma \circ \tau \circ \sigma'^T} \approx M_\sigma M_\tau M_{\sigma'^T}$ is only an approximation rather than an equality. Consider the difference $M_\sigma M_\tau M_{\sigma'^T} - M_{\sigma \circ \tau \circ \sigma'^T}$. The graph matrix $M_{\sigma \circ \tau \circ \sigma'^T}$ decomposes (by definition) into a sum over injective maps $\varphi : V(\sigma \circ \tau \circ \sigma'^T) \to [n]$. Also by expanding definitions, the product $M_\sigma M_\tau M_{\sigma'^T}$ expands into a sum over triples of injective maps $(\varphi_1, \varphi_2, \varphi_3)$, where $\varphi_1 : V(\sigma) \to [n], \varphi_2 : V(\tau) \to [n], \varphi_3 : V(\sigma') \to [n]$ where $\varphi_1$ and $\varphi_2$ agree on $V_\sigma = U_\tau$ and $\varphi_2$ and $\varphi_3$ agree on $V_\tau = U_{\sigma'^T}$.

If they are combined into one map $\varphi : V(\sigma \cup \tau \cup \sigma') \to [n]$, the resulting $\varphi$ may not be injective because $\varphi_1(V(\sigma)), \varphi_2(V(\tau)), \varphi_3(V(\sigma'^T))$ may have nontrivial intersection (beyond $\varphi_1(V_\sigma)$ and $\varphi_2(V_\tau)$). We call the resulting terms *intersection terms* and handling them properly is a major part of the technical analysis.

**Remark C.26.** *Actually, the approximation $M_\alpha = M_{\sigma \circ \tau \circ \sigma'^T} \approx M_\sigma M_\tau M_{\sigma'^T}$ is also off by a multiplicative constant because there is also a subtle issue involving the automorphism groups of these shapes. For now, we ignore this issue. For details about this issue, see Lemma F.81*

## C.3 Shape Coefficient Matrices

The idea for our analysis is as follows. Given a matrix-valued function $\Lambda$ which is symmetric under permutations of $[n]$, we write $\Lambda = \sum_\alpha \lambda_\alpha M_\alpha$. We then break each shape $\alpha$ up into left, middle, and right parts $\sigma$, $\tau$, and $\sigma'^T$.

For this analysis, we use *shape coefficient matrices* $H_\tau$ whose rows and columns are indexed by left shapes and whose entries depend on the coefficients $\lambda_\alpha$. We choose these matrices so that

$$\Lambda = \sum_\tau H_\tau(\sigma, \sigma') M_{\sigma \circ \tau \circ \sigma'^T} \approx \sum_\tau H_\tau(\sigma, \sigma') M_\sigma M_\tau M_{\sigma'^T}$$

To set this up, we separate the possible middle parts $\tau$ into groups based on the size of $U_\tau$ and whether or not they are trivial.

**Definition C.27.** *We define $\mathcal{I}_{mid}$ to be the set of all possible $U_\tau$. Here $\mathcal{I}_{mid}$ is the set of tuples of unspecified vertices of the form $U = (u_1, \ldots, u_k)$ where $0 \leq k \leq d$.*

**Definition C.28.** *We say that a proper middle shape $\tau$ is trivial if $E(\tau) = \emptyset$ and $|U_\tau \cap V_\tau| = |U_\tau| = |V_\tau|$ (i.e. $V_\tau$ is a permutation of $U_\tau$).*

For simplicity, the only proper trivial middle parts $\tau$ we consider are shapes $Id_U$ corresponding to identity matrices.

**Definition C.29.** *Given a tuple of unspecified vertices $U = (u_1, \ldots, u_{|U|})$ We define $Id_U$ to be the shape where $V(Id_U) = U$, $U_{Id_U} = V_{Id_U} = U$, and $E(Id_U) = \emptyset$.*

We group all of the proper non-trivial middle parts $\tau$ into sets $\mathcal{M}_U$ based on the size of $U_\tau$.

**Definition C.30.** *Given a tuple of unspecified vertices $U = (u_1, \ldots, u_{|U|})$, we define $\mathcal{M}_U$ to be the set of proper non-trivial middle parts $\tau$ such that $U_\tau$ and $V_\tau$ have the same size as $U$. Note that $U_\tau$ and $V_\tau$ may intersect each other arbitrarily.*

With these definitions, we can now define our shape coefficient matrices.

**Definition C.31.** *Given $U \in \mathcal{I}_{mid}$, we define $\mathcal{L}_U$ to be the set of left shapes $\sigma$ such that $|V_\sigma| = |U|$.*

**Definition C.32.** *For each $U \in \mathcal{I}_{mid}$, we define the shape coefficient matrix $H_{Id_U}$ to be the matrix indexed by left shapes $\sigma, \sigma' \in \mathcal{L}_U$ with entries $H_{Id_U}(\sigma, \sigma') = \frac{1}{|U|!} \lambda_{\sigma \circ \sigma'^T}$*

**Definition C.33.** *For each $U \in \mathcal{I}_{mid}$, for each $\tau \in \mathcal{M}_U$, we define the shape coefficient matrix $H_\tau$ to be the matrix indexed by left shapes $\sigma, \sigma' \in \mathcal{L}_U$ with entries $H_\tau(\sigma, \sigma') = \frac{1}{(|U|!)^2} \lambda_{\sigma \circ \tau \circ \sigma'^T}$*

With these shape coefficient matrices, we have the following decomposition of $\Lambda = \sum_\alpha \lambda_\alpha M_\alpha$.

**Lemma C.34.** $\Lambda = \sum_{U \in \mathcal{I}_{mid}} \sum_{\sigma, \sigma' \in \mathcal{L}_U} H_{Id_U}(\sigma, \sigma') M_{\sigma \circ \sigma'^T} + \sum_{U \in \mathcal{I}_{mid}} \sum_{\tau \in \mathcal{M}_U} \sum_{\sigma, \sigma' \in \mathcal{L}_U} H_\tau(\sigma, \sigma') M_{\sigma \circ \tau \circ \sigma'^T}$

We defer the proof of this lemma to Lemma F.84.

For technical reasons, we need to define one more operation to handle intersection terms. We call this operation *the $-\gamma, \gamma$ operation*.

**Definition C.35.** *Given $U, V \in \mathcal{I}_{mid}$ where $|U| > |V|$, we define $\Gamma_{U,V}$ to be the set of left parts $\gamma$ such that $|U_\gamma| = |U|$ and $|V_\gamma| = |V|$.*

**Definition C.36.** *Given $U, V \in \mathcal{I}_{mid}$ where $|U| > |V|$, a shape coefficient matrix $H_{Id_V}$, and a $\gamma \in \Gamma_{U,V}$, we define the shape coefficient matrix $H_{Id_V}^{-\gamma;\gamma}$ to be the matrix indexed by left shapes $\sigma, \sigma' \in \mathcal{L}_U$ with entries $H_{Id_V}^{-\gamma;\gamma}(\sigma, \sigma') = H(\sigma \circ \gamma, \sigma' \circ \gamma)$*

## C.4 Informal Theorem Statement

We are now ready to state a qualitative version of our main theorem. For the quantitative version of our main theorem, see Theorem F.101.

**Theorem C.37.** *There exist functions $f(\tau) : \mathcal{M}_U \to \mathbb{R}$ and $f(\gamma) : \Gamma_{U,V} \to \mathbb{R}$ depending on $n$ and other parameters such that if $\Lambda = \sum_\alpha \lambda_\alpha M_\alpha$ and the following conditions hold:*

1. (PSD mass) For all $U \in \mathcal{I}_{mid}$, $H_{Id_U} \succeq 0$

2. (Middle shape bounds) For all $U \in \mathcal{I}_{mid}$ and all $\tau \in \mathcal{M}_U$,

$$
\left[ \begin{array}{cc} H_{Id_U} & f(\tau)H_\tau \\ f(\tau)H_\tau^T & H_{Id_U} \end{array} \right] \succeq 0
$$

3. (Intersection term bounds) For all $U, V \in \mathcal{I}_{mid}$ such that $|U| > |V|$ and all $\gamma \in \Gamma_{U,V}$,
$H_{Id_{V_\gamma}}^{-\gamma,\gamma} \preceq f(\gamma)H_{Id_{U_\gamma}}$

then with probability at least $1 - o(1)$ over $G \sim \{\pm 1\}^{\binom{n}{2}}$ it holds that $\Lambda(G) \succeq 0$.

**Remark C.38.** *Condition 1 of Theorem C.37 will follow from condition 2 but we state it explicitly since it will correspond to the dominating terms of the approximate PSD decomposition.*

**Remark C.39.** *As we will demonstrate in the remainder of this paper, the theorem works well when the coefficients $\lambda_\alpha$ has some decay for each vertex or edge in the shape. In many settings, this can be done quite easily by adding noise to the distribution, such as resampling part of the input, or by lowering the parameters slightly, such as $m \leq n^{k/4-\varepsilon}$ instead of $m \leq n^{k/4}$.*

## C.5 An application to planted clique

Before we move on, we present an informal example.

**Example C.40.** *When the pseudo-calibration method is applied to prove an SoS lower bound for the planted clique problem in $n$ node graphs with clique size $k$, as in [19], the matrix-valued function which results is $\Lambda = \sum_{\alpha \,:\, |V(\alpha)| \leq t} \left(\frac{k}{n}\right)^{|V(\alpha)|} M_\alpha$ where $t \approx \log(n)$. One may then compute that the matrices $H_{Id_U}$ and $H_\tau$ are as follows (at least so long as $|V(\sigma)|, |V(\tau)|, |V(\sigma')| \ll t$; we ignore this detail for now). For all $r \in [0, \frac{d}{2}]$,*

1. *For $U$ with $|U| = r$, $H_{Id_U}(\sigma, \sigma') = \left(\frac{k}{n}\right)^{|V(\sigma)|+|V(\sigma')|-r}$*

2. *For all proper, non-trivial middle shapes $\tau$ such that $|U_\tau| = |V_\tau| = r$,*

$$
H_\tau(\sigma, \sigma') = \left(\frac{k}{n}\right)^{|V(\sigma)|+|V(\sigma')|+|V(\tau)|-2r}
$$

*Defining $v_r$ to be the vector such that $v_r(\sigma) = \left(\frac{k}{n}\right)^{|V(\sigma)|-\frac{r}{2}}$, we have that*

1. *For $U$ with $|U| = r$, $H_{Id_U} = v_{|U|}v_{|U|}^T$*

2. *For all proper, non-trivial middle shapes $\tau$ such that $|U_\tau| = |V_\tau| = r$, $H_\tau = \left(\frac{k}{n}\right)^{|V(\tau)|-r} v_r v_r^T$*

3. *For all left parts $\gamma$, $H_{Id_{V_\gamma}}^{-\gamma,\gamma} = \left(\frac{k}{n}\right)^{2|V(\gamma)|-|U_\gamma|-|V_\gamma|} v_{|U_\gamma|}v_{|U_\gamma|}^T$*

*It turns out in this setting that we can take $f(\tau)$ to be $\tilde{O}(n^{\frac{|V(\tau)|-|U_\tau|}{2}})$ and $f(\gamma)$ to be $\tilde{O}(n^{|V(\gamma)\setminus U_\gamma|})$. Thus, as long as $k \ll \sqrt{n}$,*

1. *For any $U$ and all $\tau$ such that $V_\tau \neq U_\tau$ with $|U_\tau| = |V_\tau| = |U|$, $f(\tau)H_\tau \preceq H_{Id_U}$.*

2. *For all non-trivial left parts $\gamma$, $H_{Id_{V_\gamma}}^{-\gamma,\gamma} \preceq f(\gamma)H_{Id_{U_\gamma}}$*

**Remark C.41.** *This does not quite satisfy the conditions of Theorem C.37 because there are $\tau$ such that $V_\tau = U_\tau$ but which are non-trivial because $E(\tau) \neq \emptyset$. For these $\tau$, condition 2 of Theorem C.37 fails. [19] handle this issue by grouping together all of the $\tau$ where $V_\tau = U_\tau$ into the indicator function for whether $V_\tau = U_\tau$ is a clique. Since this issue is specific to planted clique, we don't try to incorporate it into our theorem to avoid losing generality.*

## C.6 Further definitions needed for our applications

We will describe some more notations and definitions that will be useful to us to describe the qualitative bounds for our applications. We make these modifications because sometimes the input is from a distribution $\Omega$ which is not $\{-1, 1\}$. If the entries are labeled by more than 2 indices such as Tensor PCA where we can have order-3 tensors, then we use hyperedges instead of edges. The other modification is that we take an orthonormal basis for $\Omega$ and give each edge a label corresponding to the basis element. Finally, there may be $t$ types of indices rather than just one, so the symmetry group will be $S_{n_1} \times \ldots \times S_{n_t}$ rather than $S_n$. To handle this, we will have shapes with different types of vertices.

### C.6.1 Tensor PCA

We consider the input to be a tensor $A \in \mathbb{R}^{[n]^k}$. The input entries are now sampled from the distribution $\mathcal{N}(0, 1)$ instead of $\{-1, 1\}$. So, we will work with the Hermite basis of polynomials. Let the standard unnormalized Hermite polynomials be denoted as $h_0(x) = 1, h_1(x) = x, h_2(x) = x^2 - 1, \ldots$. Then, we work with the basis $h_a(A) := \prod_{e \in [n]^k} h_e(A_e)$ over $a \in \mathbb{N}^{[n]^k}$. Accordingly, we will modify the graphs that represent ribbons (and by extension, shapes), to have labeled hyperedges of arity $k$. So, an hyperedge $e$ with a label $t$ will correspond to the hermite polynomial $h_t(A_e)$.

**Definition C.42** (Hyperedges). *Instead of standard edges, we will have labeled hyperedges of arity $k$ in the underlying graphs for our ribbons as well as shapes. The label for an hyperedge $e$, denoted $l_e$, is an element of $\mathbb{N}$ which will correspond to the Hermite polynomial being evaluated on that entry.*

Note that our hyperedges are ordered since the tensor $A$ is not necessarily symmetric. For variables $x_1, \ldots, x_n$, the rows and columns of our moment matrix will now correspond to monomials of the form $\prod_{i \leq n} x_i^{p_i}$ for $p_i \geq 0$. To capture this, we use the notion of index shape pieces and index shapes. Informally, we split the above monomial product into groups based on their powers and each such group will form an index shape piece.

**Definition C.43** (Index shape piece). *An index shape piece $U_i = ((U_{i,1}, \ldots, U_{i,t}), p_i)$ is a tuple of indices $(U_{i,1}, \ldots, U_{i,t})$ along with a power $p_i \in \mathbb{N}$. Let $V(U_i)$ be the set $\{U_{i,1}, \ldots, U_{i,t}\}$ of vertices of this index shape piece. When clear from context, we use $U_i$ instead of $V(U_i)$.*

If we realize $U_{i,1}, \ldots, U_{i,t}$ to be indices $a_1, \ldots, a_t \in [n]$, then, this realization of this index shape piece corresponds to the monomial $\prod_{j \leq t} x_{a_j}^{p_i}$.

**Definition C.44** (Index shape). *An index shape $U$ is a set of index shape pieces $U_i$ that have different powers. Let $V(U)$ be the set of vertices $\cup_i V(U_i)$. When clear from context, we use $U$ instead of $V(U)$.*

Observe that each realization of an index shape corresponds to a row or column of the moment matrix.

**Definition C.45.** *For two index shapes $U, V$, we write $U \equiv V$ if for all powers $p$, the index shape pieces of power $p$ in $U$ and $V$ have the same length.*

**Definition C.46.** *Define $\mathcal{I}_{mid}$ to be the set of all index shapes $U$ that contain only index shape pieces of power $1$.*

In the definition of shapes, the distinguished set of vertices should now be replaced by index shapes.

**Definition C.47** (Shapes). *Shapes are tuples $\alpha = (H_\alpha, U_\alpha, V_\alpha)$ where $H_\alpha$ is a graph with hyperedges of arity $k$ and $U_\alpha, V_\alpha$ are index shapes such that $U_\alpha, V_\alpha \subseteq V(H_\alpha)$.*

**Definition C.48** (Proper shape). *A shape $\alpha$ is proper if it has no isolated vertices outside $U_\alpha \cup V_\alpha$, no multi-edges and all the edges have a nonzero label.*

To define the notion of vertex separators, we modify the notion of paths for hyperedges.

**Definition C.49** (Path). *A path is a sequence of vertices $u_1, \ldots, u_t$ such that $u_i, u_{i+1}$ are in the same hyperedge, for all $i \leq t - 1$.*

The notions of vertex separator and decomposition into left, middle and right parts are identically defined with the above notion of hyperedges and paths. In the definition of trivial shape $\tau$, we now

require $U_\tau \equiv V_\tau$. For $U \in \mathcal{I}_{mid}$, $\mathcal{M}_U$ will be the set of proper non-trivial middle parts $\tau$ with $U_\tau \equiv V_\tau \equiv U$ and $\mathcal{L}_U$ will be the set of left parts $\sigma$ such that $V_\sigma \equiv U$. Similarly, for $U, V \in \mathcal{I}_{mid}$, $\mathcal{L}_{U,V}$ will be the set of left parts $\gamma$ such that $U_\gamma \equiv U$ and $V_\gamma \equiv V$.

In order to define the moment matrix, we need to truncate our shapes based on the number of vertices and the labels on our hyperedges. So, we make the following definition.

**Definition C.50** (Truncation parameters). *For integers $D_{sos}, D_V, D_E \geq 0$, say that a shape $\alpha$ satisfies the truncation parameters $D_{sos}, D_V, D_E$ if*

- *The degrees of the monomials that $U_\alpha$ and $V_\alpha$ correspond to, are at most $\frac{D_{sos}}{2}$*

- *The left part $\sigma$, the middle part $\tau$ and the right part $\sigma'^T$ of $\alpha$ satisfy $|V(\sigma)|, |V(\tau)|, |V(\sigma'^T)| \leq D_V$*

- *For each $e \in E(\alpha)$, $l_e \leq D_E$.*

### C.6.2 Sparse PCA

We consider the $m$ vectors $v_1, \ldots, v_m \in \mathbb{R}^d$ to be the input. Similar to Tensor PCA, we will work with the Hermite basis of polynomials since the entries are sampled from the distribution $\mathcal{N}(0,1)$. In particular, if we denote the unnormalized Hermite polynomials by $h_0(x) = 1, h_1(x) = x, h_2(x) = x^2 - 1, \ldots$, then, we work with the basis $h_a(v) := \prod_{i \in [m], j \in [n]} h_{a_{i,j}}(v_{i,j})$ over $a \in \mathbb{N}^{m \times n}$. To capture this basis, we will modify the graphs that represent ribbons (and by extension, shapes), to be bipartite graphs with two types of vertices, and have labeled edges that go across vertices of different types. So, an edge $(i, j)$ with label $t$ between a vertex $i$ of type 1 and a vertex $j$ of type 2 will correspond to $h_t(v_{i,j})$.

**Definition C.51** (Vertices). *We will have two types of vertices, the vertices corresponding to the $m$ input vectors that we call type 1 vertices and the vertices corresponding to ambient dimension of the space that we call type 2 vertices.*

**Definition C.52** (Edges). *Edges will go across vertices of different types, thereby forming a bipartite graph. An edge between a type 1 vertex $i$ and a type 2 vertex $j$ corresonds to the input entry $v_{i,j}$. Each edge will have a label in $\mathbb{N}$ corresponding to the Hermite polynomial evaluated on that entry.*

We will have variables $x_1, \ldots, x_n$ in our SoS program, so we will work with index shape pieces and index shapes as in Tensor PCA, since the rows and columns of our moment matrix will now correspond to monomials of the form $\prod_{i \leq n} x_i^{p_i}$ for $p_i \geq 0$. But since in our decompositions into left, right and middle parts, we will have type 2 vertices as well in the vertex separators, we will define a generalized notion of index shape pieces and index shapes.

**Definition C.53** (Index shape piece). *An index shape piece $U_i = ((U_{i,1}, \ldots, U_{i,t}), t_i, p_i)$ is a tuple of indices $(U_{i,1}, \ldots, U_{i,t})$ along a type $t_i \in \{1, 2\}$ with a power $p_i \in \mathbb{N}$. Let $V(U_i)$ be the set $\{U_{i,1}, \ldots, U_{i,t}\}$ of vertices of this index shape piece. When clear from context, we use $U_i$ instead of $V(U_i)$.*

For an index shape piece $((U_{i,1}, \ldots, U_{i,t}), t_i, p_i)$ with type $t_i = 2$, if we realize $U_{i_1}, \ldots, U_{i_t}$ to be indices $a_1, \ldots, a_t \in [n]$, then, this index shape pieces correspond this to the monomial $\prod_{j \leq n} x_{a_j}^{p_i}$.

**Definition C.54** (Index shape). *An index shape $U$ is a set of index shape pieces $U_i$ that have either have different types or different powers. Let $V(U)$ be the set of vertices $\cup_i V(U_i)$. When clear from context, we use $U$ instead of $V(U)$.*

Observe that each realization of an index shape corresponds to a row or column of the moment matrix. For our moment matrix, the only nonzero rows correspond to index shapes that have only index shape pieces of type 2, since the only SoS variables are $x_1 \ldots, x_n$, but in order to do our analysis, we need to work with the generalized notion of index shapes that allow index shape pieces of both types.

**Definition C.55.** *For two index shapes $U, V$, we write $U \equiv V$ if for all types $t$ and all powers $p$, the index shape pieces of type $t$ and power $p$ in $U$ and $V$ have the same length.*

**Definition C.56.** *Define $\mathcal{I}_{mid}$ to be the set of all index shapes $U$ that contain only index shape pieces of power 1.*

Since we are working with standard graphs, the notion of path and vertex separator need no modifications, but we will now use the minimum weight vertex separator instead of the minimum vertex separator where we define the weight as follows.

**Definition C.57** (Weight of an index shape). *Suppose we have an index shape $U = \{U_1, U_2\} \in \mathcal{I}_{mid}$ where $U_1 = ((U_{1,1}, \ldots, U_{1,|U_1|}), 1, 1)$ is an index shape piece of type 1 and $U_2 = ((U_{2,1}, \ldots, U_{2,|U_2|}), 2, 1)$ is an index shape piece of type 2. Then, define the weight of this index shape to be $w(U) = \sqrt{m}^{|U_1|}\sqrt{n}^{|U_2|}$.*

We now give the modified definition of shapes.

**Definition C.58** (Shapes). *Shapes are tuples $\alpha = (H_\alpha, U_\alpha, V_\alpha)$ where $H_\alpha$ is a graph with two types of vertices, has labeled edges only across vertices of different types and $U_\alpha, V_\alpha$ are index shapes such that $U_\alpha, V_\alpha \subseteq V(H_\alpha)$.*

**Definition C.59** (Proper shape). *A shape $\alpha$ is proper if it has no isolated vertices outside $U_\alpha \cup V_\alpha$, no multi-edges and all the edges have a nonzero label.*

In Appendix F, we will show that with this new definition of weight and shapes, any shape $\alpha$ has a unique decomposition into $\sigma \circ \tau \circ \sigma'^T$ where $\sigma, \tau, \sigma'^T$ are left, middle and right parts respectively. Here, $\tau$ may possibly be improper.

In the definition of trivial shape $\tau$, we now require $U_\tau \equiv V_\tau$. For $U \in \mathcal{I}_{mid}$, $\mathcal{M}_U$ will be the set of proper non-trivial middle parts $\tau$ with $U_\tau \equiv V_\tau \equiv U$ and $\mathcal{L}_U$ will be the set of left parts $\sigma$ such that $V_\sigma \equiv U$. Similarly, for $U, V \in \mathcal{I}_{mid}$, $\mathcal{L}_{U,V}$ will be the set of left parts $\gamma$ such that $U_\gamma \equiv U$ and $V_\gamma \equiv V$.

Finally, in order to define the moment matrix, we need to truncate our shapes based on the number of vertices and the labels on our edges. So, we make the following definition.

**Definition C.60** (Truncation parameters). *For integers $D_{sos}, D_V, D_E \geq 0$, say that a shape $\alpha$ satisfies the truncation parameters $D_{sos}, D_V, D_E$ if*

- *The degrees of the monomials that $U_\alpha$ and $V_\alpha$ correspond to, are at most $\frac{D_{sos}}{2}$*

- *The left part $\sigma$, the middle part $\tau$ and the right part $\sigma'^T$ of $\alpha$ satisfy $|V(\sigma)|, |V(\tau)|, |V(\sigma'^T)| \leq D_V$*

- *For each $e \in E(\alpha)$, $l_e \leq D_E$.*

### C.6.3 Relaxing the third condition

In Theorem C.37, the third qualitative condition we'd like to show is as follows: For all $U, V \in \mathcal{I}_{mid}$ such that $|U| > |V|$ and all $\gamma \in \Gamma_{U,V}$, $H_{Id_{V_\gamma}}^{-\gamma,\gamma} \preceq f(\gamma) H_{Id_{U_\gamma}}$. For technical reasons, we won't be able to show this directly. To handle this, we instead work with a slight modification of $H_{Id_{U_\gamma}}$, a matrix $H'_\gamma$ that's very close to $H_{Id_{U_\gamma}}$. So, what we will end up showing is: For all $U, V \in \mathcal{I}_{mid}$ such that $|U| > |V|$ and all $\gamma \in \Gamma_{U,V}$, $H_{Id_{V_\gamma}}^{-\gamma,\gamma} \preceq f(\gamma) H'_\gamma$.

Let $D_V$ be the truncation parameter. A canonical choice for $H'_\gamma$ is to take

1. $H'_\gamma(\sigma, \sigma') = H_{Id_U}(\sigma, \sigma')$ whenever $|V(\sigma \circ \gamma)| \leq D_V$ and $|V(\sigma' \circ \gamma)| \leq D_V$.
2. $H'_\gamma(\sigma, \sigma') = 0$ whenever $|V(\sigma \circ \gamma)| > D_V$ or $|V(\sigma' \circ \gamma)| > D_V$.

With this choice, $H'_\gamma$ is the same as $H_{Id_{U_\gamma}}$ upto truncation error.

## D  Application: Tensor PCA

We first decompose the moment matrix into graph matrices and then show the qualitative bounds needed.

### D.1  Decomposition into graph matrices

Define the degree of SoS to be $D_{sos} = n^{C_{sos}\varepsilon}$ for some constant $C_{sos} > 0$ that we choose later. And define the truncation parameters to be $D_V = n^{C_V\varepsilon}, D_E = n^{C_E\varepsilon}$ for some constants $C_V, C_E > 0$.

**Remark D.1** (Choice of parameters). *We first set $\varepsilon$ to be a sufficiently small constant. Based on the choice of $\varepsilon$, we will set the constant $C_\Delta > 0$ sufficiently small so that the planted distribution is well defined. Based on these choices, we choose $C_V, C_E, C_{sos}$ in that order.*

The underlying graphs for the graph matrices have the following structure; There will be $n$ vertices of a single type and the edges will be ordered hyperedges of arity $k$. For the analysis of Tensor PCA, we will use the following notation. For an index shape $U$ and a vertex $i$, define $deg^U(i)$ as follows: If $i \in V(U)$, then it is the power of the unique index shape piece $A \in U$ such that $i \in V(A)$. Otherwise, it is 0. Also define $deg(U) = \sum_{i \in V(U)} deg^U(i)$. This is also the degree of the monomial that $U$ corresponds to. For a shape $\alpha$ and vertex $i$ in $\alpha$, let $deg^\alpha(i) = \sum_{i \in e \in E(\alpha)} l_e$ and let $deg(\alpha) = deg(U_\alpha) + deg(V_\alpha)$.

We will now describe the decomposition of the moment matrix $\Lambda$ using Lemma B.1.

**Definition D.2.** *If a shape $\alpha$ is proper, satisfies the truncation parameters $D_{sos}, D_V, D_E$ and is such that $deg^\alpha(i) + deg^{U_\alpha}(i) + deg^{V_\alpha}(i)$ is even for all $i \in V(\alpha)$, define*

$$\lambda_\alpha = \Delta^{|V(\alpha)|} \left( \frac{1}{\sqrt{\Delta n}} \right)^{deg(\alpha)} \prod_{e \in E(\alpha)} \left( \frac{\lambda}{(\Delta n)^{\frac{k}{2}}} \right)^{l_e}$$

*Otherwise, define $\lambda_\alpha = 0$.*

**Corollary D.3.** $\Lambda = \sum \lambda_\alpha M_\alpha$.

## D.2 Qualitative bounds

We prove the PSD mass condition and the middle shape and intersection term bounds, by first stating them and then introducing appropriate notation to prove them all in a unified manner.

**Lemma D.4** (PSD mass). *For all $U \in \mathcal{I}_{mid}$, $H_{Id_U} \succeq 0$*

We define the following quantities to capture the contribution of the vertices within $\tau, \gamma$ to the Fourier coefficients.

**Definition D.5.** *For $U \in \mathcal{I}_{mid}$ and $\tau \in \mathcal{M}_U$, if $deg^\tau(i)$ is even for all vertices $i \in V(\tau) \setminus U_\tau \setminus V_\tau$, define*

$$S(\tau) = \Delta^{|V(\tau)| - |U_\tau|} \prod_{e \in E(\tau)} \left( \frac{\lambda}{(\Delta n)^{\frac{k}{2}}} \right)^{l_e}$$

*Otherwise, define $S(\tau) = 0$. For all $U, V \in \mathcal{I}_{mid}$ where $w(U) > w(V)$ and $\gamma \in \Gamma_{U,V}$, if $deg^\gamma(i)$ is even for all vertices $i$ in $V(\gamma) \setminus U_\gamma \setminus V_\gamma$, define*

$$S(\gamma) = \Delta^{|V(\gamma)| - \frac{|U_\gamma| + |V_\gamma|}{2}} \prod_{e \in E(\gamma)} \left( \frac{\lambda}{(\Delta n)^{\frac{k}{2}}} \right)^{l_e}$$

*Otherwise, define $S(\gamma) = 0$.*

We now state the bounds in terms of these quantities.

**Lemma D.6** (Middle shape bounds). *For all $U \in \mathcal{I}_{mid}$ and $\tau \in \mathcal{M}_U$,*

$$\begin{bmatrix} \frac{S(\tau)}{|Aut(U)|} H_{Id_U} & H_\tau \\ H_\tau^T & \frac{S(\tau)}{|Aut(U)|} H_{Id_U} \end{bmatrix} \succeq 0$$

We again use the canonical definition of $H'_\gamma$ from Appendix C.6.3.

**Lemma D.7** (Intersection term bounds). *For all $U, V \in \mathcal{I}_{mid}$ where $w(U) > w(V)$ and all $\gamma \in \Gamma_{U,V}$, $\frac{|Aut(V)|}{|Aut(U)|} \cdot \frac{1}{S(\gamma)^2} H_{Id_V}^{-\gamma, \gamma} \preceq H'_\gamma$.*

### D.2.1 Proof of PSD mass condition

We introduce some notation which makes it easy to show these bounds and which also sheds light on the structure of the coefficient matrices. When we compose shapes $\sigma, \sigma'$, from Definition D.2 in order for $\lambda_{\sigma \circ \sigma'}$ to be nonzero, observe that all vertices $i$ in $\lambda_{\sigma \circ \sigma'}$ should have $deg^{\sigma \circ \sigma'}(i) + deg^{U_{\sigma \circ \sigma'}}(i) + deg^{V_{\sigma \circ \sigma'}}(i)$ to be even. To partially capture this notion conveniently, we will introduce the notion of parity vectors.

**Definition D.8.** *Define a parity vector $\rho$ to be a vector whose entries are in $\{0, 1\}$. For $U \in \mathcal{I}_{mid}$, define $\mathcal{P}_U$ to be the set of parity vectors $\rho$ whose coordinates are indexed by $U$.*

**Definition D.9.** *For a left shape $\sigma$, define $\rho_\sigma \in \mathcal{P}_{V_\sigma}$, called the parity vector of $\sigma$, to be the parity vector such that for each vertex $i \in V_\sigma$, the $i$-th entry of $\rho_\sigma$ is the parity of $deg^{U_\sigma}(i) + deg^\sigma(i)$, that is $(\rho_\sigma)_i \equiv deg^{U_\sigma}(i) + deg^\sigma(i) \pmod 2$. For $U \in \mathcal{I}_{mid}$ and $\rho \in \mathcal{P}_U$, let $\mathcal{L}_{U,\rho}$ be the set of all left shapes $\sigma \in \mathcal{L}_U$ such that $\rho_\sigma = \rho$, that is, the set of all left shapes with parity vector $\rho$.*

For a shape $\tau$, for a $\tau$ coefficient matrix $H_\tau$ and parity vectors $\rho \in \mathcal{P}_{U_\tau}, \rho' \in \mathcal{P}_{V_\tau}$, define the $\tau$-coefficient matrix $H_{\tau,\rho,\rho'}$ as $H_{\tau,\rho,\rho'}(\sigma, \sigma') = H_\tau(\sigma, \sigma')$ if $\sigma \in \mathcal{L}_{U_\tau,\rho}, \sigma' \in \mathcal{L}_{V_\tau,\rho'}$ and 0 otherwise. The following proposition is immediate.

**Proposition D.10.** *For any shape $\tau$ and $\tau$-coefficient matrix $H_\tau$, $H_\tau = \sum_{\rho \in \mathcal{P}_{U_\tau}, \rho' \in \mathcal{P}_{V_\tau}} H_{\tau,\rho,\rho'}$*

**Proposition D.11.** *For any $U \in \mathcal{I}_{mid}$, $H_{Id_U} = \sum_{\rho \in \mathcal{P}_U} H_{Id_U,\rho,\rho}$*

*Proof.* For any $\sigma, \sigma' \in \mathcal{L}_U$, using Definition D.2 note that in order for $H_{Id_U}(\sigma, \sigma')$ to be nonzero, we must have $\rho_\sigma = \rho_{\sigma'}$. ∎

We define the following quantity to capture the contribution of the vertices within $\sigma$ to the Fourier coefficients.

**Definition D.12.** *For a shape $\sigma \in \mathcal{L}$, if $deg^\sigma(i) + deg^{U_\sigma}(i)$ is even for all vertices $i \in V(\sigma) \setminus V_\sigma$, define*

$$T(\sigma) = \Delta^{|V(\sigma)| - \frac{|V_\sigma|}{2}} \left( \frac{1}{\sqrt{\Delta n}} \right)^{deg(U_\sigma)} \prod_{e \in E(\sigma)} \left( \frac{\lambda}{(\Delta n)^{\frac{k}{2}}} \right)^{l_e}$$

*Otherwise, define $T(\sigma) = 0$. For $U \in \mathcal{I}_{mid}$ and $\rho \in \mathcal{P}_U$, define $v_\rho$ to be the vector indexed by $\sigma \in \mathcal{L}$ such that $v_\rho(\sigma)$ is $T(\sigma)$ if $\sigma \in \mathcal{L}_{U,\rho}$ and 0 otherwise.*

With this notation, the PSD mass condition is easily shown.

*Proof of the PSD mass condition Lemma D.4.* For all $U \in \mathcal{I}_{mid}, \rho \in \mathcal{P}_U$, Definition D.2 implies $H_{Id_U,\rho,\rho} = \frac{1}{|Aut(U)|} v_\rho v_\rho^T$. Therefore, $H_{Id_U} = \sum_{\rho \in \mathcal{P}_U} H_{Id_U,\rho,\rho} = \frac{1}{|Aut(U)|} \sum_{\rho \in \mathcal{P}_U} v_\rho v_\rho^T \succeq 0$. ∎

### D.2.2 Middle shape bounds

The next proposition captures the fact that when we compose shapes $\sigma, \tau, \sigma'^T$, in order for $\lambda_{\sigma \circ \tau \circ \sigma'^T}$ to be nonzero, the parities of the degrees of the merged vertices should add up correspondingly.

**Proposition D.13.** *For all $U \in \mathcal{I}_{mid}$ and $\tau \in \mathcal{M}_U$, there exist two sets of parity vectors $P_\tau, Q_\tau \subseteq \mathcal{P}_U$ and a bijection $\pi : P_\tau \to Q_\tau$ such that $H_\tau = \sum_{\rho \in P_\tau} H_{\tau,\rho,\pi(\rho)}$.*

*Proof.* Using Definition D.2 in order for $H_\tau(\sigma, \sigma')$ to be nonzero, in $\sigma \circ \tau \circ \sigma'$, we must have that for all $i \in U_\tau \cup V_\tau, deg^{U_\sigma}(i) + deg^{U_{\sigma'}}(i) + deg^{\sigma \circ \tau \circ \sigma'^T}(i)$ must be even. In other words, for any $\rho \in \mathcal{P}_U$, there is at most one $\rho' \in \mathcal{P}_U$ such that if we take $\sigma \in \mathcal{L}_{U,\rho}, \sigma' \in \mathcal{L}_U$ with $H_\tau(\sigma, \sigma')$ nonzero, then the parity of $\sigma'$ is $\rho'$. Also, observe that $\rho'$ determines $\rho$. We then take $P_\tau$ to be the set of $\rho$ such that $\rho'$ exists, $Q_\tau$ to be the set of $\rho'$ and in this case, we define $\pi(\rho) = \rho'$. ∎

A straightforward verification of the conditions of Definition D.2 implies the following proposition.

**Proposition D.14.** *For any $U \in \mathcal{I}_{mid}$ and $\tau \in \mathcal{M}_U$, suppose we take $\rho \in P_\tau$. Let $\pi$ be the bijection from Proposition D.13 so that $\pi(\rho) \in Q_\tau$. Then, $H_{\tau,\rho,\pi(\rho)} = \frac{1}{|Aut(U)|^2} S(\tau) v_\rho v_{\pi(\rho)}^T$.*

We can now prove the middle shape bounds.

*Proof of the middle shape bounds Lemma D.6.* Let $P_\tau, Q_\tau, \pi$ be from Proposition D.13. For $\rho, \rho' \in \mathcal{P}_U$, let $W_{\rho,\rho'} = v_\rho (v_{\rho'})^T$. Then, $H_{Id_U} = \sum_{\rho \in \mathcal{P}_U} H_{Id_U,\rho,\rho} = \frac{1}{|Aut(U)|} \sum_{\rho \in \mathcal{P}_U} W_{\rho,\rho}$ and $H_\tau = \sum_{\rho \in P_\tau} H_{\tau,\rho,\pi(\rho)} = \frac{1}{|Aut(U)|^2} S(\tau) \sum_{\rho \in P_\tau} W_{\rho,\pi(\rho)}$. We have

$$\begin{bmatrix} \frac{S(\tau)}{|Aut(U)|} H_{Id_U} & H_\tau \\ H_\tau^T & \frac{S(\tau)}{|Aut(U)|} H_{Id_U} \end{bmatrix} = \frac{S(\tau)}{|Aut(U)|^2} \begin{bmatrix} \sum_{\rho \in \mathcal{P}_U} W_{\rho,\rho} & \sum_{\rho \in P_\tau} W_{\rho,\pi(\rho)} \\ \sum_{\rho \in P_\tau} W_{\rho,\pi(\rho)}^T & \sum_{\rho \in \mathcal{P}_U} W_{\rho,\rho} \end{bmatrix}$$

We have $\frac{S(\tau)}{|Aut(U)|^2} \geq 0$ and the matrix is just

$$\begin{bmatrix} \sum_{\rho \in \mathcal{P}_U \setminus P_\tau} W_{\rho,\rho} & 0 \\ 0 & \sum_{\rho \in \mathcal{P}_U \setminus Q_\tau} W_{\rho,\rho} \end{bmatrix} + \begin{bmatrix} \sum_{\rho \in P_\tau} W_{\rho,\rho} & \sum_{\rho \in P_\tau} W_{\rho,\pi(\rho)} \\ \sum_{\rho \in P_\tau} W_{\rho,\pi(\rho)}^T & \sum_{\rho \in P_\tau} W_{\pi(\rho),\pi(\rho)} \end{bmatrix}$$

We have $\sum_{\rho \in \mathcal{P}_U \setminus P_\tau} W_{\rho,\rho} = \sum_{\rho \in \mathcal{P}_U \setminus P_\tau} v_\rho v_\rho^T \succeq 0$. Similarly, $\sum_{\rho \in \mathcal{P}_U \setminus Q_\tau} W_{\rho,\rho} \succeq 0$ and so, the first term in the above expression, $\begin{bmatrix} \sum_{\rho \in \mathcal{P}_U \setminus P_\tau} W_{\rho,\rho} & 0 \\ 0 & \sum_{\rho \in \mathcal{P}_U \setminus Q_\tau} W_{\rho,\rho} \end{bmatrix}$ is positive semidefinite. For the second term,

$$\begin{bmatrix} \sum_{\rho \in P_\tau} W_{\rho,\rho} & \sum_{\rho \in P_\tau} W_{\rho,\pi(\rho)} \\ \sum_{\rho \in P_\tau} W_{\rho,\pi(\rho)}^T & \sum_{\rho \in P_\tau} W_{\pi(\rho),\pi(\rho)} \end{bmatrix} = \sum_{\rho \in P_\tau} \begin{bmatrix} v_\rho v_\rho^T & v_\rho (v_{\pi(\rho)})^T \\ v_{\pi(\rho)} (v_\rho)^T & v_{\pi(\rho)} (v_{\pi(\rho)})^T \end{bmatrix} \qquad \succeq 0$$

∎

### D.2.3 Intersection term bounds

Similar to Proposition D.13, the next proposition captures the fact that when we compose shapes $\sigma, \gamma, \gamma^T, \sigma'^T$, in order for $\lambda_{\sigma \circ \gamma \circ \gamma^T \circ \sigma'^T}$ to be nonzero, the parities of the degrees of the merged vertices should add up correspondingly.

We use the following notation. For all $U, V \in \mathcal{I}_{mid}$ where $w(U) > w(V)$, for $\gamma \in \Gamma_{U,V}$ and parity vectors $\rho, \rho' \in \mathcal{P}_U$, define the $\gamma \circ \gamma^T$-coefficient matrix $H_{Id_V,\rho,\rho'}^{-\gamma,\gamma}$ as $H_{Id_V,\rho,\rho'}^{-\gamma,\gamma}(\sigma,\sigma') = H_{Id_V}^{-\gamma,\gamma}(\sigma,\sigma')$ if $\sigma \in \mathcal{L}_{U,\rho}, \sigma' \in \mathcal{L}_{U,\rho'}$ and 0 otherwise.

**Proposition D.15.** *For all $U, V \in \mathcal{I}_{mid}$ where $w(U) > w(V)$, for all $\gamma \in \Gamma_{U,V}$, there exists a set of parity vectors $P_\gamma \subseteq \mathcal{P}_U$ such that $H_{Id_V}^{-\gamma,\gamma} = \sum_{\rho \in P_\gamma} H_{Id_V,\rho,\rho}^{-\gamma,\gamma}$.*

*Proof.* Take any $\rho \in \mathcal{P}_U$. For $\sigma \in \mathcal{L}_{U,\rho}, \sigma' \in \mathcal{L}_U$, since $H_{Id_V}^{-\gamma,\gamma}(\sigma,\sigma') = \frac{\lambda_{\sigma \circ \gamma \circ \gamma^T \circ \sigma'^T}}{|Aut(V)|}$, $H_{Id_V}^{-\gamma,\gamma}(\sigma,\sigma')$ is nonzero precisely when $\lambda_{\sigma \circ \gamma \circ \gamma^T \circ \sigma'^T}$ is nonzero. For this quantity to be nonzero, using Definition D.2, we get that it is necessary, but not sufficient, that the parity vector of $\sigma'$ must also be $\rho$. And also observe that there exists a set $P_\gamma$ of parity vectors $\rho$ for which $H_{Id_V,\rho,\rho}^{-\gamma,\gamma}$ is nonzero and their sum is precisely $H_{Id_V}^{-\gamma,\gamma}$. ∎

For all $U, V \in \mathcal{I}_{mid}$ where $w(U) > w(V)$, for all $\gamma \in \Gamma_{U,V}$ and parity vector $\rho \in \mathcal{P}_U$, define the matrix $H'_{\gamma,\rho,\rho}$ as $H'_{\gamma,\rho,\rho}(\sigma,\sigma') = H'_\gamma(\sigma,\sigma')$ if $\sigma, \sigma' \in \mathcal{L}_{U,\rho}$ and 0 otherwise. The following proposition is immediate from the definition.

**Proposition D.16.** *For all $U, V \in \mathcal{I}_{mid}$ where $w(U) > w(V)$, for $\gamma \in \Gamma_{U,V}$, $H'_\gamma = \sum_{\rho \in P_\gamma} H'_{\gamma,\rho,\rho}$.*

**Proposition D.17.** *For all $U, V \in \mathcal{I}_{mid}$ where $w(U) > w(V)$, for all $\gamma \in \Gamma_{U,V}$ and $\rho \in P_\gamma$, $H_{Id_V,\rho,\rho}^{-\gamma,\gamma} = \frac{|Aut(U)|}{|Aut(V)|} S(\gamma)^2 H'_{\gamma,\rho,\rho}$.*

*Proof.* Fix $\sigma, \sigma' \in \mathcal{L}_{U,\rho}$ such that $|V(\sigma \circ \gamma)|, |V(\sigma' \circ \gamma)| \leq D_V$. Note that $|V(\sigma)| - \frac{|V_\sigma|}{2} + |V(\sigma')| - \frac{|V_{\sigma'}|}{2} + 2(|V(\gamma)| - \frac{|U_\gamma| + |V_\gamma|}{2}) = |V(\sigma \circ \gamma \circ \gamma^T \circ \sigma'^T)|$. Using Definition D.2, we can easily verify that $\lambda_{\sigma \circ \gamma \circ \gamma^T \circ \sigma'^T} = T(\sigma) T(\sigma') S(\gamma)^2$. Therefore, $H_{Id_V,\rho,\rho}^{-\gamma,\gamma}(\sigma,\sigma') = \frac{|Aut(U)|}{|Aut(V)|} S(\gamma)^2 H_{Id_U,\rho,\rho}(\sigma,\sigma')$. Since $H'_{\gamma,\rho,\rho}(\sigma,\sigma') = H_{Id_U,\rho,\rho}(\sigma,\sigma')$ whenever $|V(\sigma \circ \gamma)|, |V(\sigma' \circ \gamma)| \leq D_V$, this completes the proof. ∎

With this, we can prove the intersection term bounds.

*Proof of intersection term bounds Lemma D.7.* We have

$$\frac{|Aut(V)|}{|Aut(U)|} \cdot \frac{1}{S(\gamma)^2} H_{Id_V}^{-\gamma,\gamma} = \sum_{\rho \in P_\gamma} \frac{|Aut(V)|}{|Aut(U)|} \cdot \frac{1}{S(\gamma)^2} H_{Id_V,\rho,\rho}^{-\gamma,\gamma} = \sum_{\rho \in P_\gamma} H'_{\gamma,\rho,\rho} \preceq \sum_{\rho \in \mathcal{P}_U} H'_{\gamma,\rho,\rho} = H'_\gamma$$

where we used the fact that for all $\rho \in \mathcal{P}_U$, we have $H'_{\gamma,\rho,\rho} \succeq 0$. ∎

# E   Application: Sparse PCA

Just as in the earlier application, we decompose the moment matrix into graph matrices and then show the necessary qualitative bounds.

## E.1   Decomposition into graph matrices

Define the degree of SoS to be $D_{sos} = d^{C_{sos}\varepsilon}$ for some constant $C_{sos} > 0$ that we choose later. Define the truncation parameters to be $D_V = d^{C_V\varepsilon}, D_E = d^{C_E\varepsilon}$ for some constants $C_V, C_E > 0$. Regarding the choice of parameters, Remark D.1 directly applies.

The underlying graphs for the graph matrices have the following structure: There will be two types of vertices - $d$ type 1 vertices corresponding to the dimensions of the space and $m$ type 2 vertices corresponding to the different input vectors. The shapes will correspond to bipartite graphs with edges going between across of different types.

For the analysis of Sparse PCA, we will use the following notation. For a shape $\alpha$ and type $t \in \{1, 2\}$, let $V_t(\alpha)$ denote the vertices of $V(\alpha)$ that are of type $t$ and let $|\alpha|_t = |V_t(\alpha)|$. For an index shape $U$ and a vertex $i$, define $deg^U(i)$ as follows: If $i \in V(U)$, then it is the power of the unique index shape piece $A \in U$ such that $i \in V(A)$. Otherwise, it is 0. Also, let $deg(U) = \sum_{i \in V(U)} deg^U(i)$. This is also the degree of the monomial $p_U$. For a shape $\alpha$ and vertex $i$ in $\alpha$, let $deg^\alpha(i) = \sum_{i \in e \in E(\alpha)} l_e$ and let $deg(\alpha) = deg(U_\alpha) + deg(V_\alpha)$. For an index shape $U \in \mathcal{I}_{mid}$ and type $t \in \{1, 2\}$, let $U_t \in U$ denote the index shape piece of type $t$ in $U$ if it exists, otherwise define $U_t$ to be $\emptyset$. Also, denote by $|U|_t$ the length of the tuple $U_t$.

We will now describe the decomposition of the moment matrix $\Lambda$, where we apply Lemma B.2.

**Definition E.1.** *If a shape $\alpha$ is proper, satisfies the truncation parameters $D_{sos}, D_V, D_E$ and is such that both $U_\alpha, V_\alpha$ only contain index shape pieces of type 1 and $deg^\alpha(i) + deg^{U_\alpha}(i) + deg^{V_\alpha}(i)$ is even for all $i \in V(\alpha)$, define*

$$\lambda_\alpha = \left(\frac{1}{\sqrt{k}}\right)^{deg(\alpha)} \left(\frac{k}{d}\right)^{|\alpha|_1} \Delta^{|\alpha|_2} \prod_{j \in V_2(\alpha)} (deg^\alpha(j) - 1)!! \prod_{e \in E(\alpha)} \frac{\sqrt{\lambda}^{l_e}}{\sqrt{k}^{l_e}}$$

*Otherwise, define $\lambda_\alpha = 0$.*

**Corollary E.2.** $\Lambda = \sum \lambda_\alpha M_\alpha$.

## E.2   Qualitative bounds

In this section, we will prove the main PSD mass condition and obtain bounds of the other two conditions. As in the prior section, we will state the bounds first, introduce notation and then prove them all in a unified manner.

**Lemma E.3** (PSD mass). *For all $U \in \mathcal{I}_{mid}$, $H_{Id_U} \succeq 0$*

We define the following quantities to capture the contribution of the vertices within $\tau, \gamma$ to the Fourier coefficients.

**Definition E.4.** *For $U \in \mathcal{I}_{mid}$ and $\tau \in \mathcal{M}_U$, if $deg^\tau(i)$ is even for all vertices $i \in V(\tau) \setminus U_\tau \setminus V_\tau$, define*

$$S(\tau) = \left(\frac{k}{d}\right)^{|\tau|_1 - |U_\tau|_1} \Delta^{|\tau|_2 - |U_\tau|_2} \prod_{j \in V_2(\tau) \setminus U_\tau \setminus V_\tau} (deg^\tau(j) - 1)!! \prod_{e \in E(\tau)} \frac{\sqrt{\lambda}^{l_e}}{\sqrt{k}^{l_e}}$$

*Otherwise, define $S(\tau) = 0$. For all $U, V \in \mathcal{I}_{mid}$ where $w(U) > w(V)$ and $\gamma \in \Gamma_{U,V}$, if $deg^\gamma(i)$ is even for all vertices $i$ in $V(\gamma) \setminus U_\gamma \setminus V_\gamma$, define*

$$S(\gamma) = \left(\frac{k}{d}\right)^{|\gamma|_1 - \frac{|U_\gamma|_1 + |V_\gamma|_1}{2}} \Delta^{|\gamma|_2 - \frac{|U_\gamma|_2 + |V_\gamma|_2}{2}} \prod_{j \in V_2(\gamma) \setminus U_\gamma \setminus V_\gamma} (deg^\gamma(j) - 1)!! \prod_{e \in E(\gamma)} \frac{\sqrt{\lambda}^{l_e}}{\sqrt{k}^{l_e}}$$

*Otherwise, define $S(\gamma) = 0$.*

For getting the best bounds, it will be convenient to discretize the Normal distribution. The following fact follows from standard results on Gaussian quadrature, see for e.g. [34, Lemma 4.3].

**Fact E.5** (Discretizing the Normal distribution). *There is an absolute constant $C_{disc}$ such that, for any positive integer $D$, there exists a distribution $\mathcal{E}$ over the real numbers supported on $D$ points $p_1, \ldots, p_D$, such that $|p_i| \leq C_{disc}\sqrt{D}$ for all $i \leq D$ and $\mathbb{E}_{g \sim \mathcal{E}}[g^t] = \mathbb{E}_{g \sim \mathcal{N}(0,1)}[g^t]$ for all $t = 0, 1, \ldots, 2D - 1$.*

**Definition E.6.** *For any shape $\tau$, suppose $U' = (U_\tau)_2, V' = (V_\tau)_2$ are the type 2 vertices in $U_\tau, V_\tau$ respectively. Define $R(\tau) = (C_{disc}\sqrt{D_E})^{\sum_{j \in U' \cup V'} deg^\tau(j)}$.*

We can now state our bounds.

**Lemma E.7** (Middle shape bounds). *For all $U \in \mathcal{I}_{mid}$ and $\tau \in \mathcal{M}_U$,*

$$\begin{bmatrix} \frac{S(\tau)R(\tau)}{|Aut(U)|} H_{Id_U} & H_\tau \\ H_\tau^T & \frac{S(\tau)R(\tau)}{|Aut(U)|} H_{Id_U} \end{bmatrix} \succeq 0$$

We again use the canonical definition of $H'_\gamma$ from Appendix C.6.3

**Lemma E.8** (Intersection term bounds). *For all $U, V \in \mathcal{I}_{mid}$ where $w(U) > w(V)$ and all $\gamma \in \Gamma_{U,V}$,* $\frac{|Aut(V)|}{|Aut(U)|} \cdot \frac{1}{S(\gamma)^2 R(\gamma)^2} H_{Id_V}^{-\gamma,\gamma} \preceq H'_\gamma$.

### E.2.1 Proof of the PSD mass condition

Most of the notation and analysis here are similar to the case of Tensor PCA, we just need to appropriately modify them since there are two types of vertices in the Sparse PCA application. When we compose shapes $\sigma, \sigma'$, from Definition E.1, in order for $\lambda_{\sigma \circ \sigma'}$ to be nonzero, observe that all vertices $i$ in $\lambda_{\sigma \circ \sigma'}$ should have $deg^{\sigma \circ \sigma'}(i) + deg^{U_{\sigma \circ \sigma'}}(i) + deg^{V_{\sigma \circ \sigma'}}(i)$ to be even. To capture this notion conveniently, we again use the notion of parity vectors.

**Definition E.9.** *Define a parity vector $\rho$ to be a vector whose entries are in $\{0, 1\}$. For $U \in \mathcal{I}_{mid}$, define $\mathcal{P}_U$ to be the set of parity vectors $\rho$ whose coordinates are indexed by $U_1$ followed by $U_2$.*

**Definition E.10.** *For a left shape $\sigma$, define $\rho_\sigma \in \mathcal{P}_{V_\sigma}$, called the parity vector of $\sigma$, to be the parity vector such that for each vertex $i \in V_\sigma$, the $i$-th entry of $\rho_\sigma$ is the parity of $deg^{U_\sigma}(i) + deg^\sigma(i)$, that is, $(\rho_\sigma)_i \equiv deg^{U_\sigma}(i) + deg^\sigma(i) \pmod{2}$. For $U \in \mathcal{I}_{mid}$ and $\rho \in \mathcal{P}_U$, let $\mathcal{L}_{U,\rho}$ be the set of all left shapes $\sigma \in \mathcal{L}_U$ such that $\rho_\sigma = \rho$, that is, the set of all left shapes with parity vector $\rho$.*

For a shape $\tau$, for a $\tau$ coefficient matrix $H_\tau$ and parity vectors $\rho \in \mathcal{P}_{U_\tau}, \rho' \in \mathcal{P}_{V_\tau}$, define the $\tau$-coefficient matrix $H_{\tau,\rho,\rho'}$ as $H_{\tau,\rho,\rho'}(\sigma, \sigma') = H_\tau(\sigma, \sigma')$ if $\sigma \in \mathcal{L}_{U_\tau,\rho}, \sigma' \in \mathcal{L}_{V_\tau,\rho'}$ and 0 otherwise. This immediately implies the following proposition.

**Proposition E.11.** *For any shape $\tau$ and $\tau$-coefficient matrix $H_\tau$, $H_\tau = \sum_{\rho \in \mathcal{P}_{U_\tau}, \rho' \in \mathcal{P}_{V_\tau}} H_{\tau,\rho,\rho'}$*

**Proposition E.12.** *For any $U \in \mathcal{I}_{mid}$, $H_{Id_U} = \sum_{\rho \in \mathcal{P}_U} H_{Id_U,\rho,\rho}$*

*Proof.* For any $\sigma, \sigma' \in \mathcal{L}_U$, using Definition E.1, note that in order for $H_{Id_U}(\sigma, \sigma')$ to be nonzero, we must have $\rho_\sigma = \rho_{\sigma'}$. $\blacksquare$

We now discretize the normal distribution while matching the first $2D_E - 1$ moments.

**Definition E.13.** *Let $\mathcal{D}$ be a distribution over the real numbers obtained by setting $D = D_E$ in Fact E.5. So, in particular, for any $x$ sampled from $\mathcal{D}$, we have $|x| \leq C_{disc}\sqrt{D_E}$ and for $t \leq 2D_E - 1$, $\mathbb{E}_{x \sim \mathcal{D}}[x^t] = (t-1)!!$.*

We define the following quantities to capture the contribution of the vertices within $\sigma$ to the Fourier coefficients.

**Definition E.14.** *For a shape $\sigma \in \mathcal{L}$, if $deg^{\sigma}(i) + deg^{U_{\sigma}}(i)$ is even for all vertices $i \in V(\sigma) \setminus V_{\sigma}$, define*

$$T(\sigma) = \left(\frac{1}{\sqrt{k}}\right)^{deg(U_{\sigma})} \left(\frac{k}{d}\right)^{|\sigma|_1 - \frac{|V_{\sigma}|_1}{2}} \Delta^{|\sigma|_2 - \frac{|V_{\sigma}|_2}{2}} \prod_{j \in V_2(\sigma) \setminus V_{\sigma}} (deg^{\sigma}(j) - 1)!! \prod_{e \in E(\sigma)} \frac{\sqrt{\lambda}^{l_e}}{\sqrt{k}^{l_e}}$$

*Otherwise, define $T(\sigma) = 0$.*

**Definition E.15.** *Let $U \in \mathcal{I}_{mid}$. Let $x_i$ for $i \in U_2$ be variables. Denote them collectively as $x_{U_2}$. For $\rho \in \mathcal{P}_U$, define $v_{\rho, x_{U_2}}$ to be the vector indexed by left shapes $\sigma \in \mathcal{L}$ such that the $\sigma$th entry is $T(\sigma) \prod_{i \in U_2} x_i^{deg^{\sigma}(i)}$ if $\sigma \in \mathcal{L}_{U,\rho}$ and $0$ otherwise.*

The following proposition is obvious and immediately implies the PSD mass condition.

**Proposition E.16.** *For any $U \in \mathcal{I}_{mid}, \rho \in \mathcal{P}_U$, suppose $x_i$ for $i \in U_2$ are random variables sampled from $\mathcal{D}$. Then, $H_{Id_U, \rho, \rho} = \frac{1}{|Aut(U)|} \mathbb{E}_x[v_{\rho, x_{U_2}} v_{\rho, x_{U_2}}^T]$.*

*Proof.* Observe that for $\sigma, \sigma' \in \mathcal{L}_{U,\rho}$ and $t \in \{1, 2\}$, $(|\sigma|_t - \frac{|V_{\sigma}|_t}{2}) + (|\sigma'|_t - \frac{|V_{\sigma'}|_t}{2}) = |\sigma \circ \sigma'|_t$. The result follows by verifying the conditions of Definition E.1 and using Definition E.13. ∎

*Proof of the PSD mass condition Lemma E.3.* We have $H_{Id_U} = \sum_{\rho \in \mathcal{P}_U} H_{Id_U, \rho, \rho} \succeq 0$ because of the above proposition. ∎

### E.2.2 Middle shape bounds

The next proposition captures the fact that when we compose shapes $\sigma, \tau, \sigma'^T$, in order for $\lambda_{\sigma \circ \tau \circ \sigma'^T}$ to be nonzero, the parities of the degrees of the merged vertices should add up correspondingly.

**Proposition E.17.** *For all $U \in \mathcal{I}_{mid}$ and $\tau \in \mathcal{M}_U$, there exist two sets of parity vectors $P_{\tau}, Q_{\tau} \subseteq \mathcal{P}_U$ and a bijection $\pi : P_{\tau} \to Q_{\tau}$ such that $H_{\tau} = \sum_{\rho \in P_{\tau}} H_{\tau, \rho, \pi(\rho)}$.*

*Proof.* Using Definition E.1, in order for $H_{\tau}(\sigma, \sigma')$ to be nonzero, we must have that, in $\sigma \circ \tau \circ \sigma'$, for all $i \in U_{\tau} \cup V_{\tau}$, $deg^{U_{\sigma}}(i) + deg^{U_{\sigma'}}(i) + deg^{\sigma \circ \tau \circ \sigma'^T}(i)$ must be even. In other words, for any $\rho \in \mathcal{P}_U$, there is at most one $\rho' \in \mathcal{P}_U$ such that if we take $\sigma \in \mathcal{L}_{U,\rho}, \sigma' \in \mathcal{L}_U$ with $H_{\tau}(\sigma, \sigma')$ nonzero, then the parity of $\sigma'$ is $\rho'$. Also, observe that $\rho'$ determines $\rho$. We then take $P_{\tau}$ to be the set of $\rho$ such that $\rho'$ exists, $Q_{\tau}$ to be the set of $\rho'$ and in this case, we define $\pi(\rho) = \rho'$. ∎

**Proposition E.18.** *For any $U \in \mathcal{I}_{mid}$ and $\tau \in \mathcal{M}_U$, suppose we take $\rho \in P_{\tau}$. Let $\pi$ be the bijection from Proposition E.17 so that $\pi(\rho) \in Q_{\tau}$. Let $U' = (U_{\tau})_2, V' = (V_{\tau})_2$ be the type 2 vertices in $U_{\tau}, V_{\tau}$ respectively. Let $x_i$ for $i \in U' \cup V'$ be random variables independently sampled from $\mathcal{D}$. Define $x_{U'}$ (resp. $x_{V'}$) to be the subset of variables $x_i$ for $i \in U'$ (resp. $i \in V'$). Then,*

$$H_{\tau, \rho, \pi(\rho)} = \frac{1}{|Aut(U)|^2} S(\tau) \mathbb{E}_x \left[ v_{\rho, x_{U'}} \left( \prod_{i \in U' \cup V'} x_i^{deg^{\tau}(i)} \right) v_{\pi(\rho), x_{V'}}^T \right]$$

*Proof.* For $\sigma \in L_{U,\rho}, \sigma' \in \mathcal{L}_{U, \pi(\rho)}$ and $t \in \{1, 2\}$, we have $(|\tau|_t - |U_{\tau}|_t) + (|\sigma|_t - \frac{|V_{\sigma}|_t}{2}) + (|\sigma'|_t - \frac{|V_{\sigma'}|_t}{2}) = |\sigma \circ \tau \circ \sigma'|_t$. The result then follows by a straightforward verification of the conditions of Definition E.1 using Definition E.13. ∎

We are ready to show the middle shape bounds.

*Proof of the middle shape bounds Lemma E.7.* Let $P_{\tau}, Q_{\tau}, \pi$ be from Proposition E.17. Let $U' = (U_{\tau})_2, V' = (V_{\tau})_2$ be the type 2 vertices in $U_{\tau}, V_{\tau}$ respectively. Let $x_i$ for $i \in U' \cup V'$ be random variables independently sampled from $\mathcal{D}$. Define $x_{U'}$ (resp. $x_{V'}$) to be the subset of variables $x_i$ for $i \in U'$ (resp. $i \in V'$).

For $\rho \in \mathcal{P}_U$, define $W_{\rho,\rho} = \mathbb{E}_{y_{U_2} \sim \mathcal{D}^{U_2}}[v_{\rho,y_{U_2}} v_{\rho,y_{U_2}}^T]$ so that $H_{Id_U,\rho,\rho} = \frac{1}{|Aut(U)|} W_{\rho,\rho}$. Observe that $W_{\rho,\rho} = \mathbb{E}[v_{\rho,x_{U'}} v_{\rho,x_{U'}}^T] = \mathbb{E}[v_{\rho,x_{V'}} v_{\rho,x_{V'}}^T]$ because $x_{U'}$ and $x_{V'}$ are also sets of variables sampled from $\mathcal{D}$ and, $U', V'$ have the same size as $U_2$ because $U_\tau = V_\tau = U$.

For $\rho, \rho' \in \mathcal{P}_U$, define $Y_{\rho,\rho'} = \mathbb{E}\left[ v_{\rho,x_{U'}} \left( \prod_{i \in U' \cup V'} x_i^{deg^\tau(i)} \right) v_{\pi(\rho),x_{V'}}^T \right]$. Then, $H_\tau = \sum_{\rho \in P_\tau} H_{\tau,\rho,\pi(\rho)} = \frac{1}{|Aut(U)|^2} S(\tau) \sum_{\rho \in P_\tau} Y_{\rho,\pi(\rho)}$. We have

$$\begin{bmatrix} \frac{S(\tau)R(\tau)}{|Aut(U)|} H_{Id_U} & H_\tau \\ H_\tau^T & \frac{S(\tau)R(\tau)}{|Aut(U)|} H_{Id_U} \end{bmatrix} = \frac{S(\tau)}{|Aut(U)|^2} \begin{bmatrix} R(\tau) \sum_{\rho \in \mathcal{P}_U} W_{\rho,\rho} & \sum_{\rho \in P_\tau} Y_{\rho,\pi(\rho)} \\ \sum_{\rho \in P_\tau} Y_{\rho,\pi(\rho)}^T & R(\tau) \sum_{\rho \in \mathcal{P}_U} W_{\rho,\rho} \end{bmatrix}$$

We hae $\frac{S(\tau)}{|Aut(U)|^2} \geq 0$ and the matrix is just

$$R(\tau) \begin{bmatrix} \sum_{\rho \in \mathcal{P}_U \backslash P_\tau} W_{\rho,\rho} & 0 \\ 0 & \sum_{\rho \in \mathcal{P}_U \backslash Q_\tau} W_{\rho,\rho} \end{bmatrix} + \begin{bmatrix} R(\tau) \sum_{\rho \in P_\tau} W_{\rho,\rho} & \sum_{\rho \in P_\tau} Y_{\rho,\pi(\rho)} \\ \sum_{\rho \in P_\tau} Y_{\rho,\pi(\rho)}^T & R(\tau) \sum_{\rho \in P_\tau} W_{\pi(\rho),\pi(\rho)} \end{bmatrix}$$

We have $\sum_{\rho \in \mathcal{P}_U \backslash P_\tau} W_{\rho,\rho} = \sum_{\rho \in \mathcal{P}_U \backslash P_\tau} \mathbb{E}[v_{\rho,x_U} v_{\rho,x_U}^T] \succeq 0$. Similarly, $\sum_{\rho \in \mathcal{P}_U \backslash Q_\tau} W_{\rho,\rho} \succeq 0$. Also, $R(\tau) \geq 0$ and so, the first term in the above expression is positive semidefinite. And the second term is just

$$\sum_{\rho \in P_\tau} \mathbb{E} \begin{bmatrix} R(\tau) v_{\rho,x_{U'}} v_{\rho,x_{U'}}^T & v_{\rho,x_{U'}} \left( \prod_{i \in U' \cup V'} x_i^{deg^\tau(i)} \right) v_{\pi(\rho),x_{V'}}^T \\ v_{\rho,x_{U'}}^T \left( \prod_{i \in U' \cup V'} x_i^{deg^\tau(i)} \right) v_{\pi(\rho),x_{V'}} & R(\tau) v_{\pi(\rho),x_{V'}} v_{\pi(\rho),x_{V'}}^T \end{bmatrix}$$

We will prove that the term inside the expectation is positive semidefinite for each $\rho \in P_\tau$ and each sampling of the $x_i$ from $\mathcal{D}$, which will complete the proof. Fix $\rho \in P_\tau$ and any sampling of the $x_i$ from $\mathcal{D}$. Let $w_1 = v_{\rho,X_{U'}}$, $w_2 = v_{\pi(\rho),x_{V'}}$. Let $E = \prod_{i \in U' \cup V'} x_i^{deg^\tau(i)}$. We would like to prove that $\begin{bmatrix} R(\tau) w_1 w_1^T & E w_1 w_2^T \\ E w_1^T w_2 & R(\tau) w_2 w_2^T \end{bmatrix} \succeq 0$. For all $y$ sampled from $\mathcal{D}$, $|y| \leq C_{disc}\sqrt{D_E}$ and so, $|E| \leq (C_{disc}\sqrt{D_E})^{\sum_{j \in U' \cup V'} deg^\tau(j)} = R(\tau)$.

If $E \geq 0$, then

$$\begin{bmatrix} R(\tau) w_1 w_1^T & E w_1 w_2^T \\ E w_1^T w_2 & R(\tau) w_2 w_2^T \end{bmatrix} = (R(\tau) - E) \begin{bmatrix} w_1 w_1^T & 0 \\ 0 & w_2 w_2^T \end{bmatrix} + E \begin{bmatrix} w_1 w_1^T & w_1 w_2^T \\ w_1^T w_2 & w_2 w_2^T \end{bmatrix} \succeq 0$$

since $R(\tau) - E \geq 0$ And if $E < 0$,

$$\begin{bmatrix} R(\tau) w_1 w_1^T & E w_1 w_2^T \\ E w_1^T w_2 & R(\tau) w_2 w_2^T \end{bmatrix} = (R(\tau) + E) \begin{bmatrix} w_1 w_1^T & 0 \\ 0 & w_2 w_2^T \end{bmatrix} - E \begin{bmatrix} w_1 w_1^T & -w_1 w_2^T \\ -w_1^T w_2 & w_2 w_2^T \end{bmatrix} \succeq 0$$

since $R(\tau) + E \geq 0$. $\blacksquare$

### E.2.3 Intersection term bounds

Just as in Proposition E.17, the next proposition captures the fact that when we compose shapes $\sigma, \gamma, \gamma^T, \sigma'^T$, in order for $\lambda_{\sigma \circ \gamma \circ \gamma^T \circ \sigma'^T}$ to be nonzero, the parities of the degrees of the merged vertices should add up correspondingly. Just as in the tensor PCA application, we similarly define $H_{Id_V,\rho,\rho'}^{-\gamma,\gamma}$ and $H'_{\gamma,\rho,\rho}$. The following propositions are simple and proved the same way.

**Proposition E.19.** *For all $U, V \in \mathcal{I}_{mid}$ where $w(U) > w(V)$, for all $\gamma \in \Gamma_{U,V}$, there exists a set of parity vectors $P_\gamma \subseteq \mathcal{P}_U$ such that $H_{Id_V}^{-\gamma,\gamma} = \sum_{\rho \in P_\gamma} H_{Id_V,\rho,\rho}^{-\gamma,\gamma}$.*

**Proposition E.20.** *For all $U, V \in \mathcal{I}_{mid}$ where $w(U) > w(V)$, for $\gamma \in \Gamma_{U,V}$, $H'_\gamma = \sum_{\rho \in P_\gamma} H'_{\gamma,\rho,\rho}$.*

We will now define vectors which are truncations of $v_{\rho,x_{U_2}}$. This definition and the following proposition are mostly a matter of technicality and they are essentially similar to the PSD mass condition analysis.

**Definition E.21.** *Let $U, V \in \mathcal{I}_{mid}$ where $w(U) > w(V)$, and let $\gamma \in \Gamma_{U,V}$. Let $x_i$ for $i \in U_2$ be variables. Denote them collectively as $x_{U_2}$. For $\rho \in \mathcal{P}_U$, define $v_{\rho,x_{U_2}}^{-\gamma}$ to be the vector indexed by left shapes $\sigma \in \mathcal{L}$ such that the $\sigma$th entry is $v_{\rho,x_{U_2}}(\sigma)$ if $|V(\sigma \circ \gamma)| \leq D_V$ and 0 otherwise.*

With this, we can decompose each slice $H_{Id_V,\rho,\rho}^{-\gamma,\gamma}$.

**Proposition E.22.** *For any $U,V \in \mathcal{I}_{mid}$ where $w(U) > w(V)$, and for any $\gamma \in \Gamma_{U,V}$, suppose we take $\rho \in P_\gamma$. When we compose $\gamma$ with $\gamma^T$ to get $\gamma \circ \gamma^T$, let $U' = (U_{\gamma \circ \gamma^T})_2, V' = (V_{\gamma \circ \gamma^T})_2$ be the type 2 vertices in $U_{\gamma \circ \gamma^T}, V_{\gamma \circ \gamma^T}$ respectively. And let $W'$ be the set of type 2 vertices in $\gamma \circ \gamma^T$ that were identified in the composition when we set $V_\gamma = U_\gamma^T$. Let $x_i$ for $i \in U' \cup W' \cup V'$ be random variables independently sampled from $\mathcal{D}$. Define $x_{U'}$ (resp. $x_{V'}, x_{W'}$) to be the subset of variables $x_i$ for $i \in U'$ (resp. $i \in V', i \in W'$). Then,*

$$H_{Id_V,\rho,\rho}^{-\gamma,\gamma} = \frac{1}{|Aut(V)|} S(\gamma)^2 \, \mathbb{E}_x \left[ (v_{\rho,x_{U'}}^{-\gamma}) \left( \prod_{i \in U' \cup W' \cup V'} x_i^{deg^{\gamma \circ \gamma^T}(i)} \right) (v_{\rho,x_{V'}}^{-\gamma})^T \right]$$

*Proof.* Fix $\sigma, \sigma' \in \mathcal{L}_{U,\rho}$ such that $|V(\sigma \circ \gamma)|, |V(\sigma' \circ \gamma)| \leq D_V$. Note that for $t \in \{1,2\}$, $|\sigma|_t - \frac{|V_\sigma|_t}{2} + |\sigma'|_t - \frac{|V_{\sigma'}|_t}{2} + 2(|\gamma|_t - \frac{|U_\gamma|_t + |V_\gamma|_t}{2}) = |\sigma \circ \gamma \circ \gamma^T \circ \sigma'^T|_t$. We can easily verify the equality using Definition E.1 and Definition E.13. ∎

**Proposition E.23.** *For any $U,V \in \mathcal{I}_{mid}$ where $w(U) > w(V)$, and for any $\gamma \in \Gamma_{U,V}$, suppose we take $\rho \in \mathcal{P}_U$. Then, $H'_{\gamma,\rho,\rho} = \frac{1}{|Aut(U)|} \mathbb{E}_{y_{U_2} \sim \mathcal{D}^{U_2}} \left[ (v_{\rho,y_{U_2}}^{-\gamma})(v_{\rho,y_{U_2}}^{-\gamma})^T \right]$.*

We can finally show the intersection term bounds.

*Proof of the intersection term bounds Lemma E.8.* Let $U', V', W'$ be as in Proposition E.22. We have

$$\frac{|Aut(V)|}{|Aut(U)|} \cdot \frac{1}{S(\gamma)^2 R(\gamma)^2} H_{Id_V}^{-\gamma,\gamma} = \sum_{\rho \in P_\gamma} \frac{|Aut(V)|}{|Aut(U)|} \cdot \frac{1}{S(\gamma)^2 R(\gamma)^2} H_{Id_V,\rho,\rho}^{-\gamma,\gamma}$$

$$= \sum_{\rho \in P_\gamma} \frac{1}{|Aut(U)|} \cdot \frac{1}{R(\gamma)^2} \mathbb{E}_x \left[ (v_{\rho,x_{U'}}^{-\gamma}) \left( \prod_{i \in U' \cup W' \cup V'} x_i^{deg^{\gamma \circ \gamma^T}(i)} \right) (v_{\rho,x_{V'}}^{-\gamma})^T \right]$$

We will now prove that, for all $\rho \in P_\gamma$,

$$\frac{1}{|Aut(U)|} \cdot \frac{1}{R(\gamma)^2} \mathbb{E}_x \left[ (v_{\rho,x_{U'}}^{-\gamma}) \left( \prod_{i \in U' \cup W' \cup V'} x_i^{deg^{\gamma \circ \gamma^T}(i)} \right) (v_{\rho,x_{V'}}^{-\gamma})^T \right] \preceq H'_{\gamma,\rho,\rho}$$

which reduces to proving that

$$\frac{2}{R(\gamma)^2} \mathbb{E}_x \left[ (v_{\rho,x_{U'}}^{-\gamma}) \left( \prod_{i \in U' \cup W' \cup V'} x_i^{deg^{\gamma \circ \gamma^T}(i)} \right) (v_{\rho,x_{V'}}^{-\gamma})^T \right] \preceq 2 \mathbb{E}_{y_{U_2} \sim \mathcal{D}^{U_2}} \left[ (v_{\rho,y_{U_2}}^{-\gamma})(v_{\rho,y_{U_2}}^{-\gamma})^T \right]$$

$$= \mathbb{E}_x \left[ (v_{\rho,x_{U'}}^{-\gamma})(v_{\rho,x_{U'}}^{-\gamma})^T + (v_{\rho,x_{V'}}^{-\gamma})(v_{\rho,x_{V'}}^{-\gamma})^T \right]$$

where the last equality followed from linearity of expectation and the fact that $U' \equiv V' \equiv U_2$. Since $H_{Id_V,\rho,\rho}^{-\gamma,\gamma}$ is symmetric, we have

$$\mathbb{E}_x \left[ (v_{\rho,x_{U'}}^{-\gamma}) \left( \prod_{i \in U' \cup W' \cup V'} x_i^{deg^{\gamma \circ \gamma^T}(i)} \right) (v_{\rho,x_{V'}}^{-\gamma})^T \right] = \mathbb{E}_x \left[ (v_{\rho,x_{V'}}^{-\gamma}) \left( \prod_{i \in U' \cup W' \cup V'} x_i^{deg^{\gamma \circ \gamma^T}(i)} \right) (v_{\rho,x_{U'}}^{-\gamma})^T \right]$$

So, it suffices to prove

$$\frac{1}{R(\gamma)^2} \mathbb{E}_x \left[ (v_{\rho,x_{U'}}^{-\gamma}) \left( \prod_{i \in U' \cup W' \cup V'} x_i^{deg^{\gamma \circ \gamma^T}(i)} \right) (v_{\rho,x_{V'}}^{-\gamma})^T + (v_{\rho,x_{V'}}^{-\gamma}) \left( \prod_{i \in U' \cup W' \cup V'} x_i^{deg^{\gamma \circ \gamma^T}(i)} \right) (v_{\rho,x_{U'}}^{-\gamma})^T \right]$$

$$\preceq \mathbb{E}_x \left[ (v_{\rho,x_{U'}}^{-\gamma})(v_{\rho,x_{U'}}^{-\gamma})^T + (v_{\rho,x_{V'}}^{-\gamma})(v_{\rho,x_{V'}}^{-\gamma})^T \right]$$

We will prove that for every sampling of the $x_i$ from $\mathcal{D}$, we have

$$\frac{1}{R(\gamma)^2}\left((v_{\rho,x_{U'}}^{-\gamma})\left(\prod_{i\in U'\cup W'\cup V'} x_i^{deg^{\gamma\circ\gamma^T}(i)}\right)(v_{\rho,x_{V'}}^{-\gamma})^T + (v_{\rho,x_{V'}}^{-\gamma})\left(\prod_{i\in U'\cup W'\cup V'} x_i^{deg^{\gamma\circ\gamma^T}(i)}\right)(v_{\rho,x_{U'}}^{-\gamma})^T\right)$$
$$\preceq (v_{\rho,x_{U'}}^{-\gamma})(v_{\rho,x_{U'}}^{-\gamma})^T + (v_{\rho,x_{V'}}^{-\gamma})(v_{\rho,x_{V'}}^{-\gamma})^T$$

Then, taking expectations will give the result. Indeed, fix a sampling of the $x_i$ from $\mathcal{D}$. Let $E = \prod_{i\in U'\cup W'\cup V'} x_i^{deg^{\gamma\circ\gamma^T}(i)}$ and let $w_1 = v_{\rho,x_{U'}}^{-\gamma}, w_2 = v_{\rho,x_{V'}}^{-\gamma}$. Then, the inequality we need to show is $\frac{E}{R(\gamma)^2}(w_1 w_2^T + w_2 w_1^T) \preceq w_1 w_1^T + w_2 w_2^T$. Now, since $|x_i| \leq C_{disc}\sqrt{D_E}$ for all $i$, we have $|E| \leq \prod_{i\in U'\cup W'\cup V'}(C_{disc}\sqrt{D_E})^{deg^{\gamma\circ\gamma^T}(i)} = R(\gamma)^2$. If $E \geq 0$, using $\frac{E}{R(\gamma)^2}(w_1 - w_2)(w_1 - w_2)^T \succeq 0$ gives

$$\frac{E}{R(\gamma)^2}(w_1 w_2^T + w_2 w_1^T) \preceq \frac{E}{R(\gamma)^2}(w_1 w_1^T + w_2 w_2^T) \preceq w_1 w_1^T + w_2 w_2^T$$

since $0 \leq E \leq R(\gamma)^2$. And if $E < 0$, using $\frac{-E}{R(\gamma)^2}(w_1 + w_2)(w_1 + w_2)^T \succeq 0$ gives

$$\frac{E}{R(\gamma)^2}(w_1 w_2^T + w_2 w_1^T) \preceq \frac{-E}{R(\gamma)^2}(w_1 w_1^T + w_2 w_2^T) \preceq w_1 w_1^T + w_2 w_2^T$$

since $0 \leq -E \leq R(\gamma)^2$. Finally, we use the fact that for all $\rho \in \mathcal{P}_U$, we have $H'_{\gamma,\rho,\rho} \succeq 0$ which can be proved the same way as the proof of Lemma E.3. Therefore,

$$\frac{|Aut(V)|}{|Aut(U)|} \cdot \frac{1}{S(\gamma)^2 R(\gamma)^2} H_{Id_V}^{-\gamma,\gamma} \preceq \sum_{\rho\in P_\gamma} H'_{\gamma,\rho,\rho} \preceq \sum_{\rho\in\mathcal{P}_U} H'_{\gamma,\rho,\rho} = H'_\gamma$$

■

## E.3 Intuition for quantitative bounds

In this section, we will give some intuition for the bounds that appear in our main theorem Theorem A.1, which is formally proved in Appendix K. Informally, the theorem states that when $m \leq \frac{d}{\lambda^2}$ and $m \leq \frac{k^2}{\lambda^2}$, then $\Lambda \succeq 0$ with high probability.

We will try and understand why the inequality $\lambda_{\sigma\circ\tau\circ\sigma'^T}^2 \|M_\tau\|^2 \leq \lambda_{\sigma\circ\sigma^T}\lambda_{\sigma'\circ\sigma'^T}$ holds. Assume for simplicity that $d < n$ and consider the shapes in Fig. 3. The assumption $d < n$ is used in this example since otherwise, if $d > n$, the decomposition differs from what's shown in the figure.

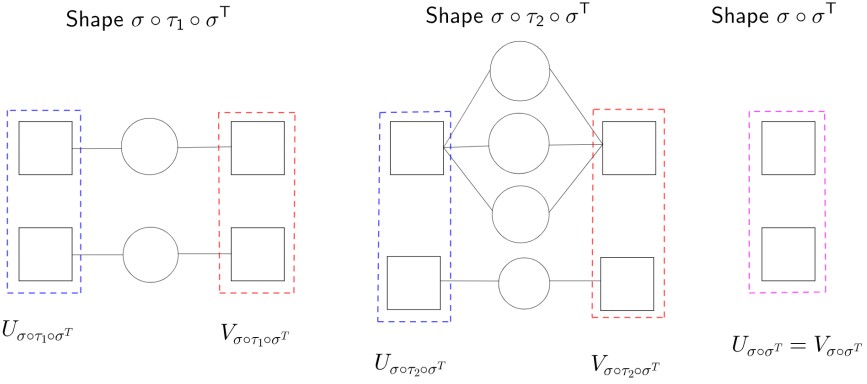

Figure 3: Shapes $\sigma \circ \tau_1 \circ \sigma^T, \sigma \circ \tau_2 \circ \sigma^T$ and $\sigma \circ \sigma^T$.

Firstly, the shape $\sigma \circ \sigma^T$ has a coefficient of $\lambda_{\sigma\circ\sigma^T} \approx \left(\frac{1}{\sqrt{k}}\right)^4 \left(\frac{k}{d}\right)^2$. The first shape $\sigma \circ \tau_1 \circ \sigma^T$ has a coefficient of $\lambda_{\sigma\circ\tau_1\circ\sigma^T} \approx \left(\frac{1}{\sqrt{k}}\right)^4 \left(\frac{k}{d}\right)^4 \left(\frac{\sqrt{\lambda}}{\sqrt{k}}\right)^4$ and with high probability, upto lower order

terms, $\|M_{\tau_1}\| \le md$ (these norm bounds follow from [1]). So, the inequality $\lambda_{\sigma\circ\tau_1\circ\sigma^T}^2 \|M_{\tau_1}\|^2 \le \lambda_{\sigma\circ\sigma^T}\lambda_{\sigma\circ\sigma^T}$ rearranges to $m \le \frac{d}{\lambda^2}$. But this is precisely one of the assumptions on $m$. Moreover, this also confirms that we need this assumption on $m$ in order for our strategy to go through.

The second shape $\sigma\circ\tau_2\circ\sigma^T$ has a coefficient of $\lambda_{\sigma\circ\tau_2\circ\sigma^T} \approx \left(\frac{1}{\sqrt{k}}\right)^4 \left(\frac{k}{d}\right)^4 \left(\frac{\sqrt{\lambda}}{\sqrt{k}}\right)^8$ and with high probability, upto lower order terms, $\|M_{\tau_2}\| \le m^2 d$. So, the inequality $\lambda_{\sigma\circ\tau_2\circ\sigma^T}^2 \|M_{\tau_2}\|^2 \le \lambda_{\sigma\circ\sigma^T}\lambda_{\sigma\circ\sigma^T}$ rearranges to $m^2 \le \frac{k^2 d}{\lambda^4}$. But this is obtained simply by multiplying our assumptions on $m$, namely $m \le \frac{k^2}{\lambda^2}$ and $m \le \frac{d}{\lambda^2}$.

Moreover, consider a shape of the form $\sigma\circ\tau_3\circ\sigma^T$ where $\tau_3$ is similar to $\tau_2$ except it has $t$ (instead of 3) different circle vertices that are common neighbors to the top 2 square vertices. Analyzing our required inequality, we get for our strategy to go through, $m$ has to satisfy $m \le \frac{k^2}{\lambda^2} \cdot \left(\frac{d}{k^2}\right)^{\frac{2}{t+1}}$. By taking $t$ arbitrarily large, we can see that the condition $m \le \frac{k^2}{\lambda^2}$ is needed.

So, we get that for our analysis to go through, the assumptions $m \le \frac{d}{\lambda^2}$ and $m \le \frac{k^2}{\lambda^2}$ are necessary. We will prove that in fact, these are sufficient. To do this, we use an argument that exploits the special structure of the shapes $\alpha$ that appear in our decomposition of $\Lambda$ and their coefficients $\lambda_\alpha$, as we obtained in Definition E.1. For details, see Appendix K.

# F   Definitions and Quantitative Main Theorem Statement

## F.1   Section Introduction

In this section, we make our definitions and results more precise. We also generalize our definitions and results to handle problems where one or more of the following is true:

1. The input entries correspond to hyperedges rather than edges.
2. We have different types of indices.
3. $\Omega$ is a more complicated distribution than $\{-1, +1\}$.
4. We have to consider matrix indices which are not multilinear.

Throughout this section and the remainder of this manuscript, we give the reader a choice for the level of generality. In particular, we will first recall our definition for the simpler case when our input is $\{-1, +1\}^{\binom{n}{2}}$ and we only consider multilinear indices. We will then discuss how this simpler definition generalizes. We denote these generalizations with an asterix $*$.

### F.1.1   Additional Parameters for the General Case*

In the general case we will need a few additional parameters which we define here.
**Definition F.1.**

1. *We define $k$ to be the arity of the hyperedges corresponding to the input.*
2. *We define $t_{max}$ to be the number of different types of indices. We define $n_i$ to be the number of possibilities for indicies of type $i$ and we define $n = \max\{n_i : i \in [t_{max}]\}$.*

## F.2   Indices, Input Entries, Vertices, and Edges

Note: For this section, we use $X$ to denote the input, we use $x$ to denote entries of the input and we use $y$ to denote solution variables.
**Definition F.2** (Vertices: Simplified Case). *When the input and solution variables are indexed by one type of index which takes values in $[n]$ then we represent the index $i$ by a vertex labeled $i$.*

*If we want to leave an index unspecified, we instead represent it by a vertex labeled with a variable (we will generally use $u$, $v$, or $w$ for these variables).*
**Definition F.3** (Vertices: General Case*). *When the input and solution variables are indexed by several types of indices where indices of type $t$ take values in $[n_t]$, we represent an index of type $t$ with value $i$ as a vertex labeled by the tuple $(t, i)$. We say that such a vertex has type $t$.*

*If we want to leave an index of type $t$ unspecified, we instead represent it by a vertex labeled with a tuple $(t, ?)$ where $?$ is a variable (which will generally be $u$, $v$, or $w$).*

**Definition F.4** (Edges: Simplified Case). *When the input is $X \in \{-1, +1\}^{\binom{n}{2}}$, we represent the entries of the input by the undirected edges $\{(i, j) : i < j \in [n]\}$. Given an edge $e = (i, j)$, we take $x_e = x_{ij}$ to be the input entry corresponding to $e$.*

**Definition F.5** (Edges: General Case*). *In general, we represent the entries of the input by hyperedges whose form depends on nature of the input. We still take $x_e$ to be the input entry corresponding to $e$.*

**Example F.6.** *If the input is an $n_1 \times n_2$ matrix $X$ then we will have two types of indices, one for the row and one for the column. Thus, we will have the vertices $\{(1, i) : i \in [n_1]\} \cup \{(2, j) : j \in [n_2]\}$. In this case, we have an edge $((1, i), (2, j))$ for each entry $x_{ij}$ of the input.*

**Example F.7.** *If the input is an $n \times n$ matrix $X$ which is not symmetric then we only need the indices $[n]$. In this case, we have a directed edge $(i, j)$ for each entry $x_{ij}$ where $i \neq j$. If the entries $x_{ii}$ are also part of the input than we also have loops $(i, i)$ for these entries.*

**Example F.8.** *If our input is a symmetric $n \times n \times n$ tensor $X$ (i.e. $x_{ijk} = x_{ikj} = x_{jik} = x_{jki} = x_{kij} = x_{kji}$) and $x_{ijk} = 0$ whenever $i, j, k$ are not distinct then we only need the indices $[n]$. In this case, we have an undirected hyperedge $e = (i, j, k)$ for each entry $x_e = x_{ijk}$ of the input where $i, j, k$ are distinct.*

**Example F.9.** *If the input is an $n_1 \times n_2 \times n_3$ tensor $X$ then we will have three types of indices. Thus, we will have the vertices $\{(1, i) : i \in [n_1]\} \cup \{(2, j) : j \in [n_2]\} \cup \{(3, k) : k \in [n_3]\}$. In this case, we have a hyperedge $e = ((1, i), (2, j), (3, k))$ for each entry $x_e = x_{ijk}$ of the input.*

### F.3 Matrix Indices and Monomials

In this subsection, we discuss how our matrices are indexed and how we associate matrix indices with monomials. We also describe the automorphism groups of matrix indices.

**Definition F.10** (Matrix Indices: Simplified Case). *If there is only one type of index and we have the constraints $y_i^2 = 1$ or $y_i^2 = y_i$ on the solution variables then we define a matrix index $A$ to be a tuple of indices $(a_1, \ldots, a_{|A|})$. We make the following definitions about matrix indices:*

1. *We associate the monomial $\prod_{j=1}^{|A|} y_{a_j}$ to $A$.*

2. *We define $V(A)$ to be the set of vertices $\{a_i : i \in [|A|]\}$. For brevity, we will often write $A$ instead of $V(A)$ when it is clear from context that we are referring to $A$ as a set of vertices rather than a matrix index.*

3. *We take the automorphism group of $A$ to be $Aut(A) = S_{|A|}$ (the permutations of the elements of $A$)*

**Example F.11.** *The matrix index $A = (4, 6, 1)$ represents the monomial $y_4 y_6 y_1 = y_1 y_4 y_6$ and $Aut(A) = S_3$*

**Remark F.12.** *We take $A$ to be an ordered tuple rather than a set for technical reasons.*

In general, we need a more intricate definition for matrix indices. We start by defining matrix index pieces

**Definition F.13** (Matrix Index Piece Definition*). *We define a matrix index piece $A_i = ((a_{i1}, \ldots, a_{i|A_i|}), t_i, p_i)$ to be a tuple of indices $(a_{i1}, \ldots, a_{i|A_i|})$ together with a type $t_i$ and a power $p_i$. We make the following definitions about matrix index pieces:*

1. *We associate the monomial $p_{A_i} = \prod_{j=1}^{|A_i|} y_{t_i j}^{p_i}$ with $A_i$.*

2. *We define $V(A_i)$ to be the set of vertices $\{(t_i, a_{ij}) : j \in [|A_i|]\}$.*

3. *We take the automorphism group of $A_i$ to be $Aut(A_i) = S_{|A_i|}$*

4. *We say that $A_i$ and $A_j$ are disjoint if $V(A_i) \cap V(A_j) = \emptyset$ (i.e. $t_i \neq t_j$ or $\{a_{i1}, \ldots, a_{i|A_i|}\} \cap \{a_{j1}, \ldots, a_{j|A_j|}\} = \emptyset$)*

**Definition F.14** (General Matrix Index Definition*). *We define a matrix index $A = \{A_i\}$ to be a set of disjoint matrix index pieces. We make the following definitions about matrix indices:*

1. We associate the monomial $p_A = \prod_{A_i \in A} p(A_i)$ with $A$.

2. We define $V(A)$ to be the set of vertices $\cup_{A_i \in A} V(A_i)$. For brevity, we will often write $A$ instead of $V(A)$ when it is clear from context that we are referring to $A$ as a set of vertices rather than a matrix index.

3. We take the automorphism group of $A$ to be $Aut(A) = \prod_{A_i \in A} Aut(A_i)$

**Example F.15** (*). If $A_1 = ((2), 1, 1)$, $A_2 = ((3,1), 1, 2)$, and $A_3 = ((1,2,3), 2, 1)$ then $A = \{A_1, A_2, A_3\}$ represesents the monomial $p = y_{12} y_{13}^2 y_{11}^2 y_{21} y_{22} y_{23}$ and we have $Aut(A) = S_1 \times S_2 \times S_3$

## F.4 Fourier Characters and Ribbons

A key idea is to analyze Fourier characters of the input.

**Definition F.16** (Simplified Fourier Characters). *If the input distribution is $\Omega = \{-1, 1\}$ then given a multi-set of edges $E$, we define $\chi_E(X) = \prod_{e \in E} x_e$.*

**Example F.17.** *If the input is a graph $G \in \{-1, 1\}^{\binom{n}{2}}$ and $E$ is a set of potential edges of $G$ (with no multiple edges) then $\chi_E(G) = (-1)^{|E \setminus E(G)|}$.*

In general, the Fourier characters are somewhat more complicated.

**Definition F.18** (Orthonormal Basis for $\Omega$*). *We define the polynomials $\{h_i : i \in \mathbb{Z} \cap [0, |supp(\Omega)| - 1]\}$ to be the unique polynomials (which can be found through the Gram-Schmidt process) such that*

1. $\forall i, E_\Omega[h_i^2(x)] = 1$

2. $\forall i \neq j, E_\Omega[h_i(x) h_j(x)] = 0$

3. *For all $i$, the leading coefficient of $h_i(x)$ is positive.*

**Example F.19.** *If $\Omega$ is the normal distribution then the polynomials $\{h_i\}$ are the Hermite polynomials with the appropriate normalization so that for all $i$, $E_\Omega[h_i^2(x)] = 1$. In particular, $h_0(x) = 1$, $h_1(x) = x$, $h_2(x) = \frac{x^2 - 1}{\sqrt{2!}}$, $h_3(x) = \frac{x^3 - 3x}{\sqrt{3!}}$, etc.*

**Definition F.20** (General Fourier Characters*). *Given a multi-set of hyperedges $E$, each of which has a label $l(e) \in [|support(\Omega)| - 1]$ (or $\mathbb{N}$ if $\Omega$ has infinite support), we define $\chi_E = \prod_{e \in E} h_{l(e)}(X_e)$.*

*We say that such a multi-set of hyperedges $E$ is proper if it contains no duplicate hyperedges, i.e. it is a set (though the labels on the hyperedges can be arbitrary non-negative integers). Otherwise, we say that $E$ is improper.*

**Remark F.21.** *The Fourier characters are $\{\chi_E : E \text{ is proper}\}$. For improper $E$, $\chi_E$ can be decomposed as a linear combination of $\chi_{E_j}$ where each $E_j$ is proper. We allow improper $E$ because it is sometimes more convenient to have improper $E$ in the middle of the analysis and then do this decomposition at the end.*

**Definition F.22** (Ribbons). *A ribbon $R$ is a tuple $(H_R, A_R, B_R)$ where $H_R$ is a multi-graph (*or multi-hypergraph with labeled edges in the general case) whose vertices are indices of the input and $A_R$ and $B_R$ are matrix indices such that $V(A_R) \subseteq V(H_R)$ and $V(B_R) \subseteq V(H_R)$. We make the following definitions about ribbons:*

1. *We define $V(R) = V(H_R)$ and $E(R) = E(H_R)$*

2. *We define $\chi_R = \chi_{E(R)}$.*

3. *We define $M_R$ to be the matrix such that $(M_R)_{A_R B_R} = \chi_R$ and $M_{AB} = 0$ whenever $A \neq A_R$ or $B \neq B_R$.*

*We say that $R$ is a proper ribbon if $H_R$ contains no isolated vertices outside of $A_R \cup B_R$ and $E(R)$ is proper. If there is an isolated vertex in $(V(R) \setminus A_R) \setminus B_R$ or $E(R)$ is improper then we say that $R$ is an improper ribbon.*

Proper ribbons are useful because they give an orthonormal basis for the space of matrix valued functions.

**Definition F.23** (Inner products of matrix functions). *For a pair of real matrices $M_1, M_2$ of the same dimension, we write $\langle M_1, M_2 \rangle = tr(M_1 M_2^T)$ (i.e. $\langle M_1, M_2 \rangle$ is the entrywise dot product of $M_1$ and $M_2$). For a pair of matrix-valued functions $M_1, M_2$ (of the same dimensions), we define*

$$\langle M_1, M_2 \rangle = E_X \left[ \langle M_1(X), M_2(X) \rangle \right]$$

**Proposition F.24.** *If $R$ and $R'$ are two proper ribbons then $\langle M_R, M_{R'} \rangle = 1$ if $R = R'$ and is $0$ otherwise.*

## F.5 Shapes

In this subsection, we describe a basis for $S$-invariant matrix valued functions where each matrix in this basis can be described by a relatively small *shape* $\alpha$. The fundamental idea behind shapes is that we keep the structure of the objects we are working with but leave the elements of the object unspecified.

### F.5.1 Simplified Index Shapes

**Definition F.25** (Simplified Index shapes). *With our simplifying assumptions, an index shape $U$ is a tuple of unspecified indices $(u_1, \cdots, u_{|U|})$. We make the following definitions about index shapes:*

1. *We define $V(U)$ to be the set of vertices $\{u_i : i \in [|U|]\}$. For brevity, we will often write $U$ instead of $V(U)$ when it is clear from context that we are referring to $U$ as a set of vertices rather than an index shape.*

2. *We define the weight of $U$ to be $w(U) = |U|$.*

3. *We take the automorphism group of $U$ to be $Aut(U) = S_{|U|}$ (the permutations of the elements of $U$)*

**Definition F.26.** *We say that a matrix index $A = (a_1, \ldots, a_{|A|})$ has index shape $U = (u_1, \ldots, u_{|U|})$ if $|U| = |A|$. Note that in this case, if we take the map $\varphi : \{u_j : j \in [|U|]\} \to [n]$ where $\varphi(u_j) = a_j$ then $\varphi(U) = (\varphi(u_1), \ldots, \varphi(u_{|U|})) = (a_1, \ldots, a_{|A|}) = A$*

**Definition F.27.** *We say that index shapes $U = (u_1, \ldots, u_{|U|})$ and $V = (v_1, \ldots, v_{|V|})$ are equivalent (which we write as $U \equiv V$) if $|U| = |V|$. If $U \equiv V$ then we can set $U = V$ by setting $v_j = u_j$ for all $j \in [|U|]$.*

**Example F.28.** *The matrix index $A = \{4, 6, 1\}$ has shape $U = \{u_1, u_2, u_3\}$ which has weight $3$.*

### F.5.2 General Index Shapes*

In general, we define general index shapes in the same way that we defined general matrix indices (just with unspecified indices)

**Definition F.29** (Index Shape Piece Definition). *We define a index shape piece $U_i = ((u_{i1}, \ldots, u_{i|U_i|}), t_i, p_i)$ to be a tuple of indices $(u_{i1}, \ldots, u_{i|A_i|})$ together with a type $t_i$ and a power $p_i$. We make the following definitions about index shape pieces:*

1. *We define $V(U_i)$ to be the set of vertices $\{(t_i, u_{ij}) : j \in [|U_i|]\}$.*

2. *We define $w(U_i) = |U_i| log_n(n_{t_i})$*

3. *We take the automorphism group of $U_i$ to be $Aut(U_i) = S_{|U_i|}$*

**Definition F.30** (General Index Shape Definition). *We define an index shape $U = \{U_i\}$ to be a set of index shape pieces such that for all $i' \neq i$, either $t_{i'} \neq t_i$ or $p_{i'} \neq p_i$. We make the following definitions about index shapes:*

1. *We define $V(U)$ to be the set of vertices $\cup_{U_i \in U} V(U_i)$. For brevity, we will often write $U$ instead of $V(U)$ when it is clear from context that we are referring to $U$ as a set of vertices rather than an index shape.*

2. *We define $w(U)$ to be $w(U) = \sum_{U_i \in U} w(U_i)$*

3. *We take the automorphism group of $U$ to be $Aut(U) = \prod_{U_i \in U} Aut(U_i)$*

**Remark F.31.** *For technical reasons, we want to ensure that if two index shapes $U$ and $U'$ have the same weight then $U$ and $U'$ have the same number of each type of vertex. To ensure this, we add an infinitesimal perturbation to each $n_i$ if necessary.*

**Definition F.32.** *We say that a matrix index $A$ has index shape $U$ if there is an assignment of values to the unspecified indices of $U$ which results in $A$. More precisely, we say that $A$ has index shape $U$ if there is a map $\varphi : \{u_{ij}\} \to \mathbb{N}$ such that if we define $\varphi(U_i)$ to be $\varphi(U_i) = ((\varphi(u_{i1}), \ldots, \varphi(u_{i|U_i|})), t_i, p_i)$ then $\varphi(U) = \{\varphi(U_i)\} = \{A_i\} = A$.*

**Definition F.33.** *If $U$ and $V$ are two index shapes, we say that $U$ is equivalent to $V$ (which we write as $U \equiv V$) if $U$ and $V$ have the same number of index shape pieces and we can order the index shape pieces of $U$ and $V$ so that writing $U = \{U_i\}$ and $V = \{V_i\}$ where $U_i = ((u_{i1}, \ldots, u_{i|U_i|}), t_i, p_i)$ and $V_i = ((v_{i1}, \ldots, v_{i|V_i|}), t_i', p_i')$, we have that for all $i$, $|V_i| = |U_i|$, $t_i' = t_i$, and $p_i' = p_i$. If $U \equiv V$ then we can set $U = V$ by setting $u_{ij} = v_{ij}$ for all $i$ and all $j \in [|U_i|]$.*

### F.5.3  Ribbon Shapes

With these definitions, we are now ready to define shapes and the matrices associated to them.

**Definition F.34** (Shapes). *A ribbon shape $\alpha$ (which we call a shape for brevity) is a tuple $\alpha = (H_\alpha, U_\alpha, V_\alpha)$ where $H_\alpha$ is a multi-graph (\*or multi-hypergraph with labeled edges in the general case) whose vertices are unspecified distinct indices of the input (\*whose type is specified in the general case) and $U_\alpha$ and $V_\alpha$ are index shapes such that $V(U_\alpha) \subseteq V(H_\alpha)$ and $V(V_\alpha) \subseteq V(H_\alpha)$. We make the following definitions about shapes:*

1. *We define $V(\alpha) = V(H_\alpha)$ (note that $V(\alpha)$ and $V_\alpha$ are not the same thing) and we define $E(\alpha) = E(H_\alpha)$.*

2. *We say that a shape $\alpha$ is proper if it contains no isolated vertices outside of $V(U_\alpha) \cup V(V_\alpha)$, $E(\alpha)$ has no multiple edges/hyperedges and edges in $E(\alpha)$ do not have label $0$. If there is an isolated vertex in $V(\alpha) \setminus V(U_\alpha) \setminus V(V_\alpha)$ or $E(\alpha)$ has a multiple edge/hyperedge then we say that $\alpha$ is an improper shape.*

*Note: For brevity, we will often write $U_\alpha$ and $V_\alpha$ instead of $V(U_\alpha)$ and $V(V_\alpha)$ when it is clear from context that we are referring to $U_\alpha$ and $V_\alpha$ as sets of vertices rather than index shapes.*

**Definition F.35** (Trivial shapes). *We say that a shape $\alpha$ is trivial if $V(\alpha) = V(U_\alpha) = V(V_\alpha)$ and $E(\alpha) = \emptyset$. Otherwise, we say that $\alpha$ is non-trivial.*

**Remark F.36.** *Note that all trivial shapes can do is permute the order of the vertices in $V(U_\alpha) = V(V_\alpha)$.*

**Definition F.37.** *Informally, we say that a ribbon $R$ has shape $\alpha$ if replacing the indices in $R$ with unspecified labels results in $\alpha$. Formally, we say that $R$ has shape $\alpha$ if there is an injective mapping $\varphi : V(\alpha) \to [n]$ (\*or $[t_{max}] \times [n]$ in the general case) such that $\varphi(\alpha) = R$, i.e. $\varphi(H_\alpha) = H_R$, $\varphi(U_\alpha) = A_R$, and $\varphi(V_\alpha) = B_R$*

**Definition F.38.** *We say that two shapes $\alpha$ and $\beta$ are equivalent (which we write as $\alpha \equiv \beta$) if they are the same up to renaming their indices. More precisely, we say that $\alpha \equiv \beta$ if there is a bijective map $\pi : V(H_\alpha) \to V(H_\beta)$ such that $\pi(H_\alpha) = H_\beta$, $\pi(U_\alpha) = U_\beta$, and $\pi(V_\alpha) = V_\beta$.*

**Definition F.39.** *Given a shape $\alpha$ and matrix indices $A, B$ of shapes $U_\alpha$ and $V_\alpha$ respectively, we define $\mathcal{R}(\alpha, A, B)$ to be the set of ribbons $R$ such that $R$ has shape $\alpha$, $A_R = A$, and $B_R = B$.*

**Definition F.40.** *For a shape $\alpha$, we define the matrix-valued function $M_\alpha$ to have entries $M_\alpha(A, B)$ given by*

$$(M_\alpha)_{A,B}(X) = \sum_{R \in \mathcal{R}(\alpha, A, B)} \chi_R(X)$$

For examples of $M_\alpha$, see [1].

**Proposition F.41.** *The $M_\alpha$'s for proper shapes $\alpha$ are an orthogonal basis for the $S$-invariant functions.*[†]

---

[†]Because of orthogonality of the underlying Fourier characters, it is not hard to check that when $\alpha \neq \alpha'$ and $M_\alpha, M_{\alpha'}$ have the same dimensions, $\langle M_\alpha, M_{\alpha'} \rangle = 0$.

**Remark F.42.** *Conceptually, one may think of forming an orthonormal basis for this space with the functions $M_\alpha / \sqrt{\langle M_\alpha, M_\alpha \rangle}$, but for technical reasons it is easiest to work with these functions without normalizing them to $1$. By orthogonality and the fact that every Boolean function is a polynomial, any S-invariant matrix-valued function $\Lambda$ is expressible as*

$$\Lambda = \sum_\alpha \frac{\langle \Lambda, M_\alpha \rangle}{\langle M_\alpha, M_\alpha \rangle} \cdot M_\alpha$$

In the proof of our main theorem, we encounter improper shapes. We can handle them by decomposing them into proper shapes using basic Fourier analysis. For now, we will illustrate how this can be done via an example.

### F.6  Composing Ribbons and Shapes

**Definition F.43** (Composing Ribbons). *We say that ribbons $R_1$ and $R_2$ are composable if $B_{R_1} = A_{R_2}$. Note that this definition is not symmetric so we may have that $R_1$ and $R_2$ are composable but $R_2$ and $R_1$ are not composable.*

*We say that $R_1$ and $R_2$ are properly composable if we also have that $V(R_1) \cap V(R_2) = V(B_{R_1}) = V(A_{R_2})$ (there are no unexpected intersections between $R_1$ and $R_2$).*

*If $R_1$ and $R_2$ are composable ribbons then we define the composition of $R_1$ and $R_2$ to be the ribbon $R_1 \circ R_2$ such that*

1. *$A_{R_1 \circ R_2} = A_{R_1}$ and $B_{R_1 \circ R_2} = B_{R_2}$*

2. *$V(R_1 \circ R_2) = V(R_1) \cup V(R_2)$*

3. *$E(R_1 \circ R_2) = E(R_1) \cup E(R_2)$ (and thus $\chi_{R_1 \circ R_2} = \chi_{R_1} \chi_{R_2}$)*

*We say that ribbons $R_1, \ldots, R_k$ are composable/properly composable if for all $j \in [k-1]$, $R_1 \circ \ldots \circ R_j$ and $R_{j+1}$ are composable/properly composable. If $R_1, \ldots, R_k$ are composable then we define $R_1 \circ \ldots \circ R_k$ to be $R_1 \circ \ldots \circ R_k = (R_1 \circ \ldots \circ R_{k-1}) \circ R_k$*

**Proposition F.44.** *Ribbon composition is associative, i.e. if $R_1, R_2, R_3$ are composable/properly composable ribbons then $R_2, R_3$ are composable/properly composable, $R_1, (R_2 \circ R_3)$ are composable/properly composable, and $R_1 \circ (R_2 \circ R_3) = (R_1 \circ R_2) \circ R_3$*

**Proposition F.45.** *If $R_1$ and $R_2$ are composable ribbons then $M_{R_1 \cup R_2} = M_{R_1} M_{R_2}$.*

We have similar definitions for composing shapes.

**Definition F.46** (Composing Shapes). *We say that shapes $\alpha$ and $\beta$ are composable if $U_\beta \equiv V_\alpha$. Note that this definition is not symmetric so we may have that $\alpha$ and $\beta$ are composable but $\beta$ and $\alpha$ are not composable.*

*If $\alpha$ and $\beta$ are composable shapes then we define the composition of $\alpha$ and $\beta$ to be the shape $\alpha \circ \beta$ such that*

1. *$U_{\alpha \circ \beta} = U_\alpha$ and $V_{\alpha \circ \beta} = V_\beta$*

2. *After setting $U_\beta = V_\alpha$, we take $V(\alpha \circ \beta) = V(\alpha) \cup V(\beta)$*

3. *$E(\alpha \circ \beta) = E(\alpha) \cup E(\beta)$*

*We say that shapes $\alpha_1, \ldots, \alpha_k$ are composable if for all $j \in [k-1]$, $\alpha_1 \circ \ldots \circ \alpha_j$ and $\alpha_{j+1}$ are composable. If $\alpha_1, \ldots, \alpha_k$ are composable then we define the shape $\alpha_1 \circ \ldots \circ \alpha_k$ to be $\alpha_1 \circ \ldots \circ \alpha_k = (\alpha_1 \circ \ldots \circ \alpha_{k-1}) \circ \alpha_k$*

**Proposition F.47.** *Shape composition is associative, i.e. if $\alpha_1, \alpha_2, \alpha_3$ are composable shapes then $\alpha_2, \alpha_3$ are composable, $\alpha_1, (\alpha_2 \circ \alpha_3)$ are composable, and $\alpha_1 \circ (\alpha_2 \circ \alpha_3) = (\alpha_1 \circ \alpha_2) \circ \alpha_3$*

**Example F.48.** *Fig. 4 illustrates an example of shape composition. We have two types of vertices that we diagrammaticaly represent by squares and circles. Observe how the shapes $\sigma \circ \sigma'^T$ and $\sigma \circ \tau \circ \sigma'^T$ are obtained from the shapes $\sigma, \tau$ and $\sigma'^T$.*

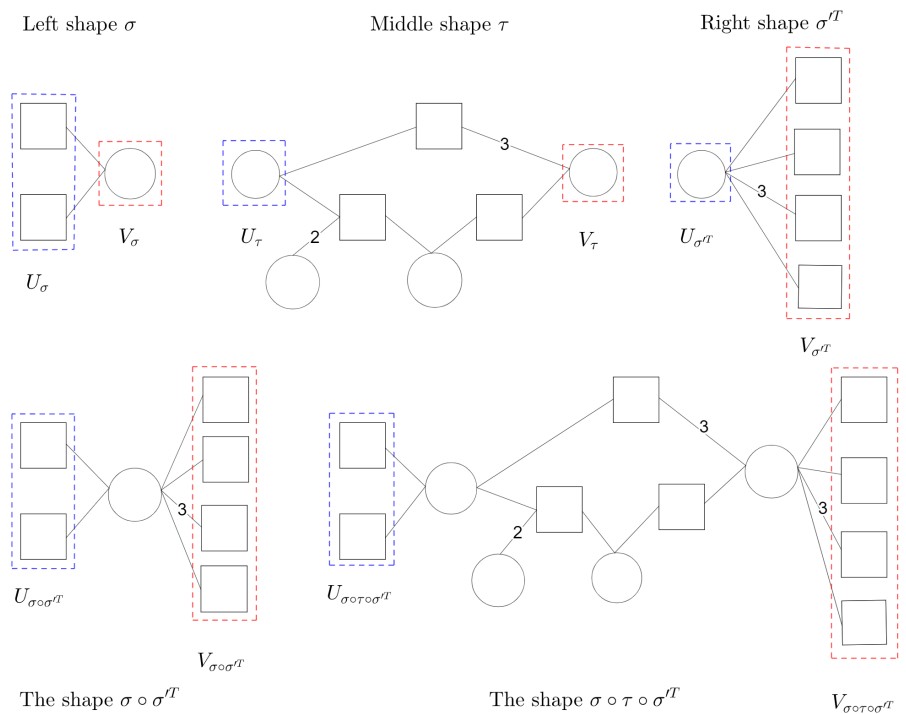

Figure 4: Illustration of shape composition and decomposition.

## F.7 Decomposition of Shapes into Left, Middle, and Right parts

In this subsection, we describe how shapes can be decomposed into left, middle, and right parts based on the leftmost and rightmost *minimum vertex separators*, which is a crucial idea for our analysis.

**Definition F.49** (Paths). *A path in a shape $\alpha$ is a sequence of vertices $v_1, \ldots, v_t$ such that $v_i, v_{i+1}$ are in some edge/hyperedge together. A pair of paths is vertex-disjoint if the corresponding sequences of vertices are disjoint.*

**Definition F.50** (Vertex separators). *Let $\alpha$ be a shape and let $U$ and $V$ be sets of vertices in $\alpha$. We say that a set of vertices $S \subseteq V(\alpha)$ is a* vertex separator *of $U$ and $V$ if every path in $\alpha$ from $U$ to $V$ contains at least one vertex in $S$. Note that any vertex separator $S$ of $U$ and $V$ must contain all of the vertices in $U \cap V$.*

*As a special case, we say that $S$ is a vertex separator of $\alpha$ if $S$ is a vertex separator of $U_\alpha$ and $V_\alpha$*

We define the weight of a set of vertices $S \subseteq V(\alpha)$ in the same way that weight is defined for index shapes.

**Definition F.51** (Simplified Weight). *When there is only one type of index, the weight of a set of vertices $S \subseteq V(\alpha)$ is simply $|S|$.*

**Definition F.52** (General Weight*). *In general, given a set of vertices $S \subseteq V(\alpha)$, writing $S = \cup_t S_t$ where $S_t$ is the set of vertices of type $t$ in $S$, we define the weight of $S$ to be $w(S) = \sum_t |S_t| log_n(n_t)$*

**Remark F.53** (*). *Again, if necessary, we add an infinitesimal perturbation to $n_1, n_2, \ldots, n_{t_{max}}$ so that if two separators $S$ and $S'$ have the same weight then $S$ and $S'$ have the same number of each type of vertex.*

**Definition F.54** (Leftmost and rightmost minimum vertex separators). *The* leftmost *minimum vertex separator is the vertex separator $S$ of minimum weight such that for every other minimum-weight vertex separator $S'$, $S$ is a separator of $U_\alpha$ and $S'$. The* rightmost *minimum vertex separator is the vertex separator $T$ of minimum weight such that for every other minimum-weight vertex separator $T'$, $T$ is a separator of $T'$ and $V_\alpha$*

We now have the following crucial idea. Every shape $\alpha$ can be decomposed into the composition of three composable shapes $\sigma, \tau, \sigma'^T$ based on the leftmost and rightmost minimum vertex separators $S, T$ of $\alpha$ together with orderings of $S$ and $T$.

**Definition F.55** (Simplified Separators With Orderings). *Under our simplifying assumptions, given a set of vertices $S \subseteq V(\alpha)$ and an ordering $O_S = s_1, \ldots, s_{|S|}$ of the vertices of $S$, we define the index shape $(S, O_S)$ to be $(S, O_S) = (s_1, \ldots, s_{|S|})$.*

**Definition F.56** (General Separators With Orderings*). *In the general case, we need to give an ordering for each type of vertex. Let $S \subseteq V(\alpha)$ be a subset of the vertices of $\alpha$ and write $S = \cup_t S_t$ where $S_t$ is the set of vertices in $S$ of type $t$. Given $O_S = \{O_t\}$ where $O_t = s_{t1}, \ldots, s_{t|S_t|}$ is an ordering of the vertices of $S_t$, we define the index shape piece $(S_t, O_t)$ to be $(S_t, O_t) = ((s_{t1}, \ldots, s_{t|S_t|}), t, 1)$ and we define the index shape $(S, O_S)$ to be $(S, O_S) = \{(S_t, O_t)\}$.*

**Proposition F.57.** *The number of possible orderings $O$ for $S$ is equal to $|Aut((S, O_S))|$*

**Definition F.58** (Shape transposes). *Given a shape $\alpha$, we define $\alpha^T$ to be the shape $\alpha$ with $U_\alpha$ and $V_\alpha$ swapped i.e. $U_{\sigma^T} = V_\sigma$ and $V_{\sigma^T} = U_\sigma$.*

**Definition F.59** (Left, middle, and right parts). *Let $\alpha$ be a shape. Let $S$ and $T$ be the leftmost and rightmost minimal vertex separators of $\alpha$ together with orderings $O_S, O_T$ of $S$ and $T$.*

- *We define the* left part $\sigma_\alpha$ *of $\alpha$ to be the shape such that*

    1. *$H_{\sigma_\alpha}$ is the induced subgraph of $H_\alpha$ on all of the vertices of $\alpha$ reachable from $U_\alpha$ without passing through $S$ (note that $H_{\sigma_\alpha}$ includes the vertices of $S$) except that we remove any edges/hyperedges which are contained entirely within $S$.*
    2. *$U_{\sigma_\alpha} = U_\alpha$ and $V_{\sigma_\alpha} = (S, O_S)$*

- *We define the* right part $\sigma'^T_\alpha$ *of $\alpha$ to be the shape such that*

    1. *$H_{\sigma'^T_\alpha}$ is the induced subgraph of $H_\alpha$ on all of the vertices of $\alpha$ reachable from $V_\alpha$ without passing through $T$ (note that $H_{\sigma'^T_\alpha}$ includes the vertices of $T$) except that we remove any edges/hyperedges which are contained entirely within $T$.*
    2. *$V_{\sigma'^T_\alpha} = V_\alpha$ and $U_{\sigma'^T_\alpha} = (T, O_T)$*

- *We define the* middle part $\tau_\alpha$ *of $\alpha$ to be the shape such that*

    1. *$H_{\tau_\alpha}$ is the induced subgraph of $H_\alpha$ on all of the vertices of $\alpha$ which are not reachable from $U_\alpha$ and $V_\alpha$ without touching $S$ and $T$ (note that $H_{\tau_\alpha}$ includes the vertices of $S$ and $T$). $H_{\tau_\alpha}$ also includes the hyperedges entirely within $S$ and the hyperedges entirely within $T$.*
    2. *$U_{\tau_\alpha} = (S, O_S)$ and $V_{\tau_\alpha} = (T, O_T)$*

.

**Example F.60.** Fig. 4 *illustrates an example decomposition. We have two types of vertices that we diagrammatically represent by squares and circles. In this example, we assume that the set containing a single circle vertex has a lower weight compared to a set of two square vertices.*

1. *If we start with the shape $\sigma \circ \sigma'^T$, then it can be decomposed uniquely in to the composition of the left shape $\sigma$, the right shape $\sigma'^T$. In this case, the middle shape (not shown in this figure) is trivial.*

2. *If we start with the shape $\sigma \circ \tau \circ \sigma'^T$, then it can be decomposed uniquely into the composition of the left shape $\sigma$, the middle shape $\tau$ and the right shape $\sigma'^T$, which are all shown in this figure.*

**Proposition F.61.** *If $\sigma, \tau, \sigma'^T$ are the left, middle, and rights parts for $\alpha$ for given orderings $O_S, O_T$ of $S$ and $T$ then $\alpha = \sigma \circ \tau \circ \sigma'^T$.*

**Remark F.62.** *One may ask which ordering(s) we should take of $S$ and $T$. The answer is that we will take all of the possible orderings of $S$ and $T$ simultaneously, giving equal weight to each.*

Based on this decomposition and the following claim, we make the following definitions for what it means for a shape to be a left, middle, or right part.

**Claim F.63** (Proved in Section 6.1 in [19]). [†]

- *Every shape $\sigma$ which is the left part of some other shape $\alpha$ has that $V_\sigma$ is its left-most and right-most minimum-weight separator.*

- *Every shape $\sigma^T$ which is the right part of some other shape $\alpha$ has that $U_{\sigma^T}$ is its left-most and right-most minimum-weight separator.*

- *Every shape $\tau$ which is the middle part of some other shape $\alpha$ has $U_\tau$ as its left-most minimum size separator and $V_\tau$ as its right-most minimum-weight separator.*

**Definition F.64.**

1. *We say that a shape $\sigma$ is a left shape if $\sigma$ is a proper shape, $V_\sigma$ is the left-most and right-most minimum-weight separator of $\sigma$, every vertex in $V(\sigma) \setminus V_\sigma$ is reachable from $U_\sigma$ without touching $V_\sigma$, and $\sigma$ has no hyperedges entirely within $V_\sigma$.*

2. *We say that a shape $\tau$ is a proper middle shape if $\tau$ is a proper shape, $U_\tau$ is the left-most minimum-weight separator of $\tau$, and $V_\tau$ is the right most minimum-weight separator of $\tau$. In the analysis, we will also need to consider improper middle shapes $\tau$ which may not be proper shapes and which may have smaller separators between $U_\tau$ and $V_\tau$.*

3. *We say that a shape $\sigma^T$ is a right shape if $\sigma^T$ is a proper shape, $U_{\sigma^T}$ is the left-most and right-most minimum-weight separator of $\sigma^T$, every vertex in $V(\sigma^T) \setminus U_{\sigma^T}$ is reachable from $V_{\sigma^T}$ without touching $U_{\sigma^T}$, and $\sigma^T$ has no hyperedges entirely within $U_{\sigma^T}$.*

**Proposition F.65.** *For all shapes $\sigma$, $\sigma$ is a left shape if and only if $\sigma^T$ is a right shape.*

**Remark F.66.** *As the reader has likely guessed, throughout this section we use $\sigma$ to denote left parts and $\tau$ to denote middle parts. Instead of having a separate letter for right parts, we express right parts as the transpose of a left part.*

## F.8 Coefficient matrices

We will have that $\Lambda = \sum_\alpha \lambda_\alpha M_\alpha$. To analyze $\Lambda$, it is extremely useful to express these coefficients in terms of matrices. To do this, we will need a few more definitions. We start by defining the sets of index shapes that can appear when analyzing $\Lambda$.

**Definition F.67.** *Given a moment matrix $\Lambda$, we define the following sets of index shapes.*

1. *We define $\mathcal{I}(\Lambda) = \{U : \exists \text{ matrix index } A : A \text{ is a row index of } \Lambda, A \text{ has shape } U\}$ to be the set of index shapes which describe row and column indices of $\Lambda$.*

2. *We define $w_{max}$ to be $w_{max} = \max \{w(U) : U \in \mathcal{I}(\Lambda)\}$.*

3. *With our simplifying assumptions, we define $\mathcal{I}_{mid}$ to be $\mathcal{I}_{mid} = \{U : |U| \leq w_{max}\}$*

3*. *In general, we define $\mathcal{I}_{mid}$ to be $\mathcal{I}_{mid} = \{U : w(U) \leq w_{max}, \forall U_i \in U, p_i = 1\}$*

We also need to define the sets of shapes which can appear when analyzing $\Lambda$.

**Definition F.68** (Truncation Parameters). *Given a moment matrix $\Lambda = \sum_\alpha \lambda_\alpha M_\alpha$, we define $D_V, D_E$ to be the smallest natural numbers such that for all shapes $\alpha$ such that $\lambda_\alpha \neq 0$, decomposing $\alpha$ as $\alpha = \sigma \circ \tau \circ \sigma'^T$,*

1. *$|V(\sigma)| \leq D_V$, $|V(\tau)| \leq D_V$, and $|V(\sigma')| \leq D_V$.*

2.* *For all edges $e \in E(\sigma) \cup E(\tau) \cup E(\sigma')$, $l_e \leq D_E$.*

**Remark F.69.** *Under our simplifying assumptions, all edges have label 1 so we will take $D_E = 1$ and ignore conditions involving $D_E$.*

**Definition F.70.** *Given a moment matrix $\Lambda$, we define the following sets of shapes:*

1. *$\mathcal{L} = \{\sigma : \sigma \text{ is a left shape}, U_\sigma \in \mathcal{I}(\Lambda), V_\sigma \in \mathcal{I}_{mid}, |V(\sigma)| \leq D_V, \forall e \in E(\sigma), l_e \leq D_E\}$*

2. *Given $V \in \mathcal{I}_{mid}$, we define $\mathcal{L}_V = \{\sigma \in \mathcal{L} : V_\sigma \equiv V\}$*

---

[†]The proof in [19] only explicitly treats the case when the shapes $\alpha$ are graphs, but the proof easily generalizes to the case when the $\alpha$ are hypergraphs.

3. *Given $U \in \mathcal{I}_{mid}$, we define $\mathcal{M}_U = \{\tau : \tau$ is a non-trivial proper middle shape, $U_\tau \equiv V_\tau \equiv U, |V(\tau)| \leq D_V, \forall e \in E(\tau), l_e \leq D_E\}$*

**Definition F.71.** *Given a moment matrix $\Lambda$, we define a $\Lambda$-coefficient matrix (which we call a coefficient matrix for brevity) to be a matrix whose rows and columns are indexed by left shapes $\sigma, \sigma' \in \mathcal{L}$.*

*We say that a coefficient matrix $H$ is SOS-symmetric if $H(\sigma, \sigma')$ is invariant under permuting the vertices of $U_\sigma$ and permuting the vertices of $U_{\sigma'}$ (\*more precisely, for the general case we permute the vertices within each index shape piece of $U_\sigma$ and permute the vertices within each index shape piece of $U_{\sigma'}$).*

**Definition F.72.** *Given a shape $\tau$, we say that a coefficient matrix $H$ is a $\tau$-coefficient matrix if $H(\sigma, \sigma') = 0$ whenever $V_\sigma \not\equiv U_\tau$ or $V_\tau \not\equiv U_{\sigma'^T}$.*

**Definition F.73.** *Given an index shape $U$, we define $Id_U$ to be the shape with $U_{Id_U} = V_{Id_U} = U$, no other vertices, and no edges.*

Given a shape $\tau$ and a $\tau$-coefficient matrix $H$, we create two different matrix-valued functions, $M_\tau^{fact}(H)$ and $M_\tau^{orth}(H)$. As we will see, we can express $\Lambda$ in terms of $M^{orth}$ but to show PSDness we will need to shift to $M^{fact}$. We analyze the difference betweem $M^{fact}$ and $M^{orth}$ in subsections G.2, G.3, and G.4.

**Definition F.74.** *Given a shape $\tau$ and a $\tau$-coefficient matrix $H$, define*

$$M_\tau^{fact}(H) = \sum_{\sigma \in \mathcal{L}_{U_\tau}, \sigma' \in \mathcal{L}_{V_\tau}} H(\sigma, \sigma') M_\sigma M_\tau M_{\sigma'}^T$$

**Proposition F.75.** *For all $A$ and $B$ with shapes in $\mathcal{I}(\Lambda)$,*

$$\left(M_\tau^{fact}(H)\right)(A, B) =$$
$$\sum_{\sigma \in \mathcal{L}_{U_\tau}, \sigma' \in \mathcal{L}_{V_\tau}} H(\sigma, \sigma') \sum_{A', B'} \sum_{\substack{R_1 \in \mathcal{R}(\sigma, A, A'), R_2 \in \mathcal{R}(\tau, A', B'), \\ R_3 \in \mathcal{R}(\sigma'^T, B', B)}} M_{R_1}(A, A') M_{R_2}(A', B') M_{R_3}(B', B)$$

If $R_1, R_2, R_3$ are properly composable then $R = R_1 \circ R_2 \circ R_3$ has the expected shape $\sigma \circ \tau \circ \sigma'^T$. Otherwise, $R_1 \circ R_2 \circ R_3$ will have a different shape. We define $M_\tau^{orth}(H)$ to be the same sum as $M_\tau^{fact}(H)$ except that it is restricted to properly composable ribbons $R_1, R_2, R_3$.

**Definition F.76.** *We define $M_\tau^{orth}(H)$ so that for all $A$ and $B$ with shapes in $\mathcal{I}(\Lambda)$,*

$$\left(M_\tau^{orth}(H)\right)(A, B)$$
$$= \sum_{\sigma \in \mathcal{L}_{U_\tau}, \sigma' \in \mathcal{L}_{V_\tau}} H(\sigma, \sigma') \sum_{A', B'} \sum_{\substack{R_1 \in \mathcal{R}(\sigma, A, A'), R_2 \in \mathcal{R}(\tau, A', B'), \\ R_3 \in \mathcal{R}(\sigma'^T, B', B), R_1, R_2, R_3 \text{ are properly composable}}} M_{R_1}(A, A') M_{R_2}(A', B') M_{R_3}(B', B)$$
$$= \sum_{\sigma \in \mathcal{L}_{U_\tau}, \sigma' \in \mathcal{L}_{V_\tau}} H(\sigma, \sigma') \sum_{A', B'} \sum_{\substack{R_1 \in \mathcal{R}(\sigma, A, A'), R_2 \in \mathcal{R}(\tau, A', B'), \\ R_3 \in \mathcal{R}(\sigma'^T, B', B), R_1, R_2, R_3 \text{ are properly composable}}} M_{R_1 \circ R_2 \circ R_3}(A, B)$$

It would be nice if we had that $M_\tau^{orth}(H) = \sum_{\sigma \in \mathcal{R}_{U_\tau}, \sigma' \in \mathcal{R}_{V_\tau}} H(\sigma, \sigma') M_{\sigma \circ \tau \circ \sigma'^T}$. However, this is not quite correct because there is an additional term related to automorphism groups.

**Definition F.77.** *Given a shape $\alpha$, define $Aut(\alpha)$ to be the set of mappings from $\alpha$ to itself which keep $U_\alpha$ and $V_\alpha$ fixed.*

**Example F.78.** *Consider the shape $\sigma$ where $U_\sigma = (u_1, u_2, u_3)$, $V_\sigma = (v_1, v_2, v_3)$, and $V(\sigma) = U_\sigma \cup V_\sigma \cup \{w_1, w_2, w_3\}$ with edges*

$$E(\alpha) = \{(u_1, w_1), (u_2, w_1), (u_3, w_1), (u_1, w_2), (u_2, w_2), (u_3, w_2), (u_1, w_3), (u_2, w_3), (u_3, w_3)\}$$
$$\cup \{(w_1, v_1), (w_1, v_2), (w_2, v_1), (w_2, v_2), (w_3, v_1), (w_3, v_2)\}$$

*where all edges have label $1$. Then, $Aut(\sigma) = Aut(\sigma^T) = S_3$ and $Aut(\sigma \circ \sigma^T) = S_3 \times S_2 \times S_3$. Note that in this case $Aut(\sigma \circ \sigma^T)/(Aut(\sigma) \times Aut(\sigma^T)) = S_2$. The last computation will be useful for the definition that follows.*

**Definition F.79.** *Given composable shapes $\sigma, \tau, \sigma'^T$, we define*

$$Decomp(\sigma, \tau, \sigma') = Aut(\sigma \circ \tau \circ \sigma')/(Aut(\sigma) \times Aut(\tau) \times Aut(\sigma'^T))$$

**Remark F.80.** *Each element $\pi \in Decomp(\sigma, \tau, \sigma')$ decomposes $\sigma \circ \tau \circ \sigma'^T$ into $\sigma$, $\tau$, and $\sigma'^T$ by specifying copies $\pi(\sigma)$, $\pi(\tau)$, $\pi(\sigma'^T)$ of $\sigma$, $\tau$, and $\sigma'^T$ such that $\pi(\sigma) \circ \pi(\tau) \circ \pi(\sigma'^T) = \pi(\sigma \circ \tau \circ \sigma'^T) = \sigma \circ \tau \circ \sigma'^T$. Thus, $|Decomp(\sigma, \tau, \sigma')|$ is the number of ways to decompose $\sigma \circ \tau \circ \sigma'^T$ into $\sigma$, $\tau$, and $\sigma'^T$.*

**Lemma F.81.**

$$M_\tau^{orth}(H) = \sum_{\sigma \in \mathcal{L}_{U_\tau}, \sigma' \in \mathcal{L}_{V_\tau}} H(\sigma, \sigma')|Decomp(\sigma, \tau, \sigma'^T)|M_{\sigma \circ \tau \circ \sigma'^T}$$

*Proof sketch.* Observe that there is a bijection between ribbons $R$ with shape $\sigma \circ \tau \circ \sigma'^T$ together with an element $\pi \in Decomp(\sigma, \tau, \sigma')$ and triples of ribbons $(R_1, R_2, R_3)$ such that

1. $R_1, R_2, R_3$ have shapes $\sigma$, $\tau$, and $\sigma'^T$, respectively.

2. $V(R_1) \cap V(R_2) = A_{R_2} = B_{R_1}$, $V(R_2) \cap V(R_3) = A_{R_3} = B_{R_2}$, and $V(R_1) \cap V(R_3) = A_{R_2} \cap B_{R_2}$

To see this, note that given such ribbons $R_1, R_2, R_3$, the ribbon $R = R_1 \circ R_2 \circ R_3$ has shape $\sigma \circ \tau \circ \sigma'^T$ and the ribbons $R_1, R_2, R_3$ specify a decomposition of $\sigma \circ \tau \circ \sigma'^T$ into $\sigma$, $\tau$, and $\sigma'^T$.

Conversely, given $R$ and an element $\pi \in Decomp(\sigma, \tau, \sigma')$, $\pi$ specifies how to decompose $R$ into ribbons $R_1, R_2, R_3$ of shapes $\sigma$, $\tau$, and $\sigma'^T$. ∎

**Remark F.82.** *As this lemma shows, we have to be very careful about symmetry groups in our analysis. For accuracy, it is safest to check that the coefficients for each individual ribbon match.*

Given a matrix-valued function $\Lambda$, we can associate coefficient matrices to $\Lambda$ as follows:

**Definition F.83.** *Given a matrix-valued function $\Lambda = \sum_{\alpha: \alpha \text{ is proper}} \lambda_\alpha M_\alpha$,*

1. *For each index shape $U \in \mathcal{I}_{mid}$ and every $\sigma, \sigma' \in \mathcal{L}_U$, we take $H_{Id_U}(\sigma, \sigma') = \frac{1}{|Aut(U)|}\lambda_{\sigma \circ \sigma'^T}$*

2. *For each $U \in \mathcal{I}_{mid}$, $\tau \in \mathcal{M}_U$ and $\sigma, \sigma' \in \mathcal{L}_U$, we take $H_\tau(\sigma, \sigma') = \frac{1}{|Aut(U_\tau)| \cdot |Aut(V_\tau)|}\lambda_{\sigma \circ \tau \circ \sigma'^T}$*

**Lemma F.84.** $\Lambda = \sum_{U \in \mathcal{I}_{mid}} M_{Id_U}^{orth}(H_{Id_U}) + \sum_{U \in \mathcal{I}_{mid}} \sum_{\tau \in \mathcal{M}_U} M_\tau^{orth}(H_\tau)$

*Proof.* We check that the coefficients for each individual ribbon $R$ match. There are two cases to consider.

If $R$ has shape $\alpha$ where $\alpha$ has a unique minimum vertex separator $S$, then there is a bijection between orderings $O_S$ for $S$ and pairs of ribbons $R_1, R_2$ such that $R_1 \circ R_2 = R$ and the shapes $\sigma, \sigma'^T$ of $R_1, R_2$ are left and right shapes respectively.

To see this, observe that when we concatenate $R_1$ and $R_2$, this assigns the matrix index $B_{R_1} = A_{R_2}$ to $S$, which is equivalent to specifying an ordering $O_S$ for $S$. Conversely, given an ordering $O_S$ for $S$, we take $R_1$ to be the part of $R$ between $A_R$ and $(S, O_S)$ and we take $R_2$ to be the part of $R$ between $(S, O_S)$ and $B_R$.

From this bijection, it follows that the coefficient of $M_R$ is $\lambda_\alpha$ on both sides of the equation.

Similarly, if $R$ has shape $\alpha$ where $\alpha$ does not have a unique minimal vertex separator, then there is a bijection between orderings $O_S, O_T$ for the leftmost and rightmost minimum vertex separators $S, T$ of $R$ and triples of ribbons $R_1, R_2, R_3$ such that $R_1 \circ R_2 \circ R_3 = R$ and the shapes $\sigma, \tau, \sigma'^T$ of $R_1, R_2, R_3$ are left, proper middle, and right shapes respectively.

To see this, observe that when we concatenate $R_1$, $R_2$, and $R_3$, this assigns the matrix index $B_{R_1} = A_{R_2}$ to $S$ and assigns the matrix index $B_{R_2} = A_{R_3}$ to $T$, which is equivalent to specifying

orderings $O_S, O_T$ for $S, T$. Conversely, given orderings $O_S, O_T$ for $S, T$, we take $R_1$ to be the part of $R$ between $A_R$ and $(S, O_S)$, we take $R_2$ to be the part of $R$ between $(S, O_S)$ and $(T, O_T)$, and we take $R_2$ to be the part of $R$ between $(T, O_T)$ and $B_R$.

From this bijection, it again follows that the coefficient of $M_R$ is $\lambda_\alpha$ on both sides of the equation. ∎

## F.9 The $-\gamma, -\gamma$ operation and qualitative theorem statement

In the intersection term analysis (see subsections [G.2], [G.3], and [G.4]), we will need to further decompose left shapes $\sigma$ as $\sigma = \sigma_2 \circ \gamma$ where $\sigma_2$ and $\gamma$ are themselves left shapes. Accordingly, we make the following definitions

**Definition F.85.** *Given a moment matrix $\Lambda$, we define the following sets of left shapes:*

1. $\Gamma = \{\gamma : \gamma \text{ is a non-trivial left shape}, U_\gamma, V_\gamma \in \mathcal{I}_{mid}, |V(\gamma)| \le D_V, \forall e \in E(\gamma), l_e \le D_E\}$

2. *Given $U, V \in \mathcal{I}_{mid}$ such that $w(U) > w(V)$, define $\Gamma_{U,V} = \{\gamma \in \Gamma : U_\gamma \equiv U, V_\gamma \equiv V\}$.*

3. *Given $U \in \mathcal{I}_{mid}$, define $\Gamma_{U,*} = \{\gamma \in \Gamma : U_\gamma \equiv U\}$*

4. *Given $V \in \mathcal{I}_{mid}$, define $\Gamma_{*,V} = \{\gamma \in \Gamma : V_\gamma \equiv V\}$*

**Remark F.86.** *Under our simplifying assumptions, $\Gamma$ is the same as $\mathcal{L}$ except that $\Gamma$ excludes the trivial shapes. In general, while $\mathcal{L}$ requires that $U_\sigma \in \mathcal{I}(\Lambda)$, $\Gamma$ requires that $U_\gamma \in \mathcal{I}_{mid}$. Note that $\mathcal{I}(\Lambda)$ and $\mathcal{I}_{mid}$ may be incomparable because*

1. *There may be index shapes $U \in \mathcal{I}_{mid}$ such that no matrix index of $\Lambda$ has shape $U$.*

2. *All index shape pieces $U_i$ for index shapes $U \in \mathcal{I}_{mid}$ must have $p_i = 1$ while this is not the case for $\mathcal{I}(\Lambda)$.*

We now state our theorem qualitatively after giving one more definition.

**Definition F.87.** *Given a shape $\tau$, left shapes $\gamma \in \Gamma_{*,U_\tau}$ and $\gamma' \in \Gamma_{*,V_\tau}$, and a $\tau$-coefficient matrix $H$, define $H^{-\gamma,\gamma'}$ to be the $(\gamma \circ \tau \circ \gamma'^T)$-coefficient matrix with entries*

1. $H^{-\gamma,\gamma'}(\sigma, \sigma') = H(\sigma \circ \gamma, \sigma' \circ \gamma')$ *if $|V(\sigma \circ \gamma)| \le D_V$ and $|V(\sigma' \circ \gamma')| \le D_V$.*

2. $H^{-\gamma,\gamma'}(\sigma, \sigma') = 0$ *if $|V(\sigma \circ \gamma)| > D_V$ or $|V(\sigma' \circ \gamma')| > D_V$.*

**Remark F.88.** *For the theorem, we will only need the case when $\gamma' = \gamma$*

Our qualitative theorem statement is as follows:

**Theorem F.89.** *Let $\Lambda = \sum_{U \in \mathcal{I}_{mid}} M_{Id_U}^{orth}(H_{Id_U}) + \sum_{U \in \mathcal{I}_{mid}} \sum_{\tau \in \mathcal{M}_U} M_\tau^{orth}(H_\tau)$ be an SOS-symmetric matrix valued function.*

*There exist functions $f(\tau)$ and $f(\gamma)$ depending on $n$ and other parameters such that if the following conditions hold:*

1. *For all $U \in \mathcal{I}_{mid}$, $H_{Id_U} \succeq 0$*

2. *For all $U \in \mathcal{I}_{mid}$ and all $\tau \in \mathcal{M}_U$,*

$$\begin{bmatrix} H_{Id_U} & f(\tau)H_\tau \\ f(\tau)H_\tau^T & H_{Id_U} \end{bmatrix} \succeq 0$$

3. *For all $U, V \in \mathcal{I}_{mid}$ where $w(U) > w(V)$ and all $\gamma \in \Gamma_{U,V}$, $H_{Id_V}^{-\gamma,\gamma} \preceq f(\gamma)H_{Id_U}$*

*then with high probability $\Lambda \succeq 0$*

**Remark F.90.** *Roughly speaking, conditions 1 and 2 give us an approximate PSD decomposition for the moment matrix $M$. Condition 3 comes from the intersection term analysis, which is the most technically intensive part of the proof.*

### F.10 Quantitative theorem statement

To state our theorem quantitatively, we will need a few more things. First, the conditions of the theorem will involve functions $B_{norm}(\alpha)$, $B(\gamma)$, $N(\gamma)$, and $c(\alpha)$. Roughly speaking, these functions will be used as follows in the analysis:

1. $B_{norm}(\alpha)$ will bound the norms of the matrices $M_\alpha$

2. $B(\gamma)$ and $N(\gamma)$ will help us bound the intersection terms (see Section G.4).

3. $c(\alpha)$ will help us sum over the possible $\gamma$ and $\tau$.

Second, for technical reasons it turns out that comparing $H_{Id_{V_\gamma}}^{-\gamma,\gamma}$ to $H_{Id_{U_\gamma}}$ doesn't quite work. Instead, we compare $H_{Id_{V_\gamma}}^{-\gamma,\gamma}$ to a matrix $H'_\gamma$ of our choice where $H'_\gamma$ is very close to $H_{Id_{U_\gamma}}$ ($H'_\gamma$ will be the same as $H_{Id_{U_\gamma}}$ up to truncation error).

**Definition F.91.** *Given a function $B_{norm}(\alpha)$, we define the distance $d_\tau(H_\tau, H'_\tau)$ between two $\tau$-coefficient matrices $H_\tau$ and $H'_\tau$ to be*

$$d_\tau(H_\tau, H'_\tau) = \sum_{\sigma \in \mathcal{L}_{U_\tau}, \sigma' \in \mathcal{L}_{V_\tau}} |H'_\tau(\sigma, \sigma') - H_\tau(\sigma, \sigma')| B_{norm}(\sigma) B_{norm}(\tau) B_{norm}(\sigma')$$

Third, we need an SOS-symmetric analogue of the identity matrix.

**Definition F.92.** *We define $Id_{Sym}$ to be the matrix such that*

1. *The rows and columns of $Id_{Sym}$ are indexed by the matrix indices $A, B$ whose index shape is in $\mathcal{I}(\Lambda)$.*

2. *$Id_{Sym}(A, B) = 1$ if $p_A = p_B$ and $Id_{Sym}(A, B) = 0$ if $p_A \neq p_B$.*

**Proposition F.93.** *If $M$ has SOS-symmetry and the rows and columns of $Id_{Sym}$ are indexed by matrix indices $A, B$ whose index shape is in $\mathcal{I}(\Lambda)$ then $M \preceq \|M\| Id_{Sym}$*

**Corollary F.94.** *For all $\tau$ and all SOS-symmetric $\tau$-coefficient matrices $H_\tau$ and $H'_\tau$,*

$$M_\tau^{fact}(H'_\tau) + M_{\tau^T}^{fact}(H'_{\tau^T}) - M_\tau^{fact}(H_\tau) - M_{\tau^T}^{fact}(H_{\tau^T}) \preceq 2d_\tau(H_\tau, H'_\tau) Id_{Sym}$$

*Note that if $\tau$, $H_\tau$ and $H'_\tau$ are all symmetric then*

$$M_\tau^{fact}(H'_\tau) - M_\tau^{fact}(H_\tau) \preceq d_\tau(H_\tau, H'_\tau) Id_{Sym}$$

Finally, we need a few more definitions about shapes $\alpha$.

**Definition F.95** ($\mathcal{M}'$)**.** *We define $\mathcal{M}'$ to be the set of all shapes $\alpha$ such that*

1. *$|V(\alpha)| \leq 3D_V$*

2.\* *$\forall e \in E(\alpha), l_e \leq D_E$*

3.\* *All edges $e \in E(\alpha)$ have multiplicity at most $3D_V$.*

**Definition F.96** ($S_\alpha$)**.** *Given a shape $\alpha$, define $S_\alpha$ to be the leftmost minimum vertex separator of $\alpha$*

**Definition F.97** ($I_\alpha$)**.** *Given a shape $\alpha$, define $I_\alpha$ to be the set of vertices in $V(\alpha) \setminus (U_\alpha \cup V_\alpha)$ which are isolated.*

Our main theorem will require the choice of several functions and parameters $q, B_{vertex}, B_{edge}(e), B_{norm}(\alpha), B(\gamma), N(\gamma), c(\alpha)$ satisfying certain conditions. $B_{edge}$ is not needed in the simplified case. For simplicity, we defer the formal conditions to the next section.

**Definition F.98** ($\varepsilon$-feasible parameters)**.** *For $\varepsilon > 0$, define $q, B_{vertex}, B_{edge}(e), B_{norm}(\alpha), B(\gamma), N(\gamma), c(\alpha)$ to be $\varepsilon$-feasible parameters if they satisfy the conditions in Theorem G.1.*

For our applications, we can work with the parameters as given by the following lemma, justified in Appendix H.

**Lemma F.99.** *For all $\varepsilon > 0$, the parameters*

1. $q = 3 \left\lceil D_V ln(n) + \frac{ln(\frac{1}{\varepsilon})}{3} + D_V ln(5) + 3D_V^2 ln(2) \right\rceil$

2. $B_{vertex} = 6D_V \sqrt[4]{2eq}$

3. $B_{norm}(\alpha) = B_{vertex}^{|V(\alpha)\setminus U_\alpha|+|V(\alpha)\setminus V_\alpha|} n^{\frac{w(V(\alpha))+w(I_\alpha)-w(S_\alpha)}{2}}$

4. $B(\gamma) = B_{vertex}^{|V(\gamma)\setminus U_\gamma|+|V(\gamma)\setminus V_\gamma|} n^{\frac{w(V(\gamma)\setminus U_\gamma)}{2}}$

5. $N(\gamma) = (3D_V)^{2|V(\gamma)\setminus V_\gamma|+|V(\gamma)\setminus U_\gamma|}$

6. $c(\alpha) = 100(3D_V)^{|U_\alpha\setminus V_\alpha|+|V_\alpha\setminus U_\alpha|+2|E(\alpha)|} 2^{|V(\alpha)\setminus(U_\alpha\cup V_\alpha)|}$

*are $\varepsilon$-feasible.*

**Remark F.100.** *In our applications, we show SoS lower bounds for $n^\varepsilon$ degrees of SoS, where input size is $n^{O(1)}$. In this setting, we take $D_V, D_E$ to be of the order of $n^{O(\varepsilon)}$. Therefore, for simplicity, we can interpret the parameters as*

$$q = n^{O(\varepsilon)}, B_{vertex} = n^{O(\varepsilon)}, B_{norm}(\alpha) = n^{O(\varepsilon)|V(\alpha)|} n^{\frac{w(V(\alpha))+w(I_\alpha)-w(S_\alpha)}{2}}$$

$$B(\gamma) = n^{O(\varepsilon)|V(\gamma)|} n^{\frac{w(V(\gamma)\setminus U_\gamma)}{2}}, N(\gamma) = n^{O(\varepsilon)|V(\gamma)|}, c(\alpha) = n^{O(\varepsilon)|V(\alpha)|}$$

We can now state our main theorem.

**Theorem F.101.** *Given the moment matrix $\Lambda = \sum_{U\in\mathcal{I}_{mid}} M_{Id_U}^{orth}(H_{Id_U}) + \sum_{U\in\mathcal{I}_{mid}} \sum_{\tau\in\mathcal{M}_U} M_\tau^{orth}(H_\tau)$, for all $\varepsilon > 0$, if we take $\varepsilon$-feasible parameters, and we have SOS-symmetric coefficient matrices $\{H_\gamma' : \gamma \in \Gamma\}$ such that the following conditions hold:*

1. *(PSD mass) For all $U \in \mathcal{I}_{mid}$, $H_{Id_U} \succeq 0$*

2. *(Middle shape bounds) For all $U \in \mathcal{I}_{mid}$ and $\tau \in \mathcal{M}_U$,*

$$\begin{bmatrix} \frac{1}{|Aut(U)|c(\tau)} H_{Id_U} & B_{norm}(\tau)H_\tau \\ B_{norm}(\tau)H_\tau^T & \frac{1}{|Aut(U)|c(\tau)} H_{Id_U} \end{bmatrix} \succeq 0$$

3. *(Intersection term bounds) For all $U, V \in \mathcal{I}_{mid}$ where $w(U) > w(V)$ and all $\gamma \in \Gamma_{U,V}$,*

$$c(\gamma)^2 N(\gamma)^2 B(\gamma)^2 H_{Id_V}^{-\gamma,\gamma} \preceq H_\gamma'$$

*then with probability at least $1 - \varepsilon$,*

$$\Lambda \succeq \frac{1}{2} \left( \sum_{U\in\mathcal{I}_{mid}} M_{Id_U}^{fact}(H_{Id_U}) \right) - 3 \left( \sum_{U\in\mathcal{I}} \sum_{\gamma\in\Gamma_{U,*}} \frac{d_{Id_U}(H_\gamma', H_{Id_U})}{|Aut(U)|c(\gamma)} \right) Id_{sym}$$

*(Truncation error bounds) If it is also true that whenever $\|M_\alpha\| \leq B_{norm}(\alpha)$ for all $\alpha \in \mathcal{M}'$,*

$$\sum_{U\in\mathcal{I}_{mid}} M_{Id_U}^{fact}(H_{Id_U}) \succeq 6 \left( \sum_{U\in\mathcal{I}} \sum_{\gamma\in\Gamma_{U,*}} \frac{d_{Id_U}(H_\gamma', H_{Id_U})}{|Aut(U)|c(\gamma)} \right) Id_{sym}$$

*then with probability at least $1 - \varepsilon$, $\Lambda \succeq 0$.*

### F.10.1 General Main Theorem

Before stating the general main theorem, we need to modify a few definitions for $\alpha$ and give a few definitions for $\Omega$

**Definition F.102** ($S_{\alpha,min}$ and $S_{\alpha,max}$). *Given a shape $\alpha \in \mathcal{M}'$, define $S_{\alpha,min}$ to be the leftmost minimum vertex separator of $\alpha$ if all edges with multiplicity at least 2 are deleted and define $S_{\alpha,max}$ to be the leftmost minimum vertex separator of $\alpha$ if all edges with multiplicity at least 2 are present.*

**Definition F.103** (General $I_\alpha$). *Given a shape $\alpha$, define $I_\alpha$ to be the set of vertices in $V(\alpha)\setminus(U_\alpha\cup V_\alpha)$ such that all edges incident with that vertex have multiplicity at least 2.*

**Definition F.104** ($B_\Omega$). *We take $B_\Omega(j)$ to be a non-decreasing function such that for all $j \in \mathbb{N}$, $E_\Omega[x^j] \leq B_\Omega(j)^j$*

**Definition F.105** ($h_j^+$). *For all $j$, we define $h_j^+$ to be the polynomial $h_j$ where we make all of the coefficients have positive sign.*

**Lemma F.106.** *If $\Omega = N(0,1)$ then we can take $B_\Omega(j) = \sqrt{j}$ and we have that*

$$h_j^+(x) \leq \frac{1}{\sqrt{j!}}(x^2 + j)^{\frac{j}{2}} \leq \left(\frac{e}{j}(x^2 + j)\right)^{\frac{j}{2}}$$

For a proof, see [1, Lemma 8.15]. We again give a choice of $\varepsilon$-feasible parameters used in our applications, justified in Appendix H.

**Lemma F.107.** *For all $\varepsilon > 0$, the parameters*

1. $q = \lceil 3D_V ln(n) + ln(\frac{1}{\varepsilon}) + (3D_V)^k ln(D_E + 1) + 3D_V ln(5) \rceil$

2. $B_{vertex} = 6qD_V$

3. $B_{edge}(e) = 2h_{l_e}^+(B_\Omega(6D_V D_E)) \max_{j \in [0, 3D_V D_E]} \left\{ \left(h_j^+(B_\Omega(2qj))\right)^{\frac{l_e}{\max\{j, l_e\}}} \right\}$

   *As a special case, if $\Omega = N(0,1)$ then we can take $B_{edge}(e) = \left(400 D_V^2 D_E^2 q\right)^{l_e}$*

4. $B_{norm}(\alpha) = 2e B_{vertex}^{|V(\alpha)\setminus U_\alpha| + |V(\alpha)\setminus V_\alpha|} \left(\prod_{e \in E(\alpha)} B_{edge}(e)\right) n^{\frac{w(V(\alpha)) + w(I_\alpha) - w(S_{\alpha,min})}{2}}$

5. $B(\gamma) = B_{vertex}^{|V(\gamma)\setminus U_\gamma| + |V(\gamma)\setminus V_\gamma|} \left(\prod_{e \in E(\gamma)} B_{edge}(e)\right) n^{\frac{w(V(\gamma)\setminus U_\gamma)}{2}}$

6. $N(\gamma) = (3D_V)^{2|V(\gamma)\setminus V_\gamma| + |V(\gamma)\setminus U_\gamma|}$

7. $c(\alpha) = 100(3t_{max}D_V)^{|U_\alpha\setminus V_\alpha| + |V_\alpha\setminus U_\alpha| + k|E(\alpha)|}(2t_{max})^{|V(\alpha)\setminus(U_\alpha \cup V_\alpha)|}$

*are $\varepsilon$-feasible.*

Similar to Remark F.100, in our applications, we can interpret the above parameters in a much simpler manner. Just as in all our applications, assume we work with the Gaussian distribution $\Omega = N(0,1)$, $k$ is a constant and we work with SoS degree $n^\varepsilon$. Then, we think of each vertex or edge of the shape $\alpha$ or $\gamma$ essentially contributing a factor of $n^\varepsilon$. Therefore, we can interpret

$$q = n^{O(\varepsilon)}, B_{vertex} = n^{O(\varepsilon)}, B_{edge} = n^{O(\varepsilon)|E(\alpha)|}$$

$$B_{norm}(\alpha) = n^{O(\varepsilon)(|V(\alpha)| + |E(\alpha)|)} n^{\frac{w(V(\alpha)) + w(I_\alpha) - w(S_{\alpha,min})}{2}}, B(\gamma) = n^{O(\varepsilon)(|V(\gamma)| + |E(\gamma)|)} n^{\frac{w(V(\gamma)\setminus U_\gamma)}{2}}$$

$$N(\gamma) = n^{O(\varepsilon)|V(\gamma)|}, c(\alpha) = n^{O(\varepsilon)(|V(\alpha)| + |E(\alpha)|)}$$

**Theorem F.108.** *Given the moment matrix $\Lambda = \sum_{U \in \mathcal{I}_{mid}} M_{Id_U}^{orth}(H_{Id_U}) + \sum_{U \in \mathcal{I}_{mid}} \sum_{\tau \in \mathcal{M}_U} M_\tau^{orth}(H_\tau)$, for all $\varepsilon > 0$, if we take $\varepsilon$-feasible parameters and we have SOS-symmetric coefficient matrices $\{H_\gamma' : \gamma \in \Gamma\}$ such that the following conditions hold:*

1. *(PSD mass) For all $U \in \mathcal{I}_{mid}$, $H_{Id_U} \succeq 0$*

2. *(Middle shape bounds) For all $U \in \mathcal{I}_{mid}$ and $\tau \in \mathcal{M}_U$,*

$$\begin{bmatrix} \frac{1}{|Aut(U)|c(\tau)} H_{Id_U} & B_{norm}(\tau) H_\tau \\ B_{norm}(\tau) H_\tau^T & \frac{1}{|Aut(U)|c(\tau)} H_{Id_U} \end{bmatrix} \succeq 0$$

3. *(Intersection term bounds) For all $U, V \in \mathcal{I}_{mid}$ where $w(U) > w(V)$ and all $\gamma \in \Gamma_{U,V}$,*

$$c(\gamma)^2 N(\gamma)^2 B(\gamma)^2 H_{Id_V}^{-\gamma,\gamma} \preceq H_\gamma'$$

*then with probability at least $1 - \varepsilon$,*

$$\Lambda \succeq \frac{1}{2}\left(\sum_{U \in \mathcal{I}_{mid}} M_{Id_U}^{fact}(H_{Id_U})\right) - 3\left(\sum_{U \in \mathcal{I}} \sum_{\gamma \in \Gamma_{U,*}} \frac{d_{Id_U}(H_\gamma', H_{Id_U})}{|Aut(U)|c(\gamma)}\right) Id_{sym}$$

(Truncation error bounds) If it is also true that whenever $\|M_\alpha\| \le B_{norm}(\alpha)$ for all $\alpha \in \mathcal{M}'$,

$$\sum_{U \in \mathcal{I}_{mid}} M_{Id_U}^{fact}(H_{Id_U}) \succeq 6 \left( \sum_{U \in \mathcal{I}_{mid}} \sum_{\gamma \in \Gamma_{U,*}} \frac{d_{Id_U}(H_\gamma', H_{Id_U})}{|Aut(U)|c(\gamma)} \right) Id_{sym}$$

then with probability at least $1 - \varepsilon$, $\Lambda \succeq 0$.

## F.11  Choosing $H_\gamma'$ and Truncation Error

A canonical choice for $H_\gamma'$ is to take

1. $H_\gamma'(\sigma, \sigma') = H_{Id_U}(\sigma, \sigma')$ whenever $|V(\sigma \circ \gamma)| \le D_V$ and $|V(\sigma' \circ \gamma)| \le D_V$.
2. $H_\gamma'(\sigma, \sigma') = 0$ whenever $|V(\sigma \circ \gamma)| > D_V$ or $|V(\sigma' \circ \gamma)| > D_V$.

With this choice, the truncation error is

$$d_{Id_{U_\gamma}}(H_{Id_{U_\gamma}}, H_\gamma') = \sum_{\substack{\sigma, \sigma' \in \mathcal{L}_{U_\gamma} : V(\sigma) \le D_V, V(\sigma') \le D_V, \\ |V(\sigma \circ \gamma)| > D_V \text{ or } |V(\sigma' \circ \gamma)| > D_V}} B_{norm}(\sigma) B_{norm}(\sigma') H_{Id_{U_\gamma}}(\sigma, \sigma')$$

## G  Proof of the Main Theorem

In this section, we prove the main theorem under the assumption that the functions $B_{norm}(\alpha)$, $B(\gamma)$, $N(\gamma)$, and $c(\alpha)$ have certain properties. More precisely, we prove the following theorem.

**Theorem G.1.** *For all $\varepsilon > 0$ and all $\varepsilon' \in (0, \frac{1}{20}]$, for any moment matrix*

$$\Lambda = \sum_{U \in \mathcal{I}_{mid}} M_{Id_U}^{orth}(H_{Id_U}) + \sum_{U \in \mathcal{I}_{mid}} \sum_{\tau \in \mathcal{M}_U} M_\tau^{orth}(H_\tau),$$

*if $B_{norm}(\alpha)$, $B(\gamma)$, $N(\gamma)$, and $c(\alpha)$ are functions such that*

1. *With probability at least $(1 - \varepsilon)$, for all shapes $\alpha \in \mathcal{M}'$, $\|M_\alpha\| \le B_{norm}(\alpha)$.*

2. *For all $\tau \in \mathcal{M}'$, $\gamma \in \Gamma_{*,U_\tau}$, $\gamma' \in \Gamma_{*,V_\tau}$, and all intersection patterns $P \in \mathcal{P}_{\gamma,\tau,\gamma'}$,*

   $$B_{norm}(\tau_P) \le B(\gamma)B(\gamma')B_{norm}(\tau)$$

   *Note: Intersection patterns and $\mathcal{P}_{\gamma,\tau,\gamma'}$ will be defined later, see Definitions G.8 and G.9*

3. *For all composable $\gamma_1, \gamma_2$, $B(\gamma_1)B(\gamma_2) = B(\gamma_1 \circ \gamma_2)$.*

4. *$\forall U \in \mathcal{I}_{mid}, \sum_{\gamma \in \Gamma_{U,*}} \frac{1}{|Aut(U)|c(\gamma)} < \varepsilon'$*

5. *$\forall V \in \mathcal{I}_{mid}, \sum_{\gamma \in \Gamma_{*,V}} \frac{1}{|Aut(U_\gamma)|c(\gamma)} < \varepsilon'$*

6. *$\forall U \in \mathcal{I}_{mid}, \sum_{\tau \in \mathcal{M}_U} \frac{1}{|Aut(U)|c(\tau)} < \varepsilon'$*

7. *For all $\tau \in \mathcal{M}'$, $\gamma \in \Gamma_{*,U_\tau} \cup \{Id_{U_\tau}\}$, and $\gamma' \in \Gamma_{*,V_\tau} \cup \{Id_{V_\tau}\}$,*

$$\sum_{j>0} \sum_{\gamma_1, \gamma_1', \cdots, \gamma_j, \gamma_j' \in \Gamma_{\gamma,\gamma',j}} \prod_{i:\gamma_i \text{ is non-trivial}} \frac{1}{|Aut(U_{\gamma_i})|} \prod_{i:\gamma_i' \text{ is non-trivial}} \frac{1}{|Aut(U_{\gamma_i'})|} \sum_{P_1, \cdots, P_j : P_i \in \mathcal{P}_{\gamma_i, \tau_{P_{i-1}}, \gamma_i'^T}} \left( \prod_{i=1}^{j} N(P_i) \right)$$

$$\le \frac{N(\gamma)N(\gamma')}{(|Aut(U_\gamma)|)^{1_{\gamma \text{ is non-trivial}}} (|Aut(U_{\gamma'})|)^{1_{\gamma' \text{ is non-trivial}}}}$$

   *Note: $\Gamma_{\gamma,\gamma',j}$ will be defined later, see Definition G.18*

*and we have SOS-symmetric coefficient matrices $\{H_\gamma' : \gamma \in \Gamma\}$ such that the following conditions hold:*

1. *For all $U \in \mathcal{I}_{mid}$, $H_{Id_U} \succeq 0$*

2. *For all $U \in \mathcal{I}_{mid}$ and $\tau \in \mathcal{M}_U$,*

$$\begin{bmatrix} \frac{1}{|Aut(U)|c(\tau)}H_{Id_U} & B_{norm}(\tau)H_\tau \\ B_{norm}(\tau)H_\tau^T & \frac{1}{|Aut(U)|c(\tau)}H_{Id_U} \end{bmatrix} \succeq 0$$

3. *For all $U, V \in \mathcal{I}_{mid}$ where $w(U) > w(V)$ and all $\gamma \in \Gamma_{U,V}$,*

$$c(\gamma)^2 N(\gamma)^2 B(\gamma)^2 H_{Id_V}^{-\gamma,\gamma} \preceq H'_\gamma$$

*then with probability at least $1 - \varepsilon$,*

$$\Lambda \succeq \frac{1}{2}\left( \sum_{U \in \mathcal{I}_{mid}} M_{Id_U}^{fact}(H_{Id_U}) \right) - 3\left( \sum_{U \in \mathcal{I}_{mid}} \sum_{\gamma \in \Gamma_{U,*}} \frac{d_{Id_U}(H'_\gamma, H_{Id_U})}{|Aut(U)|c(\gamma)} \right) Id_{sym}$$

*If it is also true that whenever $||M_\alpha|| \leq B_{norm}(\alpha)$ for all $\alpha \in \mathcal{M}'$,*

$$\sum_{U \in \mathcal{I}_{mid}} M_{Id_U}^{fact}(H_{Id_U}) \succeq 6\left( \sum_{U \in \mathcal{I}_{mid}} \sum_{\gamma \in \Gamma_{U,*}} \frac{d_{Id_U}(H'_\gamma, H_{Id_U})}{|Aut(U)|c(\gamma)} \right) Id_{sym}$$

*then with probability at least $1 - \varepsilon$, $\Lambda \succeq 0$.*

Throughout this section, we assume that we have functions $B_{norm}(\alpha)$, $B(\gamma)$, $N(\gamma)$, and $c(\alpha)$. If $\forall \alpha \in \mathcal{M}', ||M_\alpha|| \leq B_{norm}(\alpha)$ then we say that the norm bounds hold. For the other properties of these functions, we will either restate these properties in our intermediate results to highlight where these properties are needed or just state that the conditions on these functions are satisfied for brevity.

## G.1 Warm-up: Analysis with no intersection terms

In this subsection, we show how the analysis works if we ignore the difference between $M^{fact}$ and $M^{orth}$

**Theorem G.2.** *For all $\varepsilon' \in (0, \frac{1}{2}]$, if the norm bounds hold and the following conditions hold*

1. *For all $U \in \mathcal{I}_{mid}$, $H_{Id_U} \succeq 0$*

2. *For all $U \in \mathcal{I}_{mid}$ and all $\tau \in \mathcal{M}_U$*

$$\begin{bmatrix} \frac{1}{|Aut(U)|c(\tau)}H_{Id_U} & B_{norm}(\tau)H_\tau \\ B_{norm}(\tau)H_\tau^T & \frac{1}{|Aut(U)|c(\tau)}H_{Id_U} \end{bmatrix} \succeq 0$$

3. *$\forall U \in \mathcal{I}_{mid}, \sum_{\tau \in \mathcal{M}_U} \frac{1}{|Aut(U)|c(\tau)} \leq \varepsilon'$.*

*then*

$$\sum_{U \in \mathcal{I}_{mid}} M_{Id_U}^{fact}(H_{Id_U}) + \sum_{U \in \mathcal{I}_{mid}} \sum_{\tau \in \mathcal{M}_U} M_\tau^{fact}(H_\tau) \succeq (1 - 2\varepsilon') \sum_{U \in \mathcal{I}_{mid}} M_{Id_U}^{fact}(H_{Id_U}) \succeq 0$$

*Proof.* We first show how a single term $M_\sigma M_\tau M_{\sigma'^T}$ plus its transpose $M_{\sigma'} M_{\tau^T} M_{\sigma^T}$ can be bounded.

**Lemma G.3.** *If the norm bounds hold then for all $\tau \in \mathcal{M}'$ and shapes $\sigma, \sigma'$ such that $\sigma, \tau, \sigma'^T$ are composable, for all $a, b$ such that $a > 0$, $b > 0$, and $ab = B_{norm}(\tau)^2$,*

$$M_\sigma M_\tau M_{\sigma'^T} + M_{\sigma'} M_{\tau^T} M_{\sigma^T} \preceq a M_\sigma M_{\sigma^T} + b M_{\sigma'} M_{\sigma'^T}$$

*Proof.* Observe that

$$0 \preceq \left( \sqrt{a}M_\sigma - \frac{\sqrt{b}}{B_{norm}(\tau)}M_{\sigma'}M_{\tau^T} \right)\left( \sqrt{a}M_\sigma - \frac{\sqrt{b}}{B_{norm}(\tau)}M_{\sigma'}M_{\tau^T} \right)^T =$$

$$a M_\sigma M_{\sigma^T} - M_\sigma M_\tau M_{\sigma'^T} - M_{\sigma'} M_{\tau^T} M_{\sigma^T} + \frac{b}{B_{norm}(\tau)^2} M_{\sigma'} M_{\tau^T} M_\tau M_{\sigma'^T} \preceq$$

$$a M_\sigma M_{\sigma^T} - M_\sigma M_\tau M_{\sigma'^T} - M_{\sigma'} M_{\tau^T} M_{\sigma^T} + \frac{b}{B_{norm}(\tau)^2} M_{\sigma'}(B_{norm}(\tau)^2 Id)M_{\sigma'^T}$$

Thus, $M_\sigma M_\tau M_{\sigma'^T} + M_{\sigma'} M_{\tau^T} M_{\sigma^T} \preceq a M_\sigma M_{\sigma^T} + b M_{\sigma'} M_{\sigma'^T}$, as needed. ∎

Unfortunately, if we try to bound everything term by term, there may be too many terms to bound. Instead, we generalize this argument for vectors and coefficient matrices.

**Definition G.4.** *Let $\tau$ be a shape. We say that a vector $v$ is a left $\tau$-vector if the coordinates of $v$ are indexed by left shapes $\sigma \in \mathcal{L}_{U_\tau}$. We say that a vector $w$ is a right $\tau$-vector if the coordinates of $w$ are indexed by left shapes $\sigma' \in \mathcal{L}_{V_\tau}$.*

**Lemma G.5.** *For all $\tau \in \mathcal{M}'$, if the norm bounds hold, $v$ is a left $\tau$-vector, and $w$ is a right $\tau$-vector then*
$$M_\tau^{fact}(vw^T) + M_{\tau^T}^{fact}(wv^T) \preceq B_{norm}(\tau)\left(M_{Id_{U_\tau}}^{fact}(vv^T) + M_{Id_{V_\tau}}^{fact}(ww^T)\right)$$
*and*
$$-M_\tau^{fact}(vw^T) - M_{\tau^T}^{fact}(wv^T) \preceq B_{norm}(\tau)\left(M_{Id_{U_\tau}}^{fact}(vv^T) + M_{Id_{V_\tau}}^{fact}(ww^T)\right)$$

*Proof.* Observe that

$$0 \preceq \left(\sum_\sigma v_\sigma M_\sigma \mp \frac{w_\sigma M_\sigma M_{\tau^T}}{B_{norm}(\tau)}\right)\left(\sum_{\sigma'} v_{\sigma'} M_{\sigma'} \mp \frac{w_{\sigma'} M_{\sigma'} M_{\tau^T}}{B_{norm}(\tau)}\right)^T =$$

$$\sum_{\sigma,\sigma'}(v_\sigma v_{\sigma'})M_\sigma M_{\sigma'^T} \mp \sum_{\sigma,\sigma'}\frac{(v_\sigma w_{\sigma'})}{B_{norm}(\tau)}M_\sigma M_\tau M_{\sigma'}$$

$$\mp \sum_{\sigma,\sigma'}\frac{(w_\sigma v_{\sigma'})}{B_{norm}(\tau)}M_\sigma M_{\tau^T} M_{\sigma'} + \frac{1}{B_{norm}(\tau)^2}\sum_{\sigma,\sigma'}(v_\sigma v_{\sigma'})M_\sigma M_\tau M_{\tau^T} M_{\sigma'^T}$$

Further observe that $\sum_{\sigma,\sigma'}(v_\sigma v_{\sigma'})M_\sigma M_{\sigma'^T} = M_{Id_{U_\tau}}^{fact}(vv^T)$, $\quad \sum_{\sigma,\sigma'}(v_\sigma w_{\sigma'})M_\sigma M_\tau M_{\sigma'^T} = M_\tau^{fact}(vw^T)$, $\quad \sum_{\sigma,\sigma'}(w_\sigma v_{\sigma'})M_\sigma M_{\tau^T} M_{\sigma'^T} = M_{\tau^T}^{fact}(wv^T)$ and

$$\sum_{\sigma,\sigma'}(w_\sigma w_{\sigma'})M_\sigma M_\tau M_{\tau^T} M_{\sigma'^T} = \left(\sum_\sigma w_\sigma M_\sigma\right)M_\tau M_{\tau^T}\left(\sum_\sigma w_\sigma M_\sigma\right)^T$$

$$\preceq \left(\sum_\sigma w_\sigma M_\sigma\right)B_{norm}(\tau)^2 Id\left(\sum_\sigma w_\sigma M_\sigma\right)^T$$

$$= B_{norm}(\tau)^2 M_{Id_{V_\tau}}^{fact}(ww^T)$$

Putting everything together implies the result. ∎

**Corollary G.6.** *For all $\tau \in \mathcal{M}'$, if the norm bounds hold and $H_U$ and $H_V$ are matrices such that*
$$\begin{bmatrix} H_U & B_{norm}(\tau)H_\tau \\ B_{norm}(\tau)H_\tau^T & H_V \end{bmatrix} \succeq 0$$
*then $M_\tau^{fact}(H_\tau) + M_{\tau^T}^{fact}(H_{\tau^T}) \preceq M_{Id_{U_\tau}}^{fact}(H_U) + M_{Id_{V_\tau}}^{fact}(H_V)$*

*Proof.* If $\begin{bmatrix} H_U & B_{norm}(\tau)H_\tau \\ B_{norm}(\tau)H_\tau^T & H_V \end{bmatrix} \succeq 0$ then we can write it as $\sum_i (v_i, w_i)(v_i, w_i)^T$. Since the $M^{fact}$ operations are linear, the result now follows by summing the equation
$$M_\tau^{fact}(v_i w_i^T) + M_{\tau^T}^{fact}(w_i v_i^T) \preceq B_{norm}(\tau)\left(M_{Id_{U_\tau}}^{fact}(v_i v_i^T) + M_{Id_{V_\tau}}^{fact}(w_i w_i^T)\right)$$
over all $i$. ∎

Theorem G.2 now follows directly. For all $U \in \mathcal{I}_{mid}$ and all $\tau \in \mathcal{M}_U$, using Corollary G.6 with $H_U = H_V = \frac{1}{|Aut(U)|c(\tau)}H_{Id_U}$,
$$M_\tau^{fact}(H_\tau) + M_{\tau^T}^{fact}(H_{\tau^T}) \preceq \frac{1}{|Aut(U)|c(\tau)}M_{Id_U}^{fact}(H_{Id_U}) + \frac{1}{|Aut(U)|c(\tau)}M_{Id_U}^{fact}(H_{Id_U})$$
Summing this equation over all $U \in \mathcal{I}_{mid}$ and all $\tau \in \mathcal{M}_U$, we obtain that
$$\sum_{U \in \mathcal{I}_{mid}}\sum_{\tau \in \mathcal{M}_U} M_\tau^{fact}(H_\tau) \preceq 2\varepsilon' \sum_{U \in \mathcal{I}_{mid}} M_{Id_U}^{fact}(H_{Id_U})$$
as needed. ∎

## G.2 Intersection Term Analysis Strategy

As we saw in the previous subsection, the analysis works out nicely if we work with $M^{fact}$. Unfortunately, our matrices are expressed in terms of $M^{orth}$. In this subsection, we describe our strategy for analyzing the difference between $M^{fact}$ and $M^{orth}$.

Recall the following expressions for $\left(M^{fact}_\tau(H)\right)(A, B)$ and $\left(M^{orth}_\tau(H)\right)(A, B)$ where $A$ has shape $U_\tau$ and $B$ has shape $V_\tau$:

$$\left(M^{fact}_\tau(H)\right)(A, B) = \sum_{\sigma\in\mathcal{L}_{U_\tau},\sigma'\in\mathcal{L}_{V_\tau}} H(\sigma,\sigma') \sum_{A',B'} \sum_{\substack{R_1\in\mathcal{R}(\sigma,A,A'),R_2\in\mathcal{R}(\tau,A',B'),\\ R_3\in\mathcal{R}(\sigma'^T,B',B)}} M_{R_1}(A, A')M_{R_2}(A', B')M_{R_3}(B', B)$$

$$\left(M^{orth}_\tau(H)\right)(A, B)$$
$$= \sum_{\sigma\in\mathcal{L}_{U_\tau},\sigma'\in\mathcal{L}_{V_\tau}} H(\sigma,\sigma') \sum_{A',B'} \sum_{\substack{R_1\in\mathcal{R}(\sigma,A,A'),R_2\in\mathcal{R}(\tau,A',B'),\\ R_3\in\mathcal{R}(\sigma'^T,B',B),R_1,R_2,R_3\text{ are properly composable}}} M_{R_1}(A, A')M_{R_2}(A', B')M_{R_3}(B', B)$$

This implies that $\left(M^{fact}_\tau(H)\right)(A, B) - \left(M^{orth}_\tau(H)\right)(A, B)$ is equal to

$$\sum_{\sigma\in\mathcal{L}_{U_\tau},\sigma'\in\mathcal{L}_{V_\tau}} H(\sigma,\sigma') \sum_{A',B'} \sum_{\substack{R_1\in\mathcal{R}(\sigma,A,A'),R_2\in\mathcal{R}(\tau,A',B'),\text{ and }R_3\in\mathcal{R}(\sigma'^T,B',B)\\ R_1,R_2,R_3\text{ are not properly composable}}} M_{R_1}(A, A')M_{R_2}(A', B')M_{R_3}(B', B)$$

Thus, to understand the difference between $M^{fact}$ and $M^{orth}$, we need to analyze the terms $\chi_{R_1}\chi_{R_2}\chi_{R_3} = \chi_{R_1\circ R_2\circ R_3}$ for ribbons $R_1, R_2, R_3$ which are composable but not properly composable. These terms, which we call intersection terms, are not negligible and must be analyzed carefully. In particular, we decompose each resulting ribbon $R = R_1 \circ R_2 \circ R_3$ into new left, middle, and right parts. We do this as follows:

1. Let $V_*$ be the set of vertices which appear more than once in $V(R_1 \circ R_2 \circ R_3)$. In other words, $V_*$ is the set of vertices involved in the intersections between $R_1$, $R_2$, and $R_3$ (not counting the facts that $B_{R_1} = A_{R_2}$ and $B_{R_2} = A_{R_3}$ because we expect these intersections).

2. Let $A'$ be the leftmost minimum vertex separator of $A_{R_1}$ and $B_{R_1} \cup V_*$ in $R_1$. We turn $A'$ into a matrix index by specifying an ordering $O_{A'}$ for the vertices in $A'$.

3. Let $B'$ be the leftmost minimum vertex separator of $A_{R_3} \cup V_*$ and $B_{R_3}$ in $R_2$. We turn $B'$ into a matrix index by specifying an ordering $O_{B'}$ for the vertices in $B'$.

4. Decompose $R_1$ as $R_1 = R'_1 \cup R_4$ where $R'_1$ is the part of $R_1$ between $A_{R_1}$ and $A'$ and $R_4$ is the part of $R_1$ between $B'$ and $B_{R_1} = A_{R_2}$. Similarly, decompose $R_3$ as $R_3 = R_5 \cup R'_3$ where $R_5$ is the part of $R_3$ between $B_{R_1} = A_{R_2}$ and $B'$ and $R'_3$ is the part of $R_3$ between $B'$ and $B_{R_3}$.

5. Take $R'_2 = R_4 \circ R_2 \circ R_5$ and note that $R'_1 \circ R'_2 \circ R'_3 = R_1 \circ R_2 \circ R_3$. We view $R'_1, R'_2, R'_3$ as the left, middle, and right parts of $R = R_1 \circ R_2 \circ R_3$

While we will verify our analysis by checking the coefficients of the ribbons, we want to express everything in terms of shapes. We use the following conventions for the names of the shapes:

1. As usual, we let $\sigma$, $\tau$, and $\sigma'^T$ be the shapes of $R_1$, $R_2$, and $R_3$.

2. We let $\gamma$ and $\gamma'^T$ be the shapes of $R_4$ and $R_5$.

3. We let $\sigma_2$, $\tau_P$, and $\sigma_2'^T$ be the shapes of $R'_1$, $R'_2$, and $R'_3$. Here $P$ is the intersection pattern induced by $R_4$, $R_2$, and $R_5$ which we define in the next subsection.

**Remark G.7.** *A key feature of our analysis is that it will work the same way regardless of the shapes $\sigma_2, \sigma_2'^T$ of $R'_1$ and $R'_3$. In other words, if we replace $\sigma_2$ by $\sigma_{2a}$ and $\sigma'_2$ by $\sigma'_{2a}$ for a given intersection term, this just replaces $\sigma = \sigma_2 \cup \gamma$ with $\sigma_a = \sigma_{2a} \cup \gamma$ and $\sigma' = \sigma'_2 \cup \gamma'$ with $\sigma'_a = \sigma'_{2a} \cup \gamma'$. This allows us to focus on the shapes $\gamma, \tau,$ and $\gamma'^T$ and is the reason why the $-\gamma, \gamma$ operation appears in our results.*

## G.3 Intersection Term Analysis

In this section, we implement our strategy for analyzing intersection terms. We begin by defining intersection patterns which describe how the ribbons $R_1$, $R_2$, and $R_3$ intersect.

**Definition G.8** (Rough Definition of Intersection Patterns). *Given $\tau \in \mathcal{M}'$, $\gamma \in \Gamma_{*,U_\tau} \cup \{Id_{U_\tau}\}$, $\gamma' \in \Gamma_{*,V_\tau} \cup \{Id_{V_\tau}\}$, and ribbons $R_1$, $R_2$, and $R_3$ of shapes $\gamma$, $\tau$, and $\gamma'^T$ which are composable but not properly composable, we define the intersection pattern $P$ induced by $R_1$, $R_2$, and $R_3$ and the resulting shape $\tau_P$ as follows:*

1. *We take $V(P) = V(\gamma \circ \tau \circ \gamma'^T)$.*

2. *We take $E(P)$ to be the set of edges $(u,v)$ such that $u,v$ are distinct vertices in $V(\sigma \circ \tau \circ \sigma'^T)$ but $u$ and $v$ correspond to the same vertex in $R_1 \circ R_2 \circ R_3$*

3. *We define $\tau_P$ to be the shape of the ribbon $R = R_1 \circ R_2 \circ R_3$*

**Definition G.9.** *Given $\tau \in \mathcal{M}'$, $\gamma \in \Gamma_{*,U_\tau} \cup \{Id_{U_\tau}\}$, and $\gamma' \in \Gamma_{*,V_\tau} \cup \{Id_{V_\tau}\}$, we define $\mathcal{P}_{\gamma,\tau,\gamma'^T}$ to be the set of all possible intersection patterns $P$ which can be induced by ribbons $R_1$, $R_2$, and $R_3$ of shapes $\gamma$, $\tau$, and $\gamma'^T$.*

**Remark G.10.** *Note that if $\gamma = Id_{U_\tau}$ and $\gamma' = Id_{V_\tau}$ then $\mathcal{P}_{\gamma,\tau,\gamma'^T} = \emptyset$ as every intersection pattern must have an unexpected intersection so either $\gamma$ or $\gamma'$ must be non-trivial.*

It would be nice if the intersection pattern $P$ together with the ribbon $R$ allowed us to recover the original ribbons $R_1$, $R_2$, and $R_3$. Unfortunately, it is possible for different triples of ribbons to result in the same intersection pattern $P$ and ribbon $R$. That said, the number of such triples cannot be too large, and this is sufficient for our purposes.

**Definition G.11.** *Given an intersection pattern $P \in \mathcal{P}_{\gamma,\tau,\gamma'^T}$, let $R$ be a ribbon of shape $\tau_P$. We define $N(P)$ to be the number of different triples of ribbons $R_1$, $R_2$, $R_3$ such that $R_1 \circ R_2 \circ R_3 = R$ and $R_1$, $R_2$, $R_3$ induce the intersection pattern $P$.*

**Lemma G.12.** *For all intersection patterns $P \in \mathcal{P}_{\gamma,\tau,\gamma'^T}$, $N(P) \leq |V(\tau_P)|^{|V(\gamma) \setminus U_\gamma| + |V(\gamma') \setminus U_{\gamma'}|}$*

*Proof sketch.* This can be proved by making the observations that $A_{R_1} = A_R$ and $B_{R_3} = B_R$, all of the remaining vertices in $V(R_1)$ and $V(R_3)$ must be equal to some vertex in $V(R)$, and once $R_1$ and $R_3$ are determined, there is at most one ribbon $R_2$ such that $R_1, R_2, R_3$ are composable, $R = R_1 \circ R_2 \circ R_3$, and $R_1, R_2, R_3$ induce the intersection pattern $P$. ∎

With these definitions, we can now analyze the intersection terms.

**Definition G.13.** *Given a left shape $\sigma$, define $e_\sigma$ to be the vector which has a 1 in coordinate $\sigma$ and has a 0 in all other coordinates.*

**Lemma G.14.** *For all $\tau \in \mathcal{M}'$, $\sigma \in \mathcal{L}_{U_\tau}$, and $\sigma' \in \mathcal{L}_{V_\tau}$,*

$$M_\tau^{fact}(e_\sigma e_{\sigma'}^T) - M_\tau^{orth}(e_\sigma e_{\sigma'}^T) = \sum_{\sigma_2 \in \mathcal{L}, \gamma \in \Gamma: \sigma_2 \circ \gamma = \sigma} \frac{1}{|Aut(U_\gamma)|} \sum_{P \in \mathcal{P}_{\gamma,\tau,Id_{V_\tau}}} N(P) M_{\tau_P}^{orth}(e_{\sigma_2} e_{\sigma'}^T)$$

$$+ \sum_{\sigma_2' \in \mathcal{L}, \gamma' \in \Gamma: \sigma_2' \circ \gamma' = \sigma'} \frac{1}{|Aut(U_{\gamma'})|} \sum_{P \in \mathcal{P}_{Id_{U_\tau},\tau,\gamma'^T}} N(P) M_{\tau_P}^{orth}(e_\sigma e_{\sigma_2'}^T)$$

$$+ \sum_{\sigma_2 \in \mathcal{L}, \gamma \in \Gamma: \sigma_2 \circ \gamma = \sigma} \sum_{\sigma_2' \in \mathcal{L}, \gamma' \in \Gamma: \sigma_2' \circ \gamma' = \sigma'} \frac{1}{|Aut(U_\gamma)| \cdot |Aut(U_{\gamma'})|} \sum_{P \in \mathcal{P}_{\gamma,\tau,\gamma'^T}} N(P) M_{\tau_P}^{orth}(e_{\sigma_2} e_{\sigma_2'}^T)$$

*Proof sketch.* This lemma follows from the following bijection. Consider the third term

$$\sum_{\sigma_2 \in \mathcal{L}, \gamma \in \Gamma: \sigma_2 \circ \gamma = \sigma} \sum_{\sigma_2' \in \mathcal{L}, \gamma' \in \Gamma: \sigma_2' \circ \gamma' = \sigma'} \frac{1}{|Aut(U_\gamma)| \cdot |Aut(U_{\gamma'})|} \sum_{P \in \mathcal{P}_{\gamma,\tau,\gamma'^T}} N(P) M_{\tau_P}^{orth}(e_{\sigma_2} e_{\sigma_2'}^T)$$

On one side, we have the following data:

1. Ribbons $R_1$, $R_2$, and $R_3$ of shapes $\gamma, \tau, \gamma'^T$ such that $R_1, R_2, R_3$ are composable but $R_1$ and $R_2 \circ R_3$ are not properly composable (i.e. $R_1$ has an unexpected intersection with $R_2$ and/or $R_3$) and $R_1 \circ R_2$ and $R_3$ are not properly composable (i.e. $R_3$ has an unexpected intersection with $R_1$ and/or $R_2$).

2. An ordering $O_{A'}$ on the leftmost minimum vertex separator $A'$ of $A_{R_1}$ and $V_* \cup B_{R_1}$ (recall that $V_*$ is the set of vertices which appear more than once in $V(R_1 \circ R_2 \circ R_3)$).

3. An ordering $O_{B'}$ on the rightmost minimum vertex separator $B'$ of $V_* \cup A_{R_3}$ and $B_{R_3}$.

On the other side, we have the following data

1. An intersection pattern $P \in \mathcal{P}_{\gamma, \tau, \gamma'^T}$ where $\gamma$ and $\gamma'^T$ are non-trivial.

2. Ribbons $R_1'$, $R_2'$, $R_3'$ of shapes $\sigma_2, \tau_P, \sigma_2'^T$ which are properly composable

3. A number in $[N(P)]$ describing which possible triple of ribbons resulted in the intersection pattern $P$ and the ribbon $R_2'$.

To see this bijection, note that given the data on the first side, we can recover the ribbons $R_1'$, $R_2'$, and $R_3'$ as follows:

1. We decompose $R_1$ as $R_1 = R_1' \circ R_4$ where $B_{R_1'} = A_{R_4} = A'$ with the ordering $O_{A'}$.

2. We decompose $R_3$ as $R_3 = R_5 \circ R_3'$ where where $B_{R_5} = A_{R_3'} = B'$ with the ordering $O_{B'}$.

3. We take $R_2' = R_4 \circ R_2 \circ R_5$.

The intersection pattern $P$ and the number in $[N(P)]$ can be obtained from $R_1$, $R_2$, and $R_3$.

Conversely, with the data on the other side, we can recover the data on the first side as follows:

1. $R_2'$ gives an ordering $O_{A'}$ for $A' = A_{R_2'}$ and an ordering $O_{B'}$ for $B' = B_{R_2'}$.

2. The ribbon $R_2'$, intersection pattern $P$, and number in $[N(P)]$ allow us to recover $R_4$, $R_2$, and $R_5$.

3. We take $R_1 = R_1' \circ R_4$ and $R_3 = R_5 \circ R_3'$.

Thus, both sides have the same coefficient for each ribbon.

The analysis for the the first term is the same except that when $\gamma'$ is trivial, we always take $\gamma' = Id_{V_\tau}$. Thus, we always have that $B' = B_{R_2'} = B_{R_2}$ (with the same ordering) and $R_3' = R_3 = Id_{B'}$. Because of this, there is no need to specify $R_3$, $R_3'$, $R_5$, or an ordering on $B'$.

Similarly, the analysis for the the second term is the same except that when $\gamma$ is trivial, we always take $\gamma = Id_{U_\tau}$. Thus, we always have that $A' = A_{R_2'} = A_{R_2}$ (with the same ordering) and $R_1' = R_1 = Id_{A'}$. Because of this, there is no need to specify $R_1$, $R_1'$, $R_4$, or an ordering on $A'$. ∎

Applying Lemma G.14 for all $\sigma$ and $\sigma'$ simultaneously, we obtain the following corollary.

**Definition G.15.** *For all $U, V \in \mathcal{I}_{mid}$, given a $\gamma \in \Gamma_{U,V}$ and a vector $v$ indexed by left shapes $\sigma \in \mathcal{L}_V$, define $v^{-\gamma}$ to be the vector indexed by left shapes $\sigma_2 \in \mathcal{L}_U$ such that $v^{-\gamma}(\sigma_2) = v(\sigma_2 \circ \gamma)$ if $\sigma_2 \circ \gamma \in \mathcal{L}_V$ and $v^{-\gamma}(\sigma_2) = 0$ otherwise.*

**Proposition G.16.** *For all composable $\gamma_2, \gamma_1 \in \Gamma$ and all vectors $v$ indexed by left shapes in $\mathcal{L}_{V_{\gamma_1}}$, $\left(v^{-\gamma_1}\right)^{-\gamma_2} = v^{-\gamma_2 \circ \gamma_1}$*

**Corollary G.17.** *For all $\tau \in \mathcal{M}'$, for all left $\tau$-vectors $v$ and all right $\tau$-vectors $w$,*

$$M_\tau^{orth}(vw^T) = M_\tau^{fact}(vw^T) - \sum_{\gamma \in \Gamma_{*,U_\tau}} \frac{1}{|Aut(U_\gamma)|} \sum_{P \in \mathcal{P}_{\gamma,\tau,Id_{V_\tau}}} N(P) M_{\tau_P}^{orth}(v^{-\gamma}w^T)$$

$$- \sum_{\gamma' \in \Gamma_{*,V_\tau}} \frac{1}{|Aut(U_{\gamma'})|} \sum_{P \in \mathcal{P}_{Id_{U_\tau},\tau,\gamma'^T}} N(P) M_{\tau_P}^{orth}(v(w^{-\gamma})^T)$$

$$- \sum_{\gamma \in \Gamma_{*,U_\tau}} \sum_{\gamma' \in \Gamma_{*,V_\tau}} \frac{1}{|Aut(U_\gamma)| \cdot |Aut(U_{\gamma'})|} \sum_{P \in \mathcal{P}_{\gamma,\tau,\gamma'^T}} N(P) M_{\tau_P}^{orth}(v^{-\gamma}(w^{-\gamma'})^T)$$

Applying Corollary G.17 iteratively, we obtain the following theorem:

**Definition G.18.** *Given $\gamma, \gamma' \in \Gamma \cup \{Id_U : U \in \mathcal{I}_{mid}\}$ and $j > 0$, let $\Gamma_{\gamma,\gamma',j}$ be the set of all $\gamma_1, \gamma'_1, \cdots, \gamma_j, \gamma'_j \in \Gamma \cup \{Id_U : U \in \mathcal{I}_{mid}\}$ such that:*

1. *$\gamma_j, \ldots, \gamma_1$ are composable and $\gamma_j \circ \ldots \circ \gamma_1 = \gamma$*

2. *$\gamma'_j, \ldots, \gamma'_1$ are composable and $\gamma'_j \circ \ldots \circ \gamma'_1 = \gamma'$*

3. *For all $i \in [1, j]$, $\gamma_i$ or $\gamma'_i$ is non-trivial (i.e. $\gamma_i \neq Id_{U_{\gamma_i}}$ or $\gamma'_i \neq Id_{U_{\gamma'_i}}$).*

**Remark G.19.** *Note that if $\gamma = Id_U$ and $\gamma' = Id_V$ then for all $j > 0$, $\Gamma_{\gamma,\gamma',j} = \emptyset$.*

**Theorem G.20.** *For all $\tau \in \mathcal{M}'$, left $\tau$-vectors $v$, and right $\tau$-vectors $w$,*

$$M_\tau^{orth}(vw^T) = M_\tau^{fact}(vw^T) +$$

$$\sum_{\substack{\gamma \in \Gamma_{*,U_\tau} \cup \{Id_{U_\tau}\}, \gamma' \in \Gamma_{*,V_\tau} \cup \{Id_{V_\tau}\}: \\ \gamma \text{ or } \gamma' \text{ is non-trivial}}} \sum_{j > 0} (-1)^j \sum_{\gamma_1,\gamma'_1,\cdots,\gamma_j,\gamma'_j \in \Gamma_{\gamma,\gamma',j}} \prod_{i:\gamma_i \text{ is non-trivial}} \frac{1}{|Aut(U_{\gamma_i})|} \prod_{i:\gamma'_i \text{ is non-trivial}} \frac{1}{|Aut(U_{\gamma'_i})|}$$

$$\sum_{P_1,\cdots,P_j:P_i \in \mathcal{P}_{\gamma_i,\tau_{P_{i-1}},\gamma'^T_i}} \left( \prod_{i=1}^{j} N(P_i) \right) M_{\tau_{P_j}}^{fact}(v^{-\gamma}(w^{-\gamma'})^T)$$

*where we take $\tau_{P_0} = \tau$.*

## G.4  Bounding the difference between $M^{fact}$ and $M^{orth}$

In this subsection, we bound the difference between $M_\tau^{fact}(H_\tau)$ and $M_\tau^{orth}(H_\tau)$. Recall conditions $2, 5, 7$ of Theorem G.1 for $B(\gamma)$, $N(\gamma)$, and $c(\gamma)$. With these conditions, we can now bound the difference between $M^{fact}$ and $M^{orth}$.

**Lemma G.21.** *If the norm bounds and the conditions on $B(\gamma)$, $N(\gamma)$, and $c(\gamma)$ hold then for all $\tau \in \mathcal{M}'$, left $\tau$-vectors $v$, and right $\tau$-vectors $w$,*

$$\left( M_\tau^{fact}(vw^T) + M_{\tau^T}^{fact}(wv^T) \right) - \left( M_\tau^{orth}(vw^T) + M_{\tau^T}^{orth}(wv^T) \right) \preceq$$

$$\varepsilon' B_{norm}(\tau) M_{Id_{U_\tau}}^{fact}(vv^T) + 2 \sum_{\gamma \in \Gamma_{*,U_\tau}} \frac{B(\gamma)^2 N(\gamma)^2 B_{norm}(\tau) c(\gamma)}{|Aut(U_\gamma)|} M_{Id_{U_\gamma}}^{fact}(v^{-\gamma}(v^{-\gamma})^T) +$$

$$\varepsilon' B_{norm}(\tau) M_{Id_{V_\tau}}^{fact}(ww^T) + 2 \sum_{\gamma' \in \Gamma_{*,V_\tau}} \frac{B(\gamma')^2 N(\gamma')^2 B_{norm}(\tau) c(\gamma')}{|Aut(U_{\gamma'})|} M_{Id_{U_{\gamma'}}}^{fact}(w^{-\gamma'}(w^{-\gamma'})^T)$$

*Proof.* By Theorem G.20, taking $\tau_{P_0} = \tau$,

$$M_\tau^{orth}(vw^T) = M_\tau^{fact}(vw^T)+$$

$$\sum_{\substack{\gamma \in \Gamma_{*,U_\tau} \cup \{Id_{U_\tau}\}, \gamma' \in \Gamma_{*,V_\tau} \cup \{Id_{V_\tau}\}: \\ \gamma \text{ or } \gamma' \text{ is non-trivial}}} \sum_{j>0} (-1)^j \sum_{\gamma_1,\gamma_1',\cdots,\gamma_j,\gamma_j' \in \Gamma_{\gamma,\gamma',j}} \prod_{i:\gamma_i \text{ is non-trivial}} \frac{1}{|Aut(U_{\gamma_i})|} \prod_{i:\gamma_i' \text{ is non-trivial}} \frac{1}{|Aut(U_{\gamma_i'})|}$$

$$\sum_{P_1,\cdots,P_j:P_i \in \mathcal{P}_{\gamma_i,\tau_{P_{i-1}},\gamma_i'^T}} \left(\prod_{i=1}^j N(P_i)\right) M_{\tau_{P_j}}^{fact}(v^{-\gamma}(w^{-\gamma'})^T)$$

Taking the transpose of this equation gives

$$M_{\tau^T}^{orth}(wv^T) = M_{\tau^T}^{fact}(wv^T)+$$

$$\sum_{\substack{\gamma \in \Gamma_{*,U_\tau} \cup \{Id_{U_\tau}\}, \gamma' \in \Gamma_{*,V_\tau} \cup \{Id_{V_\tau}\}: \\ \gamma \text{ or } \gamma' \text{ is non-trivial}}} \sum_{j>0} (-1)^j \sum_{\gamma_1,\gamma_1',\cdots,\gamma_j,\gamma_j' \in \Gamma_{\gamma,\gamma',j}} \prod_{i:\gamma_i \text{ is non-trivial}} \frac{1}{|Aut(U_{\gamma_i})|} \prod_{i:\gamma_i' \text{ is non-trivial}} \frac{1}{|Aut(U_{\gamma_i'})|}$$

$$\sum_{P_1,\cdots,P_j:P_i \in \mathcal{P}_{\gamma_i,\tau_{P_{i-1}},\gamma_i'^T}} \left(\prod_{i=1}^j N(P_i)\right) M_{\tau_{P_j}^T}^{fact}(w^{-\gamma'}(v^{-\gamma})^T)$$

Now observe that by Lemma G.5, if the norm bounds hold,

$$\pm \left( M_{\tau_{P_j}}^{fact}(v^{-\gamma}(w^{-\gamma'})^T) + M_{\tau_{P_j}^T}^{fact}(w^{-\gamma'}(v^{-\gamma})^T)\right) =$$

$$\pm M_{\tau_{P_j}}^{fact}\left(\left(\sqrt{\frac{N(\gamma)B(\gamma)c(\gamma)}{N(\gamma')B(\gamma')c(\gamma')}}v^{-\gamma}\right)\left(\sqrt{\frac{N(\gamma')B(\gamma')c(\gamma')}{N(\gamma)B(\gamma)c(\gamma)}}(w^{-\gamma'})^T\right)\right) \pm$$

$$M_{\tau_{P_j}^T}^{fact}\left(\left(\sqrt{\frac{N(\gamma')B(\gamma')c(\gamma')}{N(\gamma)B(\gamma)c(\gamma)}}w^{-\gamma'}\right)\left(\sqrt{\frac{N(\gamma)B(\gamma)c(\gamma)}{N(\gamma')B(\gamma')c(\gamma')}}(v^{-\gamma})^T\right)\right) \preceq$$

$$B_{norm}(\tau_{P_j})\left(\frac{N(\gamma)B(\gamma)c(\gamma)}{N(\gamma')B(\gamma')c(\gamma')}M_{Id_{U_\gamma}}^{fact}(v^{-\gamma}(v^{-\gamma})^T) + \frac{N(\gamma')B(\gamma')c(\gamma')}{N(\gamma)B(\gamma)c(\gamma)}M_{Id_{U_{\gamma'}}}^{fact}(w^{-\gamma'}(w^{-\gamma'})^T)\right)$$

Combining these equations,

$$\left(M_\tau^{fact}(vw^T) + M_{\tau^T}^{fact}(wv^T)\right) - \left(M_\tau^{orth}(vw^T) + M_{\tau^T}^{orth}(wv^T)\right) \preceq$$

$$\sum_{\substack{\gamma \in \Gamma_{*,U_\tau} \cup \{Id_{U_\tau}\}, \gamma' \in \Gamma_{*,V_\tau} \cup \{Id_{V_\tau}\}: \\ \gamma \text{ or } \gamma' \text{ is non-trivial}}} \sum_{j>0} \sum_{\gamma_1,\gamma_1',\cdots,\gamma_j,\gamma_j' \in \Gamma_{\gamma,\gamma',j}} \prod_{i:\gamma_i \text{ is non-trivial}} \frac{1}{|Aut(U_{\gamma_i})|} \prod_{i:\gamma_i' \text{ is non-trivial}} \frac{1}{|Aut(U_{\gamma_i'})|}$$

$$\sum_{P_1,\cdots,P_j:P_i \in \mathcal{P}_{\gamma_i,\tau_{P_{i-1}},\gamma_i'^T}} \left(\prod_{i=1}^j N(P_i)\right) B_{norm}(\tau_{P_j})$$

$$\left(\frac{N(\gamma)B(\gamma)c(\gamma)}{N(\gamma')B(\gamma')c(\gamma')}M_{Id_{U_\gamma}}^{fact}(v^{-\gamma}(v^{-\gamma})^T) + \frac{N(\gamma')B(\gamma')c(\gamma')}{N(\gamma)B(\gamma)c(\gamma)}M_{Id_{U_{\gamma'}}}^{fact}(w^{-\gamma'}(w^{-\gamma'})^T)\right)$$

Putting these equations together,

$$\left(M_\tau^{fact}(vw^T) + M_{\tau^T}^{fact}(wv^T)\right) - \left(M_\tau^{orth}(vw^T) + M_{\tau^T}^{orth}(wv^T)\right) \preceq$$

$$\sum_{\substack{\gamma \in \Gamma_{*,U_\tau} \cup \{Id_{U_\tau}\}, \gamma' \in \Gamma_{*,V_\tau} \cup \{Id_{V_\tau}\}: \\ \gamma \text{ or } \gamma' \text{ is non-trivial}}} \frac{B(\gamma)^2 N(\gamma)^2 B_{norm}(\tau)c(\gamma)}{(|Aut(U_\gamma)|)^{1_{\gamma \text{ is non-trivial}}}(|Aut(U_{\gamma'})|)^{1_{\gamma' \text{ is non-trivial}}}c(\gamma')} M_{Id_{U_\gamma}}^{fact}(v^{-\gamma}(v^{-\gamma})^T)+$$

$$\sum_{\substack{\gamma \in \Gamma_{*,U_\tau} \cup \{Id_{U_\tau}\}, \gamma' \in \Gamma_{*,V_\tau} \cup \{Id_{V_\tau}\}: \\ \gamma \text{ or } \gamma' \text{ is non-trivial}}} \frac{B(\gamma')^2 N(\gamma')^2 B_{norm}(\tau)c(\gamma')}{(|Aut(U_\gamma)|)^{1_{\gamma \text{ is non-trivial}}}(|Aut(U_{\gamma'})|)^{1_{\gamma' \text{ is non-trivial}}}c(\gamma)} M_{Id_{U_{\gamma'}}}^{fact}(w^{-\gamma'}(w^{-\gamma'})^T)$$

Now observe that

$$\sum_{\substack{\gamma\in\Gamma_{*,U_\tau}\cup\{Id_{U_\tau}\},\,\gamma'\in\Gamma_{*,V_\tau}\cup\{Id_{V_\tau}\}:\\ \gamma \text{ or } \gamma' \text{ is non-trivial}}} \frac{B(\gamma)^2 N(\gamma)^2 B_{norm}(\tau)c(\gamma)}{(|Aut(U_\gamma)|)^{1_{\gamma \text{ is non-trivial}}}(|Aut(U_{\gamma'})|)^{1_{\gamma' \text{ is non-trivial}}}c(\gamma')} M_{Id_{U_\gamma}}^{fact}(v^{-\gamma}(v^{-\gamma})^T) \preceq$$

$$\left(\sum_{\gamma'\in\Gamma_{*,V_\tau}}\frac{1}{|Aut(U_{\gamma'})|c(\gamma')}\right)B_{norm}(\tau)M_{Id_{U_\tau}}^{fact}(vv^T)+$$

$$\sum_{\gamma\in\Gamma_{*,U_\tau}}\left(\sum_{\gamma'\in\Gamma_{*,V_\tau}\cup\{Id_{V_\tau}\}}\frac{1}{(|Aut(U_{\gamma'})|)^{1_{\gamma' \text{ is non-trivial}}}c(\gamma')}\right)\frac{B(\gamma)^2 N(\gamma)^2 B_{norm}(\tau)c(\gamma)}{(|Aut(U_\gamma)|)^{1_{\gamma \text{ is non-trivial}}}} M_{Id_{U_\gamma}}^{fact}(v^{-\gamma}(v^{-\gamma})^T) \preceq$$

$$\varepsilon' B_{norm}(\tau)M_{Id_{U_\tau}}^{fact}(vv^T) + 2\sum_{\gamma\in\Gamma_{*,U_\tau}}\frac{B(\gamma)^2 N(\gamma)^2 B_{norm}(\tau)c(\gamma)}{|Aut(U_\gamma)|}M_{Id_{U_\gamma}}^{fact}(v^{-\gamma}(v^{-\gamma})^T)$$

Following similar logic,

$$\sum_{\substack{\gamma\in\Gamma_{*,U_\tau}\cup\{Id_{U_\tau}\},\,\gamma'\in\Gamma_{*,V_\tau}\cup\{Id_{V_\tau}\}:\\ \gamma \text{ or } \gamma' \text{ is non-trivial}}} \frac{B(\gamma')^2 N(\gamma')^2 B_{norm}(\tau)c(\gamma')}{(|Aut(U_\gamma)|)^{1_{\gamma \text{ is non-trivial}}}(|Aut(U_{\gamma'})|)^{1_{\gamma' \text{ is non-trivial}}}c(\gamma)} M_{Id_{U_{\gamma'}}}^{fact}(w^{-\gamma'}(w^{-\gamma'})^T) \preceq$$

$$\varepsilon' B_{norm}(\tau)M_{Id_{V_\tau}}^{fact}(ww^T) + 2\sum_{\gamma'\in\Gamma_{*,V_\tau}}\frac{B(\gamma')^2 N(\gamma')^2 B_{norm}(\tau)c(\gamma')}{|Aut(U_{\gamma'})|}M_{Id_{U_{\gamma'}}}^{fact}(w^{-\gamma'}(w^{-\gamma'})^T)$$

Putting everything together implies the result. ∎

Using Lemma G.21 we have the following corollaries:

**Corollary G.22.** *For all $U \in \mathcal{I}_{mid}$, if the norm bounds and the conditions on $B(\gamma)$, $N(\gamma)$, and $c(\gamma)$ hold and $H_{Id_U} \succeq 0$ then*

$$M_{Id_U}^{fact}(H_{Id_U}) - M_{Id_U}^{orth}(H_{Id_U}) \preceq \varepsilon' M_{Id_U}^{fact}(H_{Id_U}) + 2\sum_{\gamma\in\Gamma_{*,U}}\frac{B(\gamma)^2 N(\gamma)^2 c(\gamma)}{|Aut(U_\gamma)|}M_{Id_{U_\gamma}}^{fact}(H_{Id_U}^{-\gamma,\gamma})$$

**Corollary G.23.** *For all $U \in \mathcal{I}_{mid}$ and all $\tau \in \mathcal{M}_U$, if the norm bounds and the conditions on $B(\gamma)$, $N(\gamma)$, and $c(\gamma)$ hold and*

$$\begin{bmatrix} \frac{1}{|Aut(U)|c(\tau)}H_{Id_U} & B_{norm}(\tau)H_\tau \\ B_{norm}(\tau)H_\tau^T & \frac{1}{|Aut(U)|c(\tau)}H_{Id_U} \end{bmatrix} \succeq 0$$

*then*

$$\left(M_\tau^{fact}(H_\tau) + M_{\tau^T}^{fact}(H_\tau^T)\right) - \left(M_\tau^{orth}(H_\tau) + M_{\tau^T}^{orth}(H_\tau^T)\right) \preceq$$

$$2\varepsilon'\frac{1}{|Aut(U)|c(\tau)}M_{Id_U}^{fact}(H_{Id_U}) + 4\sum_{\gamma\in\Gamma_{*,U}}\frac{B(\gamma)^2 N(\gamma)^2 c(\gamma)}{|Aut(U_\gamma)|\cdot|Aut(U)|c(\tau)}M_{Id_{U_\gamma}}^{fact}(H_{Id_U}^{-\gamma,\gamma})$$

### G.5 Proof of the Main Theorem

We now prove the following theorem which is a slight modification of Theorem G.1 and which implies Theorem G.1.

**Theorem G.24.** *For all $\varepsilon > 0$ and all $\varepsilon' \in (0, \frac{1}{20}]$, for any moment matrix*

$$\Lambda = \sum_{U\in\mathcal{I}_{mid}} M_{Id_U}^{orth}(H_{Id_U}) + \sum_{U\in\mathcal{I}_{mid}}\sum_{\tau\in\mathcal{M}_U} M_\tau^{orth}(H_\tau),$$

*if the parameters are $\varepsilon$-feasible and moreover, for all $\alpha \in \mathcal{M}', ||M_\alpha|| \leq B_{norm}(\alpha)$, and we have SOS-symmetric coefficient matrices $\{H_\gamma' : \gamma \in \Gamma\}$ such that the following conditions hold:*

1. For all $U \in \mathcal{I}_{mid}$, $H_{Id_U} \succeq 0$

2. For all $U \in \mathcal{I}_{mid}$ and $\tau \in \mathcal{M}_U$,

$$\begin{bmatrix} \frac{1}{|Aut(U)|c(\tau)} H_{Id_U} & B_{norm}(\tau) H_\tau \\ B_{norm}(\tau) H_\tau^T & \frac{1}{|Aut(U)|c(\tau)} H_{Id_U} \end{bmatrix} \succeq 0$$

3. For all $U, V \in \mathcal{I}_{mid}$ where $w(U) > w(V)$ and all $\gamma \in \Gamma_{U,V}$,

$$c(\gamma)^2 N(\gamma)^2 B(\gamma)^2 H_{Id_V}^{-\gamma,\gamma} \preceq H'_\gamma$$

*then*

$$\Lambda \succeq \frac{1}{2} \left( \sum_{U \in \mathcal{I}_{mid}} M_{Id_U}^{fact}(H_{Id_U}) \right) - 3 \left( \sum_{U \in \mathcal{I}_{mid}} \sum_{\gamma \in \Gamma_{U,*}} \frac{d_{Id_U}(H'_\gamma, H_{Id_U})}{|Aut(U)|c(\gamma)} \right) Id_{sym}$$

*If it is also true that*

$$\sum_{U \in \mathcal{I}_{mid}} M_{Id_U}^{fact}(H_{Id_U}) \succeq 6 \left( \sum_{U \in \mathcal{I}_{mid}} \sum_{\gamma \in \Gamma_{U,*}} \frac{d_{Id_U}(H'_\gamma, H_{Id_U})}{|Aut(U)|c(\gamma)} \right) Id_{sym}$$

*then $\Lambda \succeq 0$.*

*Proof.* We make the following observations:

1. By Theorem G.2,

$$\sum_{U \in \mathcal{I}_{mid}} M_{Id_U}^{fact}(H_{Id_U}) + \sum_{U \in \mathcal{I}_{mid}} \sum_{\tau \in \mathcal{M}_U} M_\tau^{fact}(H_\tau) \succeq (1 - 2\varepsilon') \sum_{U \in \mathcal{I}_{mid}} M_{Id_U}^{fact}(H_{Id_U})$$

2. By Corollary G.22,

$$\sum_{U \in \mathcal{I}_{mid}} \left( M_{Id_U}^{fact}(H_{Id_U}) - M_{Id_U}^{orth}(H_{Id_U}) \right) \preceq \varepsilon' \sum_{U \in \mathcal{I}_{mid}} M_{Id_U}^{fact}(H_{Id_U}) + 2 \sum_{U \in \mathcal{I}_{mid}} \sum_{\gamma \in \Gamma_{*,U}} \frac{M_{Id_{U_\gamma}}^{fact}(H'_\gamma)}{c(\gamma)|Aut(U_\gamma)|}$$

3. By Corollary G.23,

$$\sum_{U \in \mathcal{I}_{mid}} \sum_{\tau \in \mathcal{M}_U} \left( M_\tau^{fact}(H_\tau) - M_\tau^{orth}(H_\tau) \right) \preceq$$

$$\sum_{U \in \mathcal{I}_{mid}} \sum_{\tau \in \mathcal{M}_U} \left( \frac{2\varepsilon'}{|Aut(U)|c(\tau)} M_{Id_U}^{fact}(H_{Id_U}) + 4 \sum_{\gamma \in \Gamma_{*,U}} \frac{B(\gamma)^2 N(\gamma)^2 c(\gamma)}{|Aut(U_\gamma)| \cdot |Aut(U)|c(\tau)} M_{Id_{U_\gamma}}^{fact}(H_{Id_U}^{-\gamma,\gamma}) \right) \preceq$$

$$2\varepsilon'^2 \sum_{U \in \mathcal{I}_{mid}} M_{Id_U}^{fact}(H_{Id_U}) + 4\varepsilon' \sum_{U \in \mathcal{I}_{mid}} \sum_{\gamma \in \Gamma_{*,U}} \frac{M_{Id_{U_\gamma}}^{fact}(H'_\gamma)}{c(\gamma)|Aut(U_\gamma)|}$$

4.

$$\sum_{U \in \mathcal{I}_{mid}} \sum_{\gamma \in \Gamma_{*,U}} \frac{M_{Id_{U_\gamma}}^{fact}(H'_\gamma)}{c(\gamma)|Aut(U_\gamma)|} = \sum_{U \in \mathcal{I}_{mid}} \sum_{\gamma \in \Gamma_{*,U}} \frac{M_{Id_{U_\gamma}}^{fact}(H_{Id_{U_\gamma}}) + \left( M_{Id_{U_\gamma}}^{fact}(H'_\gamma) - M_{Id_{U_\gamma}}^{fact}(H_{Id_{U_\gamma}}) \right)}{c(\gamma)|Aut(U_\gamma)|} \preceq$$

$$\sum_{U \in \mathcal{I}_{mid}} \sum_{\gamma \in \Gamma_{*,U}} \frac{M_{Id_{U_\gamma}}^{fact}(H_{Id_{U_\gamma}})}{c(\gamma)|Aut(U_\gamma)|} + \left( \sum_{U \in \mathcal{I}_{mid}} \sum_{\gamma \in \Gamma_{U,*}} \frac{d_{Id_{U_\gamma}}(H'_\gamma, H_{Id_{U_\gamma}})}{|Aut(U_\gamma)|c(\gamma)} \right) Id_{sym} \preceq$$

$$\varepsilon' \sum_{U \in \mathcal{I}_{mid}} M_U^{fact}(H_{Id_U}) + \left( \sum_{U \in \mathcal{I}_{mid}} \sum_{\gamma \in \Gamma_{U,*}} \frac{d_{Id_{U_\gamma}}(H'_\gamma, H_{Id_{U_\gamma}})}{|Aut(U_\gamma)|c(\gamma)} \right) Id_{sym}$$

Putting everything together,

$$\Lambda = \sum_{U \in \mathcal{I}_{mid}} M_{Id_U}^{orth}(H_{Id_U}) + \sum_{U \in \mathcal{I}_{mid}} \sum_{\tau \in \mathcal{M}_U} M_\tau^{orth}(H_\tau) =$$

$$\sum_{U \in \mathcal{I}_{mid}} M_{Id_U}^{fact}(H_{Id_U}) + \sum_{U \in \mathcal{I}_{mid}} \sum_{\tau \in \mathcal{M}_U} M_\tau^{fact}(H_\tau) + \sum_{U \in \mathcal{I}_{mid}} \left( M_{Id_U}^{fact}(H_{Id_U}) - M_{Id_U}^{orth}(H_{Id_U}) \right) +$$

$$\sum_{U \in \mathcal{I}_{mid}} \sum_{\tau \in \mathcal{M}_U} \left( M_\tau^{fact}(H_\tau) - M_\tau^{orth}(H_\tau) \right) \succeq$$

$$(1 - 3\varepsilon' - 2\varepsilon'^2) \sum_{U \in \mathcal{I}_{mid}} M_{Id_U}^{fact}(H_{Id_U}) - (2 + 4\varepsilon') \sum_{U \in \mathcal{I}_{mid}} \sum_{\gamma \in \Gamma_{*,U}} \frac{M_{Id_{U_\gamma}}^{fact}(H'_\gamma)}{c(\gamma)|Aut(U_\gamma)|} \succeq$$

$$(1 - 5\varepsilon' - 6\varepsilon'^2) \sum_{U \in \mathcal{I}_{mid}} M_{Id_U}^{fact}(H_{Id_U}) - (2 + 4\varepsilon') \left( \sum_{U \in \mathcal{I}_{mid}} \sum_{\gamma \in \Gamma_{U,*}} \frac{d_{Id_{U_\gamma}}(H'_\gamma, H_{Id_{U_\gamma}})}{|Aut(U_\gamma)|c(\gamma)} \right) Id_{sym} \succeq$$

$$\frac{1}{2} \sum_{U \in \mathcal{I}_{mid}} M_{Id_U}^{fact}(H_{Id_U}) - 3 \left( \sum_{U \in \mathcal{I}_{mid}} \sum_{\gamma \in \Gamma_{U,*}} \frac{d_{Id_{U_\gamma}}(H'_\gamma, H_{Id_{U_\gamma}})}{|Aut(U_\gamma)|c(\gamma)} \right) Id_{sym}$$

∎

# H Choosing the functions $B_{norm}(\alpha)$, $B(\gamma)$, $N(\gamma)$, and $c(\alpha)$

In this subsection, we give functions $B_{norm}(\alpha)$, $B(\gamma)$, $N(\gamma)$, and $c(\alpha)$ which are $\varepsilon$-feasible thereby proving Lemma F.99 and Lemma F.107 and completing the proof of our main theorem.

## H.1 Choosing $B_{norm}(\alpha)$

We need matrix norm bounds which hold for all $\alpha \in \mathcal{M}'$. To obtain such norm bounds, we start with the norm bounds in the graph matrix norm bound paper. We then modify these bounds as follows:

1. We make the bounds more compatible with the conditions of our theorem. To do this, we upper bound many of the terms in the norm bound by $B_{vertex}^{|V(\alpha) \setminus U_\alpha| + |V(\alpha) \setminus V_\alpha|}$ where $B_{vertex}$ is a function of our parameters. In general, we will also need to upper bound some of the terms by $\prod_{e \in E(\alpha)} (B_{edge}(e))$ where $B_{edge}(e)$ is a function of $l_e$, $\Omega$, and our parameters.

2. We generalize the bounds so that they apply to improper shapes as well as proper shapes. Under our simplifying assumptions, all we need to do here is to take isolated vertices into account. In general, we also need to handle multi-edges.

### H.1.1 Simplified $B_{norm}(\alpha)$

Under our simplifying assumptions, we start with the following norm bound from the updated graph matrix norm bound paper [1]:

**Theorem H.1** (Simplified Graph Matrix Norm Bounds). *Under our simplifying assumptions, for all $\varepsilon > 0$ and all proper shapes $\alpha$, taking $c_\alpha = |V(\alpha) \setminus (U_\alpha \cup V_\alpha)| + |S_\alpha \setminus (U_\alpha \cap V_\alpha)|$,*

$$Pr\left( ||M_\alpha|| > (2|V_\alpha \setminus (U_\alpha \cap V_\alpha)|)^{|V(\alpha) \setminus (U_\alpha \cap V_\alpha)|} (2eq)^{\frac{c_\alpha}{2}} n^{\frac{w(V(\alpha)) - w(S_\alpha)}{2}} \right) < \varepsilon$$

*where $q = 3 \left\lceil \frac{ln(\frac{n^{w(S_\alpha)}}{\varepsilon})}{3c_\alpha} \right\rceil$*

**Corollary H.2.** *For all shapes $\alpha$ and all $\varepsilon > 0$,*

$$Pr\left( ||M_\alpha|| > \left( 2|V_\alpha| \sqrt[4]{2eq} \right)^{|V(\alpha) \setminus U_\alpha| + |V(\alpha) \setminus V_\alpha|} n^{\frac{w(V(\alpha)) + w(I_\alpha) - w(S_\alpha)}{2}} \right) < \varepsilon$$

*where $q = 3 \left\lceil \frac{ln(\frac{n^{w(S_\alpha)}}{\varepsilon})}{3c_\alpha} \right\rceil$.*

**Corollary H.3.** *For all $z \in \mathbb{N}$ and all $\varepsilon > 0$, taking $\varepsilon'' = \frac{\varepsilon}{5^z 2^{z^2}}$, with probability at least $1 - \varepsilon$ we have that for all shapes $\alpha$ such that $|V(\alpha)| \leq z$,*

$$||M_\alpha|| \leq \left( 2|V_\alpha| \sqrt[4]{2eq} \right)^{|V(\alpha) \backslash U_\alpha| + |V(\alpha) \backslash V_\alpha|} n^{\frac{w(V(\alpha)) + w(I_\alpha) - w(S_\alpha)}{2}}$$

*where $q = 3 \left\lceil \frac{ln(\frac{n^{w(S_\alpha)}}{\varepsilon''})}{3c_\alpha} \right\rceil$.*

*Proof.* This result can be proved from Corollary H.2 using a union bound and the following proposition:

**Proposition H.4.** *Under our simplifying assumptions, for all $z \in \mathbb{N}$, there are at most $5^z 2^{z^2}$ proper shapes $\alpha$ such that $V(\alpha) \leq z$.*

*Proof.* Observe that we can construct any proper shape $\alpha$ with at most $m$ vertices as follows:

1. Start with $z$ vertices $v_1, \ldots, v_z$.

2. For each vertex $v_i$, choose whether $v_i \in V(\alpha) \setminus U_\alpha \setminus V_\alpha$, $v_i \in U_\alpha \setminus V_\alpha$, $v_i \in V_\alpha \setminus U_\alpha$, $v_i \in U_\alpha \cap V_\alpha$, or $v_i \notin V(\alpha)$.

3. For each pair of vertices $v_i, v_j \in V(\alpha)$, choose whether or not $(v_i, v_j) \in E(\alpha)$

■

■

**Corollary H.5.** *For all $D_V \in \mathbb{N}$ and all $\varepsilon > 0$, taking*

$$q = 3 \left\lceil \frac{ln(\frac{5^{3D_V} 2^{9D_V^2} n^{3D_V}}{\varepsilon})}{3} \right\rceil = 3 \left\lceil D_V ln(n) + \frac{ln(\frac{1}{\varepsilon})}{3} + D_V ln(5) + 3D_V^2 ln(2) \right\rceil,$$

*$B_{vertex} = 6D_V \sqrt[4]{2eq}$, and*

$$B_{norm}(\alpha) = B_{vertex}^{|V(\alpha) \backslash U_\alpha| + |V(\alpha) \backslash V_\alpha|} n^{\frac{w(V(\alpha)) + w(I_\alpha) - w(S_\alpha)}{2}},$$

*with probability at least $(1 - \varepsilon)$ we have that for all shapes $\alpha \in \mathcal{M}'$, $||M_\alpha|| \leq B_{norm}(\alpha)$*

*Proof.* This follows from Corollary H.3 and the fact that for all $\alpha \in \mathcal{M}'$, $w(S_\alpha) \leq |V(\alpha)| \leq 3D_V$

■

### H.1.2 General $B_{norm}(\alpha)$

In general, we start with the following norm bound from the updated graph matrix norm bound paper [1]:

**Theorem H.6** (General Graph Matrix Norm Bounds). *For all $\varepsilon > 0$ and all proper shapes $\alpha$, taking $q = \lceil ln(\frac{n^{w(S_\alpha)}}{\varepsilon}) \rceil$*

$$P \left( ||M_\alpha|| > 2e(2q|V(\alpha)|)^{|V(\alpha) \backslash (U_\alpha \cap V_\alpha)|} \left( \prod_{e \in E(\alpha)} h_{l_e}^+(B_\Omega(2ql_e)) \right) n^{\frac{(w(V(\alpha)) - w(S_\alpha))}{2}} \right) < \varepsilon$$

**Corollary H.7.** *For all $\varepsilon > 0$, for all $z, l_{max}, m \in \mathbb{N}$, taking $\varepsilon'' = \frac{\varepsilon}{5^z (l_{max}+1)^{z^k}}$, with probability at least $1 - \varepsilon$, for all shapes $\alpha$ such that*

1. *$|V(\alpha)| \leq z$.*

2. *All edges in $E(\alpha)$ have label at most $l_{max}$.*

3. *All edges in $E(\alpha)$ have multiplicity at most $m$.*

,

$$||M_\alpha|| \leq 2e(2q|V(\alpha)|)^{|V(\alpha)\setminus U_\alpha|+|V(\alpha)\setminus V_\alpha|} \left( \prod_{e\in E(\alpha)} 2h_{l_e}^+(B_\Omega(2ml_{max})) \max_{j\in[0,ml_{max}]} \left\{ \left(h_j^+(B_\Omega(2qj))\right)^{\frac{l_e}{\max\{j,l_e\}}} \right\} \right)$$

$$n^{\frac{w(V(\alpha))+w(I_\alpha)-w(S_{\alpha,min})}{2}}$$

*where* $q = \left\lceil ln\left(\frac{n^{w(S_{\alpha,max})}}{\varepsilon''}\right)\right\rceil$

*Proof.* Observe that for each $\alpha$ which has multi-edges, we can write $M_\alpha = \sum_i c_i M_{\alpha_i}$ where each $\alpha_i$ has no multiple edges. We first upper bound $\sum_i |c_i|$.

**Lemma H.8.** *For any* $a_1, \ldots, a_m \in \mathbb{N} \cup \{0\}$, *taking* $p_{max} = \sum_{i=1}^m a_i$ *and writing* $\prod_{i=1}^m h_{a_i} = \sum_{k=0}^{p_{max}} c_k h_k$,

$$\sum_{k=0}^{p_{max}} |c_k| \leq (p_{max}+1) \prod_{i=1}^m h_{a_i}^+(B_\Omega(2p_{max})) \leq \prod_{i=1}^m 2h_{a_i}^+(B_\Omega(2p_{max}))$$

*Proof.* The result follows by Cauchy-Schwarz using the fact that $h_k$ form an orthonormal basis ∎

**Corollary H.9.** *For any shape* $\alpha$ *such that every edge of* $\alpha$ *has multiplicity at most* $m$ *and label at most* $l_{max}$, *if we write* $M_\alpha = \sum_i c_i M_{\alpha_i}$ *where each* $\alpha_i$ *has no multi-edges then* $\sum_i |c_i| \leq \prod_{e\in E(\alpha)} 2h_{l_e}^+(B_\Omega(2ml_{max}))$

The result now follows from Theorem H.6 and the following observations:

1. $|V(\alpha) \setminus (U_\alpha \cap V_\alpha)| \leq |V(\alpha) \setminus U_\alpha| + |V(\alpha) \setminus V_\alpha|$.

2. For any $\alpha$, writing $M_\alpha = \sum_i c_i M_{\alpha_i}$ where each $\alpha_i$ has no multi-edges, for all $\alpha_i$,

$$w(V(\alpha_i)) + w(I_{\alpha_i}) - w(S_{\alpha_i}) \leq w(V(\alpha)) + w(I_\alpha) - w(S_{\alpha,min})$$

3. For any $a_1, \ldots, a_m \in \mathbb{N} \cup \{0\}$ such that $\forall i' \in [m], a_{i'} \leq l_{max}$, for all $j \in [0, ml_{max}]$

$$h_j^+(B_\Omega(2qj)) \leq \prod_{i'=1}^m \left(h_j^+(B_\Omega(2qj))\right)^{\frac{a_{i'}}{\max\{j,a_{i'}\}}} \leq \prod_{i'=1}^m \max_{j'\in[0,ml_{max}]} \left\{ \left(h_{j'}^+(B_\Omega(2qj'))\right)^{\frac{a_{i'}}{\max\{j',a_{i'}\}}} \right\}$$

**Proposition H.10.** *For all* $z, l_{max} \in \mathbb{N}$, *there are at most* $5^z(l_{max}+1)^{z^k}$ *proper shapes* $\alpha$ *such that* $|V(\alpha)| \leq z$ *and every edge in* $E(\alpha)$.

*Proof.* This can be proved in the same way as before. ∎

∎

From this, the first condition of $\varepsilon$-feasibility follows as an easy corollary.

## H.2 Choosing $B(\gamma)$

We now describe how to choose the function $B(\gamma)$ so that conditions 2, 3 of $\varepsilon$-feasibility hold. The most important part of choosing $B(\gamma)$ is to make sure that the factors of $n$ are controlled. For this, we use the following intersection tradeoff lemma. Under our simplifying assumptions, this lemma follows from [19, Lemma 7.12]. We defer the general proof of this lemma to the end of this section.

**Lemma H.11** (Intersection Tradeoff Lemma)**.** *For all* $\gamma, \tau, \gamma'$ *and all intersection patterns* $P \in \mathcal{P}_{\gamma,\tau,\gamma'}$,

$$w(V(\tau_P)) + w(I_{\tau_P}) - w(S_{\tau_P,min}) \leq w(V(\tau)) + w(I_\tau) - w(S_{\tau,min}) + w(V(\gamma)\setminus U_\gamma) + w(V(\gamma')\setminus U_{\gamma'})$$

Based on this intersection tradeoff lemma, we can choose the function $B(\gamma)$ as follows.

**Corollary H.12.** *If we take*

$$B_{norm}(\alpha) = C \cdot B_{vertex}^{|V(\alpha)\setminus U_\alpha|+|V(\alpha)\setminus V_\alpha|} \left( \prod_{e\in E(\alpha)} B_{edge}(e) \right) n^{\frac{w(V(\alpha))+w(I_\alpha)-w(S_\alpha)}{2}}$$

*for some constant $C > 0$ and take*

$$B(\gamma) = B_{vertex}^{|V(\gamma)\setminus U_\gamma|+|V(\gamma)\setminus V_\gamma|} \left( \prod_{e\in E(\gamma)} B_{edge}(e) \right) n^{\frac{w(V(\gamma)\setminus U_\gamma)}{2}}$$

*then the 2nd and 3rd condition of $\varepsilon$-feasibility hold.*

*Proof.* We have that

$$B_{norm}(\tau_P) = B_{vertex}^{|V(\tau_P)\setminus U_{\tau_P}|+|V(\tau_P)\setminus V_{\tau_P}|} \left( \prod_{e\in E(\tau_P)} B_{edge}(e) \right) n^{\frac{w(V(\tau_P))+w(I_{\tau_P})-w(S_{\tau_P})}{2}}$$

and

$$B(\gamma)B(\gamma')B_{norm}(\tau) = B_{vertex}^{|V(\gamma)\setminus U_\gamma|+|V(\gamma)\setminus V_\gamma|+|V(\gamma')\setminus U_{\gamma'}|+|V(\gamma')\setminus V_{\gamma'}|+|V(\tau)\setminus U_\tau|+|V(\tau)\setminus V_\tau|}$$

$$\left( \prod_{e\in E(\gamma)\cup E(\gamma')\cup E(\tau)} B_{edge}(e) \right) n^{\frac{w(V(\gamma)\setminus U_\gamma)+w(V(\gamma')\setminus U_{\gamma'})+w(V(\tau))+w(I_\tau)-w(S_\tau)}{2}}$$

The first condition now follows immediately from the following observations:

1.

$$|V(\gamma) \setminus U_\gamma| + |V(\gamma) \setminus V_\gamma| + |V(\gamma') \setminus U_{\gamma'}| + |V(\gamma') \setminus V_{\gamma'}| + |V(\tau) \setminus U_\tau| + |V(\tau) \setminus V_\tau|$$
$$= |V(\gamma \circ \tau \circ \gamma'^T) \setminus U_{\gamma\circ\tau\circ\gamma'^T}| + |V(\gamma \circ \tau \circ \gamma'^T) \setminus V_{\gamma\circ\tau\circ\gamma'^T}| \geq |V(\tau_P) \setminus U_{\tau_P}| + |V(\tau_P) \setminus V_{\tau_P}|$$

2. $E(\tau_P) = E(\gamma) \cup E(\tau) \cup E(\gamma'^T)$ so $\prod_{e\in E(\tau_P)} B_{edge}(e) = \prod_{e\in E(\gamma)\cup E(\gamma')\cup E(\tau)} B_{edge}(e)$.

3. By the intersection tradeoff lemma,
$$w(V(\tau_P))+w(I_{\tau_P})-w(S_{\tau_P}) \leq w(V(\tau))+w(I_\tau)-w(S_\tau)+w(V(\gamma)\setminus U_\gamma)+w(V(\gamma')\setminus U_{\gamma'})$$

The second condition follows from the form of $B(\gamma)$. ∎

### H.3 Choosing $N(\gamma)$

To choose $N(\gamma)$, we use the following lemma:

**Lemma H.13.** *For all $D_V \in \mathbb{N}$, for all composable $\gamma, \tau, \gamma'^T$ such that $|V(\gamma)| \leq D_V$, $|V(\tau)| \leq D_V$, and $|V(\gamma')| \leq D_V$,*

$$\sum_{j>0} \sum_{\gamma_1,\gamma_1',\cdots,\gamma_j,\gamma_j'\in\Gamma_{\gamma,\gamma',j}} \prod_{i:\gamma_i \text{ is non-trivial}} \frac{1}{|Aut(U_{\gamma_i})|} \prod_{i:\gamma_i' \text{ is non-trivial}} \frac{1}{|Aut(U_{\gamma_i'})|} \sum_{P_1,\cdots,P_j:P_i\in\mathcal{P}_{\gamma_i,\tau_{P_{i-1}},\gamma_i'^T}} \left( \prod_{i=1}^j N(P_i) \right)$$

$$\leq \frac{(3D_V)^{2(|V(\gamma)\setminus V_\gamma|+|V(\gamma')\setminus V_{\gamma'}|)+(|V(\gamma)\setminus U_\gamma|+|V(\gamma')\setminus U_{\gamma'}|)}}{(|Aut(U_\gamma)|)^{1_{\gamma \text{ is non-trivial}}}(|Aut(U_{\gamma'})|)^{1_{\gamma' \text{ is non-trivial}}}}$$

*Proof.* Observe that aside from the orderings (which are canceled out by the $|Aut(U_{\gamma_i})|$ and $|Aut(U_{\gamma_i'})|$ factors), the intersection patterns $\{P_i : i \in [j]\}$ are determined by the following data on each vertex $v \in (V(\gamma) \setminus V_\gamma) \cup (V(\gamma'^T) \setminus V_{\gamma'^T})$:

1. The first $i \in [j]$ such that $v \in (V(\gamma_i) \setminus V_{\gamma_i}) \cup (V(\gamma_i'^T) \setminus V_{\gamma_i'^T})$. There are at most $j$ possibilities for this.

2. A vertex $u$ (if one exists) in $V\left(\gamma_{i-1} \circ \ldots \circ \gamma_1 \circ \tau \circ \gamma_1'^T \ldots \circ \gamma_{i-1}'^T\right)$ such that $u$ and $v$ are equal. There are at most $3D_V$ possibilities for this.

Using these observations and taking $j_{max} = |V(\gamma) \setminus V_\gamma| + |V(\gamma') \setminus V_{\gamma'}|$,

$$\sum_{j>0} \sum_{\gamma_1, \gamma_1', \cdots, \gamma_j, \gamma_j' \in \Gamma_{\gamma, \gamma', j}} \prod_{i: \gamma_i \text{ is non-trivial}} \frac{1}{|Aut(U_{\gamma_i})|} \prod_{i: \gamma_i' \text{ is non-trivial}} \frac{1}{|Aut(U_{\gamma_i'})|} \sum_{P_1, \cdots, P_j : P_i \in \mathcal{P}_{\gamma_i, \tau_{P_{i-1}}, \gamma_i'^T}} 1$$

$$\leq \sum_{j=1}^{j_{max}} \frac{(3jD_V)^{|V(\gamma) \setminus V_\gamma| + |V(\gamma') \setminus V_{\gamma'}|}}{(|Aut(U_\gamma)|)^{1_{\gamma \text{ is non-trivial}}} (|Aut(U_{\gamma'})|)^{1_{\gamma' \text{ is non-trivial}}}}$$

$$\leq j_{max} \left(\frac{2}{3}\right)^{j_{max}} \frac{(3D_V)^{2(|V(\gamma) \setminus V_\gamma| + |V(\gamma') \setminus V_{\gamma'}|)}}{(|Aut(U_\gamma)|)^{1_{\gamma \text{ is non-trivial}}} (|Aut(U_{\gamma'})|)^{1_{\gamma' \text{ is non-trivial}}}}$$

$$< \frac{(3D_V)^{2(|V(\gamma) \setminus V_\gamma| + |V(\gamma') \setminus V_{\gamma'}|)}}{(|Aut(U_\gamma)|)^{1_{\gamma \text{ is non-trivial}}} (|Aut(U_{\gamma'})|)^{1_{\gamma' \text{ is non-trivial}}}}$$

Now recall that for any $\gamma_i, \tau_{P_{i-1}}, \gamma_i'^T$ and any intersection pattern $P_i \in \mathcal{P}_{\gamma_i, \tau_{P_{i-1}}, \gamma_i'^T}$, $N(P_i) \leq |V(\tau_{P_i})|^{|V(\gamma_i) \setminus U_{\gamma_i}| + |V(\gamma_i') \setminus U_{\gamma_i'}|}$. Thus, for any $P_1, \cdots, P_j : P_i \in \mathcal{P}_{\gamma_i, \tau_{P_{i-1}}, \gamma_i'^T}$, $\prod_{i=1}^{j} N(P_i) \leq (3D_V)^{|V(\gamma) \setminus U_\gamma| + |V(\gamma') \setminus U_{\gamma'}|}$. Putting everything together, the result follows. ∎

The last condition of $\varepsilon$-feasibility follows as a direct corollary.

## H.4  Choosing $c(\alpha)$

In this section, we describe how to choose $c(\alpha)$. For simplicity, we first describe how to choose $c(\alpha)$ under our simplifying assumptions. We then describe the minor adjustments that are needed when we have hyperedges and multiple types of vertices.

**Lemma H.14.** *Under our simplifying assumptons, for all $U \in \mathcal{I}_{mid}$,*

$$\sum_{\alpha: U_\alpha \equiv U, \alpha \text{ is proper and non-trivial}} \frac{1}{|Aut(U_\alpha \cap V_\alpha)|(3D_V)^{|U_\alpha \setminus V_\alpha| + |V_\alpha \setminus U_\alpha| + 2|E(\alpha)|} 2^{|V(\alpha) \setminus (U_\alpha \cup V_\alpha)|}} < 5$$

*Proof.* In order to choose $\alpha$, it is sufficient to choose the following:

1. The number $j_1$ of vertices in $U_\alpha \setminus V_\alpha$, the number $j_2$ of vertices in $V_\alpha \setminus U_\alpha$, and the number $j_3$ of vertices in $V(\alpha) \setminus (U_\alpha \cup V_\alpha)$.

2. A mapping in $Aut(U_\alpha \cap V_\alpha)$ determining how the vertices in $U_\alpha \cap V_\alpha$ match up with each other.

3. The position of each vertex $u \in U_\alpha \setminus V_\alpha$ within $U_\alpha$ (there are at most $|U_\alpha| \leq D_V$ choices for this).

4. The position of each vertex $v \in V_\alpha \setminus U_\alpha$ within $V_\alpha$ (there are at most $|U_\alpha| \leq D_V$ choices for this).

5. The number $j_4$ of edges in $E(\alpha)$.

6. The endpoints of each edge in $E(\alpha)$.

This implies that for all $j_1, j_2, j_3, j_4 \geq 0$

$$\sum_{\substack{\alpha: U_\alpha \equiv U, |U_\alpha \setminus V_\alpha| = j_1, |V_\alpha \setminus U_\alpha| = j_2 \\ |V(\alpha) \setminus (U_\alpha \cup V_\alpha)| = j_3, |E(\alpha)| = j_4}} \frac{1}{|Aut(U_\alpha \cap V_\alpha)|(D_V)^{j_1 + j_2}(D_V)^{2j_4}} \leq 1$$

Using this, we have that

$$\sum_{\alpha:U_\alpha\equiv U,\alpha \text{ is proper and non-trivial}} \frac{1}{|Aut(U_\alpha\cap V_\alpha)|(3D_V)^{|U_\alpha\setminus V_\alpha|+|V_\alpha\setminus U_\alpha|+2|E(\alpha)|}2^{|V(\alpha)\setminus(U_\alpha\cup V_\alpha)|}}$$

$$\leq \sum_{j_1,j_2,j_3,j_4\in\mathbb{N}\cup\{0\}:j_1+j_2+j_3+j_4\geq 1} \frac{1}{3^{j_1+j_2}9^{j_4}2^{j_3}} \leq 2\left(\frac{3}{2}\right)^2\frac{9}{8}-1<5$$

■

This implies conditions $4, 5, 6$ of $\varepsilon$-feasibility.

### H.4.1  Choosing $c(\alpha)$ in general*

When we have multiple types of vertices and hyperedges of arity $k$, Lemma H.14 can be generalized as follows:

**Lemma H.15.** *Under our simplifying assumptons, for all $U \in \mathcal{I}_{mid}$,*

$$\sum_{\alpha:U_\alpha\equiv U,\alpha \text{ is proper and non-trivial}} \frac{1}{|Aut(U_\alpha\cap V_\alpha)|(3D_Vt_{max})^{|U_\alpha\setminus V_\alpha|+|V_\alpha\setminus U_\alpha|+k|E(\alpha)|}(2t_{max})^{|V(\alpha)\setminus(U_\alpha\cup V_\alpha)|}} < 5$$

*Proof sketch.* This can be proved in the same way as Lemma H.14 with the following modifications:

1. In addition to choosing the number of vertices in $U_\alpha \setminus V_\alpha$, $V_\alpha \setminus U_\alpha$, and $V(\alpha) \setminus (U_\alpha \cap V_\alpha)$, we also have to choose the types of these vertices.

2. For each hyperedge, we have to choose $k$ endpoints rather than 2 endpoints.

■

This implies the same conclusion regarding $\varepsilon$-feasibility. For technical reasons, we will need a more refined bound when the sum is over all shapes $\gamma$ of at least a prescribed size.

**Lemma H.16.** *For all $\varepsilon' > 0$, for the same choice of $c(\alpha)$ as above, for any $U \in \mathcal{I}_{mid}$ and integer $m \geq 1$, we have*

$$\sum_{\gamma\in\Gamma_{U,*}:|V(\gamma)|\geq|U|+m} \frac{1}{|Aut(U)|c(\gamma)} \leq \frac{\varepsilon'}{5\cdot 2^{m-1}}$$

*Proof sketch.* The proof is similar to the proof of $\varepsilon$-feasibility, but we now have the extra condition $j_2 + j_3 \geq m$ in the proof of Lemma H.14. Then,

$$\sum_{j_1,j_2,j_3,j_4\in\mathbb{N}\cup\{0\}:j_2+j_3\geq m} \frac{1}{3^{j_1+j_2}9^{j_4}2^{j_3}} \leq \sum_{j_1,j_4\in\mathbb{N}\cup\{0\}} \frac{1}{2^m3^{j_1}9^{j_4}} = \frac{27}{16\cdot 2^m} \leq \frac{1}{2^{m-1}}$$

■

### H.5  Proof of the Generalized Intersection Tradeoff Lemma

We now prove the generalized intersection tradeoff lemma, which in particular generalizes [19, Lemma 7.12].

**Lemma H.17.** *For all $\gamma, \tau, \gamma'$ and all intersection patterns $P \in \mathcal{P}_{\gamma,\tau,\gamma'}$,*

$$w(V(\tau_P))+w(I_{\tau_P})-w(S_{\tau_P,min}) \leq w(V(\tau))+w(I_\tau)-w(S_{\tau,min})+w(V(\gamma)\setminus U_\gamma)+w(V(\gamma')\setminus U_{\gamma'})$$

*Proof.*

**Definition H.18.**

1. *We define $I_{LM}$ to be the set of vertices which, after intersections, touch $\gamma$ and $\tau$ but not $\gamma'^T$. In particular, $I_{LM}$ consists of the vertices which result from intersecting a pair of vertices in $V(\gamma) \setminus V_\gamma$ and $V(\tau) \setminus U_\tau \setminus V_\tau$ and the vertices which are in $U_\tau \setminus V_\tau$ and are not intersected with any other vertex.*

2. We define $I_{MR}$ to be the set of vertices which, after intersections, touch $\tau$ and $\gamma'^T$ but not $\gamma$. In particular, $I_{MR}$ consists of the vertices which result from intersecting a pair of vertices in $V(\tau) \setminus U_\tau \setminus V_\tau$ and $V(\gamma'^T) \setminus U_{\gamma'^T}$ and the vertices which are in $V_\tau \setminus U_\tau$ and are not intersected with any other vertex.

3. We define $I_{LR}$ to be the set of vertices which, after intersections, touch $\gamma$ and $\gamma'^T$ but not $\tau$. In particular, $I_{LR}$ consists of the vertices which result from intersecting a pair of vertices in $V(\gamma) \setminus V_\gamma$ and $V(\gamma'^T) \setminus U_{\gamma'^T}$.

4. We define $I_{LMR}$ to be the set of vertices which, after intersections, touch $\gamma$, $\tau$, and $\gamma'^T$. In particular, $I_{LMR}$ consists of the vertices which result from intersecting a triple of vertices in $V(\gamma) \setminus V_\gamma$, $V(\tau) \setminus U_\tau \setminus V_\tau$, and $V(\gamma'^T) \setminus U_{\gamma'^T}$, intersecting a pair of vertices in $V(\gamma) \setminus V_\gamma$ and $V_\tau \setminus U_\tau$, intersecting a pair of vertices in $U_\tau \setminus V_\tau$ and $V(\gamma'^T) \setminus U_{\gamma'^T}$, and single vertices in $U_\tau \cap V_\tau$.

The main idea is as follows. A priori, any of the vertices in $I_{LM} \cup I_{MR} \cup I_{LR} \cup I_{LMR}$ could become isolated. We handle this by keeping track of the following types of flows - Flows from $U_\gamma$ to $I_{LM} \cup I_{LR} \cup I_{LMR}$, flows from $I_{LR} \cup I_{MR} \cup I_{LMR}$ to $V_{\gamma'^T}$, and flows from $I_{LM}$ to $I_{MR}$. For technical reasons, we also view vertices in $I_{LMR}$ as having flow to themselves. We then observe that flows to and from these vertices prevent these vertices from being isolated and can provide flow from $U_\gamma$ to $V_{\gamma'^T}$, which gives a lower bound on $w(S_{\tau_P})$.

We now implement this idea.

**Definition H.19** (Flow Graph). *Given a shape $\alpha$, we define the directed graph $H_\alpha$ as follows:*

1. *For each vertex $v \in V(\alpha)$, we create two vertices $v_{in}$ and $v_{out}$. We then create a directed edge from $v_{in}$ to $v_{out}$ with capacity $w(v)$*

2. *For each pair of vertices $(v, w)$ which is an edge of multiplicity $1$ in $E(\alpha)$ (or part of a hyperedge of multiplicity $1$ in $E(\alpha)$), we create a directed edge with infinite capacity from $v_{out}$ to $w_{in}$ and we create a directed edge with infinite capacity from $w_{out}$ to $v_{in}$.*

3. *We define $U_{H_\alpha}$ to be $U_{H_\alpha} = \{u_{in} : u \in U_\alpha\}$ and we define $V_{H_\alpha}$ to be $V_{H_\alpha} = \{v_{out} : v \in V_\alpha\}$*

**Lemma H.20.** *The maximum flow from $U_{H_\alpha}$ to $V_{H_\alpha}$ is equal to the minimum weight of a separator between $U_\alpha$ and $V_\alpha$.*

*Proof.* This can be proved using the max flow min cut theorem. ∎

**Definition H.21** (Modified Flow Graph). *Given a shape $\alpha$ together with a set $I_L \subseteq V(\alpha)$ of vertices in $\alpha$ (which will be the vertices in $\alpha$ which are intersected with a vertex to the left of $\alpha$) and a set $I_R \subseteq V(\alpha)$ of vertices in $\alpha$ (which will be the vertices in $\alpha$ which are intersected with a vertex to the right of $\alpha$), we define the modified flow graph $H_\alpha^{I_L, I_R}$ as follows:*

1. *We start with the flow graph $H_\alpha$*

2. *For each vertex $u \in I_L$, we delete all of the edges into $u_{in}$ and add $u_{in}$ to $U_{H_\alpha}$*

3. *For each vertex $v \in I_R$, we delete all of the edges out of $v_{out}$ and add $v_{out}$ to $V_{H_\alpha}$*

4. *We call the resulting graph $H_\alpha^{I_L, I_R}$ and the resulting sets $U_{H_\alpha^{I_L, I_R}}$ and $V_{H_\alpha^{I_L, I_R}}$*

**Lemma H.22.** *The maximum flow from $U_{H_\alpha^{I_L, I_R}}$ to $V_{H_\alpha^{I_L, I_R}}$ in $H_\alpha^{I_L, I_R}$ is at least as large as the maximum flow from $U_{H_\alpha}$ to $V_{H_\alpha}$ in $H_\alpha$*

*Proof sketch.* Observe that if we have a cut $C$ in $H_\alpha^{I_L, I_R}$ which separates $U_{H_\alpha^{I_L, I_R}}$ and $V_{H_\alpha^{I_L, I_R}}$ then $C$ separates $U_{H_\alpha}$ and $V_{H_\alpha}$ in $H_\alpha$ ∎

Before the intersections, we have the following flows. We take $F_1$ to be the maximum flow from $U_\gamma$ to $V_\gamma$ in $\gamma$. Note that $F_1$ has value $w(V_\gamma)$. We take $F_2$ to be the maximum flow from $U_\tau$ to $V_\tau$ in $\tau$. Note that $F_2$ has value $w(S_{\tau,min})$. We take $F_3$ to be the maximum flow from $U_{\gamma'^T}$ to $V_{\gamma'^T}$ in $\gamma'^T$. Note that $F_1$ has value $w(U_{\gamma'^T})$.

After the intersections, we take the following flows: We take $F_1'$ to be the maximum flow from $U_{H_\gamma^{\emptyset, I_{LM} \cup I_{LR} \cup I_{LMR}}}$ to $V_{H_\gamma^{\emptyset, I_{LM} \cup I_{LR} \cup I_{LMR}}}$ in $H_\gamma^{\emptyset, I_{LM} \cup I_{LR} \cup I_{LMR}}$. We take $F_2'$ to be the maximum flow from $U_{H_\tau^{I_{LM} \cup I_{LMR}, I_{MR} \cup I_{LMR}}}$ to $V_{H_\tau^{I_{LM} \cup I_{LMR}, I_{MR} \cup I_{LMR}}}$ in $H_\tau^{I_{LM} \cup I_{LMR}, I_{MR} \cup I_{LMR}}$ We take $F_3'$ to be the maximum flow from $U_{H_{\gamma'^T}^{I_{MR} \cup I_{LR} \cup I_{LMR}, \emptyset}}$ to $V_{H_{\gamma'^T}^{I_{MR} \cup I_{LR} \cup I_{LMR}, \emptyset}}$ in $H_{\gamma'^T}^{I_{MR} \cup I_{LR} \cup I_{LMR}, \emptyset}$.

Observe that because of how intersection patterns are defined, $val(F_1') = w(U_\gamma)$ and $val(F_3') = w(V_{\gamma'^T})$. By Lemma H.22, the value of $F_2'$ is at least as large as the value of $F_2$, so $val(F_2') \geq w(S_{\tau,min})$.

We now consider $F_1' + F_2' + F_3'$. As is, this is not a flow, but we can fix this.

**Definition H.23.** *For each vertex $v \in V(\tau_P)$, we define $f_{in}(v), f_{out}(v), f_{through}(v)$ to be the flow into $v_{in}$, flow out of $v_{out}$, flow from $v_{in}$ to $v_{out}$ respectively in $F_1' + F_2' + F_3'$. Also define $f_{imbalance}(v) = |f_{in}(v) - f_{out}(v)|$ and $f_{excess}(v) = f_{through}(v) - max\{f_{in}(v), f_{out}(v)\}$. With this information, we fix the flow $F_1' + F_2' + F_3'$ as follows. For each vertex $v \in V(\tau_P)$,*

1. *If $f_{in}(v) > f_{out}(v)$ then we create a vertex $v_{supplemental,out}$ and an edge from $v_{out}$ to $v_{supplemental,out}$ with capacity $f_{imbalance}(v)$ and we route $f_{imbalance}(v)$ of flow along this edge. We then add $v_{supplemental,out}$ to a set of vertices $V_{supplemental}$.*

2. *If $f_{in}(v) < f_{out}(v)$ then we create a vertex $v_{supplemental,in}$ and an edge from $v_{supplemental,in}$ to $v_{in}$ with capacity $f_{imbalance}(v)$ and we route $f_{imbalance}(v)$ of flow along this edge. We then add $v_{supplemental,out}$ to a set of vertices $V_{supplemental}$.*

3. *We reduce the flow on the edge from $v_{in}$ to $v_{out}$ by $f_{excess}(v)$*

*We call the resulting flow $F'$*

**Proposition H.24.** *$F'$ is a flow from $U_{H_\gamma^{\emptyset, I_{LM} \cup I_{LR} \cup I_{LMR}}} \cup U_{supplemental}$ to $V_{H_{\gamma'^T}^{I_{MR} \cup I_{LR} \cup I_{LMR}, \emptyset}} \cup V_{supplemental}$ with value $val(F') = val(F_1') + val(F_2') + val(F_3') - \sum_{v \in V(\tau)} f_{excess}(v)$*

**Corollary H.25.** *There exists a flow $F''$ from $U_{H_\gamma^{\emptyset, I_{LM} \cup I_{LR} \cup I_{LMR}}}$ to $V_{H_{\gamma'^T}^{I_{MR} \cup I_{LR} \cup I_{LMR}, \emptyset}}$ with value $val(F'') \geq val(F_1') + val(F_2') + val(F_3') - \sum_{v \in V(\tau)} (f_{excess}(v) + f_{imbalance}(v))$*

*Proof.* Consider the minimum cut $C$ between $U_{H_\gamma^{\emptyset, I_{LM} \cup I_{LR} \cup I_{LMR}}}$ and $V_{H_{\gamma'^T}^{I_{MR} \cup I_{LR} \cup I_{LMR}, \emptyset}}$. If we add all of the supplemental edges to $C$ then this gives a cut $C'$ between $U_{H_\gamma^{\emptyset, I_{LM} \cup I_{LR} \cup I_{LMR}}}$ and $V_{H_{\gamma'^T}^{I_{MR} \cup I_{LR} \cup I_{LMR}, \emptyset}}$ with capacity $capacity(C') = capacity(C) + \sum_{v \in V(\tau)} f_{imbalance}(v) \geq val(F')$. Thus, $capacity(C) \geq val(F') - \sum_{v \in V(\tau)} f_{imbalance}(v)$ so there exists a flow $F''$ from $U_{H_\gamma^{\emptyset, I_{LM} \cup I_{LR} \cup I_{LMR}}}$ to $V_{H_{\gamma'^T}^{I_{MR} \cup I_{LR} \cup I_{LMR}, \emptyset}}$ with value $val(F'') = capacity(C) \geq val(F_1') + val(F_2') + val(F_3') - \sum_{v \in V(\tau)} (f_{excess}(v) + f_{imbalance}(v))$ ∎

We now make the following observations:

**Lemma H.26.**

1. *For all vertices $v \notin I_{LM} \cup I_{MR} \cup I_{LR} \cup I_{LMR}$, $f_{excess}(v) = f_{imbalance}(v) = 0$ (and these vertices can never be isolated).*

2. *For all vertices $v \in I_{LM}$, $f_{excess}(v) + f_{imbalance}(v) \leq w(v)$. Moreover, for all vertices $v \in I_{LM}$ which are isolated, $f_{excess}(v) = f_{imbalance}(v) = 0$.*

3. *For all vertices $v \in I_{MR}$, $f_{excess}(v) + f_{imbalance}(v) \leq w(v)$. Moreover, for all vertices $v \in I_{LM}$ which are isolated, $f_{excess}(v) = f_{imbalance}(v) = 0$.*

4. *For all vertices $v \in I_{LR}$, $f_{excess}(v) + f_{imbalance}(v) \leq w(v)$. Moreover, for all vertices $v \in I_{LM}$ which are isolated, $f_{excess}(v) = f_{imbalance}(v) = 0$.*

5. *For all vertices $v \in I_{LMR}$, $f_{excess}(v) + f_{imbalance}(v) \leq 2w(v)$. Moreover, for all vertices $v \in I_{LMR}$ which are isolated, $f_{excess}(v) = w(v)$ and $f_{imbalance}(v) = 0$.*

*Proof.* For the first statement, observe that for vertices $v \notin I_{LM} \cup I_{MR} \cup I_{LR} \cup I_{LMR}$, neither $v_{in}$ nor $v_{out}$ is ever a sink or source so the flow into these vertices must equal the flow out of these vertices and thus $f_{in}(v) = f_{out}(v) = f_{through}(v)$. For the second statement, observe that for a vertex $v \in I_{LM}$,

1. $F_1'$ will have a flow of $f_{in}(v)$ into $v_{in}$ and along the edge from $v_{in}$ to $v_{out}$

2. $F_2'$ will have a flow of $f_{out}(v)$ along the edge from $v_{in}$ to $v_{out}$ and out of $v_{out}$.

Thus, $f_{excess}(v) = f_{in}(v) + f_{out}(v) - \max\{f_{in}(v), f_{out}(v)\}$. Since $f_{imbalance}(v) = |f_{in}(v) - f_{out}(v)|$, $f_{excess}(v) + f_{imbalance}(v) = f_{in}(v) + f_{out}(v) - \min\{f_{in}(v), f_{out}(v)\} \leq w(v)$. If $v$ is isolated then neither $F_1'$ nor $F_2'$ can have any flow to $v_{in}$ or out of $v_{out}$ so $f_{in}(v) = f_{through}(v) = f_{out}(v) = 0$ The third and fourth statements can be proved in the same way as the second statement. For the fifth statement, observe that for a vertex $v \in I_{LMR}$,

1. $F_1'$ will have a flow of $f_{in}(v)$ into $v_{in}$ and along the edge from $v_{in}$ to $v_{out}$.

2. $F_2'$ will have a flow of $w(v)$ along the edge from $v_{in}$ to $v_{out}$

3. $F_3'$ will have a flow of $f_{out}(v)$ along the edge from $v_{in}$ to $v_{out}$ and out of $v_{out}$.

Thus, $f_{excess}(v) = w(v) + f_{in}(v) + f_{out}(v) - \max\{f_{in}(v), f_{out}(v)\}$. Since $f_{imbalance}(v) = |f_{in}(v) - f_{out}(v)|$, $f_{excess}(v) + f_{imbalance}(v) = w(v) + f_{in}(v) + f_{out}(v) - \min\{f_{in}(v), f_{out}(v)\} \leq 2w(v)$. If $v$ is isolated then neither $F_1'$ nor $F_3'$ can have any flow to $v_{in}$ or out of $v_{out}$ so $f_{in}(v) = f_{out}(v) = 0$ and $f_{through}(v) = w(v)$. ∎

Putting everything together, we have the following corollary:

**Corollary H.27.**

$$\sum_{v \in V(\tau_P)} (f_{excess}(v) + f_{imbalance}(v)) \leq w(I_{LM}) + w(I_{LR}) + w(I_{MR}) + 2w(I_{LMR}) - (w(I_{\tau_P}) - w(I_\tau))$$

Combining this with Corollary H.25,

$$w(S_{\tau_P, min}) \geq val(F_1') + val(F_2') + val(F_3') - \sum_{v \in V(\tau_P)} (f_{excess}(v) + f_{imbalance}(v))$$

$$\geq w(U_\gamma) + w(S_{\tau, min}) + w(V_{\gamma'^T}) - w(I_{LM}) - w(I_{LR}) - w(I_{MR}) - 2w(I_{LMR}) + (w(I_{\tau_P}) - w(I_\tau))$$

Since $w(V(\tau_P)) = w(V(\tau)) + w(V(\gamma)) + w(V(\gamma')) - w(I_{LM}) - w(I_{LR}) - w(I_{MR}) - 2w(I_{LMR})$,

$$w(S_{\tau_P, min}) \geq w(U_\gamma) + w(S_{\tau, min}) + w(V_{\gamma'^T}) + w(V(\tau_P)) - w(V(\tau)) - w(V(\gamma)) - w(V(\gamma')) + (w(I_{\tau_P}) - w(I_\tau))$$

Rearranging this gives the result. ∎

# I   Bounding truncation error

Now, we illustrate one way to show truncation error bounds when we apply the general theorem. Assume $\|M_\alpha\| \leq B_{norm}(\alpha)$ for all $\alpha \in \mathcal{M}'$. We want to show

$$\sum_{U \in \mathcal{I}_{mid}} M_{Id_U}^{fact}(H_{Id_U}) \succeq 6 \left( \sum_{U \in \mathcal{I}_{mid}} \sum_{\gamma \in \Gamma_{U,*}} \frac{d_{Id_U}(H_\gamma', H_{Id_U})}{|Aut(U)|c(\gamma)} \right) Id_{sym}$$

To do this, we simply sandwich a factor of $Id_{sym}$ between the two terms. Let $D_{sos}$ be the degree of the SoS program. We will describe in Appendix I.1 how to show $\sum_{U \in \mathcal{I}_{mid}} M_{Id_U}^{fact}(H_{Id_U}) \succeq$

$\frac{1}{n^{K_1 D_{sos}^2}} Id_{sym}$ for a constant $K_1 > 0$. We also show $\sum_{U \in \mathcal{I}_{mid}} \sum_{\gamma \in \Gamma_{U,*}} \frac{d_{Id_U}(H_{Id_U}, H'_{\gamma})}{|Aut(U)|c(\gamma)} \leq \frac{n^{K_2 D_{sos}}}{2^{D_V}}$ for a constant $K_2 > 0$. Along with the fact that $Id_{Sym} \succeq 0$, we can choose $D_{sos}$ small enough so that $\frac{1}{n^{K_1 D_{sos}^2}} > \frac{n^{K_2 D_{sos}}}{2^{D_V}}$, completing the proof.

We will need the following simple bound that says that if we have sufficient decay for each vertex, then, the sum of this decay, over all shapes $\sigma \circ \sigma'$ for $\sigma, \sigma' \in \mathcal{L}'_U$, is bounded.

**Definition I.1.** *For $U \in \mathcal{I}_{mid}$, let $\mathcal{L}'_U \subset \mathcal{L}_U$ be the set of non-trivial shapes in $\mathcal{L}_U$.*

**Lemma I.2.** *Suppose $D_V = n^{C_V \varepsilon}, D_E = n^{C_E \varepsilon}$ for constants $C_V, C_E > 0$, are the truncation parameters for our shapes. For any $U \in \mathcal{I}_{mid}$,*

$$\sum_{U \in \mathcal{I}_{mid}} \sum_{\sigma, \sigma' \in \mathcal{L}'_U} \frac{1}{D_{sos}^{D_{sos}} n^{F \varepsilon |V(\sigma \circ \sigma')|}} \leq 1$$

*for a constant $F > 0$ that depends only on $C_V, C_E$. In particular, by setting $C_V, C_E$ small enough, we can make this constant arbitrarily small.*

*Proof.* For a given $j = |U|$, the number of ways to choose $U$ is at most $t_{max}^j$. For a given $U \in \mathcal{I}_{mid}$, we will bound the number of ways to choose $\sigma, \sigma' \in \mathcal{L}'_U$. This can be done similar to Lemma H.14 and implies the result. ∎

## I.1 General strategy to lower bound $\sum_{V \in \mathcal{I}_{mid}} M^{fact}(H_{Id_V})$

In this section, we describe how to show that $\sum_{V \in \mathcal{I}_{mid}} M^{fact}(H_{Id_V}) \succeq \delta Id_{Sym}$ for some $\delta > 0$ where $\delta$ will depend on $n$ and other parameters. For this, we use a similar strategy as [64]. For each $V \in \mathcal{I}_{mid}$, we choose a weight $w_V \in (0, 1]$. We then observe that since each coefficient matrix $H_{Id_V}$ is PSD,

$$\sum_{V \in \mathcal{I}_{mid}} M^{fact}(H_{Id_V}) \succeq \sum_{V \in \mathcal{I}_{mid}} w_V M^{fact}(H_{Id_V})$$

By choosing the weights $w_V$ appropriately, we can bound the off-diagonal parts by the diagonal parts, giving us $\delta Id_{Sym}$.

**Definition I.3.** *For all $V \in \mathcal{I}_{mid}$ we define $Id_{Sym,V}$ to be the matrix such that*

1. *$Id_{Sym,V}(A, B) = 1$ if $A$ and $B$ both have index shape $V$.*

2. *Otherwise, $Id_{Sym,V}(A, B) = 0$.*

**Proposition I.4.** *$Id_{Sym} = \sum_{V \in \mathcal{I}_{mid}} Id_{Sym,V}$*

**Definition I.5.** *For each $V \in \mathcal{I}_{mid}$, we define $\lambda_V = |Aut(V)| H_{Id_V}(Id_V, Id_V)$.*

**Theorem I.6.** *If $\{w_V : V \in \mathcal{I}_{mid}\}$ are weights such that for all $V \in \mathcal{I}_{mid}$ and all left shapes $\sigma \in \mathcal{L}_V$, $w_V \leq \frac{w_{U_\sigma} \lambda_{U_\sigma}}{|\mathcal{I}_{mid}| B_{norm}(\sigma)^2 c(\sigma)^2 H_{Id_V}(\sigma, \sigma)}$ then*

$$\sum_{V \in \mathcal{I}_{mid}} M^{fact}(H_{Id_V}) \succeq \frac{1}{2} \sum_{V \in \mathcal{I}_{mid}} w_V \lambda_V Id_{Sym,V} \succeq \frac{1}{2} \min_{V \in \mathcal{I}_{mid}} \{w_V \lambda_V\} Id_{Sym}$$

*Proof.* Observe that for each $V \in \mathcal{I}_{mid}$,

$$w_V \sum_{\sigma, \sigma' \in \mathcal{L}_V} H_{Id_V}(\sigma, \sigma') M_\sigma M_{\sigma'}^T = w_V \lambda_V Id_{Sym,V} + w_V \sum_{\sigma, \sigma' \in \mathcal{L}_V : \sigma \neq Id_V \text{ or } \sigma' \neq Id_V} H_{Id_V}(\sigma, \sigma') \left( \frac{M_\sigma M_{\sigma'}^T + M_{\sigma'} M_\sigma^T}{2} \right)$$

The first part of the right hand side is a diagonal part that we want to extract. We now show that we can bound the second part in terms of the diagonal parts.

**Proposition I.7.** *For all $V \in \mathcal{I}_{mid}$ and all shapes $\sigma, \sigma' \in \mathcal{L}_V$, for all $a, b > 0$ such that $ab \geq B_{norm}(\sigma)^2 B_{norm}(\sigma')^2$, if $\|M_\sigma\| \leq B_{norm}(\sigma)$ and $\|M_{\sigma'}\| \leq B_{norm}(\sigma')$ then*

$$M_\sigma M_{\sigma'}^T + M_{\sigma'} M_\sigma^T \succeq -a Id_{Sym,U_\sigma} - b Id_{Sym,U_{\sigma'}}$$

**Corollary I.8.** *If $H_{Id_V} \succeq 0$ then For any shapes $\sigma, \sigma' \in \mathcal{L}_V$,*

$$w_V H_{Id_V}(\sigma, \sigma') \left( M_\sigma M_{\sigma'}^T + M_{\sigma'} M_\sigma^T \right) \succeq -\frac{c(\sigma)}{c(\sigma')} w_V H_{Id_V}(\sigma, \sigma) B_{norm}(\sigma)^2 Id_{Sym, U_\sigma}$$
$$- \frac{c(\sigma')}{c(\sigma)} w_V H_{Id_V}(\sigma', \sigma') B_{norm}(\sigma')^2 Id_{Sym, U_{\sigma'}}$$

*Proof.* This follows from Proposition I.7 and the observation that since $H_{Id_V} \succeq 0$, for all $\sigma, \sigma' \in \mathcal{L}_V$, $H_{Id_V}(\sigma, \sigma')^2 \leq H_{Id_V}(\sigma, \sigma) H_{Id_V}(\sigma', \sigma')$ ∎

Since $w_V \leq \frac{w_{U_\sigma} \lambda_{U_\sigma}}{|\mathcal{I}_{mid}| B_{norm}(\sigma)^2 c(\sigma)^2 H_{Id_V}(\sigma, \sigma)}$ and $w_V \leq \frac{w_{U_{\sigma'}} \lambda_{U_{\sigma'}}}{|\mathcal{I}_{mid}| B_{norm}(\sigma')^2 c(\sigma')^2 H_{Id_V}(\sigma', \sigma')}$, we have that

$$\sum_{\sigma, \sigma' \in \mathcal{L}_V : \sigma \neq Id_V \text{ or } \sigma' \neq Id_V} w_V H_{Id_V}(\sigma, \sigma') \left( \frac{M_\sigma M_{\sigma'}^T + M_{\sigma'} M_\sigma^T}{2} \right) \succeq -2 \sum_{\sigma \in \mathcal{L}_V} \frac{w_{U_\sigma} \lambda_{U_\sigma} Id_{Sym, U_\sigma}}{|\mathcal{I}_{mid}| c(\sigma)} \left( \sum_{\sigma' \in \mathcal{L}_V : \sigma' \neq Id_V} \frac{1}{c(\sigma')} \right)$$
$$\succeq -\frac{1}{2|\mathcal{I}_{mid}|} \sum_{U \in \mathcal{I}_{mid}} w_U \lambda_U Id_{Sym, U}$$

Thus, for each $V \in \mathcal{I}_{mid}$,

$$w_V M_{fact}(H_{Id_V}) \succeq w_V \lambda_V Id_{Sym, V} - \frac{1}{2|\mathcal{I}_{mid}|} \sum_{U \in \mathcal{I}_{mid}} w_U \lambda_U Id_{Sym, U}$$

Summing this equation over all $V \in \mathcal{V}$, we have that

$$\sum_{V \in \mathcal{I}_{mid}} M^{fact}(H_{Id_V}) \succeq \sum_{V \in \mathcal{I}_{mid}} w_V M^{fact}(H_{Id_V}) \succeq \frac{1}{2} \sum_{V \in \mathcal{I}_{mid}} w_V \lambda_V Id_{Sym, V} \succeq \frac{1}{2} \min_{V \in \mathcal{I}_{mid}} \{w_V \lambda_V\} Id_{Sym}$$

as needed. ∎

### I.1.1 Handling Non-multilinear Matrix Indices*

If there are multilinear matrix indices, then Theorem I.6 still holds and it can be shown in a similar way, but we need to make a few adjustments.

1. We modify the definition of $Id_{Sym, V}$ as follows. For all $V \in \mathcal{I}_{mid}$ we define $Id_{Sym, V}$ to be the matrix such that
   (a) $Id_{Sym, V}(A, B) = 1$ if $A$ and $B$ have the same index shape $U$ and $U$ has the same number of each type of vertex as $V$. Note that $B$ may be a permutation of $A$ and $U$ may have different powers than $V$.
   (b) Otherwise, $Id_{Sym, V}(A, B) = 0$.
   Observe that with this modified definition, we will still have $Id_{Sym} = \sum_{V \in \mathcal{I}_{mid}} Id_{Sym, V}$.

2. Instead of taking $\lambda_V = |Aut(V)| H_{Id_V}(Id_V, Id_V)$, we define $\lambda_V$ as follows. Letting $H_{Id_V, \text{no expansion}}$ be the diagonal submatrix of $H_{Id_V}$ indexed by left shapes $\sigma$ such that $U_\sigma$ has the same number of each type of vertex as $V$ (though the powers may be different), we take
   $$\lambda_V = |Aut(V)| min\{\lambda : H_{Id_V, \text{no expansion}} \succeq \lambda Id_{Sym, V}\}$$

3. We similarly extend the definition of $c$ to left shapes $\sigma$ with multilinear indices in $U_\sigma$ so that we still have $\sum_{\sigma \in \mathcal{L}_V : (U_\sigma)_{reduced} \neq v} \frac{1}{c(\sigma)} \leq \frac{1}{10}$

## J   Tensor PCA: Quantitative bounds

In this section, we will prove the desired tradeoffs in Theorem A.2. We reuse the notation and bounds from Appendix D.

## J.1 Middle shape bounds

**Lemma J.1.** *Suppose $\lambda \leq n^{\frac{k}{4}-\varepsilon}$. For all $U \in \mathcal{I}_{mid}$ and $\tau \in \mathcal{M}_U$, suppose $deg^\tau(i)$ is even for all $i \in V(\tau) \setminus U_\tau \setminus V_\tau$, then*

$$\sqrt{n}^{|V(\tau)|-|U_\tau|} S(\tau) \leq \frac{1}{n^{0.5\varepsilon \sum_{e \in E(\tau)} l_e}}$$

*Proof.* Firstly, we claim that $\sum_{e \in E(\tau)} k l_e \geq 2(|V(\tau)| - |U_\tau|)$. For any vertex $i \in V(\tau) \setminus U_\tau \setminus V_\tau$, $deg^\tau(i)$ is even and is not 0, hence, $deg^\tau(i) \geq 2$. Any vertex $i \in U_\tau \setminus V_\tau$ cannot have $deg^\tau(i) = 0$ otherwise $U_\tau \setminus \{i\}$ is a vertex separator of strictly smaller weight than $U_\tau$, which is not possible, hence, $deg^\tau(i) \geq 1$. Therefore,

$$\sum_{e \in E(\tau)} k l_e \geq \sum_{i \in V(\tau) \setminus U_\tau \setminus V_\tau} deg^\tau(i) + \sum_{i \in U_\tau \setminus V_\tau} deg^\tau(i) + \sum_{i \in V_\tau \setminus U_\tau} deg^\tau(i) \geq 2(|V(\tau)| - |U_\tau|)$$

By choosing $C_\Delta$ sufficiently small, we have

$$\sqrt{n}^{|V(\tau)|-|U_\tau|} S(\tau) \leq \sqrt{n}^{|V(\tau)|-|U_\tau|} \Delta^{|V(\tau)|-|U_\tau|} \prod_{e \in E(\tau)} n^{(-\frac{k}{4}-0.5\varepsilon)l_e} \leq \frac{1}{n^{0.5\varepsilon \sum_{e \in E(\tau)} l_e}}$$

∎

**Corollary J.2.** *For all $U \in \mathcal{I}_{mid}$ and $\tau \in \mathcal{M}_U$, we have $c(\tau) B_{norm}(\tau) S(\tau) \leq 1$.*

*Proof.* Since $\tau$ is a proper middle shape, we have $w(I_\tau) = 0$ and $w(S_{\tau,min}) = w(U_\tau)$. This implies $n^{\frac{w(V(\tau))+w(I_\tau)-w(S_{\tau,min})}{2}} = \sqrt{n}^{|V(\tau)|-|U_\tau|}$. If $deg^\tau(i)$ is odd for any vertex $i \in V(\tau) \setminus U_\tau \setminus V_\tau$, then $S(\tau) = 0$ and the inequality is true. So, assume $deg^\tau(i)$ is even for all $i \in V(\tau) \setminus U_\tau \setminus V_\tau$. As was observed in the proof of Lemma J.1, every vertex $i \in V(\tau) \setminus U_\tau$ or $i \in V(\tau) \setminus V_\tau$ has $deg^\tau(i) \geq 1$ and hence, $|V(\tau) \setminus U_\tau| + |V(\tau) \setminus V_\tau| \leq 4 \sum_{e \in E(\tau)} l_e$. Also, $|E(\tau)| \leq \sum_{e \in E(\tau)} l_e$ and $q = n^{O(1) \cdot \varepsilon(C_V + C_E)}$. We can set $C_V, C_E$ sufficiently small so that, using Lemma J.1,

$$c(\tau) B_{norm}(\tau) S(\tau) \leq n^{O(1) \cdot \varepsilon(C_V + C_E) \cdot \sum_{e \in E(\tau)} l_e} \cdot \sqrt{n}^{|V(\tau)|-|U_\tau|} S(\tau) \leq 1$$

∎

We can now show middle shape bounds.

**Lemma J.3.** *For all $U \in \mathcal{I}_{mid}$ and $\tau \in \mathcal{M}_U$,*

$$\begin{bmatrix} \frac{1}{|Aut(U)|c(\tau)} H_{Id_U} & B_{norm}(\tau) H_\tau \\ B_{norm}(\tau) H_\tau^T & \frac{1}{|Aut(U)|c(\tau)} H_{Id_U} \end{bmatrix} \succeq 0$$

*Proof.* The expression is equal to

$$\begin{bmatrix} \left( \frac{1}{|Aut(U)|c(\tau)} - \frac{S(\tau) B_{norm}(\tau)}{|Aut(U)|} \right) H_{Id_U} & 0 \\ 0 & \left( \frac{1}{|Aut(U)|c(\tau)} - \frac{S(\tau) B_{norm}(\tau)}{|Aut(U)|} \right) H_{Id_U} \end{bmatrix}$$

$$+ B_{norm}(\tau) \begin{bmatrix} \frac{S(\tau)}{|Aut(U)|} H_{Id_U} & H_\tau \\ H_\tau^T & \frac{S(\tau)}{|Aut(U)|} H_{Id_U} \end{bmatrix}$$

By Lemma D.6, $\begin{bmatrix} \frac{S(\tau)}{|Aut(U)|} H_{Id_U} & H_\tau \\ H_\tau^T & \frac{S(\tau)}{|Aut(U)|} H_{Id_U} \end{bmatrix} \succeq 0$, so the second term above is positive semidefinite. For the first term, by Lemma D.4, $H_{Id_U} \succeq 0$ and by Corollary J.2, $\frac{1}{|Aut(U)|c(\tau)} - \frac{S(\tau) B_{norm}(\tau)}{|Aut(U)|} \geq 0$, which proves that the first term is also positive semidefinite. ∎

## J.2 Intersection term bounds

**Lemma J.4.** *Suppose $\lambda \leq n^{\frac{k}{4}-\varepsilon}$. For all $U, V \in \mathcal{I}_{mid}$ where $w(U) > w(V)$ and for all $\gamma \in \Gamma_{U,V}$,*

$$n^{w(V(\gamma)\backslash U_\gamma)} S(\gamma)^2 \leq \frac{1}{n^{B\varepsilon(|V(\gamma)\backslash(U_\gamma \cap V_\gamma)|+\sum_{e\in E(\gamma)} l_e)}}$$

*for some constant $B$ that depends only on $C_\Delta$. In particular, it is independent of $C_V$ and $C_E$.*

*Proof.* Suppose there is a vertex $i \in V(\gamma) \setminus U_\gamma \setminus V_\gamma$ such that $deg^\gamma(i)$ is odd, then $S(\gamma) = 0$ and the inequality is true. So, assume $deg^\gamma(i)$ is even for all vertices $i \in V(\gamma) \setminus U_\gamma \setminus V_\gamma$. We first claim that $k \sum_{e\in E(\gamma)} l_e \geq 2|V(\gamma) \setminus U_\gamma|$. Since $\gamma$ is a left shape, all vertices $i$ in $V(\gamma) \setminus U_\gamma$ have $deg^\gamma(i) \geq 1$. In particular, all vertices $i \in V_\gamma \setminus U_\gamma$ have $deg^\gamma(i) \geq 1$. Moreover, if $i \in V(\gamma) \setminus U_\gamma \setminus V_\gamma$, since $deg^\gamma(i)$ is even, we must have $deg^\gamma(i) \geq 2$.

Let $S'$ be the set of vertices $i \in U_\gamma \setminus V_\gamma$ that have $deg^\gamma(i) \geq 1$. Then, note that $|S'|+|U_\gamma \cap V_\gamma| \geq |V_\gamma| \implies |S'| \geq |V_\gamma \setminus U_\gamma|$ since otherwise $S' \cup (U_\gamma \cap V_\gamma)$ will be a vertex separator of $\gamma$ of weight strictly less than $V_\gamma$, which is not possible. Then,

$$\sum_{e\in E(\gamma)} k l_e \geq \sum_{i\in V(\gamma)\backslash U_\gamma\backslash V_\gamma} deg^\gamma(i) + \sum_{i\in U_\gamma\backslash V_\gamma} deg^\gamma(i) + \sum_{i\in V_\gamma\backslash U_\gamma} deg^\gamma(i) \geq 2|V(\gamma)\backslash U_\gamma|$$

Finally, note that $2|V(\gamma)| - |U_\gamma| - |V_\gamma| = |U_\gamma \setminus V_\gamma| + |V_\gamma \setminus U_\gamma| + 2|V(\gamma) \setminus U_\gamma \setminus V_\gamma| \geq |V(\gamma) \setminus (U_\gamma \cap V_\gamma)|$. By choosing $C_\Delta$ sufficiently small, we have

$$n^{w(V(\gamma)\backslash U_\gamma)} S(\gamma)^2 \leq n^{|V(\gamma)\backslash U_\gamma|} \Delta^{2|V(\gamma)|-|U_\gamma|-|V_\gamma|} \prod_{e\in E(\gamma)} n^{-(\frac{k}{2}+\varepsilon)l_e} \leq \frac{1}{n^{B\varepsilon(|V(\gamma)\backslash(U_\gamma\cap V_\gamma)|+\sum_{e\in E(\gamma)} l_e)}}$$

for a constant $B$ that depends only on $C_\Delta$. ∎

**Remark J.5.** *In the above bounds, note that there is a decay of $n^{B\varepsilon}$ for each vertex in $V(\gamma)\setminus(U_\gamma\cap V_\gamma)$. One of the main technical reasons for introducing the slack parameter $C_\Delta$ in the planted distribution was to introduce this decay.*

We can now obtain the intersection term bounds.

**Lemma J.6.** *For all $U, V \in \mathcal{I}_{mid}$ where $w(U) > w(V)$ and all $\gamma \in \Gamma_{U,V}$,*

$$c(\gamma)^2 N(\gamma)^2 B(\gamma)^2 H_{Id_V}^{-\gamma,\gamma} \preceq H'_\gamma$$

*Proof.* By Lemma D.7, we have

$$c(\gamma)^2 N(\gamma)^2 B(\gamma)^2 H_{Id_V}^{-\gamma,\gamma} \preceq c(\gamma)^2 N(\gamma)^2 B(\gamma)^2 S(\gamma)^2 \frac{|Aut(U)|}{|Aut(V)|} H'_\gamma$$

Using the same proof as in Lemma D.4, we can see that $H'_\gamma \succeq 0$. Therefore, it suffices to prove that $c(\gamma)^2 N(\gamma)^2 B(\gamma)^2 S(\gamma)^2 \frac{|Aut(U)|}{|Aut(V)|} \leq 1$. Since $U, V \in \mathcal{I}_{mid}$, $|Aut(U)| = |U|!, |Aut(V)| = |V|!$. Therefore, $\frac{|Aut(U)|}{|Aut(V)|} = \frac{|U|!}{|V|!} \leq D_V^{|U_\gamma\setminus V_\gamma|}$. Also, $|E(\gamma)| \leq \sum_{e\in E(\gamma)} l_e$ and $q = n^{O(1)\cdot\varepsilon(C_V+C_E)}$. Let $B$ be the constant from Lemma J.4. We can set $C_V, C_E$ sufficiently small so that, using Lemma J.4,

$$c(\gamma)^2 N(\gamma)^2 B(\gamma)^2 S(\gamma)^2 \frac{|Aut(U)|}{|Aut(V)|} \leq n^{O(1)\cdot\varepsilon(C_V+C_E)\cdot(|V(\gamma)\backslash(U_\gamma\cap V_\gamma)|+\sum_{e\in E(\gamma)} l_e)} \cdot n^{w(V(\gamma)\backslash U_\gamma)} S(\gamma)^2$$

$$\leq 1$$

∎

## J.3 Truncation error bounds

In this section, we will obtain the truncation error bounds using the strategy sketched in Appendix I. We also reuse the notation. First, we need the following bound on $B_{norm}(\sigma)B_{norm}(\sigma')H_{Id_U}(\sigma, \sigma')$.

**Lemma J.7.** *Suppose* $\lambda = n^{\frac{k}{4}-\varepsilon}$. *For all* $U \in \mathcal{I}_{mid}$ *and* $\sigma, \sigma' \in \mathcal{L}_U$,

$$B_{norm}(\sigma)B_{norm}(\sigma')H_{Id_U}(\sigma, \sigma') \leq \frac{1}{n^{0.5\varepsilon C_\Delta |V(\sigma \circ \sigma')|}\Delta^{D_{sos}}n^{|U|}}$$

*Proof.* Suppose there is a vertex $i \in V(\sigma) \setminus V_\sigma$ such that $deg^\sigma(i) + deg^{U_\sigma}(i)$ is odd, then $H_{Id_U}(\sigma, \sigma') = 0$ and the inequality is true. So, assume that $deg^\sigma(i) + deg^{U_\sigma}(i)$ is even for all $i \in V(\sigma) \setminus V_\sigma$. Similarly, assume that $deg^{\sigma'}(i) + deg^{U_{\sigma'}}(i)$ is even for all $i \in V(\sigma') \setminus V_{\sigma'}$. Also, if $\rho_\sigma \neq \rho_{\sigma'}$, we will have $H_{Id_U}(\sigma, \sigma') = 0$ and we'd be done. So, assume $\rho_\sigma = \rho_{\sigma'}$.

Let $\alpha = \sigma \circ \sigma'$. We will first prove that $\sum_{e \in E(\alpha)} kl_e + 2deg(\alpha) \geq 2|V(\alpha)| + 2|U|$. Firstly, note that all vertices $i \in V(\alpha) \setminus (U_\alpha \cup V_\alpha)$ have $deg^\alpha(i)$ to be even and nonzero, and hence at least 2. Moreover, in both the sets $U_\alpha \setminus (U_\alpha \cap V_\alpha)$ and $V_\alpha \setminus (U_\alpha \cap V_\alpha)$, there are at least $|U| - |U_\alpha \cap V_\alpha|$ vertices of degree at least 1, because $U$ is a minimum vertex separator. Also, note that $deg(\alpha) \geq |U_\alpha| + |V_\alpha|$. This implies that

$$\sum_{e \in E(\alpha)} kl_e + 2deg(\alpha) \geq 2|V(\alpha) \setminus (U_\alpha \cup V_\alpha)| + 2(|U| - |U_\alpha \cap V_\alpha|) + 2(|U_\alpha| + |V_\alpha|) = 2|V(\alpha)| + 2|U|$$

where we used the fact that $U_\alpha \cap V_\alpha \subseteq U$. Finally, by choosing $C_V, C_E$ sufficiently small,

$$B_{norm}(\sigma)B_{norm}(\sigma')H_{Id_U}(\sigma, \sigma') \leq \frac{1}{n^{0.5\varepsilon C_\Delta |V(\alpha)|}\Delta^{D_{sos}}n^{|U|}}$$

where we used the facts $\Delta \leq 1, deg(\alpha) \leq 2D_{sos}$. ∎

We now apply the strategy by showing the following bounds.

**Lemma J.8.** *Whenever* $\|M_\alpha\| \leq B_{norm}(\alpha)$ *for all* $\alpha \in \mathcal{M}'$,

$$\sum_{U \in \mathcal{I}_{mid}} M_{Id_U}^{fact}(H_{Id_U}) \succeq \frac{\Delta^{2D_{sos}^2}}{n^{D_{sos}}}Id_{sym}$$

*Proof.* For $V \in \mathcal{I}_{mid}$, $\lambda_V = \frac{1}{n^{|V|}}$. We then choose $w_V = \left(\frac{1}{n}\right)^{D_{sos} - |V|}$. For all left shapes $\sigma \in \mathcal{L}_V$, it's easy to verify $w_V \leq \frac{w_{U_\sigma}\lambda_{U_\sigma}}{|\mathcal{I}_{mid}|B_{norm}(\sigma)^2 c(\sigma)^2 H_{Id_V}(\sigma, \sigma)}$ using Lemma J.7. Theorem I.6 completes the proof. ∎

**Lemma J.9.** $\sum_{U \in \mathcal{I}_{mid}} \sum_{\gamma \in \Gamma_{U,*}} \frac{d_{Id_U}(H_{Id_U}, H'_\gamma)}{|Aut(U)|c(\gamma)} \leq \frac{1}{\Delta^{2D_{sos}}2^{D_V}}$.

*Proof.* Using the definition, we get

$$\sum_{U \in \mathcal{I}_{mid}} \sum_{\gamma \in \Gamma_{U,*}} \frac{d_{Id_U}(H_{Id_U}, H'_\gamma)}{|Aut(U)|c(\gamma)} \leq \sum_{U \in \mathcal{I}_{mid}} \sum_{\sigma, \sigma' \in \mathcal{L}'_U} \frac{1}{n^{0.5\varepsilon C_\Delta |V(\sigma \circ \sigma')|}\Delta^{D_{sos}}2^{\min(m_\sigma, m_{\sigma'})-1}}$$

where we used Lemma J.7. Using $n^{0.5C_\Delta |V(\sigma \circ \sigma')|} \geq n^{0.1\varepsilon C_\Delta |V(\sigma \circ \sigma')|}2^{|V(\sigma \circ \sigma')|}$,

$$\sum_{U \in \mathcal{I}_{mid}} \sum_{\gamma \in \Gamma_{U,*}} \frac{d_{Id_U}(H_{Id_U}, H'_\gamma)}{|Aut(U)|c(\gamma)} \leq \sum_{U \in \mathcal{I}_{mid}} \sum_{\sigma, \sigma' \in \mathcal{L}'_U} \frac{1}{D_{sos}^{D_{sos}}n^{0.1\varepsilon C_\Delta |V(\sigma \circ \sigma')|}\Delta^{2D_{sos}}2^{D_V}}$$

where we set $C_{sos}$ small enough so that $D_{sos} = n^{\varepsilon C_{sos}} \leq n^{c\varepsilon C_\Delta} = \frac{1}{\Delta}$. The final step will be to argue that $\sum_{U \in \mathcal{I}_{mid}} \sum_{\sigma, \sigma' \in \mathcal{L}'_U} \frac{1}{D_{sos}^{D_{sos}}n^{0.1C_\Delta \varepsilon |V(\sigma \circ \sigma')|}} \leq 1$ which will complete the proof. But this will follow from Lemma I.2 if we set $C_V, C_E$ small enough. ∎

We can finally complete the analysis of the truncation error.

**Lemma J.10.** *Whenever* $\|M_\alpha\| \leq B_{norm}(\alpha)$ *for all* $\alpha \in \mathcal{M}'$,

$$\sum_{U \in \mathcal{I}_{mid}} M_{Id_U}^{fact}(H_{Id_U}) \succeq 6\left(\sum_{U \in \mathcal{I}_{mid}} \sum_{\gamma \in \Gamma_{U,*}} \frac{d_{Id_U}(H'_\gamma, H_{Id_U})}{|Aut(U)|c(\gamma)}\right)Id_{sym}$$

*Proof.* Choose $C_{sos}$ sufficiently small so that $\frac{\Delta^{2D_{sos}^2}}{n^{D_{sos}}} \geq \frac{6}{\Delta^{2D_{sos}}2^{D_V}}$ which is satisfied by setting $C_{sos} < 0.5C_V$. Then, since $Id_{Sym} \succeq 0$, Lemma J.8 and Lemma J.9 imply the result. ∎

# K Sparse PCA: Quantitative bounds

In this section, we will verify the tradeoffs as in Theorem A.1. We already showed the relevant qualitative bounds in Appendix E. We use the bounds and also the notation from that section. In this section, let $n = \max(d, m)$. We just need to verify the conditions of Theorem F.108.

## K.1 Middle shape bounds

**Lemma K.1.** *Suppose* $0 < A < \frac{1}{4}$ *is a constant such that* $\frac{\sqrt{\lambda}}{\sqrt{k}} \leq d^{-A\varepsilon}$ *and* $\frac{1}{\sqrt{k}} \leq d^{-2A}$*. For all* $m$ *such that* $m \leq \frac{d^{1-\varepsilon}}{\lambda^2}, m \leq \frac{k^{2-\varepsilon}}{\lambda^2}$*, for all* $U \in \mathcal{I}_{mid}$ *and* $\tau \in \mathcal{M}_U$*, suppose* $deg^\tau(i)$ *is even for all* $i \in V(\tau) \setminus U_\tau \setminus V_\tau$*, then*

$$\sqrt{d}^{|\tau|_1 - |U_\tau|_1} \sqrt{m}^{|\tau|_2 - |U_\tau|_2} S(\tau) \leq \prod_{j \in V_2(\tau) \setminus U_\tau \setminus V_\tau} (deg^\tau(j) - 1)!! \cdot \frac{1}{d^{A\varepsilon \sum_{e \in E(\tau)} l_e}}$$

*Proof.* Let $r_1 = |\tau|_1 - |U_\tau|_1, r_2 = |\tau|_2 - |U_\tau|_2$. Since $\Delta \leq 1$, it suffices to prove

$$E := \sqrt{d}^{r_1} \sqrt{m}^{r_2} \left(\frac{k}{d}\right)^{r_1} \left(\frac{\sqrt{\lambda}}{\sqrt{k}}\right)^{\sum_{e \in E(\tau)} l_e} \leq \frac{1}{d^{A\varepsilon \sum_{e \in E(\tau)} l_e}}$$

We will need the following claim.

**Claim K.2.** $\sum_{e \in E(\tau)} l_e \geq 2 \max(r_1, r_2)$.

*Proof.* We will first prove $\sum_{e \in E(\tau)} l_e \geq 2r_1$. For any vertex $i \in V_1(\tau) \setminus U_\tau \setminus V_\tau$, $deg^\tau(i)$ is even and is not 0, hence, $deg^\tau(i) \geq 2$. Any vertex $i \in U_\tau \setminus V_\tau$ cannot have $deg^\tau(i) = 0$ otherwise $U_\tau \setminus \{i\}$ is a vertex separator of strictly smaller weight than $U_\tau$, which is not possible, hence, $deg^\tau(i) \geq 1$. Similarly, for $i \in V_\tau \setminus U_\tau$, $deg^\tau(i) \geq 1$. Also, since $H_\tau$ is bipartite, we have $\sum_{i \in V_1(\tau)} deg^\tau(i) = \sum_{j \in V_2(\tau)} deg^\tau(j) = \sum_{e \in E(\tau)} l_e$. Consider

$$\sum_{e \in E(\tau)} l_e \geq \sum_{i \in V_1(\tau) \setminus U_\tau \setminus V_\tau} deg^\tau(i) + \sum_{i \in (U_\tau)_1 \setminus V_\tau} deg^\tau(i) + \sum_{i \in (V_\tau)_1 \setminus U_\tau} deg^\tau(i) \geq 2r_1$$

We can similarly prove $\sum_{e \in E(\tau)} l_e \geq 2r_2$ ∎

To illustrate the main idea, we will start by proving the weaker bound $E \leq 1$. Observe that our assumptions imply $m \leq \frac{d}{\lambda^2}, m \leq \frac{k^2}{\lambda^2}$ and also, $E \leq \sqrt{d}^{r_1} \sqrt{m}^{r_2} \left(\frac{k}{d}\right)^{r_1} \left(\frac{\sqrt{\lambda}}{\sqrt{k}}\right)^{2\max(r_1, r_2)}$ where we used the fact that $\frac{\sqrt{\lambda}}{\sqrt{k}} \leq d^{-A\varepsilon} \leq 1$.

**Claim K.3.** *For integers* $r_1, r_2 \geq 0$*, if* $m \leq \frac{d}{\lambda^2}$ *and* $m \leq \frac{k^2}{\lambda^2}$*, then,*

$$\sqrt{d}^{r_1} \sqrt{m}^{r_2} \left(\frac{k}{d}\right)^{r_1} \left(\frac{\sqrt{\lambda}}{\sqrt{k}}\right)^{2\max(r_1, r_2)} \leq 1$$

*Proof.* We will consider the cases $r_1 \geq r_2$ and $r_1 < r_2$ separately. If $r_1 \geq r_2$, we have

$$\sqrt{d}^{r_1} \sqrt{m}^{r_2} \left(\frac{k}{d}\right)^{r_1} \left(\frac{\sqrt{\lambda}}{\sqrt{k}}\right)^{2r_1} \leq \sqrt{d}^{r_1} \left(\frac{\sqrt{d}}{\lambda}\right)^{r_2} \left(\frac{k}{d}\right)^{r_1} \left(\frac{\sqrt{\lambda}}{\sqrt{k}}\right)^{2r_1} = \left(\frac{\lambda}{\sqrt{d}}\right)^{r_1 - r_2} \leq \left(\frac{1}{\sqrt{m}}\right)^{r_1 - r_2} \leq 1$$

And if $r_1 < r_2$, we have

$$\sqrt{d}^{r_1} \sqrt{m}^{r_2} \left(\frac{k}{d}\right)^{r_1} \left(\frac{\sqrt{\lambda}}{\sqrt{k}}\right)^{2r_2} \leq \sqrt{d}^{r_1} \left(\frac{k}{\lambda}\right)^{r_2 - r_1} \left(\frac{\sqrt{d}}{\lambda}\right)^{r_1} \left(\frac{k}{d}\right)^{r_1} \left(\frac{\sqrt{\lambda}}{\sqrt{k}}\right)^{2r_2} = 1$$

∎

For the desired bounds, we mimic the same argument while keeping track of factors of $d^\varepsilon$.

**Claim K.4.** *For integers $r_1, r_2 \geq 0$ and an integer $r \geq 2\max(r_1, r_2)$, if $m \leq \frac{d^{1-\varepsilon}}{\lambda^2}$ and $m \leq \frac{k^{2-\varepsilon}}{\lambda^2}$, then,*

$$\sqrt{d}^{r_1} \sqrt{m}^{-r_2} \left(\frac{k}{d}\right)^{r_1} \left(\frac{\sqrt{\lambda}}{\sqrt{k}}\right)^r \leq \left(\frac{1}{d^{A\varepsilon}}\right)^r$$

The result follows by setting $r = \sum_{e \in E(\tau)} l_e$ in the above claim. ∎

**Corollary K.5.** *For all $U \in \mathcal{I}_{mid}$ and $\tau \in \mathcal{M}_U$, we have $c(\tau)B_{norm}(\tau)S(\tau)R(\tau) \leq 1$.*

*Proof.* First, note that if $deg^\tau(i)$ is odd for any vertex $i \in V(\tau) \setminus U_\tau \setminus V_\tau$, then $S(\tau) = 0$ and the inequality is true. So, assume that $deg^\tau(i)$ is even for all $i \in V(\tau) \setminus U_\tau \setminus V_\tau$. Since $\tau$ is a proper middle shape, we have $w(I_\tau) = 0$ and $w(S_{\tau,min}) = w(U_\tau)$. This implies $n^{\frac{w(V(\tau)) + w(I_\tau) - w(S_{\tau,min})}{2}} = \sqrt{d}^{|\tau|_1 - |U_\tau|_1} \sqrt{m}^{|\tau|_2 - |U_\tau|_2}$. As was observed in the proof of Lemma K.1, every vertex $i \in V(\tau) \setminus U_\tau$ or $i \in V(\tau) \setminus V_\tau$ has $deg^\tau(i) \geq 1$ and hence, $|V(\tau) \setminus U_\tau| + |V(\tau) \setminus V_\tau| \leq 4\sum_{e \in E(\tau)} l_e$. Also, $q = d^{O(1) \cdot \varepsilon(C_V + C_E)}$. We can set $C_V, C_E$ sufficiently small so that

$$c(\tau)B_{norm}(\tau)S(\tau)R(\tau) \leq d^{O(1) \cdot (C_V + C_E) \cdot \varepsilon \sum_{e \in E(\tau)} l_e} \cdot (D_V D_E)^{\sum_{e \in E(\tau)} l_e} \cdot \frac{1}{d^{A\varepsilon \sum_{e \in E(\tau)} l_e}} \leq 1$$

∎

We can now obtain our desired middle shape bounds.

**Lemma K.6.** *For all $U \in \mathcal{I}_{mid}$ and $\tau \in \mathcal{M}_U$,*

$$\begin{bmatrix} \frac{1}{|Aut(U)|c(\tau)} H_{Id_U} & B_{norm}(\tau)H_\tau \\ B_{norm}(\tau)H_\tau^T & \frac{1}{|Aut(U)|c(\tau)} H_{Id_U} \end{bmatrix} \succeq 0$$

*Proof.* We have the expression to be equal to

$$\begin{bmatrix} \left(\frac{1}{|Aut(U)|c(\tau)} - \frac{S(\tau)R(\tau)B_{norm}(\tau)}{|Aut(U)|}\right) H_{Id_U} & 0 \\ 0 & \left(\frac{1}{|Aut(U)|c(\tau)} - \frac{S(\tau)R(\tau)B_{norm}(\tau)}{|Aut(U)|}\right) H_{Id_U} \end{bmatrix}$$

$$+ B_{norm}(\tau) \begin{bmatrix} \frac{S(\tau)R(\tau)}{|Aut(U)|} H_{Id_U} & H_\tau \\ H_\tau^T & \frac{S(\tau)R(\tau)}{|Aut(U)|} H_{Id_U} \end{bmatrix}$$

By Lemma E.7, $\begin{bmatrix} \frac{S(\tau)R(\tau)}{|Aut(U)|} H_{Id_U} & H_\tau \\ H_\tau^T & \frac{S(\tau)R(\tau)}{|Aut(U)|} H_{Id_U} \end{bmatrix} \succeq 0$, so the second term above is positive semidefinite. For the first term, by Lemma E.3, $H_{Id_U} \succeq 0$ and by Corollary K.5, $\frac{1}{|Aut(U)|c(\tau)} - \frac{S(\tau)R(\tau)B_{norm}(\tau)}{|Aut(U)|} \geq 0$, which proves that the first term is also positive semidefinite. ∎

## K.2 Intersection term bounds

**Lemma K.7.** *Suppose $0 < A < \frac{1}{4}$ is a constant such that $\frac{\sqrt{\lambda}}{\sqrt{k}} \leq d^{-A\varepsilon}$, $\frac{1}{\sqrt{k}} \leq d^{-2A}$ and $\frac{k}{d} \leq d^{-A\varepsilon}$. For all $m$ such that $m \leq \frac{d^{1-\varepsilon}}{\lambda^2}, m \leq \frac{k^{2-\varepsilon}}{\lambda^2}$, for all $U, V \in \mathcal{I}_{mid}$ where $w(U) > w(V)$ and for all $\gamma \in \Gamma_{U,V}$,*

$$n^{w(V(\gamma) \setminus U_\gamma)} S(\gamma)^2 \leq \left(\prod_{j \in V_2(\gamma) \setminus U_\gamma \setminus V_\gamma} (deg^\gamma(j) - 1)!!\right)^2 \frac{1}{d^{B\varepsilon(|V(\gamma) \setminus (U_\gamma \cap V_\gamma)| + \sum_{e \in E(\gamma)} l_e)}}$$

*for some constant $B > 0$ that depends only on $C_\Delta$. In particular, it is independent of $C_V$ and $C_E$.*

*Proof.* Suppose there is a vertex $i \in V(\gamma) \setminus U_\gamma \setminus V_\gamma$ such that $deg^\gamma(i)$ is odd, then $S(\gamma) = 0$ and the inequality is true. So, assume $deg^\gamma(i)$ is even for all vertices $i \in V(\gamma) \setminus U_\gamma \setminus V_\gamma$. We have $n^{w(V(\gamma)\setminus U_\gamma)} = d^{|\gamma|_1 - |U_\gamma|_1} m^{|\gamma|_2 - |U_\gamma|_2}$. Plugging in $S(\gamma)$, we get that we have to prove

$$E := d^{|\gamma|_1 - |U_\gamma|_1} m^{|\gamma|_2 - |U_\gamma|_2} \left(\frac{k}{d}\right)^{2|\gamma|_1 - |U_\gamma|_1 - |V_\gamma|_1} \Delta^{2|\gamma|_2 - |U_\gamma|_2 - |V_\gamma|_2} \prod_{e \in E(\gamma)} \frac{\lambda^{l_e}}{k^{l_e}} \leq \frac{1}{d^{B\varepsilon(|V(\gamma)\setminus(U_\gamma \cap V_\gamma)| + \sum_{e \in E(\gamma)} l_e)}}$$

Let $S'$ be the set of vertices $i \in U_\gamma \setminus V_\gamma$ that have $deg^\gamma(i) \geq 1$. Let $e, f$ be the number of type 1 vertices and the number of type 2 vertices in $S'$ respectively. Observe that $S' \cup (U_\gamma \cap V_\gamma)$ is a vertex separator of $\gamma$. Let $g = |V_\gamma \setminus U_\gamma|_1$ (resp. $h = |V_\gamma \setminus U_\gamma|_2$) be the number of type 1 vertices (resp. type 2 vertices) in $V_\gamma \setminus U_\gamma$. We first claim that $d^e m^f \geq d^g m^h$. To see this, note that the vertex separator $S' \cup (U_\gamma \cap V_\gamma)$ has weight $\sqrt{d}^{e + |U_\gamma \cap V_\gamma|_1} \sqrt{m}^{f + |U_\gamma \cap V_\gamma|_2}$. On the other hand, $V_\gamma$ has weight $\sqrt{d}^{g + |U_\gamma \cap V_\gamma|_1} \sqrt{m}^{h + |U_\gamma \cap V_\gamma|_2}$. Since $\gamma$ is a left shape, $V_\gamma$ is the unique minimum vertex separator and hence, $\sqrt{d}^{e + |U_\gamma \cap V_\gamma|_1} \sqrt{m}^{f + |U_\gamma \cap V_\gamma|_2} \geq \sqrt{d}^{g + |U_\gamma \cap V_\gamma|_1} \sqrt{m}^{h + |U_\gamma \cap V_\gamma|_2}$ which implies $d^e m^f \geq d^g m^h$. Let $p = |V(\gamma) \setminus (U_\gamma \cup V_\gamma)|_1$ (resp. $q = |V(\gamma) \setminus (U_\gamma \cup V_\gamma)|_2$) be the number of type 1 vertices (resp. type 2 vertices) in $V(\gamma) \setminus (U_\gamma \cup V_\gamma)$. To illustrate the main idea, we will first prove the weaker inequality $E \leq 1$. Since $\Delta \leq 1$, it suffices to prove

$$d^{|\gamma|_1 - |U_\gamma|_1} m^{|\gamma|_2 - |U_\gamma|_2} \left(\frac{k}{d}\right)^{2|\gamma|_1 - |U_\gamma|_1 - |V_\gamma|_1} \prod_{e \in E(\gamma)} \frac{\lambda^{l_e}}{k^{l_e}} \leq 1$$

We have $d^{|\gamma|_1 - |U_\gamma|_1} m^{|\gamma|_2 - |U_\gamma|_2} = d^{p+g} m^{q+h} \leq n^{p + \frac{e+g}{2}} m^{q + \frac{f+h}{2}}$ since $d^e m^f \geq d^g m^h$. Also, $2|\gamma|_1 - |U_\gamma|_1 - |V_\gamma|_1 = 2p + e + g$. So, it suffices to prove

$$n^{p + \frac{e+g}{2}} m^{q + \frac{f+h}{2}} \left(\frac{k}{d}\right)^{2p+e+g} \prod_{e \in E(\gamma)} \left(\frac{\lambda}{k}\right)^{l_e} \leq 1$$

We will first prove that $\sum_{e \in E(\gamma)} l_e \geq \max(2p + e + g, 2q + f + h)$. Since $H_\gamma$ is bipartite, we have $\sum_{e \in E(\gamma)} l_e = \sum_{i \in V_1(\gamma)} deg^\gamma(i) = \sum_{i \in V_2(\gamma)} deg^\gamma(i)$. Observe that all vertices $i \in V(\gamma) \setminus U_\gamma \setminus V_\gamma$ have $deg^\gamma(i)$ nonzero and even, and hence, $deg^\gamma(i) \geq 2$. Then,

$$\sum_{e \in E(\gamma)} l_e = \sum_{i \in V_1(\gamma)} deg^\gamma(i)$$

$$\geq \sum_{i \in V_1(\gamma) \setminus U_\gamma \setminus V_\gamma} deg^\gamma(i) + \sum_{i \in (U_\gamma)_1 \setminus V_\gamma} deg^\gamma(i) + \sum_{i \in (V_\gamma)_1 \setminus U_\gamma} deg^\gamma(i)$$

$$\geq 2p + e + g$$

Similarly, $\sum_{e \in E(\gamma)} l_e \geq 2q + f + h$. Therefore, $\sum_{e \in E(\gamma)} l_e \geq \max(2p + e + g, 2q + f + h)$

Now, let $r_1 = p + \frac{e+g}{2}, r_2 = q + \frac{f+h}{2}$. Then, $\sum_{e \in E(\gamma)} l_e \geq 2\max(r_1, r_2)$ and we wish to prove $d^{r_1} m^{r_2} \left(\frac{k}{d}\right)^{2r_1} \left(\frac{\lambda}{k}\right)^{2\max(r_1, r_2)} \leq 1$ This expression simply follows by squaring Claim K.3. Now, to prove that $E \leq \frac{1}{d^{B\varepsilon(|V(\gamma)\setminus(U_\gamma \cap V_\gamma)| + \sum_{e \in E(\gamma)} l_e)}}$, we mimic this argument while keeping track of factors of $d^\varepsilon$. ∎

**Remark K.8.** *In the above bounds, note that there is a decay of $d^{B\varepsilon}$ for each vertex in $V(\gamma) \setminus (U_\gamma \cap V_\gamma)$. One of the main technical reasons for introducing the slack parameter $C_\Delta$ in the planted distribution and the conditions involving the parameter $A$ was precisely to introduce this decay.*

With this, we obtain intersection term bounds.

**Lemma K.9.** *For all $U, V \in \mathcal{I}_{mid}$ where $w(U) > w(V)$ and all $\gamma \in \Gamma_{U,V}$, $c(\gamma)^2 N(\gamma)^2 B(\gamma)^2 H_{Id_V}^{-\gamma, \gamma} \preceq H'_\gamma$*

*Proof.* Using the same proof as in Lemma E.3, we can see that $H'_\gamma \succeq 0$. Therefore, by Lemma E.8, it suffices to prove that $c(\gamma)^2 N(\gamma)^2 B(\gamma)^2 S(\gamma)^2 R(\gamma)^2 \frac{|Aut(U)|}{|Aut(V)|} \leq 1$. Let $B$ be the constant from Lemma K.7. We can set $C_V, C_E$ sufficiently small so that Lemma K.7 implies the result. ∎

### K.3 Truncation error bounds

In this section, we will obtain truncation error bounds using the strategy sketched in Appendix I. We also reuse the notation. To start with, we obtain a bound on $B_{norm}(\sigma)B_{norm}(\sigma')H_{Id_U}(\sigma, \sigma')$.

**Lemma K.10.** *Suppose $0 < A < \frac{1}{4}$ is a constant such that $\frac{\sqrt{\lambda}}{\sqrt{k}} \leq d^{-A\varepsilon}$ and $\frac{1}{\sqrt{k}} \leq d^{-2A}$. Suppose $m$ is such that $m \leq \frac{d^{1-\varepsilon}}{\lambda^2}, m \leq \frac{k^{2-\varepsilon}}{\lambda^2}$. For all $U \in \mathcal{I}_{mid}$ and $\sigma, \sigma' \in \mathcal{L}_U$,*

$$B_{norm}(\sigma)B_{norm}(\sigma')H_{Id_U}(\sigma, \sigma') \leq \frac{1}{d^{0.5A\varepsilon(|V(\sigma\circ\sigma')|+\sum_{e\in E(\alpha)}l_e)}} \cdot \frac{1}{d^{|U_\sigma|_1+|U_{\sigma'}|_1}m^{|U_{\sigma'}|_2+|U_{\sigma'}|_2}}$$

*Proof.* Suppose there is a vertex $i \in V(\sigma) \setminus V_\sigma$ such that $deg^\sigma(i) + deg^{U_\sigma}(i)$ is odd, then $H_{Id_U}(\sigma, \sigma') = 0$ and the inequality is true. So, assume that $deg^\sigma(i) + deg^{U_\sigma}(i)$ is even for all $i \in V(\sigma) \setminus V_\sigma$. Similarly, assume that $deg^{\sigma'}(i) + deg^{U_{\sigma'}}(i)$ is even for all $i \in V(\sigma') \setminus V_{\sigma'}$. Also, if $\rho_\sigma \neq \rho_{\sigma'}$, we will have $H_{Id_U}(\sigma, \sigma') = 0$ and we would be done. So, assume $\rho_\sigma = \rho_{\sigma'}$.

Let there be $e$ (resp. $f$) vertices of type 1 (resp. type 2) in $V(\sigma) \setminus U_\sigma \setminus V_\sigma$. Then, $n^{\frac{w(V(\sigma))-w(U)}{2}} = \sqrt{d}^{|V(\sigma)|_1-|U|_1}\sqrt{m}^{|V(\sigma)|_2-|U|_2} = \sqrt{d}^{|U_\sigma|_1}\sqrt{m}^{|U_\sigma|_2}\sqrt{d}^e\sqrt{m}^f$. Let there be $g$ (resp. $h$) vertices of type 1 (resp. type 2) in $V(\sigma') \setminus U_{\sigma'} \setminus V_{\sigma'}$. Then, similarly, $n^{\frac{w(V(\sigma'))-w(U)}{2}} \leq \sqrt{d}^{|U_{\sigma'}|_1}\sqrt{m}^{|U_{\sigma'}|_2}\sqrt{d}^g\sqrt{m}^h$.

Let $\alpha = \sigma \circ \sigma'$. Since all vertices in $V(\alpha) \setminus U_\alpha \setminus V_\alpha$ have degree at least 2, we have $\sum_{e\in E(\alpha)}l_e \geq \sum_{i\in V_1(\alpha)\setminus U_\alpha\setminus V_\alpha}deg^\alpha(i) \geq 2(e+g)+|U_\sigma|_1+|U_\sigma|_2$. Similarly, $\sum_{e\in E(\alpha)}l_e \geq 2(f+h)+|U_{\sigma'}|_1+|U_{\sigma'}|_2$. Therefore, by setting $r_1 = e+g, r_2 = f+h$ in Claim K.4, we have

$$\sqrt{d}^{e+g}\sqrt{m}^{f+h}\left(\frac{k}{d}\right)^{e+g}\prod_{e\in E(\alpha)}\frac{\sqrt{\lambda}^{l_e}}{\sqrt{k}^{l_e}} \leq \frac{1}{d^{A\varepsilon\sum_{e\in E(\alpha)}l_e}}$$

Also, $\left(\frac{k}{d}\right)^{|\alpha|_1} \leq \left(\frac{k}{d}\right)^{e+g+|U_\sigma|_1+|U_{\sigma'}|_1}$ and $\prod_{j\in V_2(\alpha)}(deg^\alpha(j)-1)!! \leq d^{\varepsilon C_V\sum_{e\in E(\alpha)}l_e}$. Therefore,

$$n^{\frac{w(V(\sigma))-w(U)}{2}}n^{\frac{w(V(\sigma'))-w(U)}{2}}H_{Id_U}(\sigma,\sigma')$$

$$\leq d^{O(1)D_{sos}}d^{\varepsilon C_V\sum_{e\in E(\alpha)}l_e}\sqrt{d}^{e+g}\sqrt{m}^{f+h}\left(\frac{k}{d}\right)^{e+g}\prod_{e\in E(\alpha)}\frac{\sqrt{\lambda}^{l_e}}{\sqrt{k}^{l_e}}\cdot\frac{1}{d^{|U_\sigma|_1+|U_{\sigma'}|_1}m^{|U_{\sigma'}|_2+|U_{\sigma'}|_2}}$$

$$\leq \frac{d^{\varepsilon C_V\sum_{e\in E(\alpha)}l_e}}{d^{A\varepsilon\sum_{e\in E(\alpha)}l_e}}\cdot\frac{1}{d^{|U_\sigma|_1+|U_{\sigma'}|_1}m^{|U_{\sigma'}|_2+|U_{\sigma'}|_2}}$$

By setting $C_V, C_E$ sufficiently small and plugging in the expressions for $B_{norm}(\sigma), B_{norm}(\sigma')$, we obtain the result. ∎

We can apply the the strategy now.

**Lemma K.11.** *Whenever $\|M_\alpha\| \leq B_{norm}(\alpha)$ for all $\alpha \in \mathcal{M}'$,*

$$\sum_{U\in\mathcal{I}_{mid}}M_{Id_U}^{fact}(H_{Id_U}) \succeq \frac{1}{d^{K_1D_{sos}^2}}Id_{sym}$$

*for a constant $K_1 > 0$ that can depend on $C_\Delta$.*

*Proof.* We will use Theorem I.6. For $V \in \mathcal{I}_{mid}$, $\lambda_V = \frac{\Delta^{|V|_2}}{d^{|V|_1}k^{|V|_2}}$. Let the minimum value of this quantity over all $V$ be $N$. We then choose $w_V = N/\lambda_V$ so that for all left shapes $\sigma \in \mathcal{L}_V$, Lemma K.10 implies $w_V \leq \frac{w_{U_\sigma}\lambda_{U_\sigma}}{|\mathcal{I}_{mid}|B_{norm}(\sigma)^2c(\sigma)^2H_{Id_V}(\sigma,\sigma)}$, completing the proof. ∎

**Lemma K.12.**

$$\sum_{U\in\mathcal{I}_{mid}}\sum_{\gamma\in\Gamma_{U,*}}\frac{d_{Id_U}(H_{Id_U},H'_\gamma)}{|Aut(U)|c(\gamma)} \leq \frac{d^{K_2D_{sos}}}{2^{D_V}}$$

*for a constant $K_2 > 0$ that can depend on $C_\Delta$.*

*Proof.* We do the same calculations as in the proof of Lemma J.9, until

$$\sum_{U \in \mathcal{I}_{mid}} \sum_{\gamma \in \Gamma_{U,*}} \frac{d_{Id_U}(H_{Id_U}, H'_\gamma)}{|Aut(U)|c(\gamma)} \leq \sum_{U \in \mathcal{I}_{mid}} \sum_{\sigma, \sigma' \in \mathcal{L}'_U} \frac{d^{O(1)D_{sos}}}{d^{0.5A\varepsilon|V(\sigma \circ \sigma')|}2^{\min(m_\sigma, m_{\sigma'})-1}}$$

where we used Lemma K.10. Using $d^{0.5A\varepsilon|V(\sigma \circ \sigma')|} \geq d^{0.1A\varepsilon|V(\sigma \circ \sigma')|}2^{|V(\sigma \circ \sigma')|}$,

$$\sum_{U \in \mathcal{I}_{mid}} \sum_{\gamma \in \Gamma_{U,*}} \frac{d_{Id_U}(H_{Id_U}, H'_\gamma)}{|Aut(U)|c(\gamma)} \leq \sum_{U \in \mathcal{I}_{mid}} \sum_{\sigma, \sigma' \in \mathcal{L}'_U} \frac{d^{O(1)D_{sos}}}{D_{sos}^{D_{sos}} d^{0.1A\varepsilon|V(\sigma \circ \sigma')|}2^{D_V}}$$

The final step will be to argue that $\sum_{U \in \mathcal{I}_{mid}} \sum_{\sigma, \sigma' \in \mathcal{L}'_U} \frac{1}{D_{sos}^{D_{sos}} d^{0.1A\varepsilon|V(\sigma \circ \sigma')|}} \leq 1$ which will complete the proof. But this will follow from Lemma I.2 if we set $C_V, C_E$ small enough. ∎

We can finally show that truncation errors can be handled.

**Lemma K.13.** *Whenever $\|M_\alpha\| \leq B_{norm}(\alpha)$ for all $\alpha \in \mathcal{M}'$,*

$$\sum_{U \in \mathcal{I}_{mid}} M_{Id_U}^{fact}(H_{Id_U}) \succeq 6 \left( \sum_{U \in \mathcal{I}_{mid}} \sum_{\gamma \in \Gamma_{U,*}} \frac{d_{Id_U}(H'_\gamma, H_{Id_U})}{|Aut(U)|c(\gamma)} \right) Id_{sym}$$

*Proof.* Choose $C_{sos}$ sufficiently small so that $\frac{1}{d^{K_1 D_{sos}^2}} \geq 6\frac{d^{K_2 D_{sos}}}{2^{D_V}}$ which can be satisfied by setting $C_{sos} < K_3 C_V$ for a sufficiently small constant $K_3 > 0$. Then, since $Id_{Sym} \succeq 0$, Lemma K.11 and Lemma K.12 imply the result. ∎