# OpenReview forum: "Sub-exponential time Sum-of-Squares lower bounds for Principal Components Analysis"
_NeurIPS.cc/2022/Conference — NeurIPS 2022 Accept_

### Official Review · Reviewer_Q86G · 2022-07-11

**Rating:** 7
**Confidence:** 3
**Soundness:** 2 fair
**Presentation:** 2 fair
**Contribution:** 3 good

**Summary:**

This is a theoretical work that studies the applicability of sum of squares (SoS) algorithms for sparse principal component analysis (PCA) and tensor PCA. Specifically, this work provides theoretical proof that despite the success of SOS algorithms in high dimensional statistics, they underperform when applied to sparse PCA and tensor PCA. The paper claims that sub-exponential time SOS algorithms cannot beat traditional sparse PCA algorithms, even if allowed sub-exponential time.

**Questions:**

1. For sparse PCA, it seems like this work focuses on recovering a single principal component. If so, could the authors clarify why they focus on just a single component and how one could extend this to multiple components?
2. Paper claims that the proposed theory is more general compared to existing results because the existing results assume an input distribution of +-1, whereas the proposed work relies on the distribution N(0,1). How does this make the proposed theory more general? Please provide intuition in terms of real-world examples.
3. Have SoS algorithms been used for sparse PCA in the past? If so, do they perform well? Can the authors comment on this?
4. Although the proposed theory is useful in itself, what is its practical applicability? It would be nice to see an empirical demonstration of the theory in practice. For example, demonstrate on real-world (or synthetic) data that sub-exponential time SoS algorithms are unable to recover sparse PCA solution while simple traditional algorithms succeed.
5. Repeated word "devise" in line 383. Please remove one of them.


**Limitations:**

The paper identifies relevant future directions of research based on limitations of the current work.

**Strengths And Weaknesses:**

Strengths:
1. The text is easy to follow.
2. Visual illustrations (e.g., in Figure 1 ) assist in understanding the claims of the paper.
3. The paper is theoretical and appears to propose a broad result on the usefulness of SOS algorithms for sparse PCA and tensor PCA.

Weaknesses:
1. The flow of the paper can be further improved.
2. Is the proposed theory relevant in practice? Please elaborate. For example, theorem 3.1 assumes a range for m and theorem 3.2 assumes an upper limit for \lambda. Are these valid assumptions in practice?
3. Figures are referred to as "Fig.". Please correct them.
4. Equations are not referenced. Please number them for easy reference.
5. Line 124 consists of a tensor product notation and it is not explained. Please provide an explanation.

---

> ### Author Response · Authors · 2022-08-02
> **Response to reviewer Q86G**
>
> We thank the reviewer for their feedback. We would like to clarify what appears to be an unfortunate misunderstanding in the review - **It is not the case that SoS fails while traditional methods succeed, nor do we claim this. Instead, it is the case that SoS fails when traditional methods also fail. We show precisely this in our work.** In the other direction, prior works such as [1] have shown that when traditional methods succeed, SoS also generally succeeds. Therefore, we complement their results. We apologize if any sentence or wording in the text was misleading and are happy to take specific pointers from the reviewer as to where this confusion arose.
>
> We first address the questions below
> 1. We analyze sparse PCA with one component because it is considerably simpler but is already a fascinating model to study with rich structure, and as we remark in L93-94, if SoS fails to solve this simple case then it is unlikely to do well on more complicated models. That said, finding the threshold where SoS succeeds or fails when there are multiple components is an interesting and deep question.
> If we wanted to analyze sparse PCA with multiple components, we would need to have solution variables for each component. Our techniques can be used to analyze this case, but we leave this for future work.
> 2. On one hand, our techniques can recover the +/-1 results that were claimed in [2] without proof, so they are more general. More specifically, as also stated in the short note above in a separate comment, to recover the results of [2], we simply need to change certain parameters such as B_norm(.), and the rest of the arguments go through.
> On the other hand, Gaussian N(0, 1) noise is more common than Boolean +/-1 noise in real world data (owing partly to the central limit theorem). An example is the standard usage of least squares loss for linear regression, which can be interpreted as maximizing the likelihood of the data under a linear model with additive Gaussian noise.
> 3. The work [1] uses SoS algorithms for sparse PCA, where they show that SoS performs just as well as (and sometimes better than) traditional algorithms in most settings. We highlight this in L268-278. Please see also their experiments in [1, section 9].
> 4. This is a good question. Regarding the last sentence, we would like to re-emphasize that we do not show that SoS fails when simple algorithms succeed. That said, although we would like to verify our results experimentally, the implementation will suffer from numerical and inefficiency issues. Moreover, it’s plausible that we won’t see the desired effects of the random instances (such as matrix norm bounds) until the input size is fairly large and at this level, sub-exponential time algorithms may take a long time. Therefore, we leave experimental validation for future work. For experiments illustrating success of SoS in the other regimes not considered here, please see [1, section 9]. Regarding other applications, please see the paragraph below.
> 5. Fixed, thank you.
>
> Regarding whether the assumptions are reasonable, our assumptions (almost) tightly complement parameter regimes where prior works have proposed algorithms. In practice however, it’s not clear apriori which regime applies, but an important takeaway of our work is that in practice, simple spectral algorithms such as the ones proposed in [1] (and references therein) are in a certain sense optimal. In a broader sense, as the reviewer rightly observed, the motivation of this work is theoretical in nature and serves partly as a study of information-computation tradeoffs exhibited by high-dimensional statistical models, an important research endeavor in theoretical machine learning.
>
> References:
>
> [1] T. d’Orsi, P. K. Kothari, G. Novikov, and D. Steurer. Sparse PCA: Algorithms, adversarial perturbations and certificates. 2020. https://arxiv.org/pdf/2011.06585.pdf
>
> [2] Samuel B Hopkins, Pravesh K Kothari, Aaron Potechin, Prasad Raghavendra, Tselil Schramm, and David Steurer. The power of sum-of-squares for detecting hidden structures. 2017

---

> > ### Comment · Reviewer_Q86G · 2022-08-07
> > **Reviewer response to author rebuttal**
> >
> > Thank you for your response. I stand corrected regarding my comment on the failure of SoS while traditional algorithms succeed. I believe I was misled by lines L82-86, where the claim is that SoS cannot beat traditional sparse PCA algorithms, leading me to wrongly believe that while traditional sparse PCA algorithms succeed, SoS algorithms fail. I have edited my review to reflect the same.
> >
> > I am convinced with the answers to all other questions I posed in my review. I am increasing my score to 7: Accept.

---

> > > ### Author Response · Authors · 2022-08-07
> > > **Thanks for your updated review**
> > >
> > > We kindly thank the reviewer for reading our response and updating their review and score. We are happy that the response was able to address their concerns and welcome any additional suggestions to improve the text.

---

### Official Review · Reviewer_TgBm · 2022-07-16

**Rating:** 7
**Confidence:** 5
**Soundness:** 4 excellent
**Presentation:** 2 fair
**Contribution:** 3 good

**Summary:**

Sum-of-squares is a family of algorithms that capture many known techniques for combinatorial optimization problems, and it has been recently shown to be very powerful for many ML/robust statistics problems. For the theoretical computer science community, some has seen this as a proxy to strong algorithmic impossibility result especially in the average-case setting.  This paper rigorously establishes strong Sum-of-Squares lower bounds at n^\eps degree for several problems in the colloquial “dense-graph” regime, including tensor PCA, sparse PCA both of which have been claimed by a previous work of Hopkins et al. [HKPRSS17] but without an explicit, spelled-out proof. On the technical level, it builds upon the SoS lower bound for Planted Clique by Barak et al [BHKKMP16] that they use the recursive factorization framework to show PSDness of the candidate moment matrix obtained from pseudo-calibration with intense matrix analysis. For the specific hardness this work obtains, not much is known before except deg-2 and deg-4 lower bound in the SoS realm, and this work obtains almost tight hardness matching known algorithms.


**Questions:**

1, "Their techniques fail for the Wishart model of Sparse PCA, which is more natural in practice. We overcome this shortcoming and work with the Wishart model. We emphasize that their techniques are insufficient to handle this generality and overcoming this is far from being a mere technicality."
As surprising as it may be, this is quite strong and cryptic a statement. Apology if I missed something in the supplementary material, is there a way to highlight/encapsulate the difference from your technique should one try to apply the recursive factorization framework from planted clique in a somewhat straightforward manner?

2. For Sparse PCA constraint of supporting on $k$ coordinates, is this constrained exactly satisfied by the pseudo-expectation or only on average? I suppose its the former, and it seems nice (and even important) to point out in the work if not yet considering the struggle for satisfying exact constraints in sos lower bounds.

**Ethics Review Area:**

["I don’t know"]

**Strengths And Weaknesses:**

Strength: 1) In general, this is detailedly written paper;
                2) Though the hardness result may not be too surprising, the almost tight hardness result of this work is certainly nice to have, and necessary for the community considering how universal these two problems are in ML practice;
               3) Their generalization of the framework from planted clique is likely to find applications in other problems.

Weakness:
 1) As the paper is filled with details, it'd be nice to incorporate more exposition in the technical section. It is hard to imagine someone going through this generalization; and as technical as it already is, more intuition between the technical proofs can be helpful for readability.

2) It is not immediately clear what technical challenges the authors claimed to have overcome from planted clique. They note that the method from planted clique does not immediately work, while the challenges are not explicitly stated.

---

> ### Author Response · Authors · 2022-08-02
> **Response to reviewer TgBm**
>
> We thank the reviewer for their review and positive feedback. We address the questions first
> 1. Interestingly, this requires a detailed response. Please see the short note added in a separate comment on how our work compares with Planted Clique and Hopkins et al. We will be happy to add parts of this discussion in the revision.
> 2. This is a great question. To clarify, the constraint is satisfied on average and also in fact with o(1) error, which is essentially because the planted distribution (L667-670) samples the coordinates identically and independently. We will be happy to point this out explicitly in the revision. Although it is likely possible to fix this using ad-hoc techniques as in [1, Section 7] or [2], a general technique to do this is not known and is an interesting problem for future work.
>
> Regarding the exposition, we are actively improving it. See for example the newly added section E.3 in the revised supplement that intuitively explains why we can obtain almost-tight guarantees for Sparse PCA, without diving deep into the proofs. Also, based on suggestions from other reviewers, we improved notation and added more clarifying examples and figures (e.g. the new figures Figure 2 and Figure 4 in sections C, F and the new examples C.4, C.8, F.48, F.60). We welcome additional suggestions.
>
> References:
>
> [1] Mrinalkanti Ghosh, Fernando Granha Jeronimo, Chris Jones, Aaron Potechin, Goutham Rajendran. Sum-of-Squares lower bounds for Sherrington-Kirkpatrick via Planted Affine Planes. 2020
>
> [2] Shuo Pang. SoS lower bound for exact planted clique. 2020

---

### Official Review · Reviewer_t9p5 · 2022-07-20

**Rating:** 7
**Confidence:** 3
**Soundness:** 3 good
**Presentation:** 2 fair
**Contribution:** 3 good

**Summary:**

Sparse PCA is a fundamental problem in the machine learning and statistical inference community. Suppose we are given vectors $v_1, \dots, v_m \in \mathbb{R}^d$ sampled from the mean-zero Gaussian distribution with covariance $I_d + \lambda uu^\top$ where $u$ is a $k$-sparse vector, the task is to recover the hidden vector $u$ (principal component). This is the *Wishart model*.

There are several parameters in play here: the number of samples $m$, the dimension $d$, the signal-to-noise ratio $\lambda$, and the sparsity $k$. A long line of work has established an almost complete picture of the tractability of sparse PCA in different parameter regimes; in particular, in the “hard” regime where $m \lambda^2 \ll d$ and $m \lambda^2 \ll k^2$, no known algorithm exists.

This paper shows that the powerful Sum-of-Squares (SoS) algorithm fails to solve sparse PCA in this hard regime even with degree $d^{O(1)}$. Specifically, the authors show that the natural SoS relaxation at degree $d^{O(\epsilon)}$ fails at the (easier) task of distinguishing between vectors sampled randomly vs vectors sampled from the spiked model when $m \lambda^2 \ll d^{1-\epsilon}$ and $m \lambda^2 \ll k^{2-\epsilon}$.

Their techniques also apply to tensor PCA. For $k\geq 2$, we are given an order $k$ tensor $A = \lambda u^{\otimes k} + B$ where $u\in \mathbb{R}^n$ is a unit vector and $B$ is a tensor with Gaussian entries, and the task is to recover $u$. They prove analogous SoS lower bounds for the regime $\lambda \ll n^{k/4}$.

**Techniques**

The authors showed that in the hard regime, even when given random inputs (no planted spike), the canonical SoS relaxation has a large optimal value of $m + m\lambda - o(1)$, thus failing to distinguish from the case when the inputs are drawn from the spiked Gaussian. To prove this, the authors constructed a candidate pseudo-distribution from the spiked distribution (aka planted distribution) using the pseudo-calibration method, which is by now a standard technique to prove SoS lower bounds.

As in the SoS lower bound for planted clique [18], they decompose the pseudo-moment matrix into a linear combination of “graph matrices”, which are structured random matrices that can be represented as graphs (called shapes) and whose spectral norms can be bounded by analyzing “vertex separators” in the shape.

Following planted clique [18], they factorize each shape into “left”, “right”, and “middle” shapes and then bound the “intersection terms” (the errors in the factorization). One technicality is that the coefficients in the linear combination also depend on the shapes, which they handle by analyzing the “coefficient matrix”. They show that the middle part is PSD by charging non-trivial shapes to the diagonal, which implies the whole matrix is PSD. The bulk of their proof is in fact handling the intersection terms.

**Questions:**

What are the key innovations in the proofs that are different from Hopkins et al [49]. In particular, why do their techniques fail for the Wishart model?

Where are the requirements $d^A \leq k \leq d^{1-A\epsilon}$ and $\sqrt{\lambda} / \sqrt{k} \leq d^{-A\epsilon}$ used in the proof? They seem to be buried in the details in the proof, and it would be nice if you can explain this in Appendix A.

**Other comments**
- Line 102: “tradeoff between m, n and k” should be m, \lambda, k
- Line 256: what does “sufficient decay” mean?
- Line 757: I don’t think “trivial shapes” were defined before.
- Remark C.18: “improper” not defined (it’s only defined later).

**Limitations:**

The authors adequately addressed the limitations.

**Strengths And Weaknesses:**

**Strengths**

This paper is important as it gives insights to the complexity of sparse/tensor PCA and also the power of SoS algorithms. It provides a strong piece of evidence that the hardness regimes are indeed hard.

Previous work by Hopkins et al [49] claimed SoS lower bounds for sparse PCA in the spiked Wigner model, though their proofs are not online. This paper generalizes their ideas and proves an SoS lower bound for the more natural Wishart model. I view this as the main contribution of this paper.

As in previous papers, proving SoS lower bounds requires a significant amount of work, especially for polynomial-degree. Therefore, even though many techniques in this paper are not new (e.g. factorization of shapes, charging arguments, etc), grinding out all the technical details is a strength in my opinion.

**Weaknesses**

I would like to see a detailed comparison to [49] and why their techniques fail for the Wishart model. I believe that [49] has most of the main ideas even though a full proof hasn’t been posted. There must be some technical challenges hidden in the details, but this was not explained in either the main paper or the appendix.

Finally, it is almost impossible for readers unfamiliar with graph matrices to follow the proofs. Maybe include some example graphs?

---

> ### Author Response · Authors · 2022-08-02
> **Response to reviewer t9p5**
>
> We are thankful to the reviewer for their detailed and positive feedback. To address the questions first,
> 1. Since this requires a more detailed answer, please see the short note we added in a separate comment. We will be happy to incorporate parts of this discussion in the paper.
> 2. The technical requirements show up in the intersection term bounds (lemmas J.4, K.7). We thank the reviewer for bringing this to our attention, we added clarifying remarks to this extent (please see L688-690 in Appendix A and Remarks J.5, K.8 in the revised supplement).
> 3. By sufficient decay, we essentially mean these technical decay factors from above. We added a reference in this line.
>
> We are happy to improve the exposition by adding more examples (we also have references in L920, L1694) and also more explanation/intuition for the technical sections (e.g. the newly added section E.3 in the revised supplement). In the revised supplement, please see the new figures Figure 2 and Figure 4 and the new examples C.4, C.8, F.48, F.60.
>
> > Line 102: “tradeoff between m, n and k” should be m, \lambda, k
>
> > Line 757: I don’t think “trivial shapes” were defined before.
>
> > Remark C.18: “improper” not defined (it’s only defined later).
>
> All of them have been fixed, thanks.

---

> > ### Comment · Reviewer_t9p5 · 2022-08-06
> > **Thank you for your response**
> >
> > Thank you for the comparison to Hopkins et al. in the separate comment. I do think that the coefficient matrix is very interesting, as I have not seen it in other SoS lower bounds. Your paper presented a unified approach that recovers the Wigner and Wishart model for sparse PCA as well as tensor PCA, which is a good technical contribution. I also like the figures you added.
> >
> > I stand by my original review and recommend acceptance.

---

### Official Review · Reviewer_7iRg · 2022-07-21

**Rating:** 7
**Confidence:** 4
**Soundness:** 4 excellent
**Presentation:** 3 good
**Contribution:** 3 good

**Summary:**

This paper fits into the broad area of theoretically understanding information-computation tradeoffs for statistical inference.  The abstract setup is that there is some signal x, typically in the vector space R^d.  And as input, we are given noisy observations about x -- v_1, v_2, ..., v_m.  And the goal is to infer x up to the best accuracy possible via an efficient algorithm.  When m is too small, the problem is impossible to solve, even information theoretically.  At some threshold for m the problem becomes information theoretically identifiable (information threshold) but could possibly remain computationally hard.  And at a (possibly distinct) threshold (computational threshold) for m the problem also becomes algorithmically easy to solve.

Understanding the computational threshold for average-case problems, and in particular if there is a gap in between the information and computational thresholds is a topic that has attracted a lot of attention in the theoretical computer science community.  To understand the computational threshold, one must give an algorithm along with some evidence that the problem is hard under the threshold, which usually comes in the form of lower bounds against particular algorithms (semidefinite programs, statistical queries, polynomial-based methods, eigenvalue methods, iterative algorithms, etc).  A ubiquitous phenomenon for a wide variety of such average-case algorithmic problems is that at some threshold several simple algorithms (and most notably some spectral algorithm) succeed at solving the problem, and under the threshold many seemingly more powerful algorithms fail.

This work is concerned with providing more evidence to our understanding of this computational threshold for the sparse PCA (and also tensor PCA) problems.  In the sparse PCA problem, the hidden signal x is a random d-dimensional vector supported on a much smaller number of coordinates (k coordinates) where the nonzero entries are random in {±1}.  The observations are independent samples from the gaussian distribution with covariance spiked by x: N(0, Identity + lambda xx*), where lambda is some positive scalar (to be interpreted as a "signal-to-noise" parameter).  The main result of this paper is that a certain powerful subexponential-sized semidefinite programming-based algorithm (degree-n^epsilon Sum-of-Squares algorithm) fails to solve sparse PCA better than a simple spectral algorithm, which adds to our understanding of how hard this problem should be.  Similar results are presented for the tensor PCA problem as well.

**Questions:**

1. I encourage the authors to highlight that the distinguishing problem is information theoretically solvable (given exponential time) in some interesting regime covered by the theorem, because if it isn't an SOS lower bound holds for a silly reason.
2. The discussion about SOS being "clearly" more powerful than low-degree polynomials and statistical queries is slightly questionable, and I would advise the authors to remove that line.  For a wide variety of settings we expect SOS to capture what low-degree polynomials can do, and in some settings we think low-degree lower bounds imply SOS lower bounds, but they're formally incomparable.  One example is, if you plant a random coloring in a sparse graph with signal-to-noise ratio below the so-called "Kesten-Stigum threshold", a low-degree polynomial should succeed at distinguishing this planted distribution from a random graph with constant probability, whereas the lingering belief is that SOS SDPs shouldn't solve this problem (since the intuition is that they're limited to what can be solved "with high probability").

**Limitations:**

--

**Strengths And Weaknesses:**

The current state of affairs in obtaining computational hardness for restricted classes of algorithms is that when we want to argue that it's computationally hard to beat a simple spectral, we study the following methods of hardness, which are listed in roughly increasing order of difficulty.
1) Low-degree polynomials
2) Statistical queries
3) Semidefinite programming lower bounds (Sum-of-Squares and this paper's contributions fall here)
4) Reduction-based hardness

(I must note that the gap between the difficulty of 2 and 3 is quite large.)

In the current sociological setting of the subarea, it is a generally widespread belief that a hardness result against 1 or 2 does mean that our current algorithmic techniques likely won't surmount the barriers they are facing, and likely new algorithmic ideas are needed.  An important question to answer to support this belief is to show that hardness results for 1 and 2 (which are often easy to establish) imply hardness results for other algorithms.  Such meta-theorems are hard to come by with no particularly promising attacks for the case of Sum-of-Squares, and therefore the current approach is to substantiate the conjecture via a rich set of example lower bounds, which together hopefully illuminate a path to such meta-theorems.

With respect to this paper's strengths and weaknesses I would like to touch upon two points:
A) Sparse PCA is a problem that has received a lot of attention in theoretical algorithmic statistics, and in light of that evidence for computational thresholds is interesting.  However, I would not say that the paper gave us a surprising answer as to where the computational threshold should lie; the pre-existing lower bounds in the style of #1 and #2 already told us what we think the answer should be.
B) I think the paper's main strength is in progress in substantiating the predictions #1 and #2 have for sparse PCA, since Sum-of-Squares lower bounds are only known for a handful of problems, especially since they are technically difficult to achieve.  It is worthy to note that the best reduction-based hardness (conditional on some conjecture about planted clique) only rules out quasipolynomial time algorithms, as opposed to subexponential algorithms.

Some of the other existing SOS lower bounds (planted clique, CSPs, SK model) highly use problem-specific structure, and the most desirable aspect for any new SOS lower bound is to rely less on the problem-specific structure, and try to abstract conditions needed; it is hard for me to tell whether this lower bound does this given the time constraint of the review, but it would be nice if the authors can comment on this.

Overall, I recommend acceptance.

---

> ### Author Response · Authors · 2022-08-02
> **Response to reviewer 7iRg**
>
> We thank the reviewer for their excellent summary of the field, and their positive feedback. We appreciate the observation that the reviewer brought up
> > ... the best reduction-based hardness … only rules out quasipolynomial time algorithms, as opposed to subexponential algorithms
>
> Hopefully our work fills this gap. We first address the questions raised.
> 1. We thank the reviewer for pointing out this subtle point that is worth mentioning. Indeed, exponential-time algorithms do solve the problem easily. One such algorithm is to simply brute-force through all possible signals (k-sparse boolean vectors), maximize the desired polynomial objective, and finally threshold it.
> 2. We apologize if our writing may have suggested that. To clarify, in general, SoS algorithms and other methods such as SQ or low-degree algorithms are not formally comparable, such as in the example the reviewer pointed out. We also briefly mention this point in L204-205. However, for the sake of clarity and exposition, we will carefully proofread the text and make this explicit.
>
> We now address some other points.
> > Such meta-theorems are hard to come by with no particularly promising attacks for the case of Sum-of-Squares
>
> We believe that our main theorem C.35 is a first step in this direction. Vaguely speaking, the 2nd condition (Middle Shape Bounds in L1073) is a slightly stronger assumption than the failure of low-degree algorithms. Therefore, we believe that a strengthening of our main theorem would potentially lead a way towards such a meta-theorem.
>
> > Some of the other existing SOS lower bounds (planted clique, CSPs, SK model) highly use problem-specific structure, and the most desirable aspect for any new SOS lower bound is to rely less on the problem-specific structure, and try to abstract conditions needed; it is hard for me to tell whether this lower bound does this given the time constraint of the review, but it would be nice if the authors can comment on this.
>
> We build on the point above. A contribution of our theorem C.35 is to provide a general tool for SoS lower bounds. To do this, we abstract out 3 relatively simple conditions (independent of the problem) on coefficient matrices (which can be easily obtained from pseudo-calibration) that our approach needs to show SoS lower bounds. We believe our approach is general enough to find potential applications in other high dimensional statistics problems (also noted by reviewer TgBM), and as explained earlier, a SoS version of the low-degree likelihood ratio conjecture (i.e. a meta-theorem).
>
> References:
>
> [1] Samuel B Hopkins, Pravesh K Kothari, Aaron Potechin, Prasad Raghavendra, Tselil Schramm, and David Steurer. The power of sum-of-squares for detecting hidden structures. 2017

---

### Official Review · Reviewer_xtw5 · 2022-07-22

**Rating:** 5
**Confidence:** 4
**Soundness:** 3 good
**Presentation:** 2 fair
**Contribution:** 2 fair

**Summary:**

This paper develops high-degree lower bounds for the sparse PCA and tensor PCA problems. For both of these problems, there is a well-known discrepancy between the statistical complexity (i.e. #samples or snr required to detect the hidden signal by any means feasible) and computational complexity (i.e. #samples or snr required to approximately recover the hidden signal by computationally efficient i.e. poly time algorithms). (I phrase the above in terms of detection to align with the paper, analogous definitions using recovery may be made with the obvious ordering that detection is easier than recovery). Along with the planted clique problem, sparse and tensor PCA are arguably some of the most well-studied examples of statistical problems exhibiting this computational-statistical phenomenon. While one way of developing hardness arguments is to construct problem reductions, this paper contributes to the line of unconditional hardness results by analyzing the sum-of-squares hierarchy of convex programs for sparse/tensor PCA. The first two steps (also called degrees) in this hierarchy for sparse PCA were shown ineffective by Krauthgamer et al and Wigderson and Ma respectively. This paper develops bounds for the next dimension^{a small const} many degrees.



**Questions:**

The following questions are roughly "chronological" in the paper.
1. L101: what is n for sparse pca? This should be m perhaps?
2. L251: Technically this is a lower bound (by plugging in u in the objective). It is not obvious (though it probably is) to me that this is an upper bound.
3. As I understand it, Thm 3.1 proves a lower bound on the value of the SDP applied to pure noise, i.e. the input covariance matrix is identity and has no dependence on lambda. If so, why is lambda involved in the value? This confused me for a while before I realized presumably this is because your solution (or certificate) is via pseudocalibration and explicitly has an alternative "planted" distribution involving lambda. But then why not sup over lambda in the feasible set? The same question holds for the Thm 3.2.

Some suggestions regarding notation/organization:
1. For sparse PCA it is best to keep the parameters (n, k, d, lambda) as (sample size, sparsity, dimension, snr). Similarly for tensor pca, it is probably better to keep something like (d, r, lambda)  (dimension, tensor order/rank, snr) respectively and avoid the collision use of "k". The choice in the paper appears to be first choosing dimension=n for tensor pca and then avoiding collision by choosing the somewhat confusing 'm' for sparse pca sample size. In terms of readability, this is significantly sub-optimal
2. I'd relegate the "general definition" of sos programs to the appendix, and just write the specific programs for sparse/tensor pca. Perhaps just refer to a textbook for the general definition, and it anyway collides rather unfortunately (d, n etc...) with the notational choices in 1. Alternatively use different fonts to make things obvious.
3. There are multiple points where things were also quite confusing to read in the appendix, which makes it difficult to understand/penetrate the proof techniques. For example, in L.1044 there is V(Id_U), U_{Id_U}, V_{Id_U} and E(Id_U), which is (at least all the same font) very confusing at first go. After a while I realized the first and the last were vertex/edge sets for the whole graph and the middle two were left/right parts respectively. The notation did not help though.

**Strengths And Weaknesses:**

Strengths:
1. I think the paper is an interesting and solid contribution to the hardness results on sparse PCA and tensor PCA. Both problems are (sparse PCA somewhat more than tensor PCA) well-motivated and used in applications.
2. The proof techniques are (to my mind) also potentially interesting to the theory-inclined audience. Proving sum-of-squares lower bounds tends to be a technically challenging, intricate and involved task and the authors do a good job in the beginning of the appendix explaining some of the high-level ideas and challenges.

Weaknesses:
1. At the outset, and given 2, Neurips might not be the best fit for this submission. I would expect COLT, theoretical CS conferences STOC/FOCS and of course, a number of journals, on the other hand to be quite well-suited.
2. The interesting and generalizable part of the paper seem to be some of the proof techniques, and these are relegated to the appendix. I did not have time to read the full appendix (65 pages) but the first 15-20 were useful.
3. Separately from 1, the paper is made inaccessible by a somewhat loose use of notation that is widespread. As a small example, after using 'd' for dimension, the authors also use it for 'degree' of the SOS program in the beginning of Sec. 2. For accessibility to a stats/ml audience, this paper would require significant polishing.
4. Related to 3, I trust the results despite point 2, but some of the main results in the text (Theorems 3.1 and 3.2) likely should be rephrased.

---

> ### Author Response · Authors · 2022-08-02
> **Response to reviewer xtw5**
>
> We thank the reviewer for their careful reading of the work and their detailed feedback. We are glad that they appreciate the main results as well as the proof techniques involved. We address the questions first.
> 1. We apologize for the typo. To clarify, for sparse PCA, the ambient dimension is d and the number of samples is m, but there is no n. We will proofread carefully to ensure n does not appear anywhere.
> 2. We apologize for the confusion with this informal remark. The remark as stated needs to be modified and merely serves to illustrate the $m + m \lambda$ factor. As stated in the formal and precise Theorem 3.1, we only show a lower bound on the SoS objective value as there could be other pseudo-expectation values which give a higher objective value. To be specific, when the samples come from the spiked $N(0,I_d + {\lambda}uu^T)$ and we take $x = \sqrt{k}u$, the objective value is approximately $m + m{\lambda}$. This has been clarified in the revision. To argue this claim, we essentially use the characterization of the spiked samples as in the proof of lemma B.2 and appeal to standard concentration tools. We will be happy to add this proof in our revision.
> 3. The reviewer is correct in their understanding. Our choice of presenting the result in this manner (without taking the supremum over \lambda) is to make clear the connection to the hypothesis testing variant of Sparse PCA; and also make it explicit how our results complement known algorithms. Moreover, as the reviewer rightly observed, this form is also conveniently implied from pseudo-calibration.
>
> We comment on other points the reviewer brought up
> 1. We are thankful for the useful suggestions recommended, especially with regard to notation. We are happy to incorporate them in the revision. For example, in the current revision (uploaded) of section 2, we have opted for the notation $D_{sos}$ instead of d, which also aligns with the notation used in the appendix. We are also willing to accept more feedback on how to improve the text, especially with respect to issues that may have caused the reviewer to lower their score.
> 2. Theorems 3.1 and 3.2 are written to be mathematically rigorous and precise. However, for the sake of exposition, we added their informal counterparts in Theorems 1.1 and 1.2 respectively. We will be happy to rephrase the theorems to make them more intuitive and also welcome suggestions from the reviewer.
> 3. Regarding fit, we find our results to be at the unique intersection of theoretical CS, statistics and machine learning, and therefore, a suitable fit for any of these conferences. We recognize the importance of Sparse PCA especially to the ML and statistics community, e.g., Ma and Wigderson’s popular degree-4 SoS lower bounds for Sparse PCA [1] was well-received in NIPS 2015. Moreover, many works relevant to machine learning such as clustering, signal recovery, mean/covariance estimation, etc., often employ spectral methods or semidefinite programming which are usually simpler forms of SoS algorithms. Therefore, we expect our results to be of broader interest to the NeurIPS community.
>
> References:
>
> [1] T. Ma and A. Wigderson. Sum-of-Squares lower bounds for sparse PCA. 2015.

---

### Author Response · Authors · 2022-08-02
**Short note comparing our work to Planted Clique and Hopkins et al.**

We present some differences, conceptual and technical, from the works Planted Clique [1] and Hopkins et al. [2].

**Differences from Planted Clique [1]:**

The high-level idea of pseudo-calibration and approximate factorization is roughly similar (also, many other SoS lower bounds since that work (e.g. [3, 5]) adopt a similar approach). However, there are several crucial differences and generalizations. We highlight some examples below.
1. In [1], the coefficients $\lambda_{\sigma \circ \sigma’^T}$ (stated in our language for convenience) break into product of 2 real numbers, $v_{\sigma}.v_{\sigma’^T}$. Therefore, the PSD factorization is almost explicit. However, for sparse PCA, such a factorization is not immediate. A first contribution of our work is to observe that, to conclude the same result, it’s enough for the matrix filled with $\lambda_{\sigma \circ \sigma’^T}$ to be PSD. Please see L752-754 for more on this.
2. This led us to approach this problem by defining and using coefficient matrices (like the one above) and carry out the entire analysis in terms of these matrices. To relate back, the coefficient matrix from above is rank-1 in planted clique, whereas it is not rank-1 for sparse PCA. Doing the analysis directly as in planted clique would quickly become technically unwieldy.
3. While planted clique deals with graph matrices with a single type of vertex, the Wishart model of Sparse PCA requires two types of vertices (such matrices were introduced and studied recently in [4]). When we attempt to redo the analysis, there are various technical innovations and generalizations to be made, e.g. the notion of intersection patterns in section G.3, the generalized intersection tradeoff lemma in section H.5.

**Differences from Hopkins et al. [2]:**

As several reviewers noted, [2] asserts similar results for the +/-1 variant of related Wigner model of PCA and +/-1 variant of Tensor PCA and suggests high-level ideas to show them. The main reasons these ideas don’t carry over for Gaussian inputs or the Wishart model immediately are as follows
1. We need the generalizations of [1] stated above (that were not explicit in [2]).
2. Graph matrices require two types of vertices in the Wishart model, as opposed to one type of vertex in the Wigner model. Therefore, we need completely new charging arguments to show lower bounds. See in particular section K (especially lemmas K.3 and K.7), where we explicitly carry out the analysis giving our almost-tight tradeoffs. This analysis is not present in their work, nor does it follow from their high-level ideas.
3. Their ideas are not made explicit. And as we saw in our work, there are a lot of intense technical details to be figured out, such as working with coefficient matrices, using the generalized graph matrices from [4], defining intersection patterns, generalizing intersection tradeoff lemmas, to name a few. Considering the special importance of PCA problems to the ML and TCS community, we consider it an important research endeavor to make these arguments explicit and formal. We undertake this in our work.

On a related note, we highlight that our main theorem can also recover [2]’s results. In particular, we just need to change B_norm(.) accordingly for +/-1 inputs (such bounds are available blackbox in [4]), construct the coefficient matrices and verify the conditions.

References:

[1] B. Barak, S. B. Hopkins, J. Kelner, P. Kothari, A. Moitra, and A. Potechin. A nearly tight sum-of-squares lower bound for the planted clique problem, 2016.

[2] Samuel B Hopkins, Pravesh K Kothari, Aaron Potechin, Prasad Raghavendra, Tselil Schramm, and David Steurer. The power of sum-of-squares for detecting hidden structures. 2017

[3] Mrinalkanti Ghosh, Fernando Granha Jeronimo, Chris Jones, Aaron Potechin, and Goutham Rajendran. Sum-of-Squares lower bounds for Sherrington-Kirkpatrick via Planted Affine Planes. 2020

[4] Kwangjun Ahn, Dhruv Medarametla, and Aaron Potechin. Graph matrices: Norm bounds and Applications. 2020

[5] Chris Jones, Aaron Potechin, Goutham Rajendran, Madhur Tulsiani, and Jeff Xu. Sum-of-Squares lower bounds for sparse independent set. 2021

---

### Meta-Review · Area_Chair_MztE · 2022-08-26

**Recommendation:** Accept
**Confidence:** Certain

**Metareview:**

The reviewers appreciate the solid theoretical results on the hardness of sparse and tensor PCA. The nearly sharp characterization of limitations of sum-of-square methods makes the paper stand out. The proof techniques may potentially be useful for other problems. Based on the above, I recommend acceptance. Meanwhile, please carefully revise the paper to improve presentation. As the paper is technically involved, it would be nice to better elaborate the proof ideas and highlight the key challenges.

**Award:**

No

---

### Decision · Program_Chairs · 2022-09-14

Accept